# Is a scaling factor required to obtain closure between measured and modelled atmospheric O4 absorptions? – An assessment of uncertainties of measurements and radiative transfer simulations for two days during the MAD-CAT campaign

Thomas Wagner[1], Steffen Beirle[1], Nuria Benavent[2], Tim Bösch[3], Ka Lok Chan[4], Sebastian Donner[1], Steffen Dörner[1], Caroline Fayt[5], Udo Frieß[6], David García-Nieto[2], Clio Gielen[5*], David González-Bartolome[7], Laura Gomez[7], François Hendrick[5], Bas Henzing[8], Jun Li Jin[9], Johannes Lampel[6], Jianzhong Ma[10], Kornelia Mies[1], Mónica Navarro[7], Enno Peters[3**], Gaia Pinardi[5], Olga Puentedura[7], Janis Puķīte[1], Julia Remmers[1], Andreas Richter[3], Alfonso Saiz-Lopez[2], Reza Shaiganfar[1], Holger Sihler[1], Michel Van Roozendael[5], Yang Wang[1], Margarita Yela[7]

[1] Max Planck Institute for Chemistry, Mainz, Germany
[2] Department of Atmospheric Chemistry and Climate, Institute of Physical Chemistry Rocasolano (CSIC), Spain.
[3] University of Bremen, Germany
[4] Meteorological Institute, Ludwig-Maximilians-Universität München, Germany
[5] Royal Belgian Institute for Space Aeronomy (BIRA-IASB), Brussels, Belgium
[6] University of Heidelberg, Germany
[7] Instituto Nacional de Tecnica Aeroespacial (INTA), Spain
[8] TNO, Netherlands Institute for Applied Scientific Research
[9] CMA Meteorological Observation Center, China
[10] Chinese Academy of Meteorological Science, China
* currently at the Institute of Astronomy, KU Leuven, Belgium
** Now at Institute for protection of maritime infrastructures, German Aerospace Center (DLR), Bremerhaven, Germany

**Abstract**

In this study the consistency between MAX-DOAS measurements and radiative transfer simulations of the atmospheric $O_4$ absorption is investigated on two mainly cloud-free days during the MAD-CAT campaign in Mainz, Germany, in Summer 2013. In recent years several studies indicated that measurements and radiative transfer simulations of the atmospheric $O_4$ absorption can only be brought into agreement if a so-called scaling factor (<1) is applied to the measured $O_4$ absorption. However, many studies, including such based on direct sun light measurements, came to the opposite conclusion, that there is no need for a scaling factor. Up to now, there is no broad consensus for an explanation of the observed discrepancies between measurements and simulations. Previous studies inferred the need for a scaling factor from the comparison of the aerosol optical depth derived from MAX-DOAS $O_4$ measurements with that derived from coincident sun photometer measurements. In this study a different approach is chosen: the measured $O_4$ absorption at 360 nm is directly compared to the $O_4$ absorption obtained from radiative transfer simulations. The atmospheric conditions used as input for the radiative transfer simulations were taken from independent data sets, in particular from sun photometer and ceilometer measurements at the measurement site. This study has three main goals: First all relevant error sources of the spectral analysis, the radiative transfer simulations as well as the extraction of the input parameters used for the radiative transfer simulations are quantified. One important result obtained from the analysis

of synthetic spectra is that the $O_4$ absorptions derived from the spectral analysis agree within
1% with the corresponding radiative transfer simulations at 360 nm. Based on the results from
sensitivity studies, recommendations for optimised settings for the spectral analysis and
radiative transfer simulations are given. Second, the measured and simulated results are
compared for two selected cloud free days with similar aerosol optical depth but very
different aerosol properties.: On 18 June, measurements and simulations agree within their
(rather large) uncertainties (the ratio of simulated and measured $O_4$ absorptions is found to be
1.01±0.16). In contrast, on 8 July measurements and simulations significantly disagree: For
the middle period of that day the ratio of simulated and measured $O_4$ absorptions is found to
be 0.82 ±0.10, which differs significantly from unity. Thus for that day a scaling factor is
needed to bring measurements and simulations into agreement. Third, recommendations for
further intercomparison exercises are derived. One important recommendation for future
studies is that aerosol profile data should be measured at the same wavelengths as the MAX-
DOAS measurements. Also the altitude range without profile information close to the ground
should be minimised and detailed information on the aerosol optical and/or microphysical
properties should be collected and used.
The results for both days are inconsistent, and no explanation for a $O_4$ scaling factor could be
derived in this study. Thus similar, but more extended future studies should be performed,
including more measurement days, and more instruments. Also additional wavelengths should
be included.
**1 Introduction**
Observations of the atmospheric absorption of the oxygen collision complex $(O_2)_2$ (in the
following referred to as $O_4$, see Greenblatt et al. (1990)) are often used to derive information
about atmospheric light paths from remote sensing measurements of scattered sun light (made
e.g. from ground, satellite, balloon or airplane). Since atmospheric radiative transport is
strongly influenced by scattering on aerosol and cloud particles, information on the presence
and properties of clouds and aerosols can be derived from $O_4$ absorption measurements.
Early studies based on $O_4$ measurements focussed on the effect of clouds (e.g. Erle et al.,
1995; Wagner et al., 1998; Winterrath et al., 1999; Acarreta et al., 2004; Sneep et al., 2008;
Heue et al., 2014; Gielen et al., 2014; Wagner et al., 2014), which is usually stronger than that
of aerosols. Later also aerosol properties were derived from $O_4$ measurements, in particular
from Multi-AXis- (MAX-) DOAS measurements (e.g. Hönninger et al., 2004; Wagner et al.,
2004; Wittrock et al., 2004; Friess et al., 2006; Irie et al., 2008; Clémer 2010; Friess et al.,
2016 and references therein). For the retrieval of aerosol profiles usually forward model
simulations for various assumed aerosol profiles are compared to measured $O_4$ slant column
densities (SCD, the integrated $O_4$ concentration along the atmospheric light path). The aerosol
profile associated with the best fit between the forward model and measurement results is
considered as the most probable atmospheric aerosol profile (for more details, see e.g. Frieß et
al., 2006). Note that in some cases no unique solution might exist, if different atmospheric
aerosol profiles lead to the same $O_4$ absorptions. MAX-DOAS aerosol retrievals are typically
restricted to altitudes below about 4 km; see Friess et al. (2006).
About ten years ago, Wagner et al. (2009) suggested to apply a scaling factor (SF <1) to the
$O_4$ SCDs derived from MAX-DOAS measurements at 360 nm in Milano in order to achieve
agreement with forward model simulations. They found that on a day with low aerosol load
the measured $O_4$ SCDs were larger than the model results, even if no aerosols were included
in the model simulations. If, however, the measured $O_4$ SCDs were scaled by a SF of 0.81,
good agreement with the forward model simulations (and nearby AERONET measurements)
was achieved. Similar findings were then reported by Clémer et al. (2010), who suggested a
SF of 0.8 for MAX-DOAS measurements in Beijing. Interestingly, they applied this SF to
four different $O_4$ absorption bands (360, 477, 577, and 630 nm).
While with the application of a SF the consistency between forward model and measurements
was substantially improved, both studies could not provide an explanation for the physical
mechanism behind such a SF. In the following years several research groups applied a SF in
their MAX-DOAS aerosol profile retrievals. However, a similarly large fraction of studies
(including direct sun measurements and aircraft measurements, see Spinei et al. (2015)) did
not find it necessary to apply a SF to bring measurements and forward model simulations into
agreement. An overview on the application of a SF in various MAX-DOAS publications after
2010 is provided in Table 1. Up to now, there is no community consensus on whether or not a
SF is needed for measured $O_4$ DSCDs. This is a rather unfortunate situation, because this
ambiguity directly affects the aerosol results derived from MAX-DOAS measurements and
thus the general confidence in the method.
So far, most of the studies deduced the need for a SF in a rather indirect way: aerosol
extinction profiles derived from MAX-DOAS measurements using different SF are usually
compared to independent data sets (mostly AOD from sun photometer observations) and the
SF leading to the best agreement is selected. In many cases SF between 0.75 and 0.9 were
derived.
In this study, we follow a different approach: similar to Ortega et al. (2016) we directly
compare the measured $O_4$ SCDs with the corresponding SCDs derived with a forward model
(consisting of a radiative transfer model and assumptions of the state of the atmosphere). For
this comparison, atmospheric conditions which are well characterised by independent
measurements are chosen. Such a procedure allows in particular quantifying the influence of
the uncertainties of the individual processing steps.
One peculiarity of this comparison is that the measured $O_4$ SCDs are first converted into their
corresponding air mass factors (AMF), which are defined as the ratio of the SCD and the
vertical column density (VCD, the vertically integrated concentration) (Solomon et al., 1987).

$$AMF = \frac{SCD}{VCD} \tag{1}$$

The 'measured' $O_4$ AMF is then compared to the corresponding AMF derived from radiative
transfer simulations for the atmospheric conditions during the measurements:

$$AMF_{measured} \overset{?}{=} AMF_{simulated} \tag{2}$$

The conversion of the measured $O_4$ SCDs into AMFs is carried out to ensure a simple and
direct comparison between measurements and forward model simulations. Here it should be
noted that in addition to the AMFs also so-called differential AMFs (dAMFs) will be
compared in this study. The dAMFs represent the difference between AMFs for
measurements at non-zenith elevation angles α and at 90° for the same elevation sequence:

$$dAMF_\alpha = AMF_\alpha - AMF_{90°} \tag{3}$$

For the comparison between measured and simulated $O_4$ (d)AMFs, two mostly cloud-free
days (18 June and 8 July 2013) during the Multi Axis DOAS Comparison campaign for
Aerosols and Trace gases (MAD-CAT) campaign are chosen (http://joseba.mpch-
mainz.mpg.de/mad_cat.htm). As discussed in more detail in section 4.2.2, based on the
ceilometer and sun photometer measurements, three periods on each of the two days are
selected, during which the variation of the aerosol profiles was relatively small (see Table 2).
In addition to the aerosol profiles, also other atmospheric properties are averaged during these
periods before they are used as input for the radiative transfer simulations.
The comparison is carried out for the $O_4$ absorption band at 360 nm, which is the strongest $O_4$
absorption band in the UV. In principle also other $O_4$ absorption bands (e.g. in the visible
spectral range) could be chosen, but these bands are not covered by the wavelength range of
the MPIC instrument. Thus they are not part of this study.
The comparison between measurements and simulations is performed in three different steps:
First, for two selected periods in the middle of both days, the ratios between measured and
simulated $O_4$ (d)AMFs are calculated for standard settings of the spectral retrieval and
radiative transfer simulations (for details see below). In a second step the uncertainties of the
measurements and simulations are investigated. In the final step, it is investigated whether the
ratio of measured and simulated $O_4$ (d)AMFs agree with unity taking into account these
uncertainties.
Deviations between forward model and measurements can have different reasons. In the
following an overview on these error sources and the way they are investigated in this study
are given:
a) Calculation of $O_4$ profiles and $O_4$ VCDs (eq. 1):
Profiles and VCDs of $O_4$ are derived from pressure and temperature profiles. The
uncertainties of the pressure and temperature profiles are quantified by sensitivity studies and
by the comparison of the extraction results derived from different groups/persons (see Table
169    3).
b) Calculation of $O_4$ (d)AMFs from radiative transfer simulations:
Besides differences between the different radiative transfer codes, the dominating sources of
uncertainty are those related to the input parameters. They are investigated by sensitivity
studies and by the comparison of extracted input data by different groups/persons. Also the
effects of operating different radiative transfer models by different groups are investigated.
c) Analysis of the $O_4$ (d)AMFs from MAX-DOAS measurements:
Uncertainties of the spectral analysis results are caused by errors and imperfections of the
measurements/instruments, by the dependence of the analysis results on the specific fit
settings, and the uncertainties of the $O_4$ cross sections including their temperature
dependence. They are investigated by systematic variation of the DOAS fit settings (for
measured and synthetic spectra), and by comparison of analysis results obtained from
different groups and/or instruments.
The paper is organised as follows: in section 2, information on the selected days during the
MAD-CAT campaign, on the MAX-DOAS measurements, and on the data sets from
independent measurements is provided. Section 3 presents initial comparison results for the
selected days using standard settings. In section 4 the uncertainties associated with each of the
various processing steps of the spectral analysis and the forward model simulations are
quantified by comparing them to the results for the standard settings. Section 5 presents a
summary and conclusions.
**2 MAD-CAT campaign, MAX-DOAS instruments and other data sets used in this study**
The Multi Axis DOAS Comparison campaign for Aerosols and Trace gases (MAD-CAT)
(http://joseba.mpch-mainz.mpg.de/mad_cat.htm) took place in June and July 2013 on the roof
of the Max-Planck-Institute for Chemistry in Mainz, Germany. The main aim of the campaign
was to compare MAX-DOAS retrieval results of several atmospheric trace gases like $NO_2$,
HCHO, HONO, CHOCHO as well as aerosols. The measurement location was at 150m above
sea level at the western edge of the city of Mainz.

## 2.1 MAX-DOAS instruments

During the MAD-CAT campaign, 11 MAX-DOAS instruments were operated by different groups; an overview can be found at the website http://joseba.mpch-mainz.mpg.de/equipment.htm. The main viewing direction of the MAX-DOAS instruments was towards north-west (51° with respect to North). Measurements at this viewing direction were the main focus of this study, but a few comparisons using the 'standard settings' (see section 3) were also carried out for three other azimuth angles (141°, 231°, 321°, see Fig. A2 I in appendix A1). Each elevation sequence contains the following elevation angles: 1, 2, 3, 4, 5, 6, 8, 10, 15, 30 and 90°. In this study, in addition to the MPIC instrument, also spectra from 3 other MAX-DOAS instruments were analysed. The instrumental details are given in Table 4. The spectra of the MPIC instrument are available at the website http://joseba.mpch-mainz.mpg.de/e_doc_zip.htm.

## 2.2 Additional data sets

In order to constrain the radiative transfer simulations, independent measurements and data sets were used. In particular, information on atmospheric pressure, temperature and relative humidity, as well as aerosol properties is used. In addition to local in situ measurements from air quality monitoring stations and remote sensing measurements by a ceilometer and a sun photometer, also ECMWF reanalysis data were used. An overview on these data sets is given in Table 5. The data sets used in this study are available at the websites http://joseba.mpch-mainz.mpg.de/a_doc_zip.htm and http://joseba.mpch-mainz.mpg.de/c_doc_zip.htm.

## 2.3 RTM simulations

Several radiative transfer models are used to calculate $O_4$ (d)AMFs for the selected days. As input, vertical profiles of temperature, pressure, relative humidity and aerosol extinction extracted from the independent data sets (see section 2.2 and 4) were used. The vertical resolution is high in the lowest layers and decreases with increasing altitude (see Table A1 in appendix A1). The upper boundary of the vertical grid is set to 1000 km. The lower boundary of the model grid represents the surface elevation of the instrument (150 m above sea level). For the 'standard run', a surface albedo of 5% is assumed and the aerosol optical properties are described by a Henyey-Greenstein phase function with an asymmetry parameter of 0.68 and a single scattering albedo of 0.95. Both values represent typical urban aerosols (see e.g. Dubovik et al., 2002). Ozone absorption was not considered, because it is very small at 360 nm. The MAD-CAT campaign took place around summer solstice. Thus the same dependence of the solar zenith angle (SZA) and relative azimuth angle (RAZI) on time is used for both days (see Table A2 in the appendix A1). The input data used for the radiative transfer simulations are available at the website http://joseba.mpch-mainz.mpg.de/d_doc_zip.htm. In the following sub-sections the different radiative transfer models used in this study are described.

### 2.3.1 MCARTIM

The full spherical Monte Carlo radiative transfer model MCARTIM (Deutschmann et al., 2011) explicitly simulates individual photon trajectories including the photon interactions with molecules, aerosol particles and the surface. In this study two versions of MCARTIM are used: version 1 and version 3. Version 1 is a 1-D scalar model. Version 3 can also be run in 3-

D and vector modes. In version 1 Rotational Raman scattering (RRS) is partly taken into account: the RRS cross section and phase function are explicitly considered for the determination of the photon paths, but the wavelength redistribution during the RRS events is not considered. In version 3 RRS can be fully taken into account. If operated in the same mode (1-D scalar) both models show excellent agreement.

**2.3.2 LIDORT**

In this study the LIDORT version 3.3 was used. The Linearized Discrete Ordinate Radiative Transfer (LIDORT) forward model (Spurr et al., 2001; Spurr et al., 2008) is based on the discrete ordinate method to solve the radiative transfer equation (e.g.: Chandrasekhar, 1960; Chandrasekhar, 1989; Stamnes et al., 1988). This model considers a pseudo-spherical multi-layered atmosphere including several anisotropic scatters. The formulation implemented corrects for the atmosphere curvature in the solar and single scattered beam, however the multiple scattering term is treated in the plane-parallel approximation. The properties of each of the atmospheric layers are considered homogenous in the corresponding layer. Using finite differences for the altitude derivatives, this linearized code converts the problem into a linear algebraic system. Through first order perturbation theory, it is able to provide radiance field and radiance derivatives with respect to atmospheric and surface variables (Jacobians) in a single call. LIDORT was used in several studies to derive vertical profiles of aerosols and trace gases from MAX-DOAS (e.g. Clémer et al., 2010; Hendrick et al., 2014; Franco et al., 2015).

**2.3.3 SCIATRAN**

The RTM SCIATRAN (Rozanov et al. 2014) was used in its full-spherical mode including multiple scattering but without polarization. In the operation mode used here, SCIATRAN solves the transfer equations using the discrete ordinate method. In this study, SCIATRAN was used by two groups: The IUP Bremen group used v3.8.3 for the $O_4$ dAMFs simulations (without Raman scattering). The MPIC group used v3.6.11 for the calculation of synthetic spectra (see Section 2.4) and for the $O_4$ dAMFs simulations (including Raman scattering).

**2.4 Synthetic spectra**

In addition to AMFs and dAMFs, also synthetic spectra were simulated. They are analysed in the same way as the measured spectra, which allows the investigation of two important aspects:
a) The derived $O_4$ dAMFs from the synthetic spectra can be compared to the $O_4$ dAMFs obtained directly from the radiative simulations at one wavelength (here: 360 nm) using the same settings. In this way the consistency of the spectral analysis results and the radiative transfer simulations is tested.
b) Sensitivity tests can be performed varying several fit parameters, e.g. the spectral range or the DOAS polynomial, and their effect on the derived $O_4$ dAMFs can be assessed.
Synthetic spectra are simulated using SCIATRAN taking into account rotational Raman scattering. The basic simulation settings are the same as for the RTM simulations of the $O_4$ (d)AMFs described above. In order to minimise the computational effort, for the profiles of temperature, pressure, relative humidity and aerosol extinction the input data for only two periods (18 June: 11:00 – 14:00, 8 July: 7:00 – 11:00, see Table 2) are used for the whole day.

Thus 'perfect' agreement with the measurements can only be expected for the two selected
periods. Aerosol optical properties (phase function and single scattering albedo) are taken
from AERONET measurements of the two selected days. Although the wavelength
dependencies of both quantities (and also for the aerosol extinction) are considered, it should
be noted that the associated uncertainties are probably rather large, since the optical properties
in the UV had to be extrapolated from measurements in the visible spectral range.
Spectra were simulated at a spectral resolution of 0.01 nm and convolved with a Gaussian slit
function of 0.6 nm full width at half maximum (FWHM), which is similar to those of the
measurements. For the generation of the spectra a high resolution solar spectrum (Chance and
Kurucz, 2010) and the trace gas absorptions of $O_3$, $NO_2$, HCHO, and $O_4$ are considered (see
Table A3 in appendix A1). The assumed tropospheric profiles of $NO_2$ and HCHO are similar
to those retrieved from the MAX-DOAS observations during the selected periods. Time series
of the tropospheric VCDs of $NO_2$ and HCHO for the two selected days are shown in Fig. A1
in appendix 1.
Two sets of synthetic spectra were simulated, one taking into account the temperature
dependence of the $O_4$ cross section and the other not. For the case without considering the
temperature dependence, the $O_4$ cross section for 293 K is used. In addition to spectra without
noise, also spectra with noise (sigma of the noise is assumed as $7.5 \cdot 10^{-4}$ times the intensity)
were simulated. The synthetic spectra are available at the website http://joseba.mpch-
mainz.mpg.de/f_doc_zip.htm.
**322 3 Strategies used in this studies and comparison results for 'standard settings'**
**324 3.1 Selection of days**
For the comparison of measured and simulated $O_4$ dAMFs, two mostly cloud-free days during
the MAD-CAT campaign (18 June and 8 July 2013) were selected. On both days the AOD
measured by the AERONET sun photometer at 360 nm was between 0.25 and 0.4 (see Fig. 1).
In spite of the similar AOD, very different aerosol properties at the surface were found on the
two days: on 18 June much higher concentrations of large aerosol particles ($PM_{2.5}$ and $PM_{10}$)
are found. These differences are also represented by the large differences of the Ångström
parameter for long wavelengths (440 – 870 nm) on both days. Also the aerosol height profiles
are different: On 8 July rather homogenous profiles with a layer height of about 2 km occur.
On 18 June the aerosol profiles reach to higher altitudes, but the highest extinction is found
close to the surface. Also the temporal variability of the aerosol properties, especially the
near-surface concentrations, is much larger on 18 June.
**338 3.2 Different levels of comparisons**
The comparison between the forward model and MAX-DOAS measurements is performed in
different depth for different subsets of the measurements:
a) A quantitative comparison of $O_4$ AMFs and $O_4$ dAMFs is performed for 3° elevation angle
at the standard viewing direction (51° with respect to North) for the middle periods of both
selected days. During these periods the uncertainties of the measurement and the radiative
transfer simulations are smallest because around noon the measured intensities are high and
the variation of the SZA is small. During the selected periods, also the variation of the
ceilometer profiles is relatively small. These comparisons thus constitute the core of the
comparison exercise and all sensitivity studies are performed for these two periods. The
elevation angle of 3° is selected because for such a low elevation angle the atmospheric light
paths and thus the $O_4$ absorption are rather large. Moreover, as can be seen in Fig. 2, the $O_4$
(d)AMFs for 3° are very similar to those for 1° and 6°, especially on 8 July 2013. Sensitivity
studies showed that a wrong elevation angle calibration (±0.5°) led to only small changes
(<1%) of the $O_4$ (d)AMFs. Changes of the field of view between 0.2 and 1.1° led to even
smaller differences. These findings indicate that possible uncertainties of the calibration of the
elevation angles of the instruments can be neglected. Here it is interesting to note that on 18
June even slightly lower $O_4$ (d)AMFs are found for the low elevation angles. This is in
agreement with the finding of high aerosol extinction in a shallow layer above the surface (see
Fig. 1). The azimuth angle of 51° is chosen, because it was the standard viewing direction
during the MAD-CAT campaign and measurements for this direction are available from
different instruments.
b) The quantitative comparison for 3° elevation and azimuth of 51° is also extended to the
periods prior and after the middle periods of the selected days. However, to minimise the
computational efforts, some sensitivity studies are not carried out for the first and last periods.
c) The comparison is extended to more elevation angles (1°, 3°, 6°, 10°, 15°, 30°, 90°) and
azimuth angles (51°, 141°, 231°, 321°). For this comparison only the standard settings for the
DOAS analysis and the radiative transfer simulations are applied (see Tables 6 and 7). The
comparison results for the MPIC MAX-DOAS measurements are shown in appendix A2. The
purpose of this comparison is to check whether for other viewing angles similar results are
found as for 3° elevation at 51° azimuth direction.
**3.3 Quantitative comparison for 3° elevation in standard azimuth direction**
Fig. 3 presents a comparison of the measured and simulated $O_4$ (d)AMFs for 3° elevation and
51° azimuth on both days. For the spectral analysis and the radiative transfer simulations the
respective 'standard settings' (see Tables 6 and 7) were used. On 8 July the simulated $O_4$
(d)AMFs systematically underestimate the measured $O_4$ (d)AMFs by up to 40%. Similar
results are also obtained for other elevation and azimuth angles (see appendix A2), the
differences becoming smaller towards higher elevation angles. In contrast, no systematic
underestimation is observed for most of 18 June. For some periods of that day the simulated
$O_4$ (d)AMFs are even larger than the measured $O_4$ (d)AMFs. However, here it should be
noted that the aerosol extinction profile of the 'standard settings' (using linear extrapolation
below 180 m where no ceilomter data are available) probably underestimates the aerosol
extinction close to the surface. If instead a modified aerosol profile with strongly increased
aerosol extinction below 180 m and the maximum AOD during that period is used (see Fig.
A31 in appendix A5) the corresponding (d)AMFs fall below the measured $O_4$ (d)AMFs
(green curves in Fig. A4 in appendix A2). More details on the extraction of the aerosol
extinction profiles are given in section 4.2.2 and appendix A5).
The average ratio of simulated to measured (d)AMFs (for the standard settings) during the
middle periods on both days are given in Table 8. For 18 June they are close to unity, for 8
July they are much lower (0.83 for the AMF, and 0.69 for the dAMF).
**4 Estimation of the uncertainties of the different processing steps**
There are 3 major processing steps, for which the uncertainties are quantified in this section:
a) The determination of the $O_4$ height profiles and corresponding $O_4$ vertical column densities.
b) The simulation of $O_4$ (d)AMFs by the forward model
c) The analysis of $O_4$ (d)AMFs from the MAX-DOAS measurements.
**4.1 Determination of the vertical $O_4$ profile and the $O_4$ VCD**

The $O_4$ VCD is required for conversion of measured (d)SCDs into (d)AMFs (eq. 1). $O_4$
profiles are also needed for the calculation of $O_4$ (d)AMFs. The accuracy of the calculated $O_4$
height profile and the $O_4$ VCD depends in particular on two aspects:
a) is profile information on temperature, pressure and (relative) humidity available?
b) what is the accuracy of these data sets?
Additional uncertainties are related to the details of the calculation of the $O_4$ concentration
and $O_4$ VCDs from these profiles. Both sources of uncertainties are investigated in the
following sub sections.
**4.1.1 Extraction of vertical profiles of temperature and pressure**
The procedure of extracting temperature and pressure profiles depends on the availability of
measured profile data or surface measurements. If profile data are available (e.g. from sondes
or models) they could be directly used. If only surface measurements are available, vertical
profiles of temperature and pressure could be calculated making assumptions on the lapse rate
(here we assume a value of -0.65 K / 100 m). If no measurements or model data are available,
profiles from the US standard atmosphere might be used (United States Committee on
Extension to the Standard Atmosphere, 1976). In appendix A3 the different procedures for the
extraction of pressure and temperature profiles are described in detail for the two days of the
MAD-CAT campaign. For these days the optimum choice was to combine the model data and
the surface measurements. In that way, the diurnal variation in the boundary layer could be
considered. In Fig. 4 temperature and pressure profiles extracted from the combination of in
situ measurements and ECMWF data are shown. These profiles probably best match the true
atmospheric profiles.
A comparison of temperature profiles extracted by different methods for two selected periods
on both days is shown in Fig. 5. For 8 July (right), rather good agreement is found, but for 18
June (left) the agreement is worse (differences up to 20 K). Of course, the differences between
the true and the US standard atmosphere profiles can become even larger, depending on
location and season. So the use of a fixed temperature and pressure profile should always be
the last choice. In contrast, the simple extrapolation from surface values can be very useful if
no profile data are available, because the uncertainties of this method are usually smallest at
low altitudes, where the bulk of $O_4$ is located.
**4.1.2 Calculation of $O_4$ concentration profiles and $O_4$ VCDs**
From the temperature and pressure profiles the oxygen ($O_2$) concentration is calculated. Here
also the effect of the atmospheric humidity profiles should be taken into account (see
appendix A3), because it can have a considerable effect on the near-surface layers (at least for
temperatures of about > 20°C). Finally, the square of the oxygen concentration is calculated
and used as proxy for the $O_4$ concentration consistently with assumptions made in the
determination of the absorption cross-sections (see Greenblatt et al., 1990). The uncertainties
of the derived $O_4$ concentration (and the corresponding $O_4$ VCD) caused by the uncertainty of
the input profiles is estimated by varying the input parameters (for details see appendix A3).
For both selected days during the MAD-CAT campaign the total uncertainty is estimated to
be about 1.5% assuming that the uncertainties of the individual input parameters are
independent,.
Further uncertainties arise from the procedure of the vertical integration of the $O_4$
concentration profiles. We tested the effect of using different vertical grids and altitude
ranges. It is found that the vertical grid should not be coarser than 100 m (for which a
deviation of the $O_4$ VCD of 0.3% compared to a much finer grid is found). If e.g. a vertical

grid with 500 m layers is used, the deviation increases to about 1.3%. The integration should be performed over an altitude range up to 30 km. If lower maximum altitudes are used, the $O_4$ VCD will be substantially underestimated: deviations of 0.1 %, 0.5 %, and 11% are found if the integration is performed only up to 25 km, 20 km, and 10 km, respectively. Here it should be noted that the exact consideration of the altitude of the measurement site is also very important: A deviation of 50 m already leads to a change of the $O_4$ VCD by 1%. For the MAD-CAT measurements the altitude of the instruments is 150m ±20m.

Finally, the effects of individual extraction and integration procedures are investigated by comparing the results from different groups (see Fig. 6, and Fig. A5 in appendix A3). Except for some extreme cases, the extracted temperatures typically differ by less than 3 K below 10 km. However, the deviations are typically larger for the profiles extrapolated from the surface values and in particular for the US standard atmosphere (up to > 10 K below 10 km). The variations of the extracted pressure profiles are in general rather small (< 1% below 10 km, except one obvious outlier). However, the deviations of the profiles extrapolated from the surface values and especially the US standard atmosphere are much larger (up to > 5 % below 10 km). The resulting deviations of the $O_4$ concentration from the different extractions are typically <3% below 10 km (and up to > 20 % above 10 km for the US standard atmosphere).

In Fig. 7 the $O_4$ VCDs calculated for the $O_4$ profiles extracted from the different groups and for the profiles extrapolated from the surface values and the US standard atmosphere are shown. The VCDs for the profiles extracted by the different groups agree within 2.5%. The deviations for the profiles extrapolated from the surface values are only slightly larger (typically within 3%), but show a large variability throughout the day, which is caused by the systematic increase of the surface temperature during the day (with temperature inversions in the morning on the two selected days). The deviations of the US standard atmosphere are up to 5% (but can of course be larger for other seasons and locations, see also Ortega et al. (2016).

Ultimately, the accuracy with which $O_4$ concentrations can be calculated is limited by the assumption that $O_4$ ($O_2$-$O_2$) is pure collision induced absorption. If the oxygen concentration profile is well known, the uncertainty due to bound $O_4$ is smaller than 0.14% in Earth's atmosphere (Thalman and Volkamer, 2013).

Together with the uncertainties related to the input data sets, the total uncertainty of the $O_4$ VCDs determined for both selected days is estimated as 3%.

**4.2 Uncertainties of the $O_4$ (d)AMFs derived from radiative transfer simulations**

The most important uncertainties of the simulated $O_4$ (d)AMFs are related to the uncertainties of the input parameters used for the simulations, in particular the aerosol properties. Further uncertainties are caused by imperfections of the radiative transfer models. These sources of uncertainty are discussed and quantified in the following sub sections.

**4.2.1 Uncertainties of the $O_4$ (d)AMFs caused by uncertainties of the input parameters**

In this section the effect of the uncertainties of various input parameters on the $O_4$ (d)AMFs is investigated. The general procedure is that the input parameters are varied individually and the corresponding changes of the $O_4$ (d)AMFs compared to the standard settings are quantified.

First, the effect of the $O_4$ profile shape is investigated. In contrast to the effect of the (absolute) profile shape on the $O_4$ VCD (section 4.1), here the effect of the relative profile shape on the $O_4$ AMF is investigated. The $O_4$ (d)AMFs simulated for the $O_4$ profiles extracted by the different groups (and for those derived from the US standard atmosphere and the profiles extrapolated from the surface values, see section 4.1) are compared to those for the

MPIC $O_4$ profiles (using the standard settings). The corresponding ratios are shown in Fig. A6
and Table A4 in appendix A4. For the $O_4$ profiles extracted by the different groups, and for
$O_4$ profiles extrapolated from the surface values, small variations are found (typically < 2%).
For the US standard atmosphere larger deviations (up to 7%) are derived.
Next the effect of the aerosol extinction profile is investigated. In this study, aerosol
extinction profiles are derived from the combined ceilometer and sun photometer
measurements (see Table 5). In short, the ceilometer measurements of the attenuated
backscatter are scaled by the simultaneously measured aerosol optical depth (AOD) from the
sun photometer to obtain the aerosol extinction profile. Also the self-attenuation of the aerosol
is taken into account. The different steps are illustrated in Fig. 8 and described in detail in
appendix A5. In the extraction procedure, several assumptions have to be made: First, the
ceilometer profiles have to be extrapolated for altitudes below 180 m, for which the
ceilometer is not sensitive. Furthermore, they have to be averaged over several hours and are
in addition vertically smoothed (above 2 km) to minimise the rather large scatter. Finally,
above 5 to 6 km (depending on the ceilometer profiles) the extinction is set to zero because of
the further increasing scatter and the usually small extinctions. This assumption reflects a
practical limitation of the ceilometer likely responsible for the larger variability in the profile
shape aloft by different groups. Another assumption is that the Angström exponent and the
LIDAR ratio are independent of altitude, which is typically not strictly fulfilled (the LIDAR
ratio describes the ratio between the extinction and backscatter probabilities of the molecules
and aerosol particles).
These uncertainties are quantified by sensitivity studies, in particular the effect of the
extrapolation below 180 m and the altitude above which the aerosol extinction is set to zero.
Other uncertainties, like the effect of the assumption of a constant LIDAR ratio are more
difficult to quantify without further information (see below). The effect of temporal averaging
and smoothing is probably negligible for 8 July, because similar height profiles are found for
all three periods of that day, but on 18 June the effect might be more important.
Fig. 9 shows a comparison of the aerosol extinction profiles extracted by the different groups
for the three periods on both days. Especially on 8 July systematic differences are found.
They are caused by the different altitudes, above which the aerosol extinction is set to zero. In
combination with the scaling of the profiles with the AOD obtained from the sun photometer,
this also influences the extinction values close to the surface. Deviations up to 18% are found
for the first period of 8 July. These deviations also have an effect on the corresponding $O_4$
(d)AMFs, where higher values are obtained for the profiles (INTA and IUPB 300m) which
were extracted for a larger altitude range (Fig. A7 and Table A5 in the appendix A4). Here it
is interesting to note that these differences are not related to the direct effect of the aerosol
extinction at high altitude, but to the corresponding (via the scaling with the AOD) decrease
of the aerosol extinction close to the surface. Larger deviations (up to 4%) are found for 8
July, while the deviations on 18 June are within 3%. This effect is further examined in
appendix A6.
In Fig. A8 and Table A6 in appendix A4, the effect of the different extrapolations of the
aerosol extinction profile below 180 m on the $O_4$ (d)AMFs is quantified. Similar deviations
(up to 5 %) are found for both days.
Finally, we investigated the effect of changing aerosol optical properties with altitude
(changing LIDAR ratio). Such effects are in particular important if the wavelength of the
ceilometer measurements (1064 nm) differs largely from that of the MAX-DOAS
observations (360 nm). Based on the partitioning into fine and coarse mode aerosols (derived
from the sun photometer observations) and the corresponding phase functions and optical
depths, the sensitivity of the ceilometer to fine mode aerosols were estimated (for details see
appendix A5). While for 18 June the contribution of the fine mode to the ceilometer signal is
about 32% on 8 July it is much larger (about 82 %). Thus it can be concluded that the aerosol
extinction profile derived from the ceilometer is largely representative for the fine mode
aerosols on that day. To investigate the effect of the remaining uncertainties, the shape of the
aerosol extinction profile was further modified (for details see appendix A5) taking into
account that the coarse aerosols are typically located at low altitudes. The corresponding
repartitioning of the aerosol extinction profile led to a decrease of the aerosol extinction close
to the surface which is balanced by an increase at higher altitudes (see Fig. A34). The $O_4$
dAMFs calculated for the modified profile are by about 17 % larger than those for the
standard settings (for details see appendix A5).
The effect of elevated aerosol layers (see Ortega et al., 2016) was further investigated by
systematic sensitivity studies (appendix A6). On both selected days enhanced aerosol
extinction was found at elevated layers (Fig. 9). Compared to those reported by Ortega et al.
(2016) the profiles extracted in this study reach even up to higher altitudes. For the
investigation of the effect of changes of the aerosol extinction at different altitudes, the
aerosol extinction profile on 8 July was subdivided into 3 layers (0-1.7 km; 1.7 – 4.9 km; 4.9
– 7 km), and the extinction in the individual layers was increased by +40 %. It was found that
even a strong increase of the aerosol extinction at high altitudes by 40% leads only to an
increase of the $O_4$ dAMFs by 7 %.
Also the effect of horizontal gradients should be briefly discussed. For the selected periods of
both days, the wind direction and wind speed were rather constant. On 18 June the wind
direction was between 80° and 150° with respect to North, and the wind speed was about 2
m/s. On 8 July the wind direction was between 70° and 90° (the wind came from almost the
same direction at which the instruments were looking), and the wind speed was about 3 m/s.
During the 4 hours of the selected period on 8 July, the air masses moved over a distance of
about 40 km. During the 3 hours of the selected period on 18 June, the air masses moved over
a distance of about 20 km. These distances are larger than the distances for which the MAX-
DOAS observations are sensitive (about 5 – 15 km). Since also the AOD and the aerosol
extinction profiles were rather constant during both selected periods, we conclude that for the
measurements considered here horizontal gradients can be neglected. It should also be noted
that the discrepancies between measurements and simulations were simultaneously observed
at all 4 azimuth directions.
In Fig. A9 and Table A7 in appendix A4, the effect of different single scattering albedos
(between 0.9 and 1) on the $O_4$ (d)AMFs is quantified. The effect on the $O_4$ (d)AMFs is up to 4
% on 18 June and up to 2 % on 8 July 2013.
The impact of the aerosol phase function is investigated in two ways: First, simulation results
are compared for Henyey Greenstein phase functions with different asymmetry parameters.
The corresponding results are shown in Fig. A10 and Table A8 in appendix A4. The
differences of the $O_4$ (d)AMFs for the different aerosol phase functions are rather strong: up
to 3% for the $O_4$ AMFs and up to 8% for the $O_4$ dAMFs (larger uncertainties for the dAMFs
are found because of the strong influence of the phase function on the 90° observations). Here
it should be noted that the actual deviations from the true phase function might be even larger.
In order to better estimate these uncertainties, also simulations for phase functions derived
from the sun photometer measurements based on Mie theory (in the following referred to as
Mie phase functions) were performed. A comparison of these Mie phase functions with the
Henyey Greenstein phase functions is shown in Fig. 10. Large differences, especially in
forward direction are obvious. The $O_4$ (d)AMFs for the Mie phase functions are compared to
the standard simulations (using the HG phase function for an asymmetry parameter of 0.68) in
Fig. A11 and Table A9 in appendix A4. Again rather large deviations are found, which are
larger on 18 June (up to 9 %) than on 8 July (up to 5%).

In Fig. A12 and Table A10 in appendix A4, the effect of different surface albedos on the $O_4$ (d)AMFs is quantified. For the considered variations (0.03 to 0.1) the changes of the $O_4$ (d)AMFs are within 2 %.

**4.2.2 Uncertainties of the $O_4$ (d)AMFs caused by imperfections of the radiative transfer models**

The radiative transfer models used in this study are well established and showed very good agreement in several intercomparison studies (e.g. Hendrick et al., 2006; Wagner et al., 2007; Lorente et al., 2017). Nevertheless, they are based on different methods and use different approximations (e.g. with respect to the Earth's sphericity). Thus we compared the simulated $O_4$ (d)AMFs for both days in order to estimate the uncertainties associated to these differences. In Fig. A13 and Table A11 (appendix A4), the comparison results are shown. They agree within a few percent with slightly larger differences for 18 June (up to 6 %) than for 8 July (up to 3 %).
So far, all radiative transfer simulations were carried out without considering polarisation. Thus in Fig. A14 and Table A12 in appendix A4, the results with and without considering polarisation are compared. The corresponding differences are very small (<1%).

**4.2.3 Summary of uncertainties of the $O_4$ AMF from radiative transfer simulations**

Table 9 presents an overview on the different sources of uncertainties of the simulated $O_4$ (d)AMFs derived from the comparison of the results from different groups and the sensitivity studies. The uncertainties are expressed as relative deviations from the results for the standard settings (see Table 6) derived by MPIC using MCARTIM.
In general, larger uncertainties are found for the $O_4$ dAMFs compared to the $O_4$ AMFs. This is expected because the uncertainties of the $O_4$ dAMFs contain the uncertainties of two simulations (at 90° elevation and at low elevation). Another general finding is that the uncertainties on 18 June are larger than on 8 July. This finding is mainly related to the larger uncertainties due to the aerosol phase function, which has an especially strong forward peak on 18 June. Also the uncertainties from the $O_4$ profile extraction, the choice of the radiative transfer model and the extrapolation of the aerosol extinction below 180 m are larger on 18 June than on 8 July. These higher uncertainties are probably mainly related to the high aerosol extinction close to the surface on 18 June (see section 5.1, and appendices A2 and A5).
For the total uncertainties two values are given in Table 9: The 'average deviation' is the sum of all systematic deviations of the individual uncertainties (the corresponding mean of the maximum and minimum values). The second quantity (the 'range of uncertainties) is calculated from half the individual uncertainty ranges by assuming that they are independent.
Finally, it should be noted that for some uncertainties (e.g. the effects of the surface albedo or the single scattering albedo) the given numbers probably overestimate the true uncertainties, while for others, e.g. the uncertainties related to the aerosol extinction profiles or the phase functions they possibly underestimate the true uncertainties (although reasonable assumptions were made). The two latter uncertainties are especially large for 18 June. The differences between both days are discussed in more detail in section 5.

**4.3 Uncertainties of the spectral analysis**

The uncertainties of the spectral analysis are caused by different effects:
-the specific settings of the spectral analysis like the fit window or the degree of the polynomial. Of particular interest is the effect of choosing different $O_4$ cross sections as well as their temperature dependence.

-the properties (and imperfections) of the MAX-DOAS instruments
-the effect of different analysis software and implementations
-the effect of the wavelength dependence of the AMF across the fit window.
These uncertainties are discussed and quantified in the following sub sections.

**4.3.1 Comparison of $O_4$ (d)AMFs derived from the synthetic spectra with $O_4$ (d)AMFs directly obtained from the radiative transfer simulations**


Synthetic spectra for both selected days were simulated using the radiative transfer model
SCIATRAN (for details see section 2.4 and Table A3 in appendix A1). While spectra for the
whole day are simulated (for the viewing geometry see Table A2 in appendix A1) it should be
noted that the aerosol properties during the middle periods are used also for the whole day (to
minimise the computational efforts). The spectra are analysed using the standard settings and
the derived $O_4$ (d)SCDs are converted to $O_4$ (d)AMFs using eq. 1. In addition to the spectra,
also $O_4$ (d)AMFs at 360 nm are simulated directly by the RT models using exactly the same
settings. These $O_4$ (d)AMFs are used to test whether the spectral retrieval results are indeed
representative for the simulated $O_4$ (d)AMFs at 360 nm.
Spectra are simulated with and without considering the temperature dependence of the $O_4$
cross section. Also one version of synthetic spectra with added random noise is processed.
First, the synthetic spectra are analysed using the standard settings (see Table 7). Examples of
the $O_4$ fits for synthetic (and measured) spectra are shown in Fig. 11. Here it is interesting to
note that the ratios of the results for the measured and the simulated spectra are between 0.68
and 0.74, similar to ratio for the dAMFs on 8 July shown in Table 8.
In Fig. 12 the ratios of the $O_4$ (d)AMFs derived from the synthetic spectra versus those
directly obtained from the radiative transfer simulations at 360 nm are shown. In the upper
part (a) the results for synthetic spectra considering the temperature dependence of the $O_4$
cross section are presented (without noise). Systematically enhanced ratios are found in the
morning and evening, while for most of the day the ratios are close to unity. The higher
values in the morning and evening are probably partly caused by the increased light paths
through higher atmospheric layers (with lower temperatures) when the solar zenith angle is
high. Interestingly, if the temperature dependence of the $O_4$ cross section is not taken into
account (Fig. 12 b), still slightly enhanced ratios during the morning and evening are found,
which can not be explained anymore by the temperature dependence of the $O_4$ cross section.
Thus we speculate that part of the enhanced values at high SZA are probably caused by the
wavelength dependence of the $O_4$ AMFs. Nevertheless, for most of the day the ratio is very
close to unity indicating that for SZA < 75° the $O_4$ (d)AMFs obtained from the spectral
analysis are almost identical to the $O_4$ (dAMFs) directly obtained from the radiative transfer
simulations (at 360 nm).
In Fig. 12 c results for spectra with added random noise (without consideration of the
temperature dependence of the $O_4$ cross section) are shown. On average similar results as for
the spectra without noise (Fig. 12 b) are found but the results now show a large scatter. From
these results and also the spectral analyses (Fig. 11) we conclude that the noise added to the
synthetic spectra overestimates that of the real measurements. For the sensitivity studies
discussed in section 4.3.2 only synthetic spectra without noise were used.
In Table A13 in appendix A4 the average ratios for the middle periods on both selected days
are shown. They deviate from unity by up to 2% indicating that the wavelength dependence of
the $O_4$ (d)AMF is negligible for the considered cases for SZA < 75°.

**4.3.2 Sensitivity studies for different fit parameters**

In this section the effect of the choice of several fit parameters on the derived $O_4$ (d)AMFs is investigated using both measured and synthetic spectra. It should be noted that in the following only synthetic spectra without noise were used, because for the sensitivity studies we are interested in the systematic effects. Only one fit parameter is varied for each individual test, and the results are compared to those for the standard fit parameters (see Table 7).

First the fit window is varied. Besides the standard fit window (352 to 387 nm), which contains two $O_4$ bands, also two fit windows towards shorter wavelengths are tested: 335 – 374 nm (including two $O_4$ bands) and 345 – 374 nm (including one $O_4$ band at 360 nm). The ratios of the derived $O_4$ (d)AMFs versus those for the standard analysis are shown in Fig. A15 and Table A14 in appendix A2. On 18 June rather large deviations of the $O_4$ (d)AMFs are found for both measured (-12%) and synthetic spectra (-5%) for the spectral range 335 to 374 nm. On 8 July the corresponding differences are smaller (-6% and -2% for measured and synthetic spectra, respectively). For the spectral range 345 – 374 nm, smaller differences of only up to 1% are found for both days. The reason for the larger deviations on 18 June for the spectral range 335 – 374 nm is not clear. One possible reason could be the differences of the Ångström parameters (see Fig. 1) and phase functions (see Fig 10).

In Fig. A16 and Table A15 the results for different degrees of the polynomial used in the spectral analysis are shown. For the measured spectra systematically higher $O_4$ (d)AMFs (up to 6%) than for the standard analysis are found when using lower polynomial degrees. For the synthetic spectra the effect is smaller (<3%).

In Fig. A17 and Table A16 the results for different intensity offsets are shown. Again, for the measured spectra systematically higher $O_4$ (d)AMFs (up to 16%) than for the standard analysis are found when reducing the order of the intensity offset, while for the synthetic spectra the effect is smaller (<3%). Higher order intensity offsets might compensate for wavelength dependent offsets (e.g. spectral straylight), which can be important for real measurements, while the synthetic spectra do not contain such contributions. In Fig. A18 and Table A17 the results for spectral analyses with only one Ring spectrum are shown. In contrast to the standard analysis, which includes two Ring spectra (one for clear and one for cloudy sky, see Wagner et al., 2009), only the Ring spectrum for clear sky is used. For both selected days, only small deviations (within 2%) compared to the standard analysis are found.

### 4.3.3 Sensitivity studies using different trace gas absorption cross sections

In this section the impact of different trace gas absorption cross sections on the derived $O_4$ (d)AMFs is investigated.

In Fig. A19 and Table A18 the results for using two $NO_2$ cross sections (294 and 220 K) compared to the standard analysis (using only a $NO_2$ cross section for 294 K) are shown. The results are almost the same as for the standard analysis.

In Fig. A20 and Table A19 the results for using an additional wavelength-dependent $NO_2$ cross section compared to the standard analysis (using only one $NO_2$ cross section) are shown. The second $NO_2$ cross section is calculated by multiplying the original cross section with wavelength (Pukite et al., 2010). Again, only small deviations of the results from the standard analysis (1% for the measured spectra, and 2% for the synthetic spectra are found.

In Fig. A21 and Table A20 results for using and additional wavelength-dependent $O_4$ cross sections compared to the standard analysis (using only one $O_4$ cross section) are shown. The second $O_4$ cross section is calculated like for $NO_2$, but also an orthogonalisation with respect to the original $O_4$ cross section (at 360 nm) is performed. The derived $O_4$ (d)AMFs are almost identical to those from the standard analysis (within 1%).

For the spectral retrieval of HONO in a similar spectral range, a significant impact of water vapour absorption around 363 nm was found in Wang et al. (2017c) and Lampel et al. (2017). In Fig. A22 and Table A21 the $O_4$ results for including a $H_2O$ cross section (Polyansky et al.,

2018) compared to the standard analysis (using no $H_2O$ cross section) are shown. The results are almost identical to those from the standard analysis (within 1%).

In Fig. A23 and Table A22 the results for including a HCHO cross section compared to the standard analysis (using no HCHO cross section) are shown. Especially for 18 June a large systematic effect is found: the $O_4$ dAMFs are by 4 % or 6 % smaller than for the standard analysis for measured and synthetic spectra, respectively. On 8 July the underestimation is smaller (2% and 3% for measured and synthetic spectra, respectively).

**4.3.4 Effect of using different $O_4$ cross sections**

In Fig. A24 and Table A23 the results for different $O_4$ cross sections are compared to the standard analysis (using the Thalman $O_4$ cross section). The results for both days are almost identical. For the real measurements, the derived $O_4$ dAMFs using the Hermans and Greenblatt cross sections are by 3% smaller or 8 % larger than those for the standard analysis, respectively. However, if the Greenblatt $O_4$ cross section is allowed to shift during the spectral analysis, the overestimation can be largely reduced to only +3 %. This confirms findings from earlier studies (e.g. Pinardi et al., 2013) that the wavelength calibration of the original data sets is not very accurate.

For the synthetic spectra slightly different results than for the real measurements are found for the Hermans $O_4$ cross section. The reason for these differences is not clear. However, here it should be noted that the temperature dependent $O_4$ absorption in the synthetic spectra does probably not exactly represent the true atmospheric $O_4$ absorption.

**4.3.5 Effect of the temperature dependence of the $O_4$ cross section**

The new set of $O_4$ cross sections provided by Thalman and Volkamer (2013) allows to investigate the temperature dependence of the atmospheric $O_4$ absorptions in detail. They provide $O_4$ cross sections measured at five temperatures (203, 233, 253, 273, 293 K) covering the range of temperatures relevant for atmospheric applications. Using these cross sections, the effect of the temperature dependence of the $O_4$ absorptions is investigated in two ways:

a) In a first test, synthetic spectra are simulated for different surface temperatures assuming a fixed lapse rate. These spectra are then analysed using the $O_4$ cross section for 293K (which is usually used for the spectral analysis of $O_4$). From this study the magnitude of the effect of the temperature dependence of the $O_4$ cross section on MAX-DOAS measurements can be quantified.

b) In a second test, measured and synthetic spectra for both selected days are analysed with $O_4$ cross sections for different temperatures. From this study it can be seen to which degree the temperature dependence of the $O_4$ cross section can be already corrected during the spectral analysis (if two $O_4$ cross sections are used simultaneously).

For the first study, MAX-DOAS spectra are simulated in a simplified way:

-Atmospheric temperature profiles are constructed for surface temperatures between 220 K and 310 K in steps of 10 K assuming a fixed laps rate of –0.656 K / 100 m.

-For each altitude layer (vertical extension: 20 m below 500m, 100 m between 500 m and 2 km, 200 m between 2 km and 12 km, 1 km above) the $O_4$ concentrations (calculated from the US standard atmosphere) are multiplied with the corresponding differential box-AMFs calculated for typical atmospheric conditions and viewing geometries (see Fig. A25 in appendix A4).

-High resolution absorption spectra are calculated by applying the Beer-Lambert-law for each height layer using the $O_4$ cross section of the respective temperature (interpolated between the two adjacent temperatures of the Thalman and Volkamer data set).

-The derived high resolution spectra are convolved with the instrument slit function (FWHM
of 0.6 nm).
-The logarithm of the ratio of the spectra for the low elevation and zenith is calculated and
analysed using the $O_4$ cross section for 293 K.
-The derived $O_4$ dAMFs are divided by the corresponding dAMFs directly obtained from the
radiative transfer simulations.
These calculated ratios as function of the surface temperature are shown in Fig. 13. A strong
and systematic dependence on the surface temperature is found (15 % for a change of the
surface temperature between 240 and 310 K). However, except for measurements at polar
regions, the deviations are usually small. Since for both selected days the temperatures were
rather high (indicated by the two coloured horizontal bars in the figure), the effect of the
temperature dependence of the $O_4$ absorption for the middle periods of both days is very small
(-1 to -2% for 18 June, and 0 to +1% on 8 July). It should be noted that the results shown in
Fig. 13 are obtained for generalised settings of the radiative transfer simulations. Thus it is
recommended that future studies should investigate the effect of the temperature dependence
in more detail and using the exact viewing geometry for individual observations. However,
since the temperatures on both selected days were rather high, for this study the
simplifications of the radiative transfer simulations have no strong influence on the derived
results.
In the second test the measured and synthetic spectra are analysed using $O_4$ cross sections for
different temperatures. The corresponding results are shown in Fig. A26 and Table A24.
If only the $O_4$ cross section at low temperature (203 K) is used, the derived $O_4$ AMFs and
dAMFs are by about 16% and 30% smaller than for the standard analysis (using the $O_4$ cross
section for 293 K). These results are consistently obtained for the measured and synthetic
spectra. If, however, two $O_4$ cross sections (for 203 and 293 K) are simultaneously included in
the analysis, different results are obtained for the measured and synthetic spectra: for the
measured spectra the derived $O_4$ (d)AMFs agree within 4% with those from the standard
analysis. In contrast, for the synthetic spectra, the derived $O_4$ (d)AMFs are systematically
smaller (by about 6 to 18 %). This finding was not expected, because exactly the same cross
sections were used for both the simulation and the analysis of the synthetic spectra. Detailed
investigations (see appendix A4) led to the conclusion that there is a slight inconsistency in
the temperature dependence of the $O_4$ cross sections from Thalman and Volkamer (2013):
The ratio of the peak values of the cross section at 360 and 380 nm changes in a non-
continuous way between 253 and 233 K (see Fig. A27 in appendix A4), see also Fig. S2
(values for 380nm) in the supplementary material of Thalman and Volkamer (2013). The
reason for this inconsistency is currently not known. If these two $O_4$ bands are included in the
spectral analysis (as for the standard settings), the convergence of the spectral analysis
strongly depends on the ability to fit both $O_4$ bands well. Thus the fit results for both $O_4$ cross
sections are mainly determined by the relative strengths of both $O_4$ bands (see Fig. A27 in
appendix A4). If instead a smaller wavelength range is used containing only one absorption
band (345 – 374 nm), the derived $O_4$ (d)AMFs are in rather good agreement with the results
of the analysis (using only the $O_4$ cross section for 293 K), see Table A25 in appendix A4. In
that case, the convergence of the fit mainly depends on the temperature dependence of the line
width. It should be noted that the non-continuous temperature dependence of the $O_4$
absorption cross section only affects the analysis of the synthetic spectra, because for the
simulation of the spectra all $O_4$ cross sections for temperatures between 233 and 293 K were
used. For the measured spectra, no problems are found, because in the spectral analysis only
the $O_4$ cross sections for 233 and 293 K were used.
In Fig. A28 in appendix A4 the ratios of both fit coefficients (for 203 and 293 K) as well as
the derived effective temperatures for the analyses of measured and synthetic spectra are
shown. For the measured spectra the ratios are close to zero and the derived temperatures are

close to 300K most of the time (except in early morning and evening), because the effective atmospheric temperature for both days is close to the temperature of the high temperature $O_4$ cross section (293 K) (see Fig. 13). Similar results (at least around noon) are also obtained for the synthetic spectra if the narrow spectral range (345 – 374 nm) is used. For the standard fit range (including two $O_4$ bands), however, the ratios are much higher again indicating the effect of the inconsistency of the temperature dependence of the $O_4$ cross sections (see Fig. A27 in appendix A4).

**4.3.6 Results from different instruments and analyses by different groups**

In this section the effects of using measurements from different instruments and having these spectra analysed by different groups are investigated. For that purpose three different procedures are followed: First, MPIC spectra are analysed by other groups; second, the spectra from other instruments are analysed by MPIC; third, the spectra from non-MPIC instruments are analysed by the respective group.

In Fig. 14a and Table A25 (in appendix A4) the comparison results of the analysis of MPIC spectra by other groups versus the analysis of MPIC spectra by MPIC are shown. Especially for 18 June rather large differences (between –6% / +5%) to the MPIC standard analysis are found. Interestingly the largest differences are found in the morning when the aerosol extinction close to the surface was strongest. On 8 July smaller differences (between –6% and –1%) are found.

In Fig. 14b and Table A25 (in appendix A4) the comparison results of the analysis of spectra from other instruments by MPIC versus the analysis of MPIC spectra by MPIC are shown. For this comparison all analyses are performed in the spectral range 335 – 374 nm, because the standard spectral range (352 – 387 nm) is not covered by all instruments. Again, the largest differences are found for 18 June (up to ±11%). For 8 July the differences reach up to ±6%, but for this day only a few measurements in the morning are available.

In Fig. 14c and Table A25 (in appendix A4) the comparison results of the analysis of spectra from other instruments by the respective group versus the MPIC analysis by MPIC (standard analysis) is shown. From this exercise the combined effects of different instrumental properties and retrievals can be estimated. Interestingly, the observed differences are only slightly larger than those for the analysis of the spectra from the different instruments by MPIC (Fig. 14b). This indicates that the largest uncertainties are related to the differences of the different instruments and not to the settings and implementations of the different retrievals. For the middle period of 18 June the uncertainties are within 12%. This range is also assumed for 8 July. Here it is interesting to note that the derived uncertainties of the spectral analysis are probably not representative for most recent measurement campaigns. For example, during the CINDI-2 campaign (http://www.tropomi.eu/data-products/cindi-2) the deviations of the $O_4$ spectral analysis results were much smaller than for the selected days during the MAD-CAT campaign (Kreher et al., 2019).

**4.3.7 Summary of uncertainties of the $O_4$ AMF from the spectral analysis**

Table 10 presents an overview on the different sources of uncertainties of the measured $O_4$ (d)AMFs obtained in the previous sub-sections. The uncertainties are expressed as relative deviations from the results for the standard settings (see Table 7) derived by MPIC from spectra of the MPIC instrument.

Like for the simulation results, in general, larger uncertainties are found for the $O_4$ dAMFs compared to the $O_4$ AMFs. This is expected because the uncertainties of the $O_4$ dAMFs contain the uncertainties of two analyses (at 90° elevation and at low elevation). Also, the uncertainties on 18 June are again larger than on 8 July. This finding was not expected, but is

possibly related to the higher trace gas abundances (see Fig. 1 and Table A3 in appendix A1)
and the higher aerosol extinction close to the surface on 18 June.
Another interesting finding is that the uncertainties of the spectral analysis of $O_4$ are
dominated by the effect of instrumental properties up to ±12% in the morning of 18 June.
Further important uncertainties are associated with the choice of the wavelength range, the
degree of the polynomial and the intensity offset. In contrast, the exact choices of the trace
gas cross sections (including their wavelength- and temperature dependencies) play only a
minor role (up to a few percent). Excellent agreement (within ±1%) is in particular found for
the $O_4$ analysis of the synthetic spectra using the standard settings and the directly simulated
$O_4$ (d)AMFs at 360 nm. This indicates that the $O_4$ (d)AMFs retrieved in the wavelength range
352 – 387 nm are indeed representative for radiative transfer simulations at 360 nm.
As for the uncertainties of the simulated $O_4$ (d)AMFs, the uncertainties of the spectral
analysis are also split into a systematic and a random term: the systematic deviations of the $O_4$
dAMFs from those of the standard settings are about +1% and –1.5% for 18 June and 8 July,
respectively. The range of uncertainty is calculated from the uncertainty ranges of the
different contributions by assuming that they are all independent. The random uncertainty
ranges for 18 June and 8 July are calculated as ±12.5% and ±10.8%, respectively.
**4.4 Recommendations derived from the sensitivity studies**
In this section a short summary of the most important findings from the sensitivity studies is
given.
**Temperature and pressure profiles**
Temperature and pressure profiles from sondes or model data should be used if available.
Alternatively, temperature and pressure profiles extrapolated from surface measurements
could be used. Typical uncertainties of the $O_4$ VCD derived from such profiles are still < 2%.
For high temperatures (>20°C) the atmospheric humidity should be considered. If no
measurements are available, prescribed profiles, e.g. from the US standard atmosphere or
climatologies of temperature and pressure profiles can be used. However, depending on
location and season the uncertainties of the resulting $O_4$ VCD can be rather large (see also
Ortega et al., 2016).
**Integration of the $O_4$ VCD**
The integration should be performed on a vertical grid with at least 100 m resolution up to an
altitude of 30 km. The surface altitude should be taken into account with an accuracy of at
least 20 m.
**Measurements and spectral analysis**
Instruments should have a small FOV (≤1°), an accurate elevation calibration (better than
0.5°), and a small and preferably well characterised stray light level. For the data analysis the
standard settings as provided in Table 7 should be used. From the analysis of synthetic spectra
it was found that the results for these settings are consistent with simulated $O_4$ (d)AMFs
within 1 %.
**Information on aerosols**
Aerosol profiles should be obtained from LIDARs or ceilometers using similar wavelengths
as the MAX-DOAS measurements if available (see e.g. Ortega et al., 2016). Preferred LIDAR
types are HSRL or Raman LIDARs, which directly provide profiles of aerosol extinction and
thus need no assumptions on the LIDAR ratio. They should also have high signal to noise
ratios and shallow blind region at the surface in order to cover a large altitude range.

Information on aerosol optical properties and size distributions from sun photometers or in situ measurements should be used.

**RTM simulations**

Radiative transfer models should use Mie phase functions and aerosol single scattering albedo e.g. derived from sun photometer observations. The consideration of polarisation and rotational Raman scattering is not necessary.

In summary, if the optimised settings described above are used, the uncertainties of the radiative transfer simulations and spectral analysis can be largely reduced: the uncertainties of the $O_4$ dAMFs related to radiative transfer simulations can be reduced from about $\pm 8$ % as in this study to about $\pm 4$ %; those related to the spectral analysis can be reduced from about $\pm 10$ % to about $\pm 6$ %.

**4.4.1 Preferred scenarios for future studies**

In addition to the recommendations given above, future campaigns should aim to cover different meteorological conditions (e.g. low temperatures), viewing geometries (e.g. low SZA), surface albedos (e.g. snow and ice) and wavelengths (e.g. 477, 577, and 630 nm). Also different aerosol scenarios including those with low aerosol optical depths should be covered. MAX-DOAS measurements should be performed by at least 2, preferably more instruments. In order to minimise the effects of instrumental properties, the instruments should be well calibrated and should have low straylight levels. Measurements during the CINDI-2 campaign are probably well suited for a similar study.

**5 Comparison of measurements and simulations**

The comparison results for both days are different: On 18 June (except in the evening) measurements and simulations agree within uncertainties (the ratio of simulated and measured $O_4$ dAMFs for the middle period of that day is $1.01 \pm 0.16$). In contrast, on 8 July measurements and simulations significantly disagree: Taking into account the uncertainties of the VCD calculation (3%), the radiative transfer simulations ($+16 \pm 6.4$%) and the spectral analysis ($-1.5 \pm 10.8$%) for the middle period of that day results in a ratio of simulated and measured $O_4$ dAMFs of $0.82 \pm 0.10$, which differs significantly from unity.

**5.1 Important differences between both days**

On both selected days similar aerosol AOD were measured. Also the diurnal variation of the SZA was similar because of the proximity to summer solstice. However, also many differences are found for the two days, which are discussed below.

a) temperature, pressure, wind:
On 18 June surface pressure was lower by about 13 hPa and surface temperature was higher by about 7K than on 8 July, respectively. These differences were explicitly taken into account in the calculation of the $O_4$ profiles / VCDs, the radiative transfer simulations and the interpretation of the spectral analyses. Thus they can very probably not explain the different comparison results on the two days.

On both days, wind was mainly blowing from East-North-East, but on 18 June it was blowing from West before about 08:00 and after 20:00 UTC. Wind speeds were lower on 18 June (between 1 and 2 m/s) than on 8 July (between 1 and 3 m/s).

b) aerosol properties:
The in situ aerosol measurements show very different abundances and properties of aerosols close to the ground for the selected days. On 18 June much higher concentrations of larger aerosol particles are found, which cannot be measured by the ceilometer due to the blindness for the lowest 180m. Thus it can be concluded that the enhanced aerosol concentration on 18 June is confined to a shallow layer at the surface. In general the aerosol concentrations close to the surface are more variable on 18 June than on 8 July. The high aerosol concentrations close to the surface probably also affect the LIDAR ratio, which is thus probably more variable on 18 June. Similarly, also the phase function derived from the sun photometer (for the integrated aerosol profile) is probably less representative for the low elevation angles on 18 June because different aerosol size distributions probably existed at different altitudes. Finally, the Ãngström parameter derived from AERONET observations is different for both days, especially for large wavelengths, which is in qualitative agreement with the higher in situ aerosol concentrations of large particles on 18 June. Also a larger forward peak of the derived aerosol phase function is found for 18 June. Both effects probably cause larger uncertainties on 18 June.

c) spectral analysis
Larger uncertainties of the spectral analysis are found for 18 June compared to 8 July. This finding was surprising, but was also partly reproduced by the analysis of the synthetic spectra. One possible explanation is the smaller wavelength dependence of aerosol scattering at low altitudes on 18 June, which mainly affects measurements at low elevation angles. When analysed versus a zenith reference, for which the broad band wavelength dependency is much stronger (because of the larger contribution from Rayleigh scattering), larger deviations can be expected (e.g. because of differences of instrumental straylight, or the different detector saturation levels). On 18 June also higher (about doubled) $NO_2$ and HCHO concentrations are present compared to 8 July possibly leading to increased spectral interferences with the $O_4$ absorption, but this effect is expected to be small.

**5.2 Which conditions would be needed to bring measurements and simulations on 8 July into agreement**

This section tentatively describes possible (although generally unrealistic) changes of the atmospheric scenario, the instrument properties or the input parameters, which could bring measurements and simulations on 08 July into agreement. If e.g. the whole aerosol extinction profile was scaled by 0.65, the corresponding $O_4$ dAMFs would almost perfectly match the measured ones.
Similarly good agreement could also be achieved if about 27% of the total AOD would be shifted from low layers (below 1.68 km) to high layers (above 4.9 km, see appendix A6). However, in this scenario, about 73% of the total aerosol extinction would be above 1.68 km. Such a scenario would not be in agreement with the AERONET inversion products and would also lead to an underestimation of the diurnal variation of the $O_4$ AMFs measured in zenith direction.
Also horizontal gradients of the aerosol extinction could in principle explain the discrepancy. While we are not able to quantify them, they surely would have to be of the order of several ten percent per 10 km. Such persistent horizontal gradients are not supported by the almost

constant AOD during the day (and also by the consistent aerosol in situ observations at the different sites). Also the finding that mismatch between measurements and simulations is found for all azimuth angles indicates that horizontal gradients can not explain the observed discrepancies.

Another possibility would be aerosol phase functions with very high asymmetry parameters ($>> 0.75$). Also systematic errors of the $O_4$ cross section could explain the observed discrepancies. Finally, an overcorrection of spectrograph straylight (or any other intensity offset) could explain the discrepancies. However, a rather high overcorrection (by about 20%) would be needed, which is probably unrealistic.

## 6 Conclusions

We compared MAX-DOAS observations of the atmospheric $O_4$ absorption with corresponding radiative transfer simulations for two mainly cloud-free days during the MAD-CAT campaign. A large part of this study is dedicated to the extraction of input information for the radiative transfer simulations and the quantification of the associated uncertainties of the radiative transfer simulations and spectral retrievals. An important result from the sensitivity studies is that the $O_4$ results derived from the analysis of synthetic spectra using the standard settings are consistent with the simulated $O_4$ air mass factors within 1%. Also recommendations for the settings of the radiative transfer simulations, in particular on the extraction of aerosol and $O_4$ profiles are given. Another important result is that the extent and quality of the aerosol data sets is crucial to constrain the radiative transfer simulations. For example, it is recommended that LIDAR instruments are operated at wavelengths close to those of the MAX-DOAS measurements (see Ortega et al., 2016) and have a small sensitivity gap close to the surface. Further aerosol properties (e.g. size distributions, phase functions) should be available from sun photometer and/or in situ measurements. If such aerosol data are available the corresponding uncertainties of the radiative transfer simulations could be largely reduced to about ±5%. Similar uncertainties can also be expected for optimum instrument operations and data analyses.

The comparison results for both days are different: On 18 June (except in the evening) measurements and simulations agree within uncertainties (the a ratio of simulated and measured $O_4$ dAMFs for the middle period of that day is 1.01±0.16). In contrast, on 8 July measurements and simulations significantly disagree: Taking into account the uncertainties of the VCD calculation (3%), the radiative transfer simulations (+16±6.4%) and the spectral analysis (-1.5±10.8%) for the middle period of that day results in a ratio of simulated and measured $O_4$ dAMFs of 0.81 ±0.10, which differs significantly from unity. So far no plausible explanation for the observed discrepancies on 8 July was found.

However, as long as the reason for this deviation is not understood, it is unclear how representative these findings are for other measurements (e.g. from other platforms, at other locations/seasons, for other aerosol loads, and other wavelengths). Thus further studies spanning a larger variety of measurement conditions and also including other wavelengths are recommended. The MAX-DOAS measurements collected during the recent CINDI-2 campaign are probably well suited for that purpose.


**Acknowledgments**



We are thankful for several external data sets which were used in this study: Temperature and
pressure profiles from the ERAInterim reanalysis data set were provided by the European
Centre for Medium-Range Weather Forecasts. In situ measurements of trace gas and aerosol
concentrations as well as meteorological data were performed by the environmental
monitoring services of the States of Rhineland-Palatinate and Hesse (http://www.luft-rlp.de
and https://www.hlnug.de/themen/luft/luftmessnetz.html). We thank M. O. Andreae and
Günther Schebeske for operating the Ceilometer and the AERONET instrument at the Max
Planck Institute for Chemistry. We also would like to thank Rainer Volkamer, Theodore K.
Koenig and Ivan Ortega for very helpful comments.

**Tables**


Table 1 Overview on studies which did not apply a scaling factor (upper part) or did apply a
scaling factor (lower part) to the measured $O_4$ dSCDs. Besides the initial studies proposing a
scaling factor (Wagner et al., 2009; Clémer et al., 2010) only studies after 2010 are listed.

| Reference | Measurement type | Location and period | $O_4$ band (nm) | Scaling factor |
|---|---|---|---|---|
| **Studies which did not apply a scaling factor\*** | | | | |
| Thalmann and Volkamer, 2010 | CE-DOAS | Laboratory | 477 | 1 |
| Frieß et al., 2011 | MAX-DOAS | Barrow, Alaska (Feb-Apr 2009) | 360 | 1 |
| Peters et al., 2012a | MAX-DOAS | Western Pacific Ocean (Oct 2009) | 360, 477 | 1 |
| Spinei et al. 2015 | Direct sun DOAS | JPL, USA (Jul 2007) Pullman, USA (Sep – Nov 2007, Jul – Nov 2011) Fairbanks, USA (Mar-Apr 2011) Huntsville, USA (Aug 2008) Richland, USA (Apr-Jun 2008) Greenbelt, USA (May 2007, 2012-2014) Cabauw, The Netherlands (Jun-Jul 2009) | 360, 477 | 1 |
| Spinei et al., 2015 / | Airborne DOAS | Subtropical Pacific Ocean (Jan 2012) | 360, 477 | 1 |
| Volkamer et al., 2015 | Airborne DOAS | Subtropical Pacific Ocean (Jan 2012) | 360, 477 | 1 |
| Ortega et al., 2016 | MAX-DOAS | Cape Cod, USA (Jul 2012) | 360, 477 | 1 |
| Schreier et al., 2016 | MAX-DOAS | Zugspitze, Germany (Apr-Jul 2003) Pico Espeio, Venezuela (2004 - | 360 | 1 |

| | | | | |
|---|---|---|---|---|
| | | 2009) | | |
| Seyler et al., 2017 | MAX-DOAS | German Bight (2013-2016) | 360, 477 | 1 |
| Wang et al., 2017a,b | MAX-DOAS | Wuxi, China (2011 - 2014) | 360 | 1 |
| Gielen et al., 2017 | MAX-DOAS | Bujumbura, Burundi (2013-2015) | 360, 477 | 1 |
| Franco et al., 2015 | MAX-DOAS | Jungfraujoch (2010 –2012) | 360 | 1 |
| | | | | |
| **Studies which did apply a scaling factor** | | | | |
| Wagner et al., 2009 | MAX-DOAS | Milano, Italy Sep 2013 (FORMAT II) | 360 | 0.81 |
| Clemer et al., 2010 | MAX-DOAS | Beijing, China Jul 2008 – Apr 2009 | 360, 477, 577, 630 | 0.80 |
| Irie et al., 2011 | MAX-DOAS | Cabauw, The Netherlands Jul-Jun 2009 (CINDI-I) | 360, 477 | 0.75±0.1 |
| Merlaud et al., 2011 | Airborne DOAS | Arctic Apr 2008 POLARCAT) | 360 | 0.89 |
| Vlemmix et al., 2011 | MAX-DOAS | Cabauw, The Netherlands Jul-Oct 2009 (CINDI-I) | 477 | 0.8 |
| Zieger et al., 2011 | Overview on MAX-DOAS | Cabauw, The Netherlands Jul-Oct 2009 (CINDI-I) | 360 (MPIC) 477 (BIRA) 477 (IUPHD) 477 (JAMSTEC) | 0.83 0.75 0.8 0.8* |
| Wang et al., 2014 | MAX-DOAS | Xianghe, China (2010 - 2013) | 360 | 0.8 |
| Kanaya et al., 2014 | MAX-DOAS | Cape Hedo, Japan (2007 – 2012) Fukue, Japan (2008 – 2012) Yokosuda, Japan (2007 – 2012) Gwangju, Korea (2008 – 2012) Hefei, China (2008 – 2012) Zvenigorod; Russia (2009 – 2012) | 477 477 477 477 477 477 | 0.8 0.8 0.8 0.8 0.8 0.8 |
| Hendrick et al., 2014 | MAX-DOAS | Beijing, China (2008 - 2009) Xianghe, China (2010 – 2012) | 360 | 0.8 |
| Vlemmix et al., 2015 | MAX-DOAS | Beijing, China (2008 - 2009) Xianghe, China (2010 – 2012) | 360, 477 | 0.8 |
| Irie et al., 2015 | MAX-DOAS | Tsukuba, Japan (Oct 2010) | 477 | elevation dependent scaling factor** |
| Wang et al., 2016 | MAX-DOAS | Madrid, Spain (Mar – Sep 2015) | 360 | 0.83 |
| Friess et al., 2016 | MAX-DOAS | Cabauw, The Netherlands Jul-Jul 2009 (CINDI-I) | 477 (AOIFM) 477 (BIRA) 477 (IUPHD) 477 (JAMSTEC) 360 (MPIC) | 0.8 0.8 1 0.8*** 0.77 |

*The authors of part of these studies were probably not aware that a scaling factor was applied by other groups.
**SF = 1 / (1 + EA/60)
***SF is varied during profile inversion

Table 2 Periods on both selected days, which are used for the comparisons.

| day | 1st period | 2nd period | 3rd period |
|---|---|---|---|
| 18 June 2013 | 8:00 – 11:00 UTC | 11:00 – 14:00 UTC | 14:00 – 19:00 UTC |
| 8 July 2013 | 4:00 – 7:00 UTC | 7:00 – 11:00 UTC | 11:00 – 19:00 UTC |

Table 3 Participation of the different groups in the different analysis steps

| Abreviation | Institution | Determination of the $O_4$ profile and VCD | Extraction of aerosol profiles | Radiative transfer simulations | Spectral analysis |
|---|---|---|---|---|---|
| BIRA | BIRA/IASB, Brussels, Belgium | | | | ● |
| CMA | Meteorological Observation Center, Beijing, China | | | ● | ● |
| CSIC | Department of Atmospheric Chemistry and Climate, Institute of Physical Chemistry Rocasolano (CSIC), Spain. | ● | | | ● |
| INTA | Instituto Nacional de Tecnica Aeroespacial, Spain | ● | ● | ● | ● |
| IUP-B | University of Bremen, Germany | | ● | ● | ● |
| IUP-HD | University of Heidelberg, Germany | | | | ● |
| LMU | Ludwig-Maximilians-Universität München, Germany | ● | ● | | |
| MPIC | MPI for chemistry, Mainz, Germany | ● | ● | ● | ● |

Table 4 Overview on properties of MAX-DOAS instruments participating in this study

| Institute / Instrument type | Spectral range (nm) | Spectral resolution (FWHM, nm) | Spectral range per detector pixel (nm) | Detector type / temperature | Integration time of individual spectra (s) | Reference |
|---|---|---|---|---|---|---|
| BIRA / 2-D scanning MAX-DOAS | 300 - 386 | 0.49 | 0.04 | 2-D back-illuminated CCD, 2048 x 512 pixels / -40 °C | 60 | Clémer et al., 2010 |
| IUP-Bremen / 2-D scanning MAX-DOAS | 308 - 376 | 0.43 | 0.05 | 2-D back-illuminated CCD, 1340 x 400 pixels / -35 °C | 20 | Peters et al., 2012b |
| IUP-Heidelberg / 1-D scanning MAX-DOAS | 294 - 459 | 0.59 | 0.09 | AvaSpec-ULS 2048 pixels back-thinned Hamamatsu CCD S11071-1106 / 20°C | 60 | Lampel et al., 2015 |
| MPIC / 4-azimuth MAX-DOAS | 320 – 457 | 0.67 | 0.14 | 2-D back-illuminated CCD, 1024 x 255 Pixels / -30°C | 10 s | Krautwurst, 2010 |


Table 5 Independent data sets used to constrain the atmospheric properties during both
selected days.

| Measurement / data set | Measured quantities | Derived quantities | Temporal / spatial resolution | Source / reference |
|---|---|---|---|---|
| Ceilometer | Attenuated backscatter profiles* at 1064 nm | Aerosol extinction pofiles at 360 nm | 30s** / 15 m | Wiegner and Geiß, 2012 |
| AERONET sun photometer | Solar irradiances, Sky radiances | Aerosol optical depth, single scattering albedo, phase function | Typical integration time: 2 to 15 min | Holben et al., 2001, https://aeronet.gsfc.nasa.gov/ |
| Surface measurements air quality stations in Mainz Mombach | temperature, pressure, rel. humidity | | 1h | http://www.luft-rlp.de |
| Surface measurements air quality stations in Mainz and Wiesbaden | $pm_{2.5}$ $pm_{10}$ | | 1h (Mainz stations)  30 min (Wiesbaden stations)*** | http://www.luft-rlp.de  https://www.hlnug.de/themen/luft/luftmessnetz.html |
| ECMWF ERA-Interim reanalysis | temperature, Pressure, rel. humidity | | Average over the area 49.41°-50.53° N, 7.88°-9.00° E, every 6 h | (Dee et al., 2011) |

*no useful signal below 180m due to limited overlap
**Here 15 min averages are used.
***Stations in Mainz: Parcusstrasse, Zitadelle, Mombach; Stations in Wiesbaden: Schierstein,
Ringkirche, Süd
Table 6 Standard settings for the radiative transfer simulations

| Parameter | Standard setting |
|---|---|
| Temperature and pressure profile | MPIC extraction |
| $O_4$ profile | MPIC extraction |
| Surface albedo | 5 % |
| Aerosol single scattering albedo | 0.95 |
| Aerosol phase function | HG model with asymmetry parameter of 0.68 |
| Aerosol extinction profile | MPIC extraction with linear interpolation < 180 m |
| Polarisation | Not considered |
| Raman scattering | Partly considered for synthetic spectra |


Table 7 Standard settings for the DOAS analysis of $O_4$.

| Parameter | Value, Remark / Reference |
|---|---|
| Spectral range | 352 – 387 nm |
| Degree of DOAS polynomial | 5 |
| Degree of intensity offset polynomial | 2 |
| Fraunhofer reference spectrum | 08 July, 10:05:35, SZA: 32.37°, elevation angle: 90° (this spectrum is used for both days) |
| Wavelength calibration | Fit to high resolution solar spectrum using Gaussian slit function |
| Shift / squeeze | The measured spectrum is shifted and squeezed against all other spectra |
| Ring spectrum 1 | Normal Ring spectrum calculated from DOASIS |
| Ring spectrum 2 | Ring spectrum 1 multiplied by $\lambda^{-4}$ |
| $O_3$ cross section | 223 K, Bogumil et al. (2003) |
| $NO_2$ cross section | 294 K, Vandaele et al. (1997) |
| BrO cross section | 223 K, Fleischmann et al. (2004) |
| $O_4$ cross section | 293 K, Thalman and Volkamer (2013) |

Table 8 Average ratios (simulation results divided by measurements) of the $O_4$ (d)AMFs for
both middle periods of the selected days.

| Period | 18.06.2013, 11:00 – 14:00 | 08.07.2013, 7:00 – 11:00 |
|---|---|---|
| AMF ratio | 0.97 | 0.83 |
| DAMF ratio | 0.94 | 0.69 |


Table 9 Summary of uncertainties of the simulated $O_4$ (d)AMFs for the middle periods of
both selected days. The two numbers left and right of the '/' indicate the minimum and
maximum deviations. The columns with label 'Optimum' indicate the uncertainties which
could be reached if optimum information on the measurement conditions was available (e.g.
height profiles of temperature, pressure and aerosol extinction as well as well aerosol
microphysical or optical properties).

| | $O_4$ AMF | | | | $O_4$ dAMF | | |
|---|---|---|---|---|---|---|---|
| | 18 June | 8 July | Optimum settings | | 18 June | 8 July | Optimum settings |
| **Effects of RTM** | | | | | | | |
| Radiative transfer model | -1% / +2% | 0% / +1% | ±1% | | -1% / +5% | 0% / +3% | ±1% |
| Polarisation | 0% / 0% | 0% / 0% | 0% | | 0% / 0% | 0% / +1% | 0% |
| **Effects of input parameters** | | | | | | | |
| $O_4$ profile extraction | 0% / + 2% | 0% / + 1% | ±1% | | 0% / + 4% | 0% / + 2% | ±1% |
| Single scattering albedo | -1% / + 3% | -1% / + 1% | 0% | | -1% / + 3% | -1% / + 1% | 0% |
| Phase function | -3% / +3% | -2% / 0% | ±1% | | -5% / + 9% | -5% / +2% | ±1.5% |
| Aerosol profile extraction | -1% / + 1%* | -2% / + 2% | ±1% | | -2% / + 1%* | -4% / + 4% | ±1.5% |
| Extrapolation below 180 m | 0% / + 2% | -1% / + 1% | 0% | | -1% / + 4% | -2% / + 2% | 0% |
| LIDAR ratio & wrong wavelength | ? | +5% / +6% | ±2%** | | ? | +13% / +18% | ±3%** |
| Surface albedo | 0% / + 2% | 0% / + 1% | 0% | | 0% / + 2% | -1% / + 0% | 0% |
| **Total uncertainty** | | | | | | | |
| Average deviation (from results for standard settings) | +4.5% | +6% | | | +8.5% | +16.5% | |
| Range of uncertainty | ±4.4%* | ±2.8% | ±2.8%** | | ±8.7%* | ±6.4% | ±3.8%** |

*this uncertainty does not contain the contribution from variation of aerosol properties with
altitude, see text
**if LIDAR profiles at the same wavelength and without gaps in the troposphere were
available.

Table 10 Summary of uncertainties of the measured $O_4$ (d)AMFs for the middle periods of
both selected days. The two numbers left and right of the '/' indicate the minimum and
maximum deviations. The columns with label 'Optimum' indicate the uncertainties which
could be reached if optimum instrumental performance was ensured and optimum cross
section were availble.

| | $O_4$ AMF | | | | $O_4$ dAMF | | |
|---|---|---|---|---|---|---|---|
| | 18 June | 8 July | Optimum | | 18 June | 8 July | Optimum |
| **Consistency spectral analysis versus RTM** | | | | | | | |
| Analysis of synthetic spectra | -1% / +1% | -1% / 0% | ±1% | | 0% / 0% | 0% / +1% | ±1% |
| **Fit settings** | | | | | | | |
| Spectral range | -7% / -3% | -3% / 0% | ±1% | | -12% / -1% | -6% / -1% | ±1% |
| Degree of polynomial | +0% / +4% | 0% / + 3% | ±1% | | 0% / +6% | 0% / +6% | ±1% |
| Intensity offset* | +1% / +5% | +1% / +3% | ±1% | | +3% / +11% | +2% / +4% | ±1.5% |
| Ring | +1% / +2% | -1% / +1% | ±1% | | +1% / +1% | -1% / +1% | ±1.5% |
| Temperature dependence of $NO_2$ absorption | 0% / 0% | 0% / 0% | 0% | | 0% / 0% | 0% / 0% | 0% / 0% |
| Wavelength dependence of $NO_2$ absorption | -1% / 0% | 0% / 0% | 0% | | -2% / -1% | -1% / 0% | 0% |
| Wavelength dependence of $O_4$ absorption | -1% / 0% | -1% / -1% | 0% | | 0% / +1% | -1% / -1% | 0% |
| Including $H_2O$ cross section | 0% / 0% | 0% / 0% | 0% | | +1% / +1% | +1% / +1% | 0% |
| Including HCHO cross section | -3% / 0% | -1% / 0% | 0% | | -6% / -4% | -3% / -2% | 0% |
| Different $O_4$ cross sections♣ | -2% / +1% | -2% / +1% | ±2% | | -3% / +3% | -3% / +3% | ±2% |
| **Temperature dependence of the $O_4$ absorption** | | | | | | | |
| Analysis using two $O_4$ cross sections for different temperatures♥ | 0% / 0% | +2% / +2% | ±1% | | +4% / +4% | +1% / +1% | ±1.5% |
| Analysis of synthetic spectra for different surface temperatures | -1% / 0% | -1% / +2% | | | +4% / +4% | +1% / +1% | |

| Analysis from different instruments and groups | | | | | | | |
|---|---|---|---|---|---|---|---|
| Different groups and analyses♦ | -6% / + 5% | -6% / + 5% | ±3%♠ | | -12% / +7% | -12% / +7% | ±4.5% |
| **Total uncertainty** | | | | | | | |
| Average deviation (from results for standard settings) | -4.5% | -0.5% | | | +1% | -1.5% | |
| Range of uncertainty | ±7.0% | ±6.5% | ±4.2% | | ±12.5% | ±10.8% | ±5.7% |

*here the case 'no offset' is not considered
♣here the case of the non-shifted Greenblatt $O_4$ cross section is not considered
♥here only the results for the measured spectra in the spectral range 352 – 387 nm are
considered. (temperatures on 18 June: 27–31 °C; 8 July: 20–30 °C)
♦The results for 18 June are also taken for 8 July due to the lack of measurements on 8 July
♠see Kreher et al., 2019

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

**Figures**


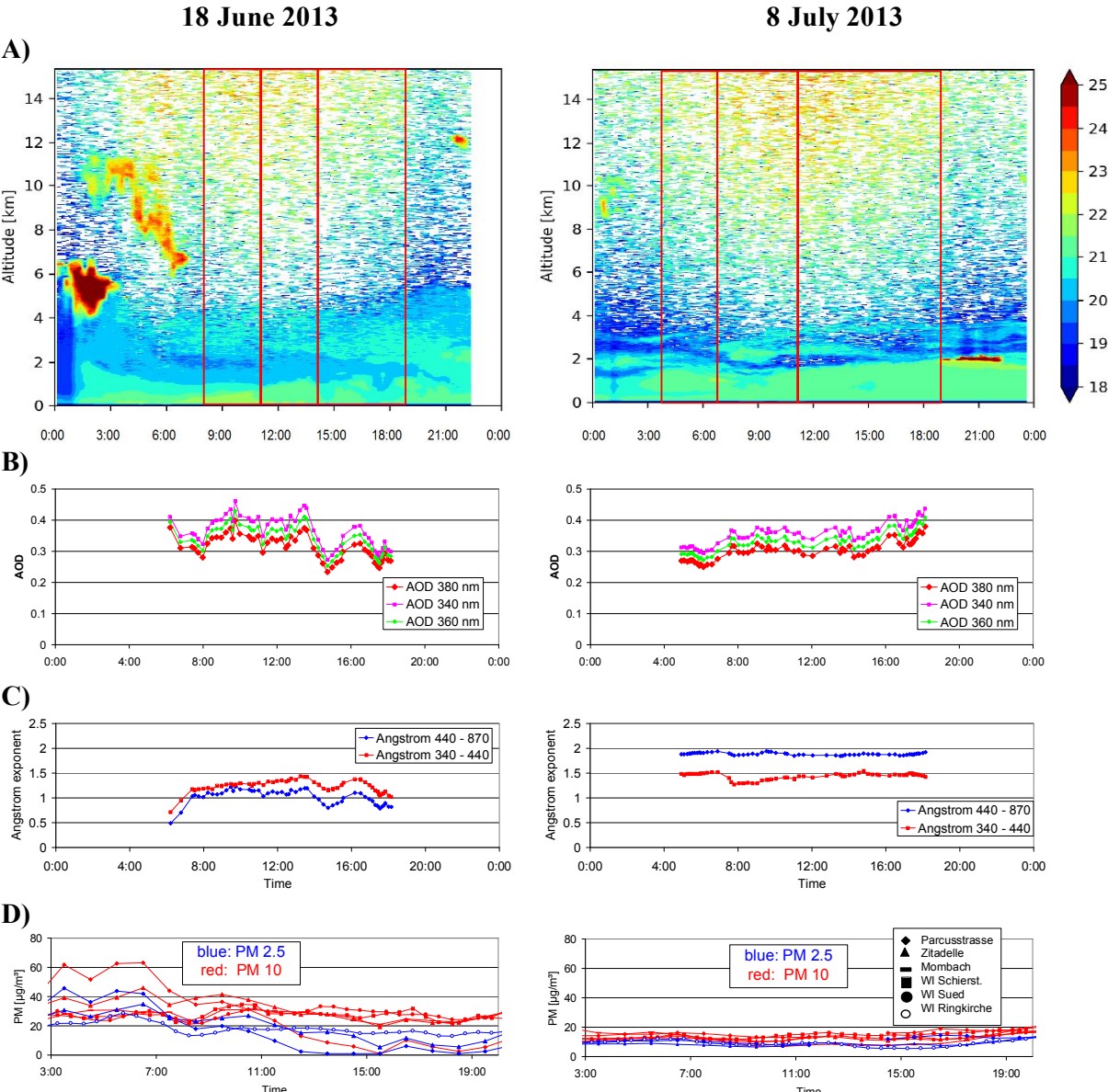

Fig. 1 Various aerosol properties on the two selected days (left: 18 June 2013; right: 8 July 2013). A) Aerosol backscatter profiles from ceilometer measurements; B) AOD at 340, 360, and 380 nm (360 values are interpolated from 340 and 380 nm) from AERONET sun photometer measurements; C) Ångström parameters for two wavelength pairs (340 – 440 nm and 440 – 870 nm) from AERONET sun photometer measurements; D) Surface in situ measurements of $PM_{2.5}$ and $PM_{10}$ measured at different air quality monitoring stations in Mainz and the nearby city of Wiesbaden .


**18 June 2013**    **8 July 2013**

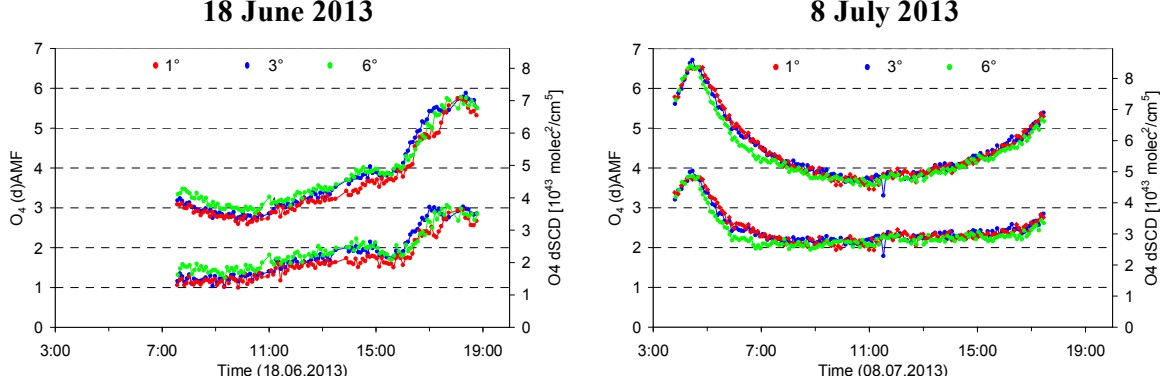

Fig. 2 $O_4$ AMFs (upper lines) and dAMFs (lower lines) for 1°, 3°, and 6° elevation angles
derived from the MPIC MAX-DOAS measurements on the two selected days. Interestingly,
on 18 June the lowest values are in general found for the lowest elevation angles, which is an
indication for the high aerosol load close to the surface. The y-axis on the right side shows the
corresponding $O_4$ (d)SCDs for $O_4$ VCDs of $1.23 \cdot 10^{43}$ molec²/cm⁵ and of $1.28 \cdot 10^{43}$
molec²/cm⁵ for 18 June and 08 July, respectively (see section 4.1.2).

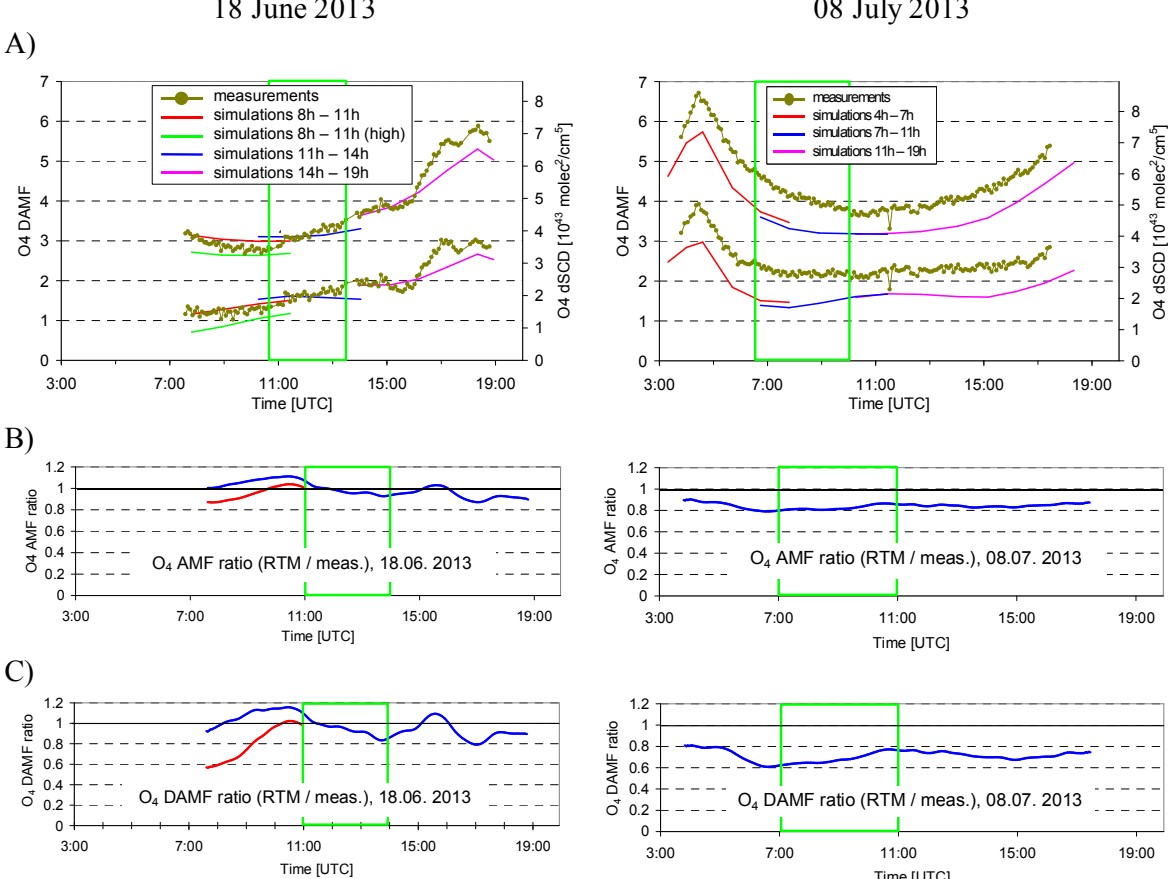

Fig. 3 A) Comparison of O$_4$ (d)AMFs from MAX-DOAS measurements and forward model simulations for the two selected days. The green rectangle indicates the middle periods on both days, which are the focus of the quantitative comparison. The green line on 18 June represents forward model results for a modified aerosol profile (see text). The y-axis on the right side shows the corresponding O$_4$ (d)SCDs for O$_4$ VCDs of $1.23 \cdot 10^{43}$ molec²/cm$^5$ and of $1.28 \cdot 10^{43}$ molec²/cm$^5$ for 18 June and 08 July, respectively (see section 4.1.2). In B) and C) the ratios of the simulated and measured AMFs and dAMFs are shown, respectively. The red line on 18 June represents the ratios for the modified aerosol scenario.

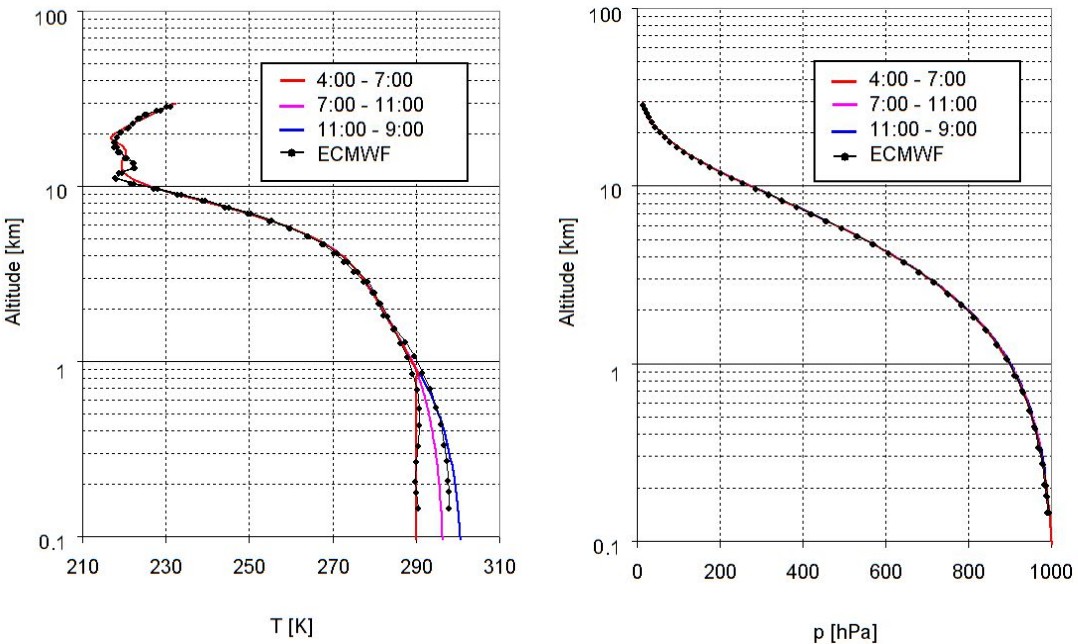

Fig. 4 Extracted temperature (left) and pressure (right) profiles for the three periods on 8 July
2013. Also shown are ECMWF profiles above Mainz for 6:00 and 18:00. To better account
for the diurnal variation of the temperatures near the surface, below 1 km the temperature is
linearly interpolated between the surface measurements and the ECMWF temperatures at 1
km (for details see text). Note that the altitude is given relative to the height of the
measurement site (150 m).

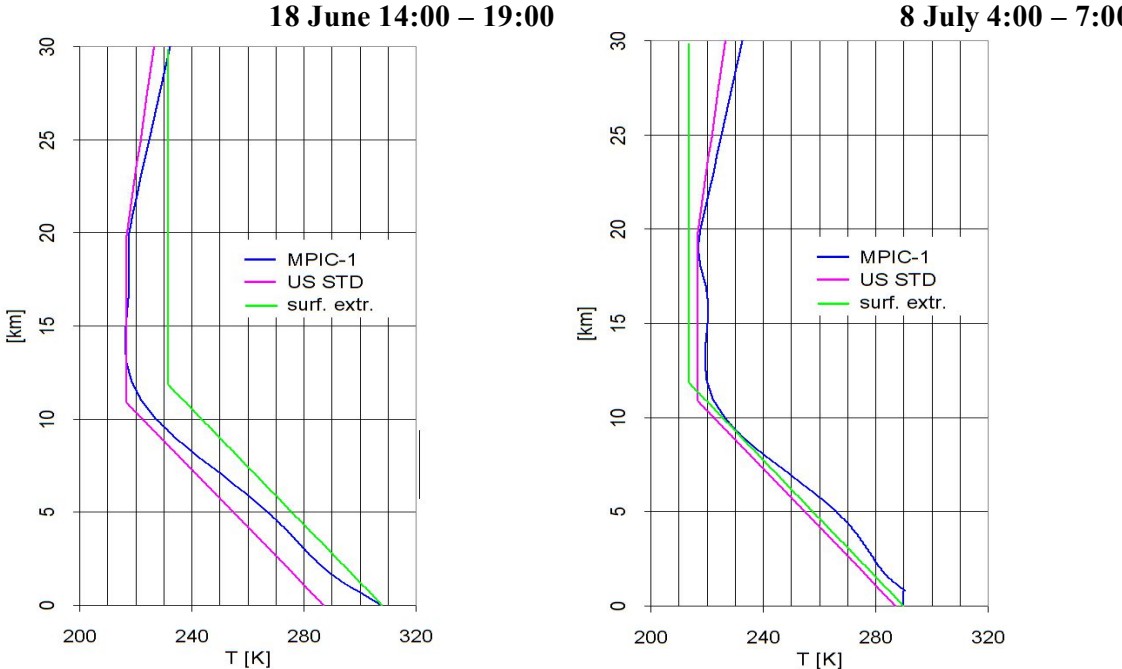

Fig. 5 Temperature profiles extracted in different ways for two periods (Left: 18 June 14:00 –
19:00; right: 8 July 4:00 – 7:00). The blue profiles are extracted from in situ measurements
and ECMWF profiles as described in the text. The green profiles are extracted from the
surface temperatures and assuming a constant lapse rate of –6.5K / km up to 12 km and a
constant temperature above. The pink curves represent the temperature profile from the US
standard atmosphere.

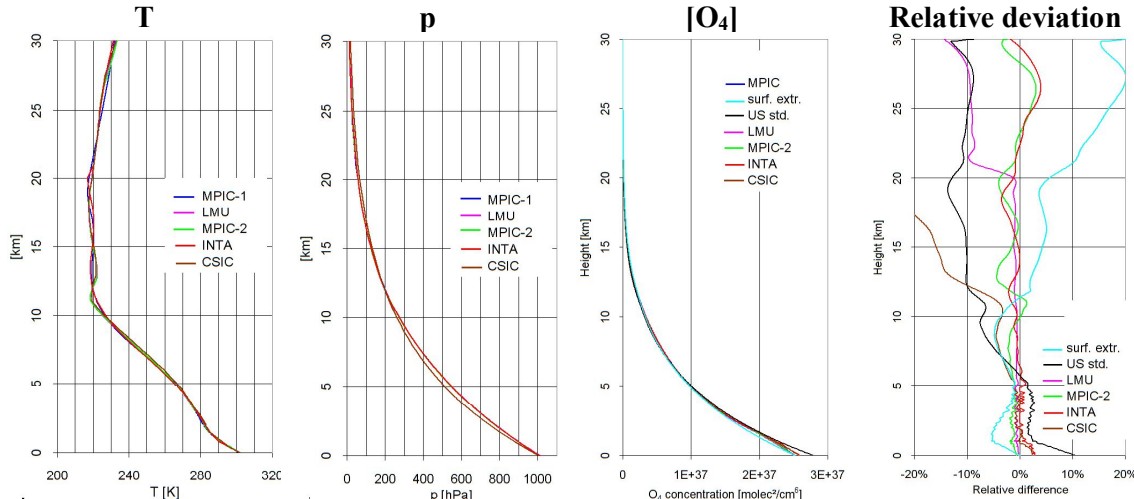

Fig. 6 Comparison of the vertical profiles of temperature, pressure and $O_4$ concentration
(expressed as the square of the $O_2$ concentration) for 8 July, 11:00 – 19 :00, extracted by the
different groups. In the right figure the relative deviations of the $O_4$ concentration compared
to the MPIC standard extraction are shown. There, also the profiles derived from the
extrapolation from the surface values and the US standard atmosphere are included.

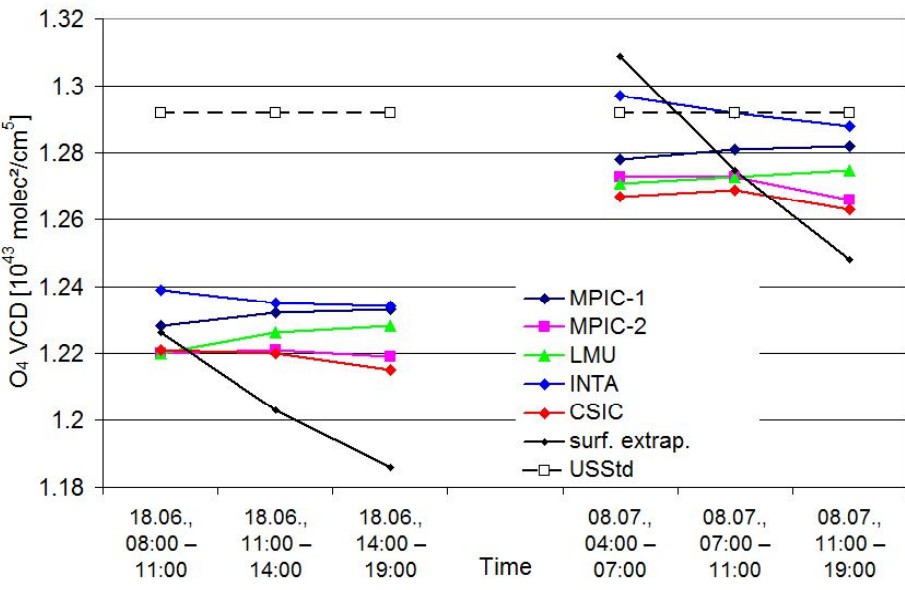


Fig. 7 Comparison of the $O_4$ VCDs for the selected periods on both days calculated from the
profiles extracted by the different groups. Also the results for the profiles extrapolated from
the surface values and the US standard atmosphere are shown.

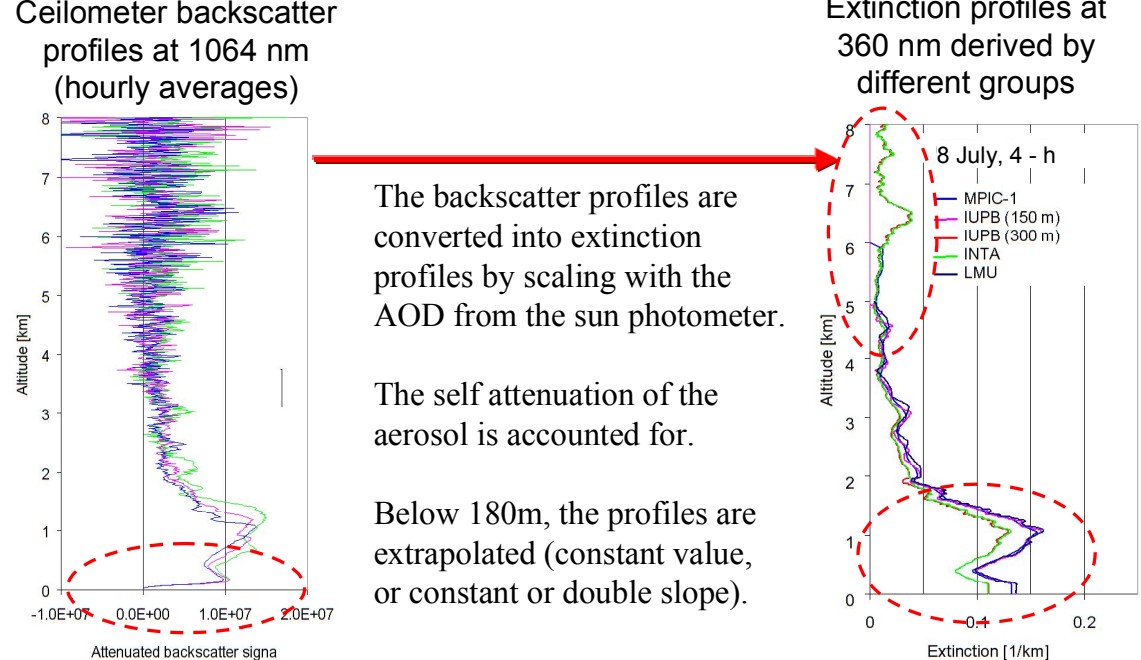

 Fig. 8 Left: Hourly averaged backscatter profiles from the ceilometer measurements for the
 period 4:00 – 7:00 on 8 July 2013. Below 180 m the values rapidly decrease to zero due to the
 missing overlap between the outgoing beam and the field of view of the telescope. Right:
 Aerosol extinction profiles extracted by the different groups from the ceilometer profiles
 (assuming a constant extinction below 180 m). The red circles indicate the height intervals
 with the larges deviations (IUPB 150 m and IUPB 300 m indicate profile extractions with
 different widths of the smoothing kernels: Hanning windows of 150 and 300 m, respectively).


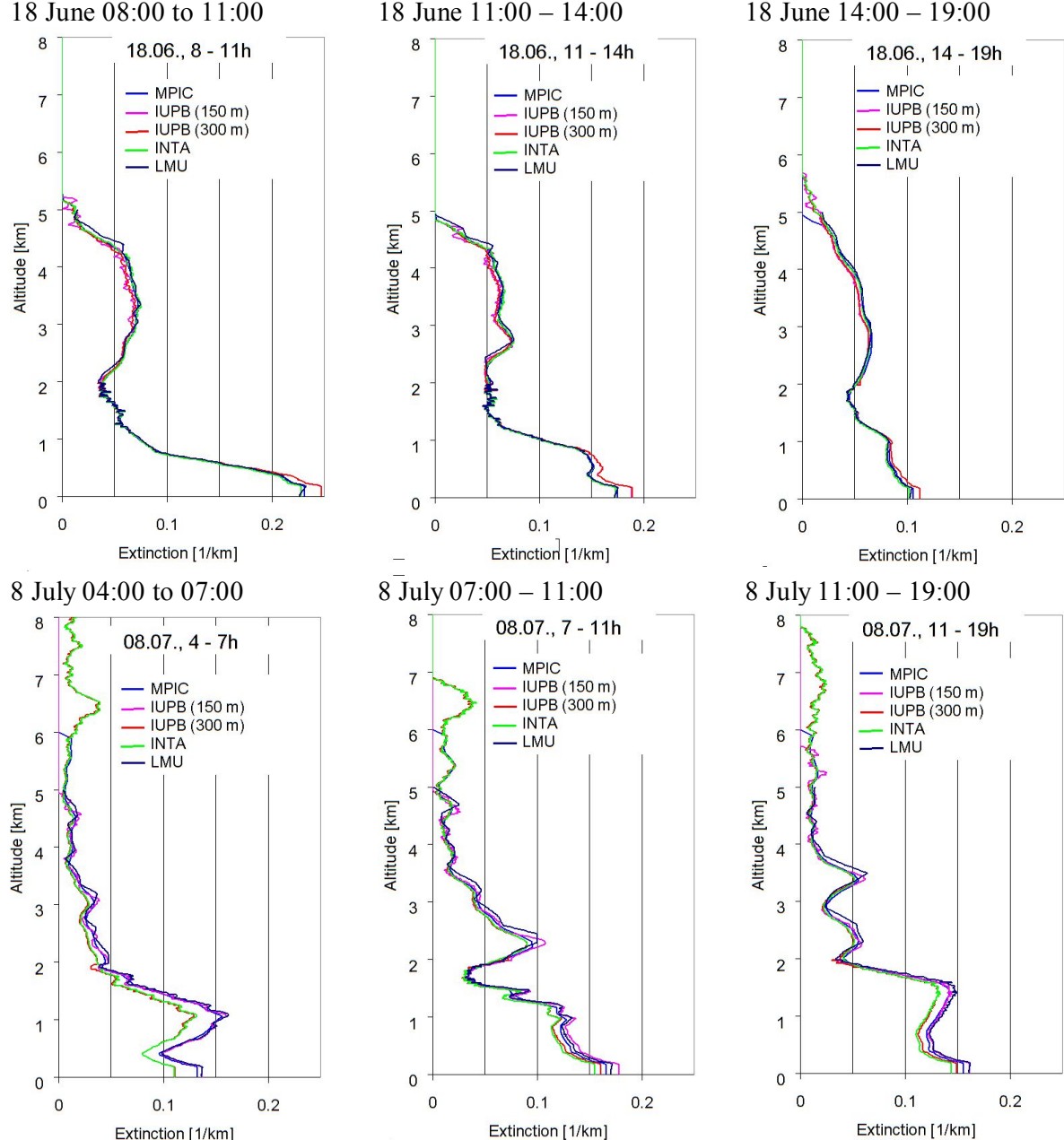

Fig. 9 Comparison of the aerosol extinction profiles extracted by the different groups for all three periods on both days.


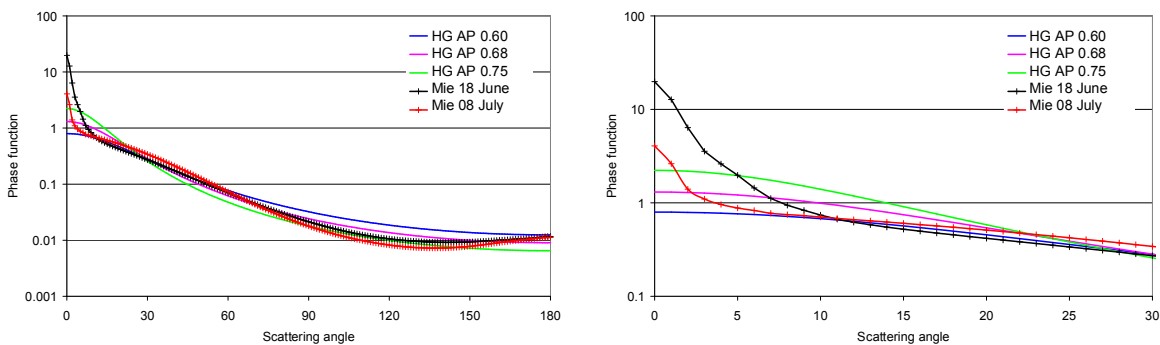

Fig. 10 Comparison of different aerosol phase functions used in the radiative transfer
simulations. The right figure is a zoom of the left figure.

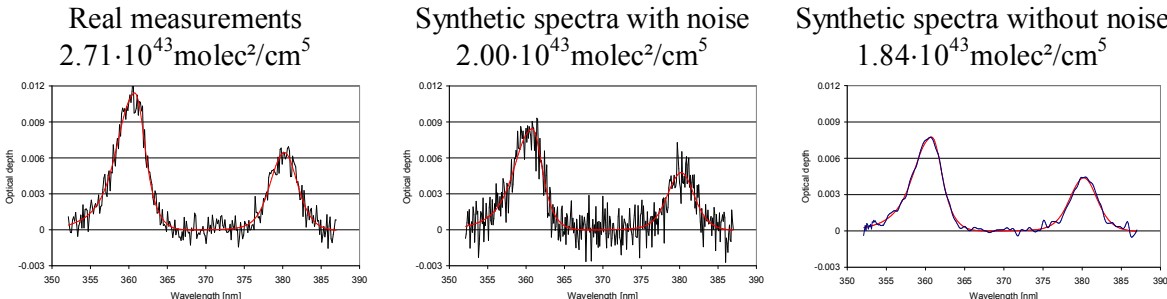

Real measurements
$2.71 \cdot 10^{43} molec^2/cm^5$

Synthetic spectra with noise
$2.00 \cdot 10^{43} molec^2/cm^5$

Synthetic spectra without noise
$1.84 \cdot 10^{43} molec^2/cm^5$

Fig. 11 Spectral analysis results for a real measurement from the MPIC instrument (left) and a
synthetic spectrum with and without noise. Spectra are taken from 8 July 2013 at 11:26
(elevation angle = 1°). The derived $O_4$ dSCD is shown above the individual plots.

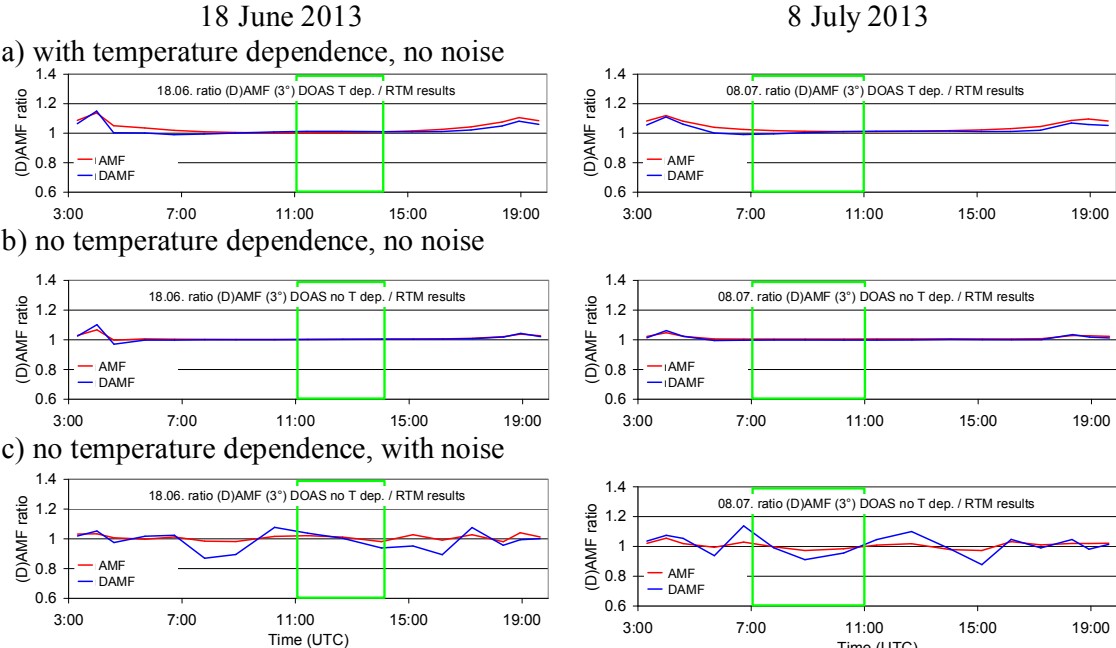

a) with temperature dependence, no noise

b) no temperature dependence, no noise

c) no temperature dependence, with noise

Fig. 12 Ratio of the O₄ (d)AMFs derived from synthetic spectra versus those obtained from radiative transfer simulations at 360 nm for both selected days.

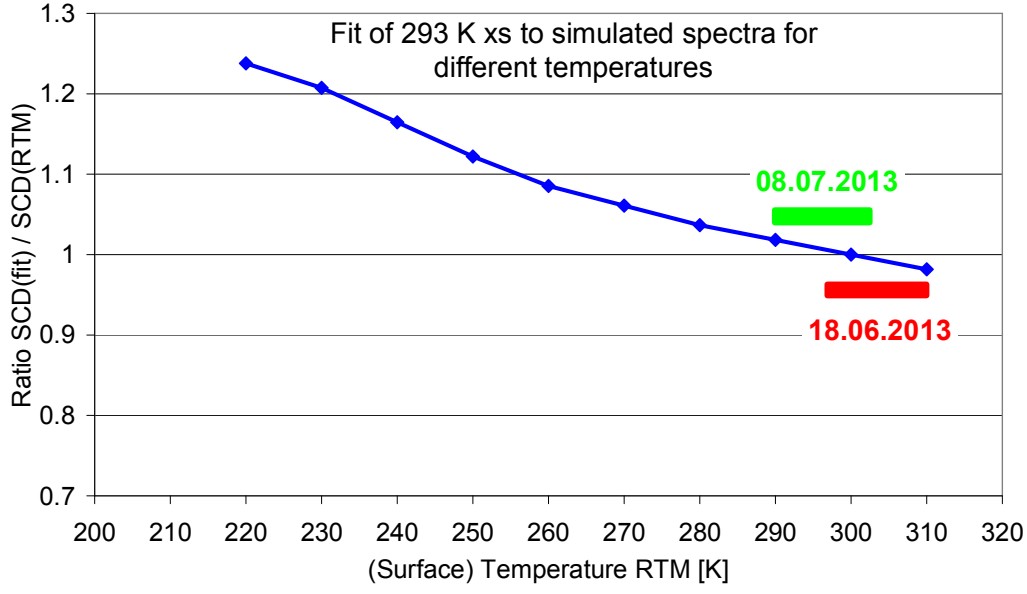

Fig. 13 Ratio of the $O_4$ dAMF obtained from simulated spectra for different surface temperatures by the corresponding $O_4$ dAMFs derived from radiative transfer simulations. The results represent MAX-DOAS observations at low elevation angles (2° to 3°).

1847
1848

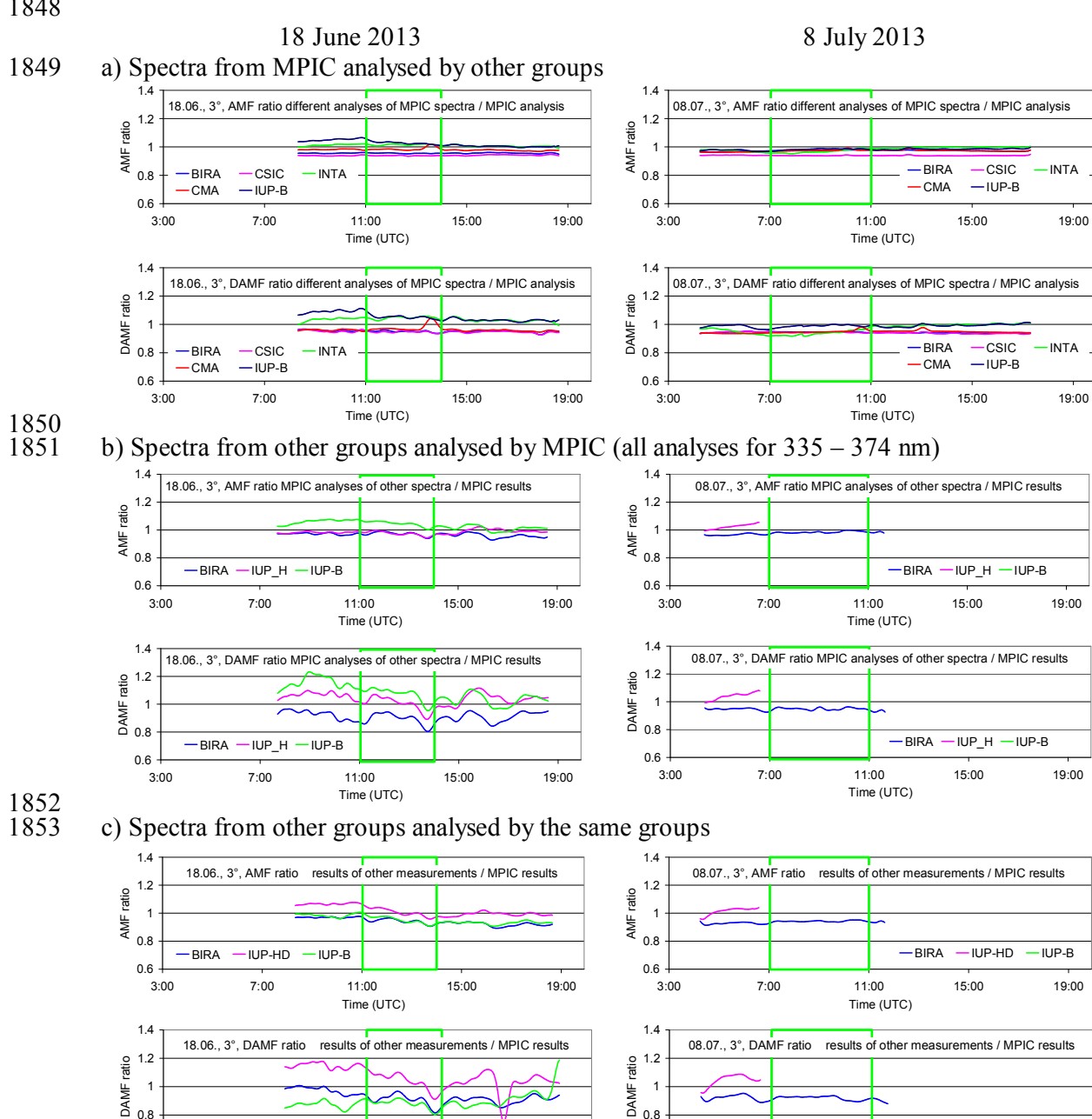

18 June 2013                    8 July 2013

a) Spectra from MPIC analysed by other groups
b) Spectra from other groups analysed by MPIC (all analyses for 335 – 374 nm)
c) Spectra from other groups analysed by the same groups
Fig. 14 a) Ratio of the $O_4$ (d)AMFs derived from MPIC spectra when analysed by other
groups versus those analysed by MPIC for both selected days; b) Ratio of the $O_4$ (d)AMFs
derived from spectra measured and analysed by other groups (using different wavelength
ranges and settings) versus those for the MPIC instrument analysed by MPIC; c) Ratio of the
$O_4$ (d)AMFs derived from spectra measured by other groups but analysed by MPIC versus
those for the MPIC instrument analysed by MPIC (using the spectral range 335 – 374 nm for
all instruments).

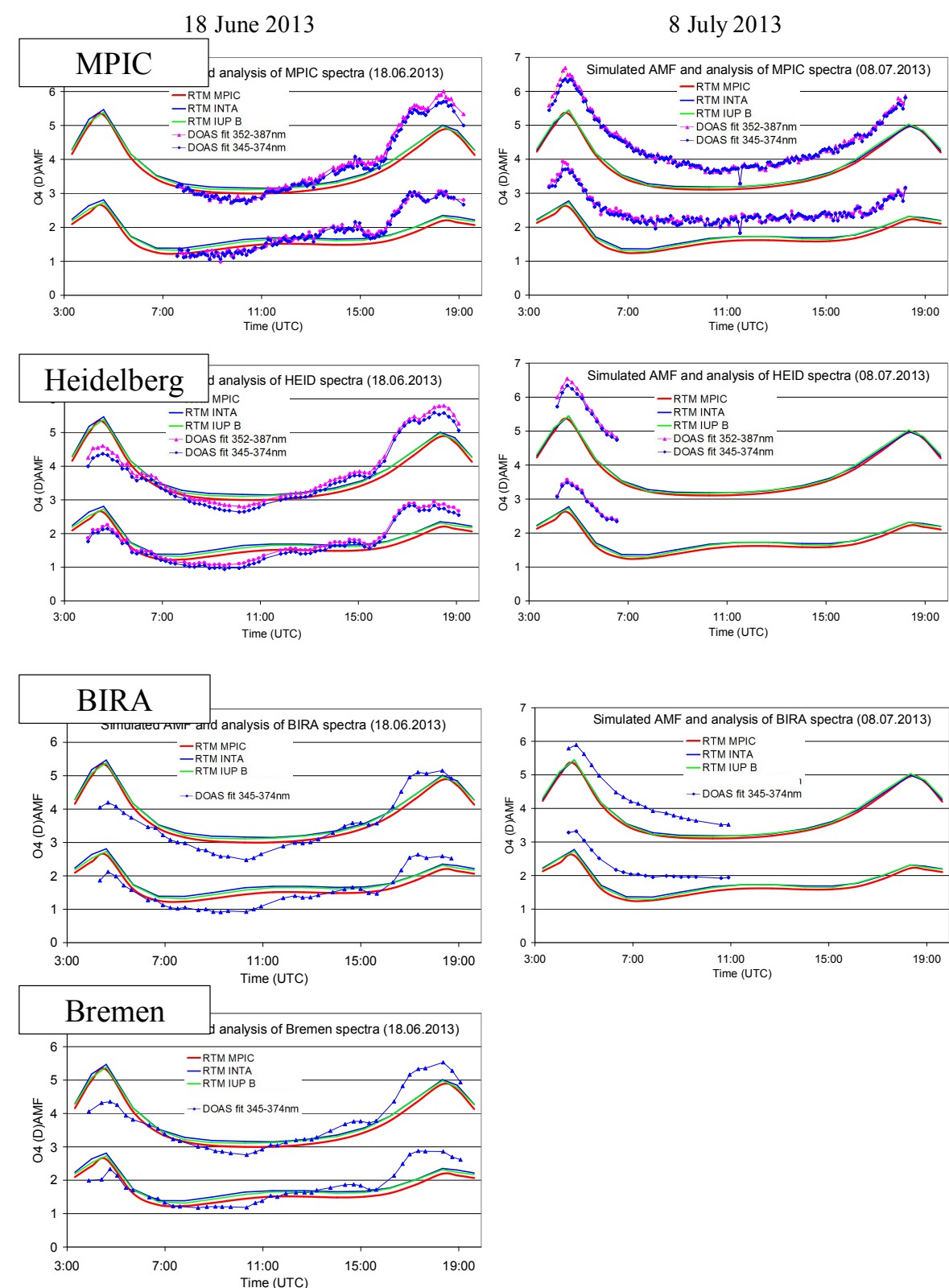

Fig. 15 Comparison of measured and simulated $O_4$ (d)AMFs for both selected days. Measurements are from 4 different instruments, but analysed by MPIC using the standard settings (see Table 7). Simulations are performed by three different groups using Mie phase functions and otherwise the standard settings (see Table 6).

**Appendix A1 Settings used for the simulation of synthetic spectra**
Table A1 Vertical resolution used in radiative transfer simulations for different altitude
ranges.

| Lower boundary [km] | Upper boundary [km] | Vertical resolution [km] |
|---|---|---|
| 0 | 0.5 | 0.02 |
| 0.5 | 2 | 0.1 |
| 2 | 12 | 0.2 |
| 12 | 25 | 1 |
| 25 | 45 | 2 |
| 45 | 100 | 5 |
| 100 | 1000 | 900 |

Table A2 Dependence of SZA and relative azimuth angle on time (UTC) for the standard
viewing direction (51° with respect to North).

| Time (UTC) | SZA | RAZI |
|---|---|---|
| 03:19 | 90 | -0.1 |
| 04:00 | 85 | 7.7 |
| 04:36 | 80 | 14.2 |
| 05:42 | 70 | 26 |
| 06:44 | 60 | 37.5 |
| 07:48 | 50 | 50.1 |
| 08:54 | 40 | 66.2 |
| 10:16 | 30 | 94.6 |
| 11:26 | 26 | 129 |
| 12:40 | 30 | 163.3 |
| 14:02 | 40 | 191.8 |
| 15:09 | 50 | 207.9 |
| 16:11 | 60 | 220.5 |
| 17:14 | 70 | 232 |
| 18:20 | 80 | 243.8 |
| 18:56 | 85 | 250.3 |
| 19:38 | 90 | 258 |


Table A3 Trace gas profiles and cross sections used for the simulation of the synthetic
spectra.

| Trace gas | Vertical profile | Cross section (reference and T) |
|---|---|---|
| $O_4$ | Derived from temperature and pressure profiles during.<br>18.06.: average profiles 11:00 – 14:00<br>08.07.: average profiles 7:00 – 11:00 | Thalman and Volkamer (2013)<br>(203, 233, 253, 273, 293 K)* |
| HCHO | 18.06.: 0-1000m, constant concentration of $2 \cdot 10^{11}$ molec/cm³ (about 8 ppb)<br>08.07.: 0-1000m, constant concentration of $1 \cdot 10^{11}$ molec/cm³ (about 4 ppb) | Meller and Moortgat (2000)<br>(298 K) |
| $NO_2$ | Troposphere<br>18.06.: 0-500m, constant concentration of $4 \cdot 10^{11}$ molec/cm³ (about 16 ppb)<br>08.07.: 0-500m, constant concentration of $2 \cdot 10^{11}$ molec/cm³ (about 8 ppb)<br>Stratosphere:<br>Gaussian profile with maximum at 25 km, and FWHM of 16 km, VCD = $5 \cdot 10^{15}$ molec/cm² | Vandaele et al. (1997)<br>(220, 294 K) |
| $O_3$ | Troposphere (0-8km):<br>constant concentration $6 \cdot 10^{11}$ molec/cm³ (about 24 ppb)<br>Stratosphere:<br>Gaussian profile with maximum at 22 km, and FWHM of 15 km, VCD = 314 DU | Serdyuchenko et al. (2014)<br>(193 – 293 K in steps of 10 K)** |

*The temperature dependence is either considered or a constant temperature of 293 K is
assumed (see text for details).
**The temperature dependence was parameterised according to Paur and Bass (1984).

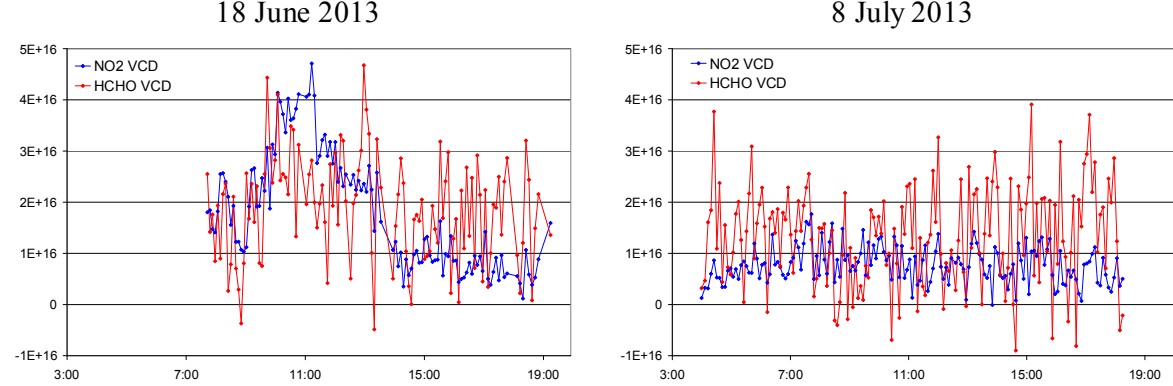

Fig. A1 Tropospheric VCDs of $NO_2$ (blue) and HCHO (red) derived from measurements at
30° elevation using the geometric approximation.

**Appendix A2 Comparison of measured and simulated $O_4$ (d)AMFs for all azimuth and**
**elevation angles of the MPIC MAX-DOAS measurements.**
The settings for the simulation of the synthetic spectra are given in Table 6 and Tables A1,
A2, and A3 in appendix 1. Measurements are analysed using the standard settings (see Table
1915 7).


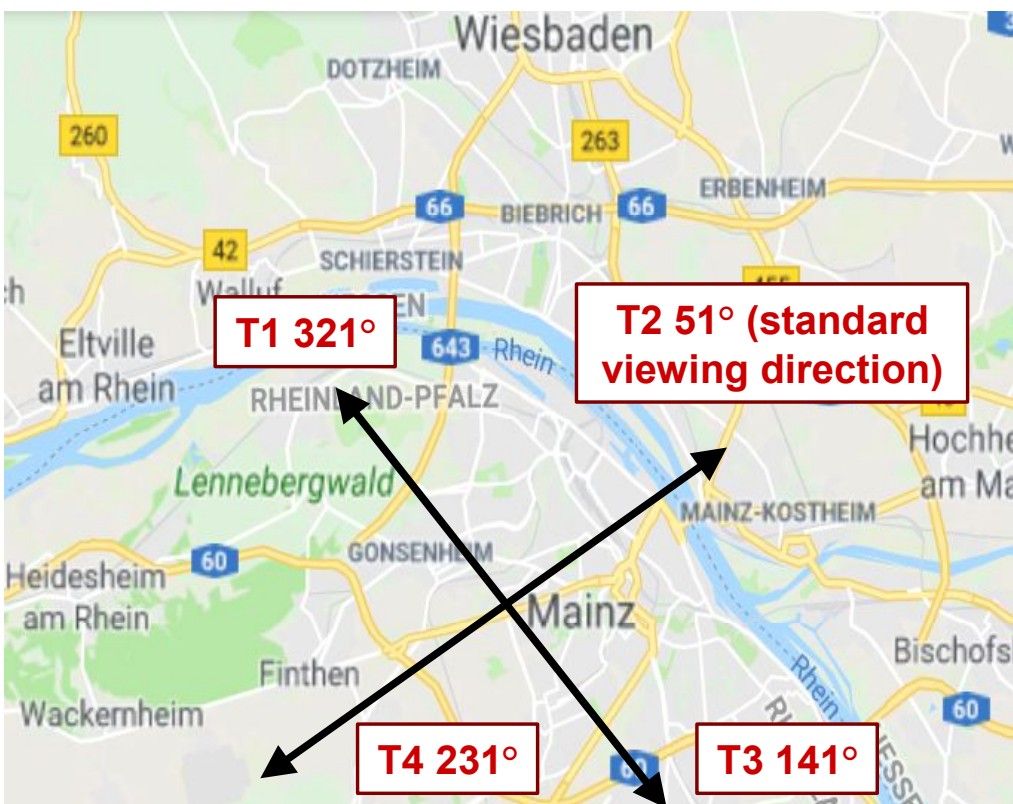

Fig. A2 Azimuth viewing directions of the 4 telescopes (T1 to T4) of the MPIC MAX-DOAS
instrument. The azimuth angles are defined with respect to North (map: © google maps).

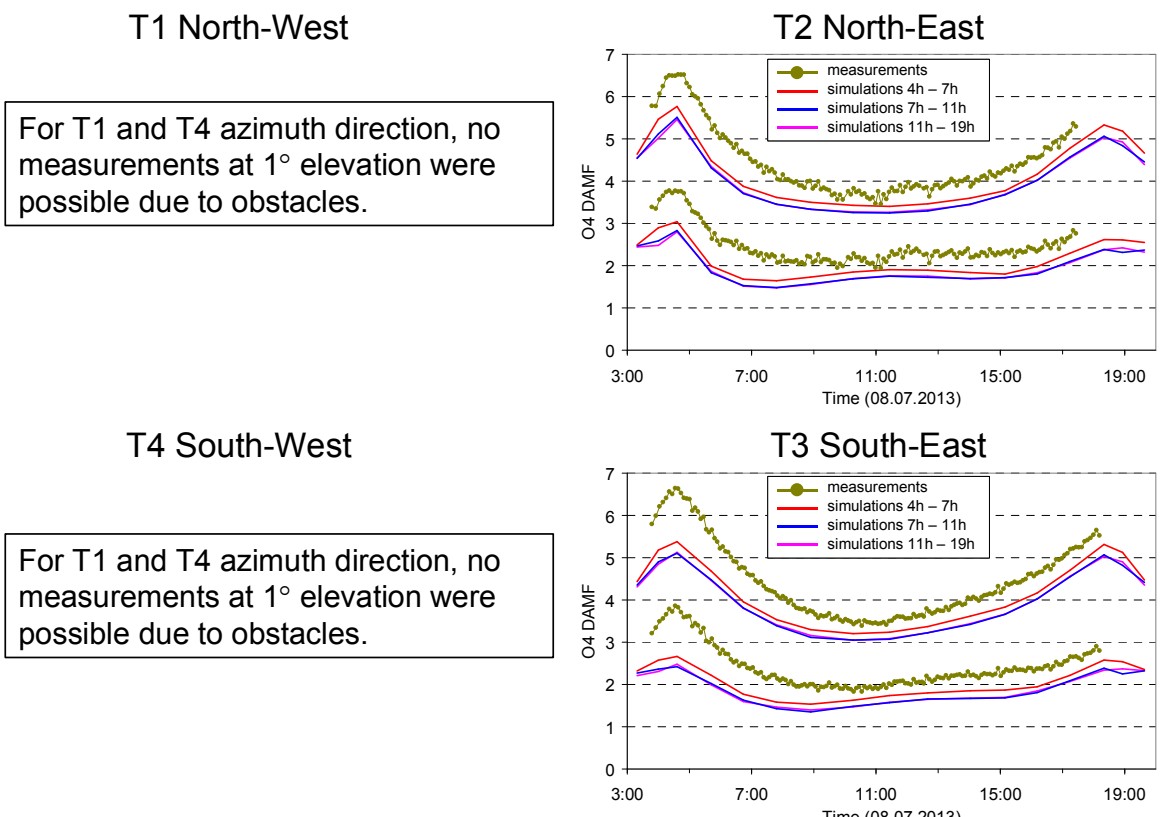

Fig. A3a Comparison results for 1° elevation angles on 8 July 2013. The upper lines indicate
the $O_4$ AMFs, the lower lines the $O_4$ dAMFs (see also Fig. 2 and 3).

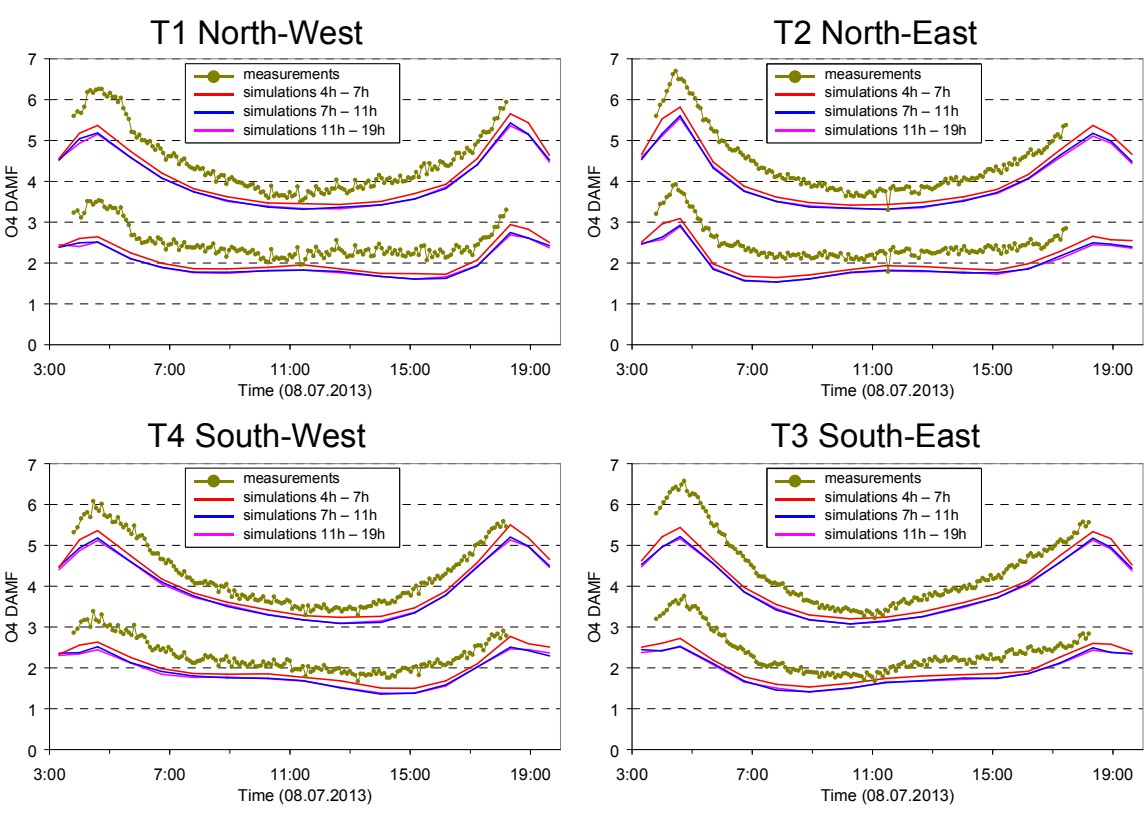

Fig. A3b Comparison results for 3° elevation angles on 8 July 2013.

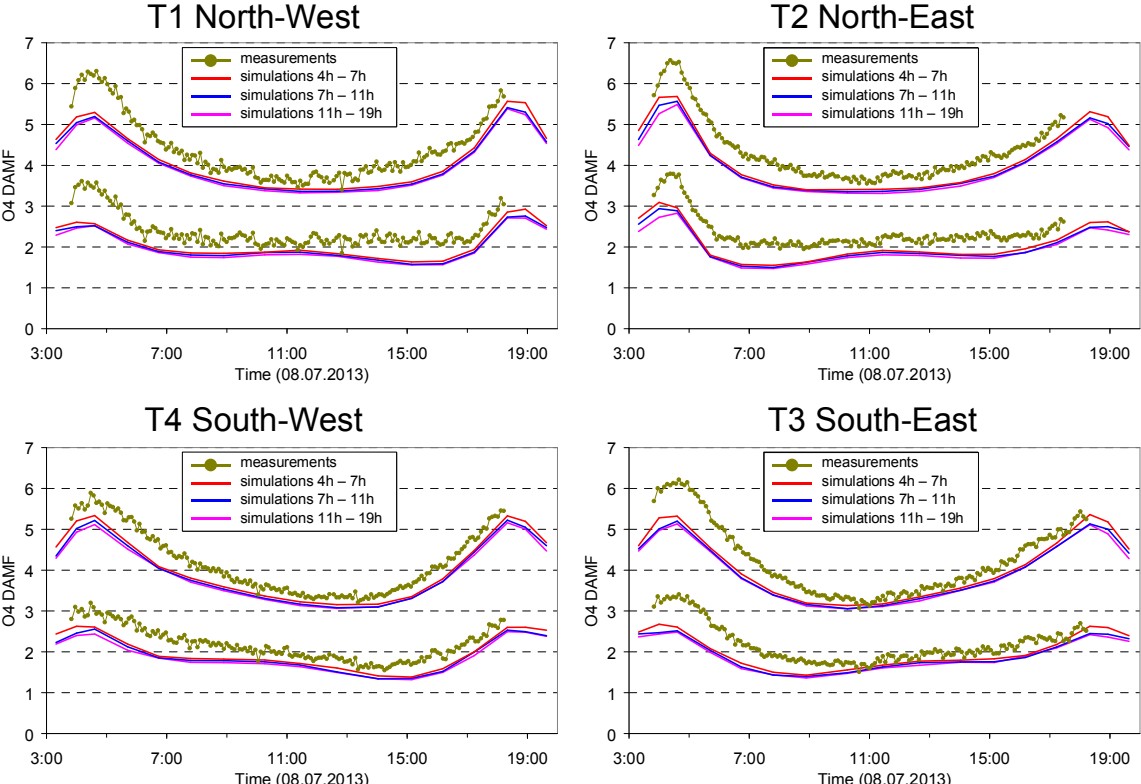

Fig. A3c Comparison results for 6° elevation angles on 8 July 2013.

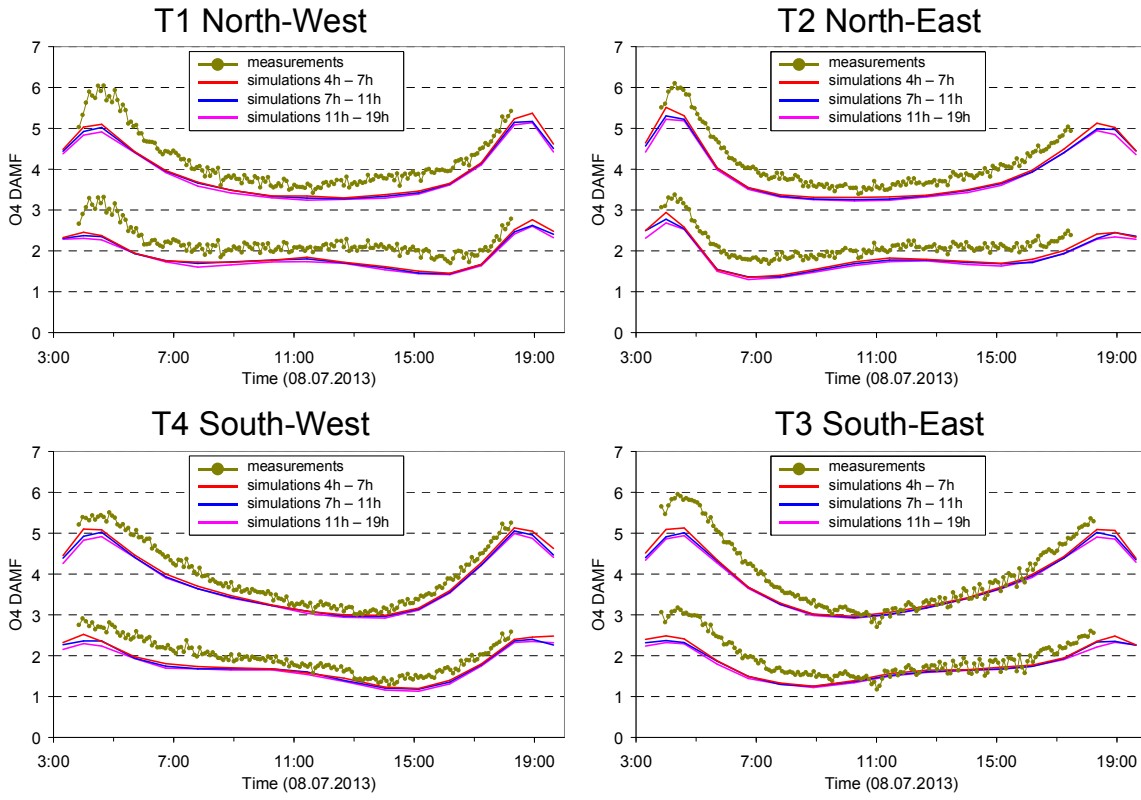

Fig. A3d Comparison results for 10° elevation angles on 8 July 2013.

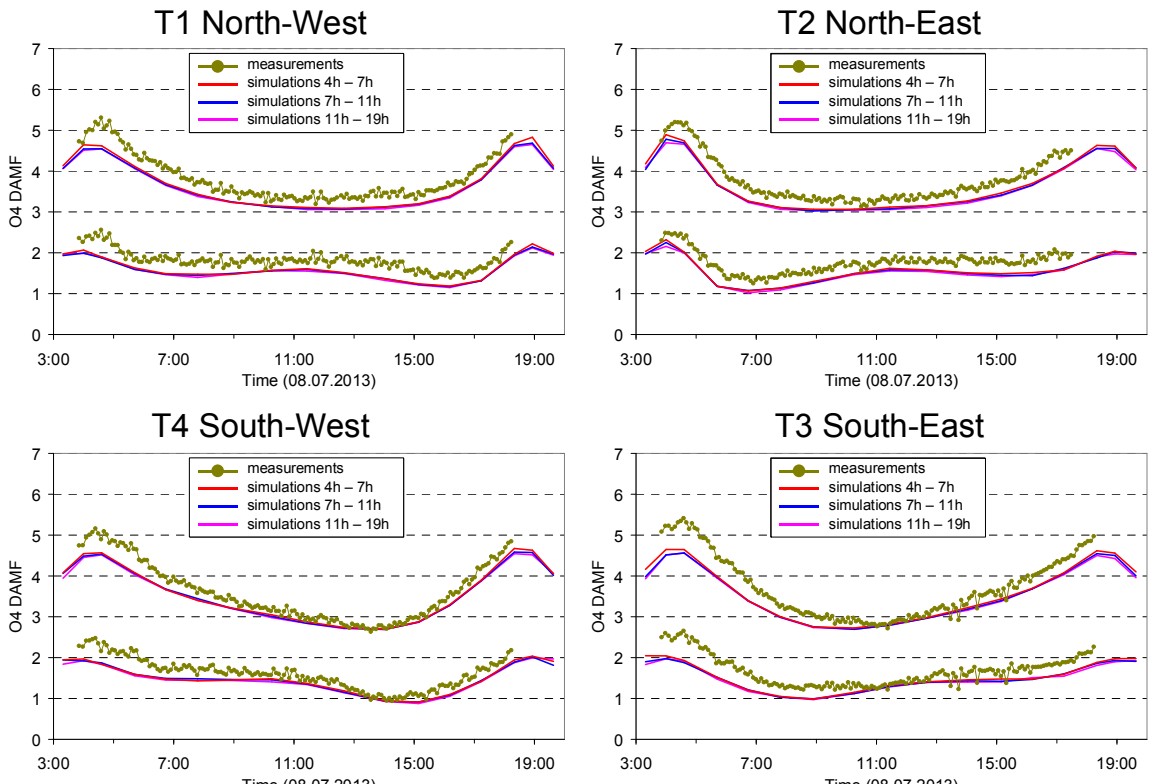

Fig. A3e Comparison results for 15° elevation angles on 8 July 2013.

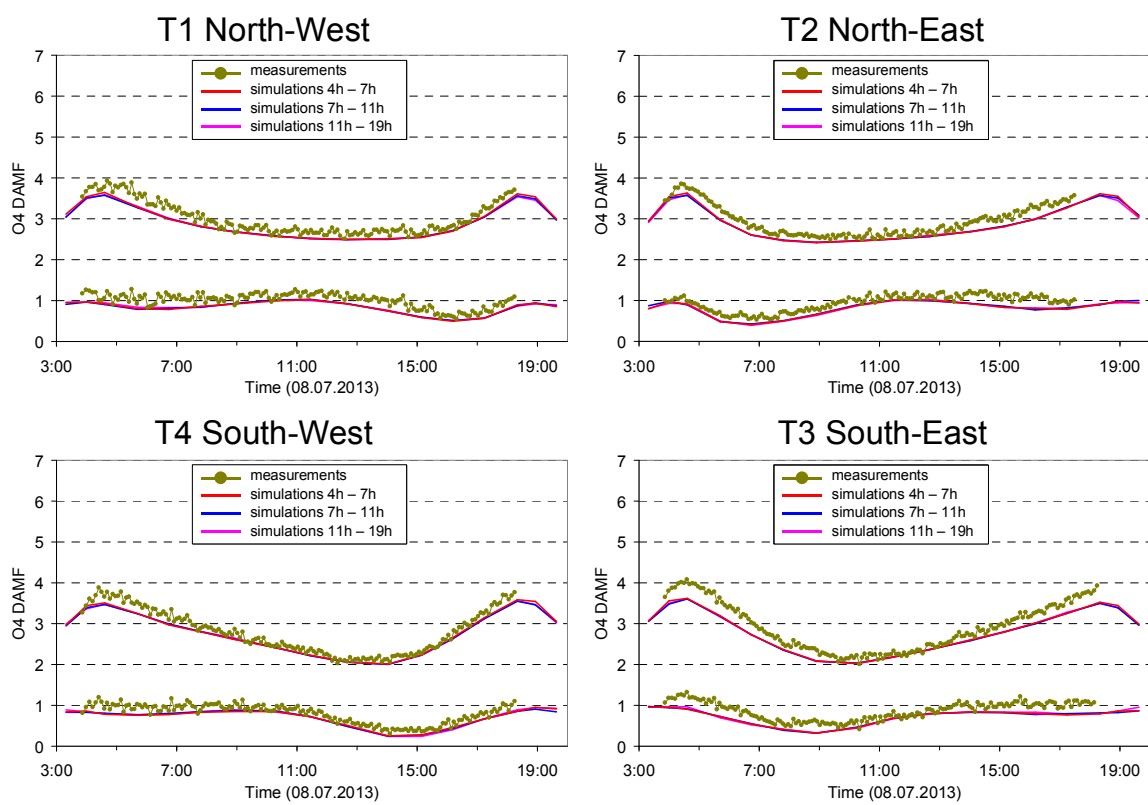

Fig. A3f Comparison results for 30° elevation angles on 8 July 2013.

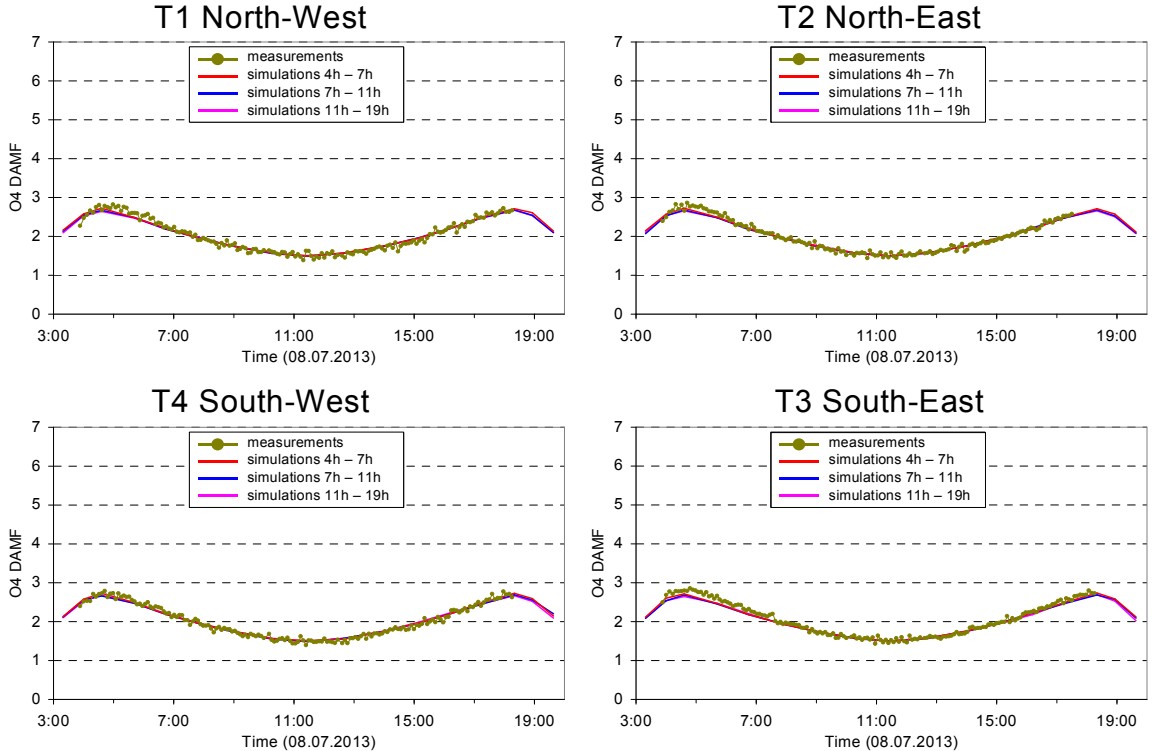

Fig. A3g Comparison results (only O₄ AMFs) for 90° elevation angles on 8 July 2013.

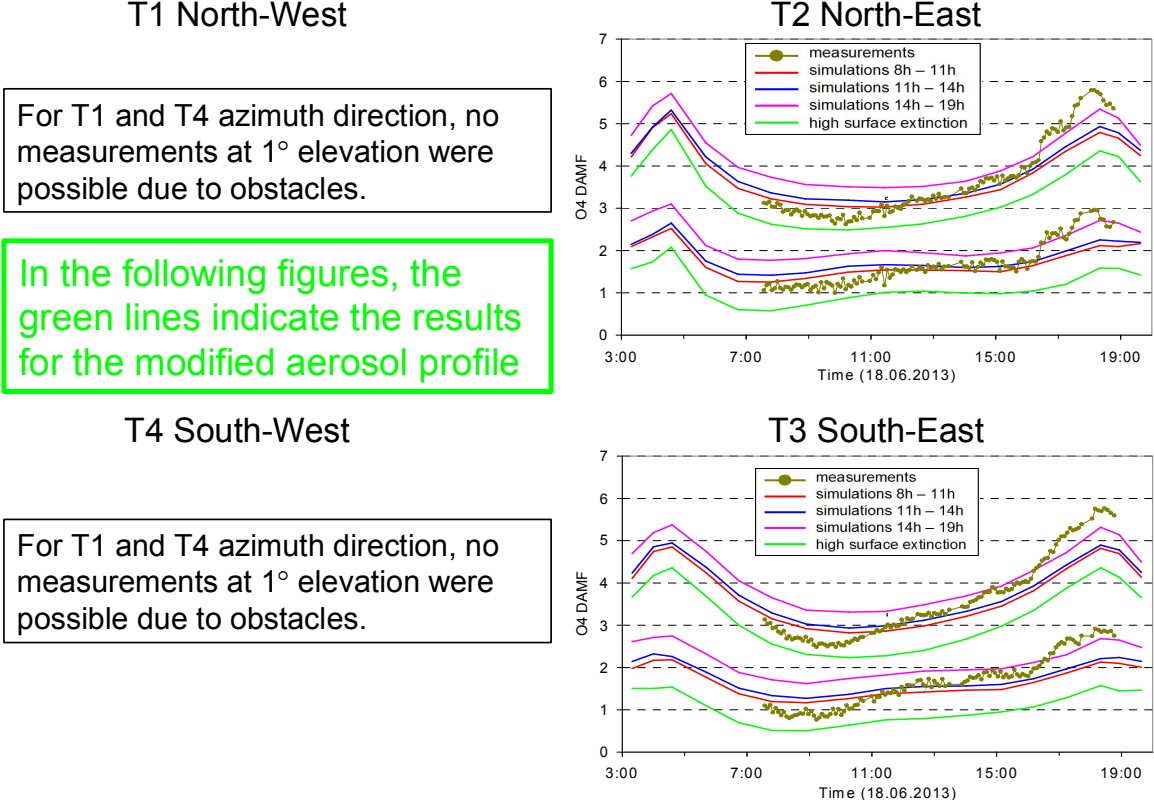

Fig. A4a Comparison results for 1° elevation angles on 18 June 2013 including the RTM
results for the modified aerosol extinction profile (green line).

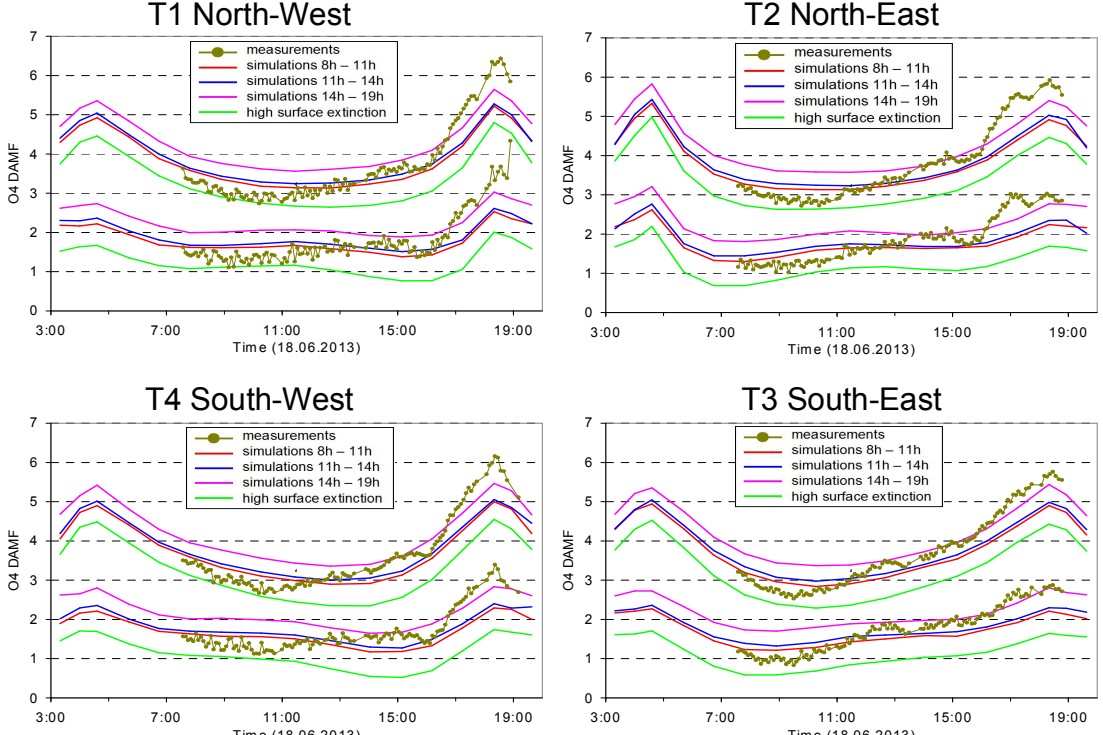

Fig. A4b Comparison results for 3° elevation angles on 18 June 2013 including the RTM
results for the modified aerosol extinction profile (green line)..

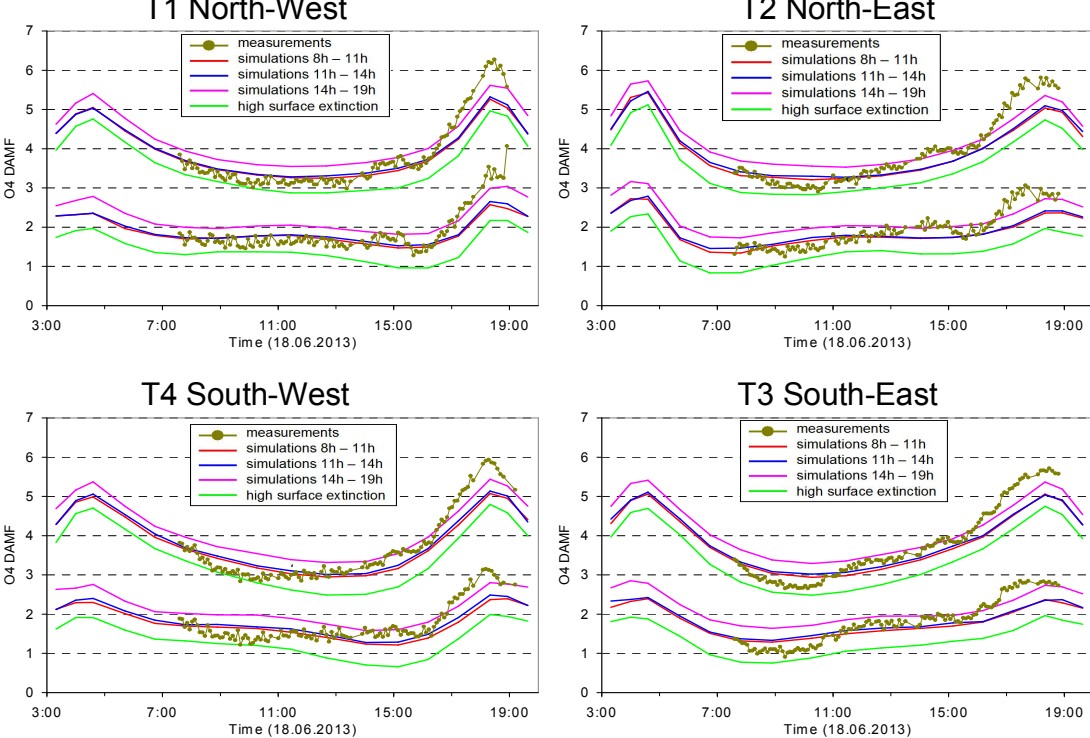

Fig. A4c Comparison results for 6° elevation angles on 18 June 2013 including the RTM
results for the modified aerosol extinction profile (green line).

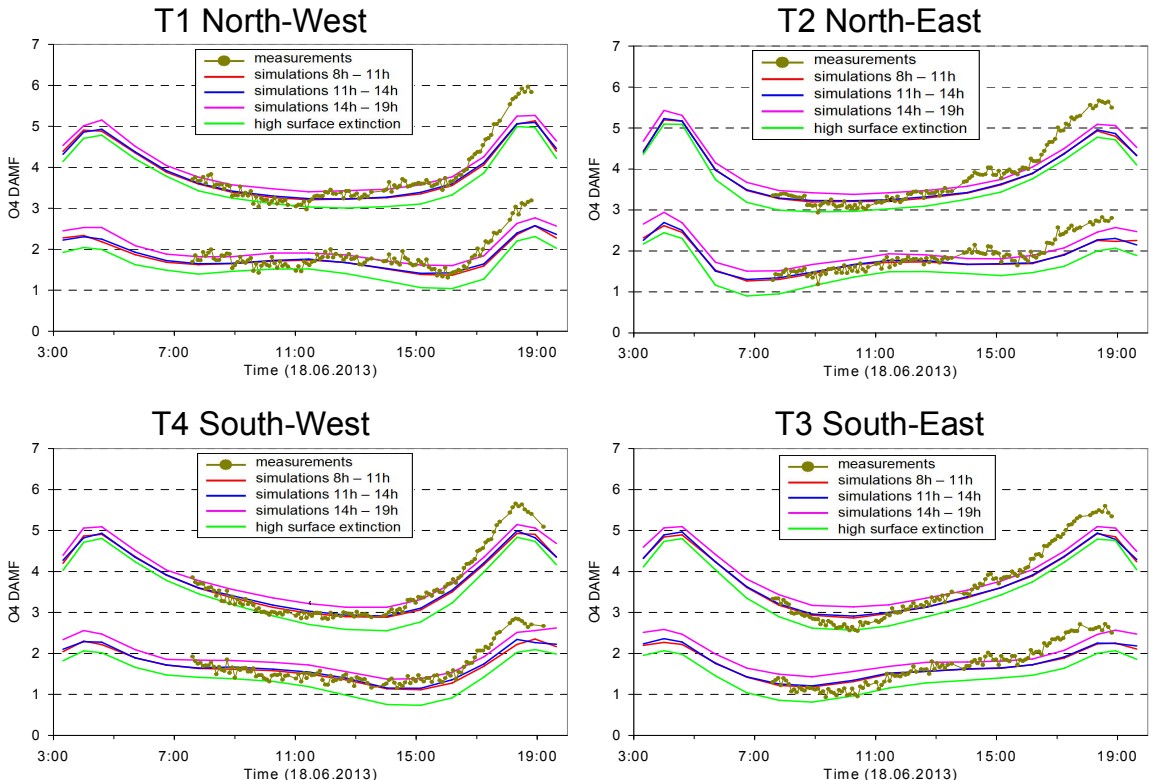

Fig. A4d Comparison results for 10° elevation angles on 18 June 2013 including the RTM
results for the modified aerosol extinction profile (green line).

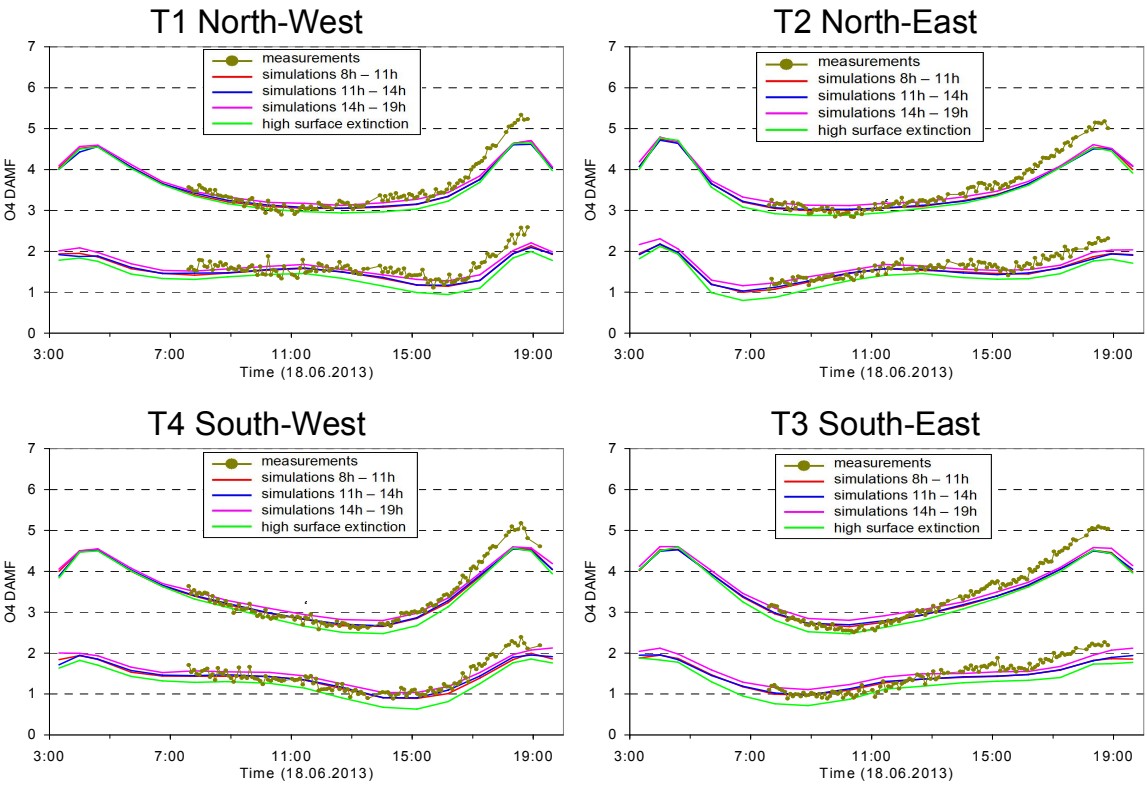

Fig. A4e Comparison results for 15° elevation angles on 18 June 2013 including the RTM
results for the modified aerosol extinction profile (green line)..

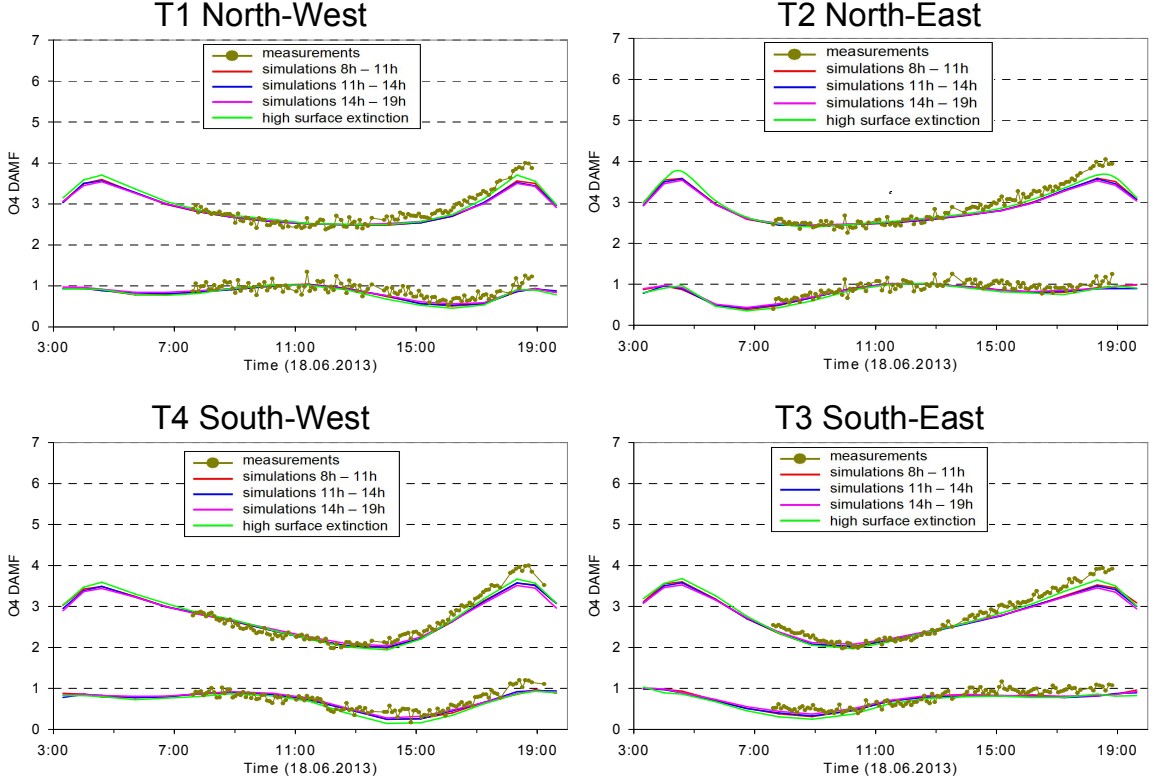

Fig. A4f Comparison results for 30° elevation angles on 18 June 2013 including the RTM
results for the modified aerosol extinction profile (green line)..

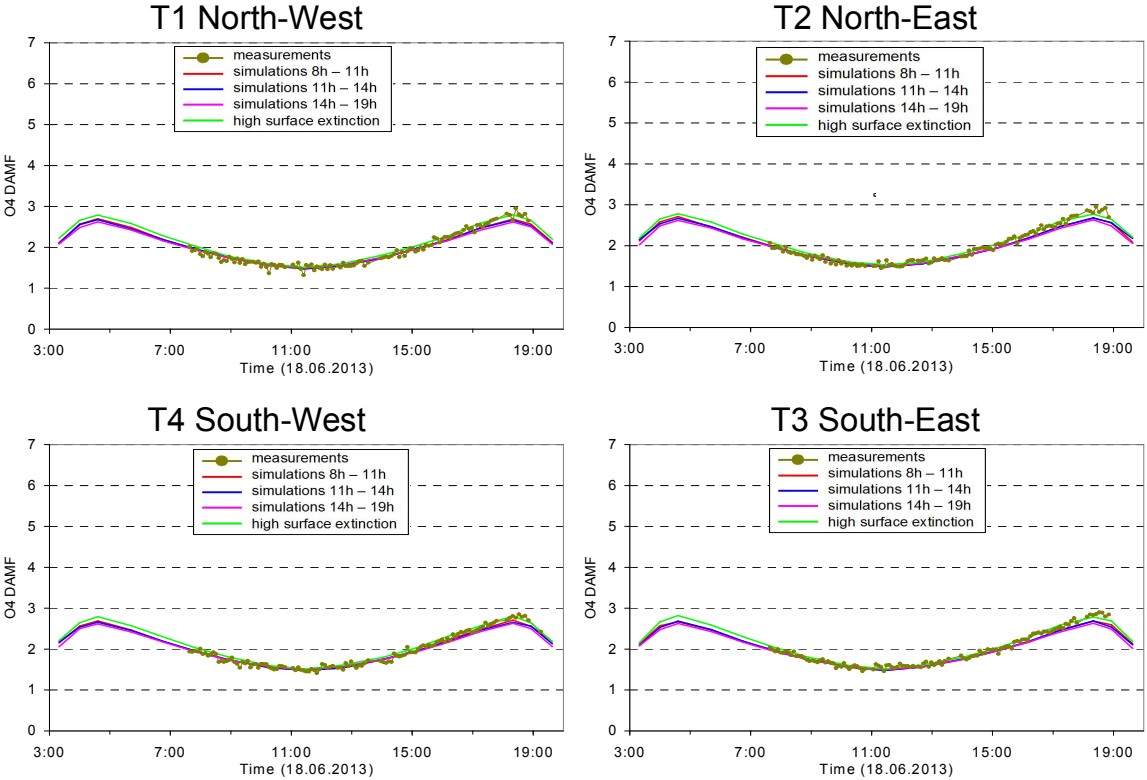

Fig. A4g Comparison results (only O$_4$ AMFs) for 90° elevation angles on 18 June 2013
including the RTM results for the modified aerosol extinction profile (green line).
**Appendix A3 Comparison of the different procedures to extract height profiles of**
**temperature, pressure and $O_4$ concentration**
**Extraction of temperature and pressure profiles**
For the two selected days during the MAD-CAT campaign two data sets of temperature and
pressure are available: surface measurements close to the measurement site and vertical
profiles from ECMWF ERA-Interim re-analysis data (see Table 5). Both data sets are used to
derive the $O_4$ concentration profiles for the three selected periods on both days. The general
procedure is that first the temperature profiles are determined. In a second step, the pressure
profiles are derived from the temperature profiles and the measured surface pressure. For the
temperature profile extraction, three height layers are treated differently:
-below 1 km
Between the surface (~150 m above sea level) and 1 km, the temperature is linearly
interpolated between the average of the in situ measurements of the respective period and the
ECMWF data at 1 km (see next paragraph). This procedure is used to account for the diurnal
variation of the temperature close to the surface. Here it is important to note that for this
surface-near layer the highest accuracy is required, because a) the maximum $O_4$ concentration
is located near the surface, and b) the MAX-DOAS measurements are most sensitive close to
the surface.
-1 km to 20 km
In this altitude range, the diurnal variation of the temperature becomes very small. Thus the
average of the four ECMWF profiles of each day is used (for simplicity, a $6^{th}$ order
polynomial is fitted to the ECMWF data).
-Above 20 km
In this altitude range the accuracy of the temperature profile is not critical and thus the
ECMWF temperature profile for 00:00 UTC of the respective day is used for simplicity.
The temperature profiles for 8 July 2013 extracted in this way are shown in Fig. 4 (left). Close
to the surface the temperature variation during the day is about 10 K.
In the next step, the pressure profiles are determined from the surface pressure (obtained from
the in situ measurements) and the extracted temperature profiles according to the ideal gas
law. In principle the effect of atmospheric humidity could also be taken into account, but the
effect is very small for surface-near layers and is thus ignored here. The derived pressure
profiles for 8 July 2013 are shown in Fig. 4 (right). Excellent agreement with the
corresponding ECMWF pressure profiles is found.
Here it should be noted that in principle also the ECMWF pressure profiles could be used.
However, we chose to determine the pressure profiles from the surface pressure and the
extracted temperature profiles, because this procedure can also be applied if no ECMWF data
(or other information on temperature and pressure profiles) is available.
If no profile data (e.g. from ECMWF) are available, temperature and pressure profiles can
also be extrapolated from surface measurements e.g. by assuming a constant lapse rate of
-0.65 K / 100 m for the altitude range between the surface and 12 km, and a constant
temperature above 12 km (as stated above, uncertainties at this altitude range have only a
negligible effect on the $O_4$ VCD). If no measurements or model data are available at all, a
fixed temperature and pressure profile can be used, e.g. the US standard atmosphere (United
States Committee on Extension to the Standard Atmosphere, 1976).

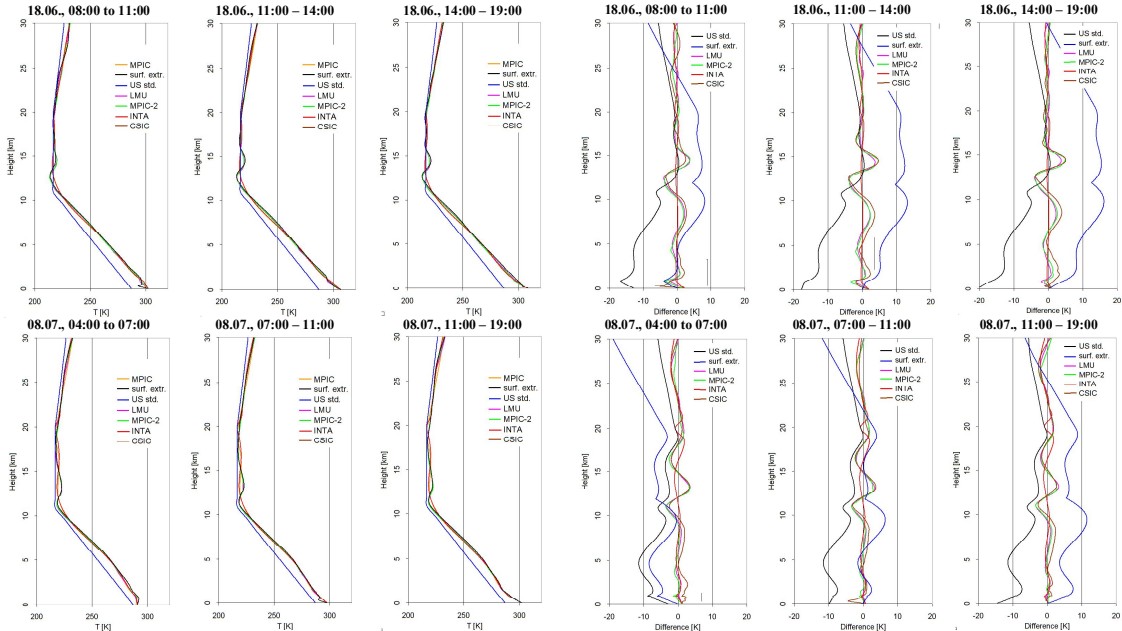

Fig. A5a Left: Comparison of temperature profiles extracted by the different groups (also
shown are the profiles from the US standard atmosphere and the profiles extrapolated from
the surface measurements). Right: Differences of these profiles compared to the MPIC
standard extraction.

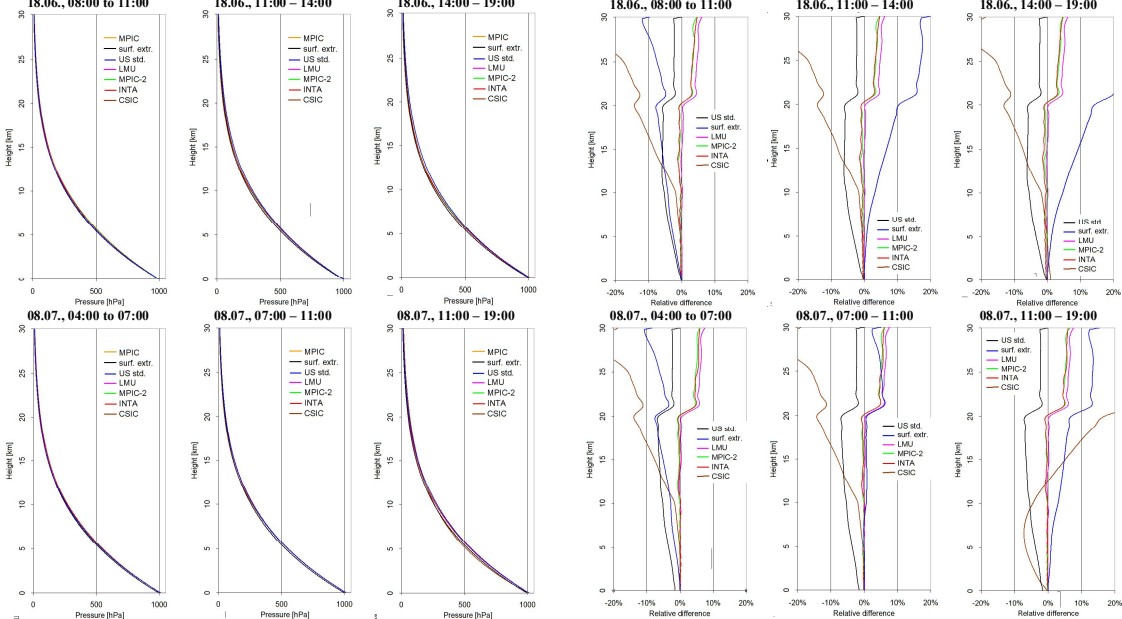

Fig. A5b Left: Comparison of pressure profiles extracted by the different groups (also shown
are the profiles from the US standard atmosphere and the profiles extrapolated from the
surface measurements). Right: Differences of these profiles compared to the MPIC standard
extraction.

**Determination of the uncertainties of the $O_4$ profiles and $O_4$ VCDs caused by uncertainties of the input parameters**

The uncertainties of the $O_4$ profiles and $O_4$ VCDs are derived by varying the input parameters according to their uncertainties. The following results are obtained:

-The variation of the temperature (whole profile) by about 2K leads to variations of the $O_4$ concentration (or $O_4$ VCD) by about 0.8%.

-The variation of the surface pressure by about 3 hPa leads to variations of the $O_4$ concentration (or $O_4$ VCD) by about 0.7%.

-The effect of uncertainties of the relative humidity depends strongly on temperature: For surface temperatures of 0°C, 10°C, 20°C, 30°C, and 35°C a variation of the relative humidity of 30% leads to variations of the $O_4$ concentration (or $O_4$ VCDs) of about 0.15%, 0.3%, 0.6%, 1.2%, and 1.6%, respectively. If the effect of atmospheric humidity is completely ignored (dry air is assumed), the resulting $O_4$ concentrations (or $O_4$ VCDs) are systematically overestimated by about 0.3%, 0.7%, 1.3%, 2.5%, and 4% for surface temperatures of 0°C, 10°C, 20°C, 30°C, and 35°C, respectively (assuming a relative humidity of 70%). In this study we used the relative humidity measured by the in situ sensors. We took these values not only for the surface layers, but also for the whole troposphere. Here it should be noted that the related uncertainties of the absolute humidity decrease quickly with altitude because the absolute humidity itself decreases quickly with altitude. Since both selected days were warm or even hot summer days, we estimate the uncertainty of the $O_4$ concentration and $O_4$ VCDs due to uncertainties of the relative humidity to 1% and 0.4% on 18 June and 8 July, respectively.

Assuming that the uncertainties of the three input parameters are independent, the total uncertainty related to these parameters is estimated to be about 1.5%.

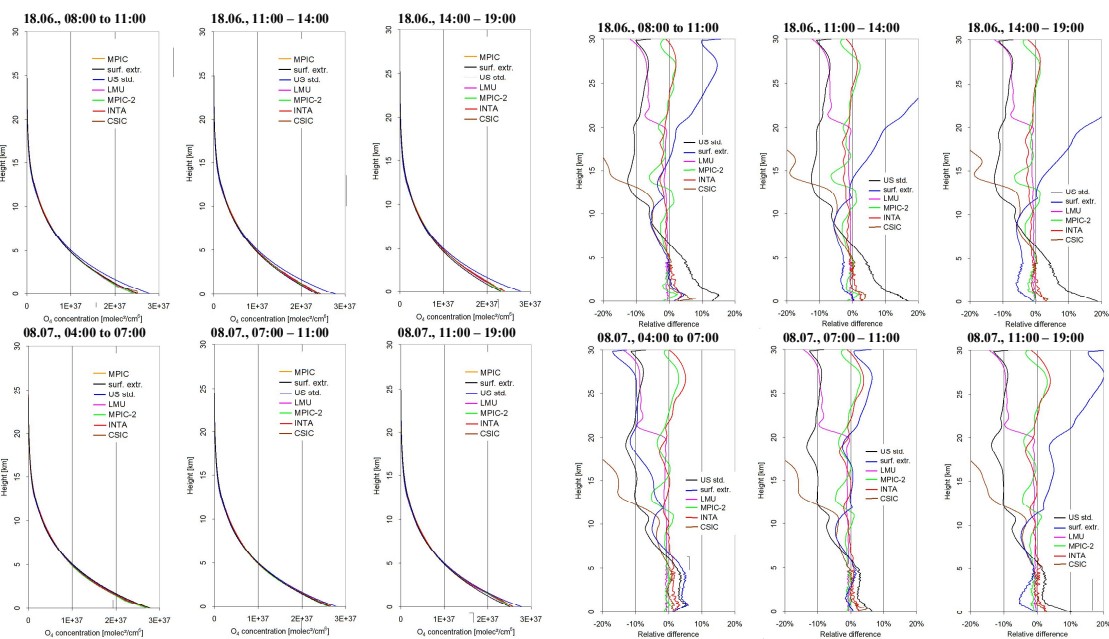

Fig. A5c Left: Comparison of $O_4$ concentration profiles extracted by the different groups (also shown are the profiles from the US standard atmosphere and the profiles extrapolated from the surface measurements). Right: Differences of these profiles compared to the MPIC standard extraction.

**Appendix A4 Results of the sensitivity studies of simulated and measured O$_4$ (d)MFs**

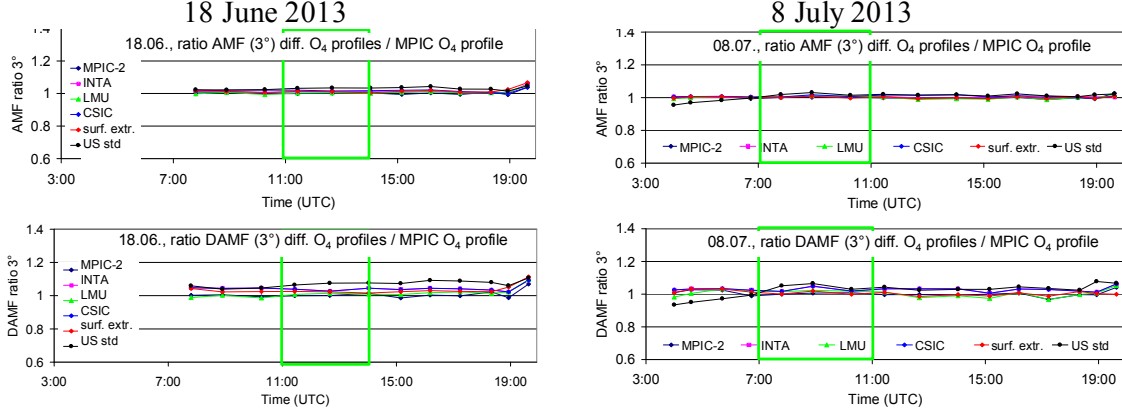

Fig. A6 Ratio of the O$_4$ AMFs (top) and O$_4$ dAMFs (bottom) derived for different O$_4$ profiles
versus the standard O$_4$ profile (MPIC) for both selected days. Besides the O$_4$ profiles
extracted by the different groups, also the O$_4$ profiles derived from the US standard
atmosphere and for the extrapolation of the surface values are included.

Table A4 Average ratios of $O_4$ (d)AMFs simulated for different $O_4$ profiles versus the results
for the standard settings (using the MPIC $O_4$ profiles) for the two middle periods on both
selected days.

| $O_4$ profile extraction | AMF ratios | | | dAMF ratios | |
|---|---|---|---|---|---|
| | 18 June 2013, 11:00 – 14:00 | 8 July 2013, 7:00 – 11:00 | | 18 June 2013, 11:00 – 14:00 | 8 July 2013, 7:00 – 11:00 |
| MPIC-2 | 1.00 | 1.00 | | 1.00 | 1.00 |
| INTA | 1.01 | 1.01 | | 1.02 | 1.01 |
| LMU | 1.00 | 1.00 | | 1.01 | 1.02 |
| CSIC | 1.02 | 1.01 | | 1.04 | 1.02 |
| Lapse rate | 1.01 | 1.00 | | 1.02 | 1.01 |
| US std. atm. | 1.03 | 1.02 | | 1.07 | 1.04 |


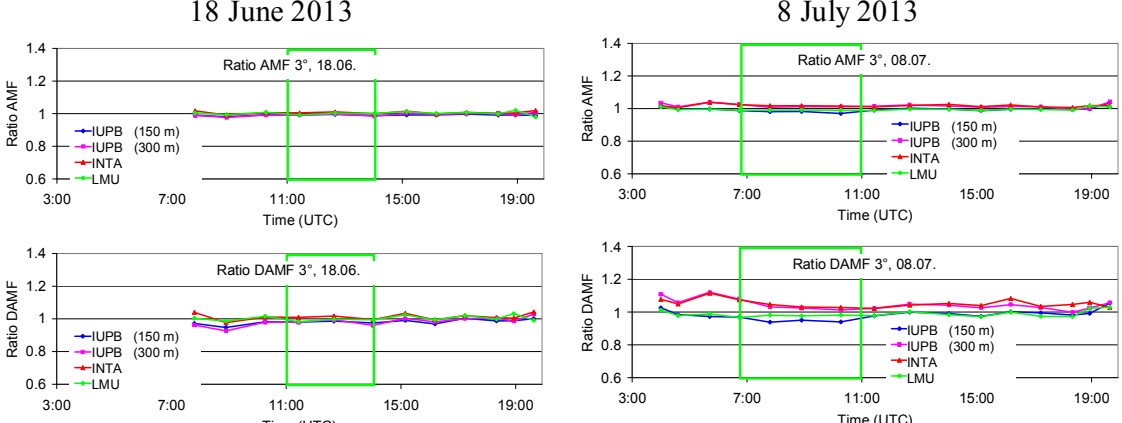

Fig. A7 Ratio of the $O_4$ AMFs (top) and $O_4$ dAMFs (bottom) derived for aerosol extinction
profiles extracted by different groups versus the standard aerosol extinction profiles (MPIC)
for both selected days.
Table A5 Average ratios of $O_4$ (d)AMFs simulated for different aerosol extinction profiles
versus the results for the standard settings (using the MPIC aerosol extinction profiles) for the
two middle periods on both selected days.

| Aerosol profile extraction | AMF ratios | | | dAMF ratios | |
|---|---|---|---|---|---|
| | 18 June 2013, 11:00 – 14:00 | 8 July 2013, 7:00 – 11:00 | | 18 June 2013, 11:00 – 14:00 | 8 July 2013, 7:00 – 11:00 |
| INTA | 1.01 | 1.02 | | 1.01 | 1.04 |
| IUP-B 150 m | 0.99 | 0.98 | | 0.98 | 0.96 |
| IUP-B 300 m | 0.99 | 1.01 | | 0.98 | 1.03 |
| LMU | 1.00 | 0.99 | | 0.99 | 0.98 |


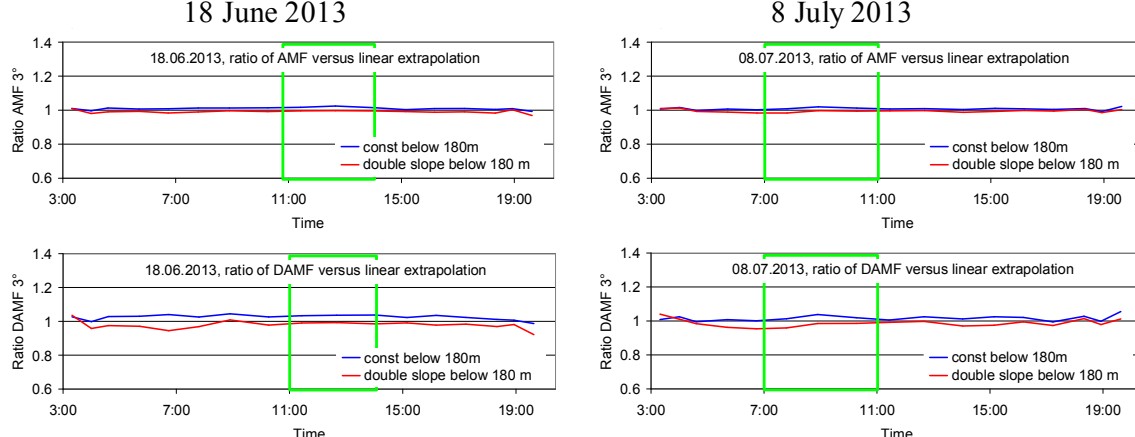

Fig. A8 Ratio of the O$_4$ AMFs (top) and O$_4$ dAMFs (bottom) derived for different
extrapolations of the aerosol extinction profiles below 180 m versus those for the standard
settings (linearly extrapolated profiles) for both selected days.

Table A6 Average ratios of O$_4$ (d)AMFs simulated for aerosol extinction profiles with
different extrapolations below 180 m versus the results for the standard settings (linear
extrapolation) for the two middle periods on both selected days.

| | **AMF ratios** | | | **dAMF ratios** | |
|---|---|---|---|---|---|
| Extrapolation below 180 m | 18 June 2013, 11:00 – 14:00 | 8 July 2013, 7:00 – 11:00 | | 18 June 2013, 11:00 – 14:00 | 8 July 2013, 7:00 – 11:00 |
| Constant extinction | 1.02 | 1.01 | | 1.04 | 1.02 |
| Double slope | 1.00 | 0.99 | | 0.99 | 0.98 |


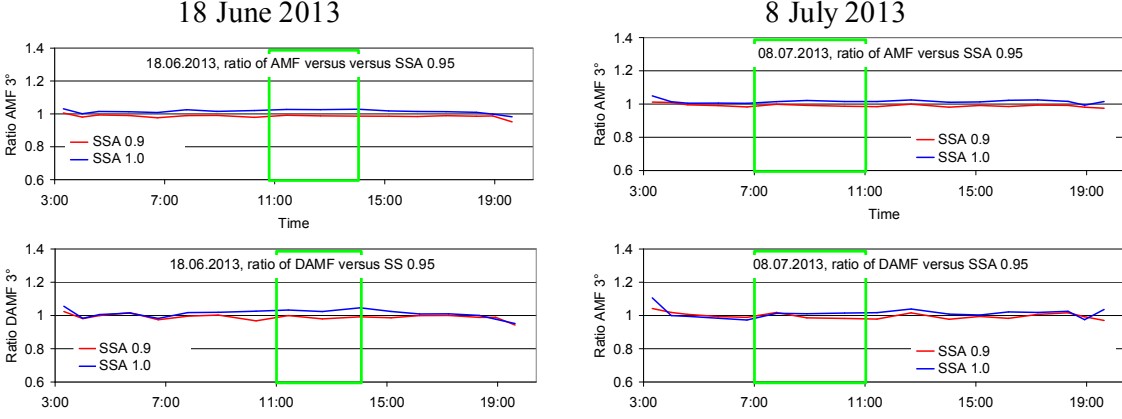

Fig. A9 Ratio of the O$_4$ AMFs (top) and O$_4$ dAMFs (bottom) derived for different aerosol
single scattering albedos versus those for the standard settings (single scattering albedo of
0.95) for both selected days.


Table A7 Average ratios of O$_4$ (d)AMFs simulated for different aerosol single scattering
albedos (SSA) versus the results for the standard settings (single scattering albedo of 0.95) for
the two middle periods on both selected days.

| | **AMF ratios** | | | **dAMF ratios** | |
|---|---|---|---|---|---|
| Single scattering albedo | 18 June 2013, 11:00 – 14:00 | 8 July 2013, 7:00 – 11:00 | | 18 June 2013, 11:00 – 14:00 | 8 July 2013, 7:00 – 11:00 |
| 0.9 | 0.99 | 0.99 | | 0.99 | 0.99 |
| 1.0 | 1.03 | 1.01 | | 1.03 | 1.01 |


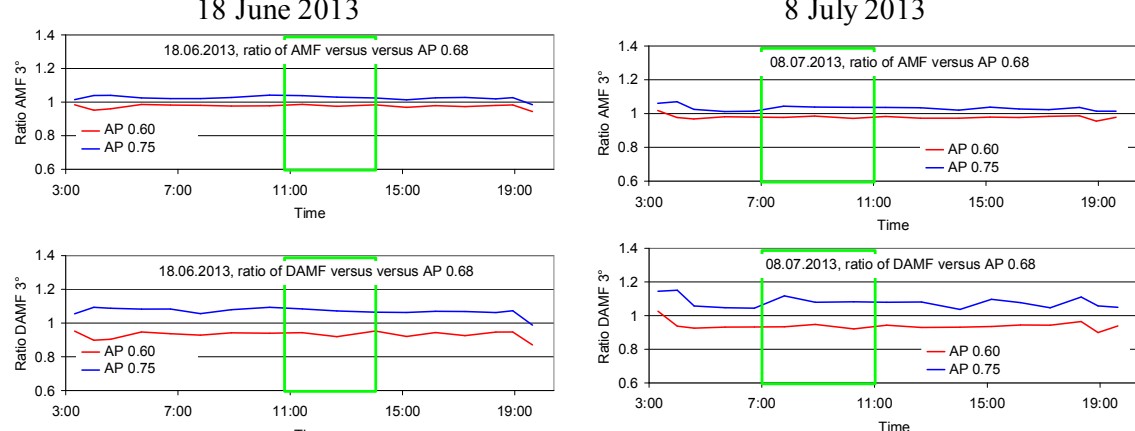

Fig. A10 Ratio of the O$_4$ AMFs (top) and O$_4$ dAMFs (bottom) derived for different aerosol
phase functions (HG-parameterisation with different asymmetry parameters) versus those for
the standard settings (asymmetry parameter of 0.68) for both selected days.
Table A8 Average ratios of O$_4$ (d)AMFs simulated for different aerosol phase functions (HG-
parameterisation with different asymmetry parameters (AP) versus the results for the standard
settings (asymmetry parameter of 0.68) for the two middle periods on both selected days.

| | **AMF ratios** | | | **dAMF ratios** | |
|---|---|---|---|---|---|
| Asymmetry parameter | 18 June 2013, 11:00 – 14:00 | 8 July 2013, 7:00 – 11:00 | | 18 June 2013, 11:00 – 14:00 | 8 July 2013, 7:00 – 11:00 |
| 0.6 | 0.98 | 0.98 | | 0.94 | 0.94 |
| 0.75 | 1.03 | 1.03 | | 1.08 | 1.07 |


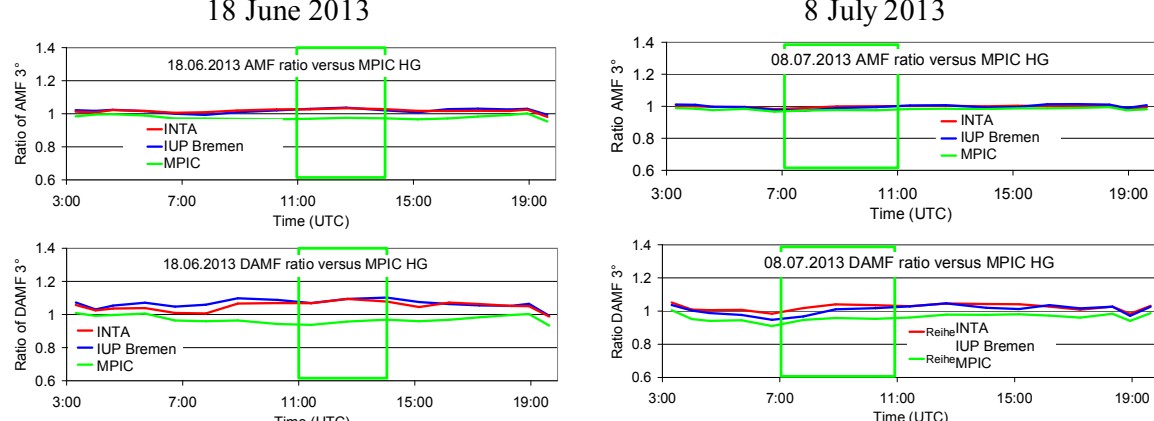

Fig. A11 Ratio of the O$_4$ AMFs (top) and O$_4$ dAMFs (bottom) simulated by INTA and IUP-Bremen and MPIC (SCIATRAN) for phase functions derived from the sun photometer measurements versus those simulated by MPIC using the Henyey Greenstein phase function for asymmetry parameter of 0.68 for both selected days.

Table A9 Average ratios of O$_4$ (d)AMFs simulated by INTA and IUP-Bremen and MPIC (SCIATRAN) for phase functions derived from the sun photometer measurements versus those simulated by MPIC using the Henyey Greenstein phase function for asymmetry parameter of 0.68 for the two middle periods on both selected days.

| | **AMF ratios** | | | **dAMF ratios** | |
|---|---|---|---|---|---|
| Group (RTM) | 18 June 2013, 11:00 – 14:00 | 8 July 2013, 7:00 – 11:00 | | 18 June 2013, 11:00 – 14:00 | 8 July 2013, 7:00 – 11:00 |
| INTA (LIDORT) | 1.03 | 1.00 | | 1.09 | 1.02 |
| IUP-Bremen (SCIATRAN) | 1.03 | 0.99 | | 1.08 | 0.99 |
| MPIC (SCIATRAN) | 0.97 | 0.98 | | 0.95 | 0.95 |

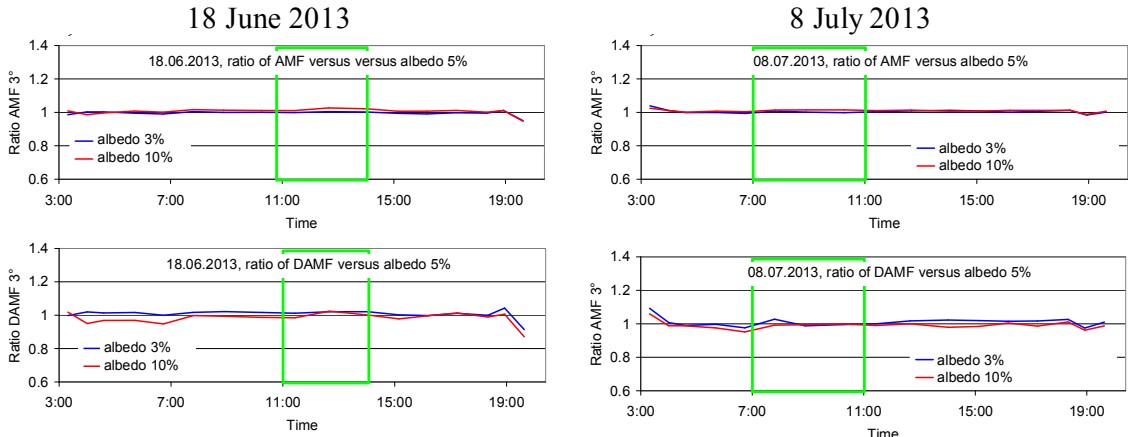

Fig. A12 Ratio of the O$_4$ AMFs (top) and O$_4$ dAMFs (bottom) for different surface albedos versus those for an albedo of 5 % for both selected days.

Table A10 Average ratios of $O_4$ (d)AMFs for different surface albedos versus those for an
albedo of 5 % for the two middle periods on both selected days.

| | AMF ratios | | | dAMF ratios | |
|---|---|---|---|---|---|
| Surface albedo | 18 June 2013, 11:00 – 14:00 | 8 July 2013, 7:00 – 11:00 | | 18 June 2013, 11:00 – 14:00 | 8 July 2013, 7:00 – 11:00 |
| 3 % | 1.00 | 1.00 | | 1.02 | 1.00 |
| 10 % | 1.02 | 1.01 | | 1.00 | 0.99 |


18 June 2013          8 July 2013

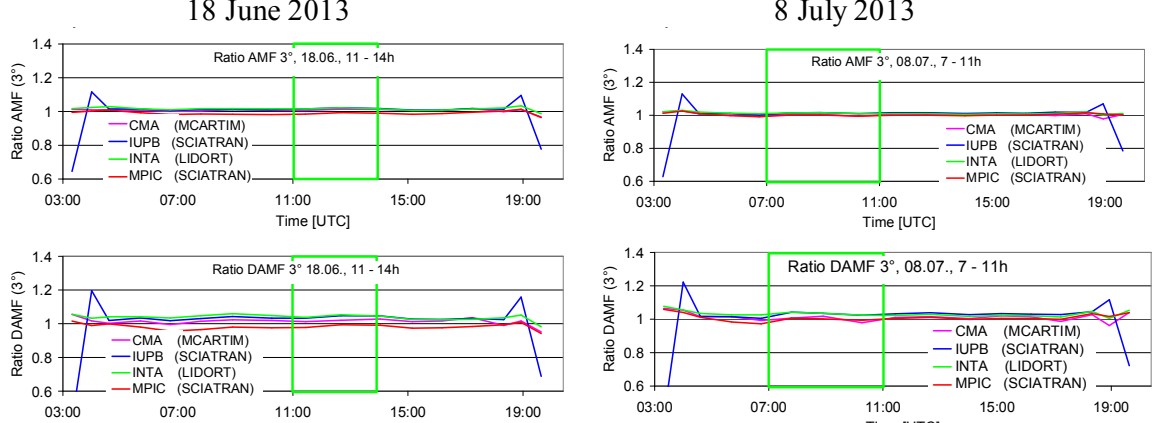

Fig. A13 Ratio of the $O_4$ AMFs (top) and $O_4$ dAMFs (bottom) simulated by different groups
using different radiative transfer models versus those for the MPIC simulations using
MCARTIM for both selected days.

Table A11 Average ratios of $O_4$ (d)AMFs simulated by different groups using different
radiative transfer models versus those for the MPIC simulations using MCARTIM for the two
middle periods on both selected days.

| | AMF ratios | | | dAMF ratios | |
|---|---|---|---|---|---|
| Group (RTM) | 18 June 2013, 11:00 – 14:00 | 8 July 2013, 7:00 – 11:00 | | 18 June 2013, 11:00 – 14:00 | 8 July 2013, 7:00 – 11:00 |
| CMA (MCARTIM) | 1.01 | 1.00 | | 1.02 | 1.00 |
| IUP-Bremen (SCIATRAN) | 1.02 | 1.01 | | 1.04 | 1.03 |
| INTA (LIDORT) | 1.02 | 1.01 | | 1.05 | 1.03 |
| MPIC (SCIATRAN) | 0.99 | 1.00 | | 0.99 | 1.00 |



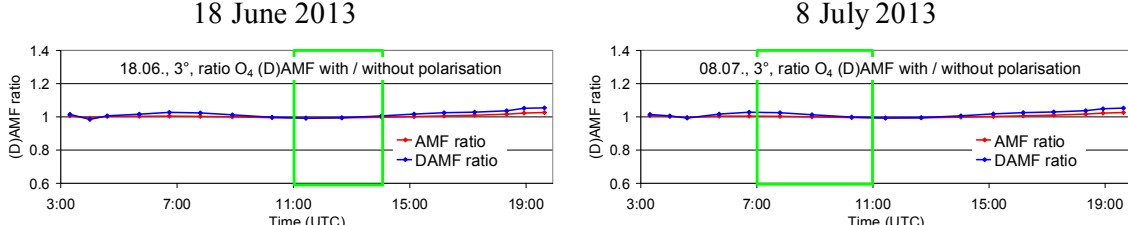

Fig. A14 Ratio of the O$_4$ (d)AMFs considering polarisation versus those without considering
polarisation for both selected days.
Table A12 Average ratios of O$_4$ (d)AMFs considering polarisation versus those without
considering polarisation for the two middle periods on both selected days.

|  | AMF ratios | |  | dAMF ratios | |
|---|---|---|---|---|---|
|  | 18 June 2013, 11:00 – 14:00 | 8 July 2013, 7:00 – 11:00 |  | 18 June 2013, 11:00 – 14:00 | 8 July 2013, 7:00 – 11:00 |
| Considering polarisation | 1.00 | 1.00 |  | 1.00 | 1.01 |

Table A13 Average ratios of O$_4$ (d)AMFs derived from synthetic spectra versus those
obtained from radiative transfer simulations at 360 nm for the two middle periods on both
selected days.

|  | AMF ratios | |  | dAMF ratios | |
|---|---|---|---|---|---|
| Temperature dependence / noise | 18 June 2013, 11:00 – 14:00 | 8 July 2013, 7:00 – 11:00 |  | 18 June 2013, 11:00 – 14:00 | 8 July 2013, 7:00 – 11:00 |
| T dep. considered / no noise | 1.01 | 1.02 |  | 1.01 | 1.00 |
| no T dep. considered / no noise | 1.00 | 1.01 |  | 1.00 | 1.00 |
| no T dep. considered / noise | 0.99 | 1.00 |  | 1.00 | 1.01 |


18 June 2013                                  8 July 2013

a) measured spectra

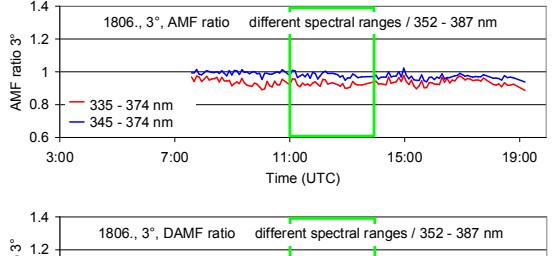
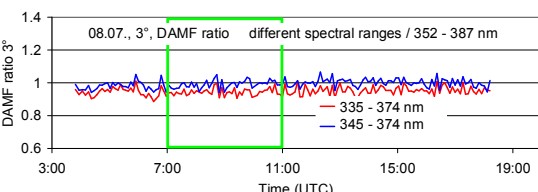

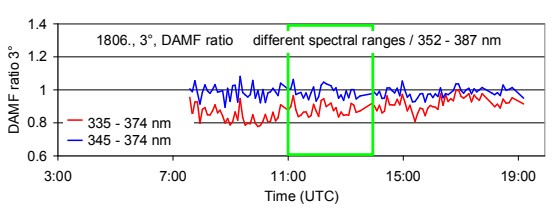
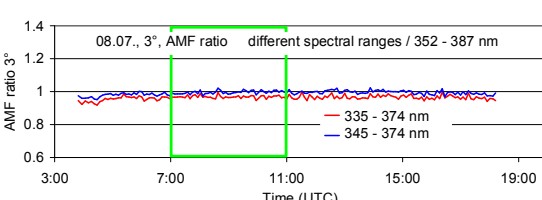

b) synthetic spectra

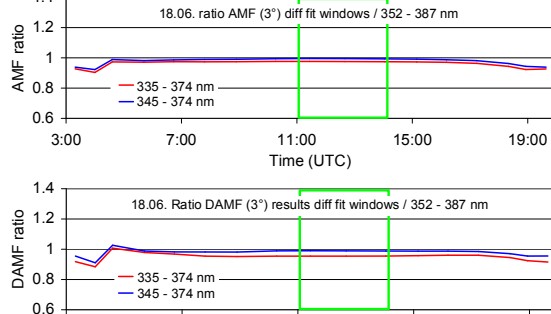
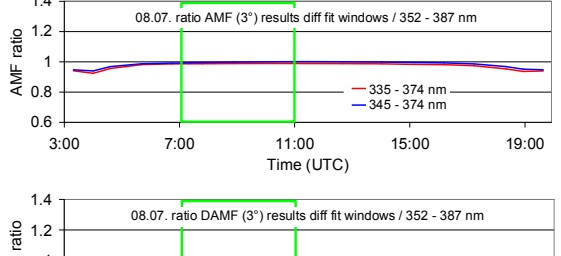

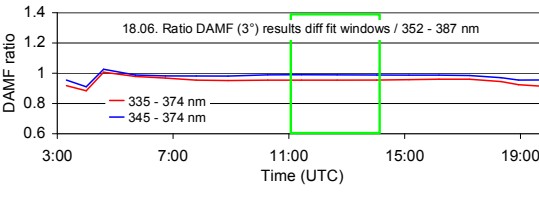
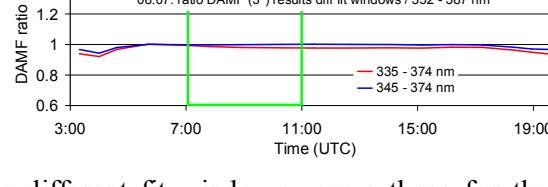

Fig. A15 Ratio of the $O_4$ (d)AMFs derived for different fit windows versus those for the
standard fit window (352 – 387 nm) for both selected days (top: results for spectra measured
by the MPIC instrument; bottom: results for synthetic spectra taking into account the
temperature dependence of the $O_4$ cross section).


Table A14 Average ratios of $O_4$ (d)AMFs derived for different fit windows versus those for
the standard fit window (352 – 387 nm) for the two middle periods on both selected days (top:
results for spectra measured by the MPIC instrument; bottom: results for synthetic spectra
taking into account the temperature dependence of the $O_4$ cross section).

| | **AMF ratios** | | | **dAMF ratios** | |
|---|---|---|---|---|---|
| Spectral range | 18 June 2013, 11:00 – 14:00 | 8 July 2013, 7:00 – 11:00 | | 18 June 2013, 11:00 – 14:00 | 8 July 2013, 7:00 – 11:00 |
| **Measured Spectra** | | | | | |
| 335 – 374 nm | 0.93 | 0.97 | | 0.88 | 0.94 |
| 345 – 374 nm | 0.98 | 1.00 | | 0.99 | 0.99 |
| **Synthetic Spectra** | | | | | |
| 335 – 374 nm | 0.98 | 0.99 | | 0.95 | 0.98 |
| 345 – 374 nm | 0.99 | 1.00 | | 0.99 | 1.00 |


<table>
<tr><td>18 June 2013</td><td>8 July 2013</td></tr>
</table>

a) measured spectra

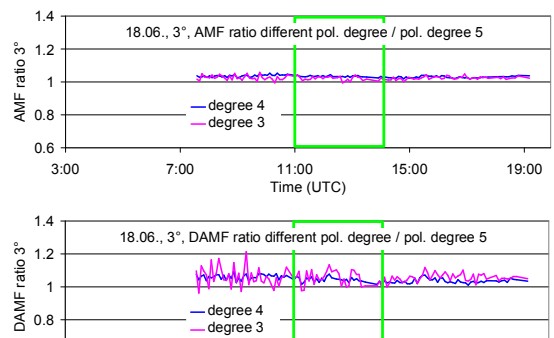
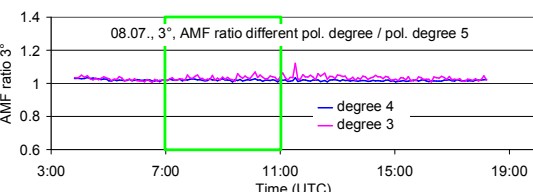
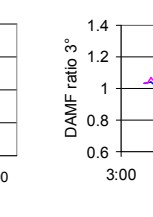

b) synthetic spectra

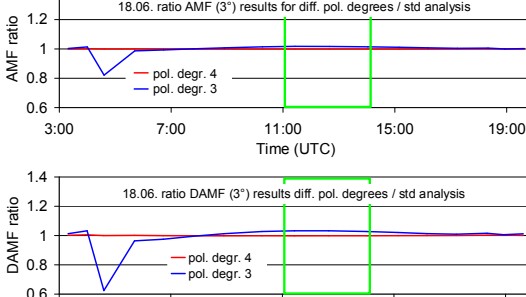
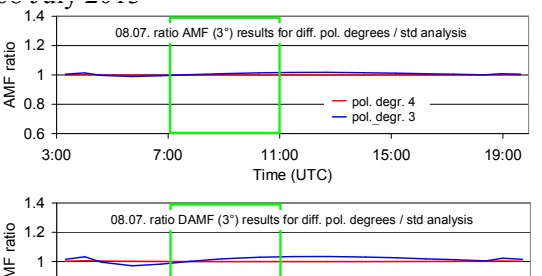
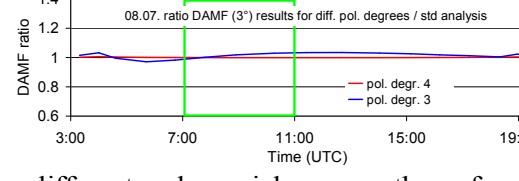

Fig. A16 Ratio of the O$_4$ (d)AMFs derived for different polynomials versus those for the
standard analysis (polynomial degree 5) for both selected days (top: results for spectra
measured by the MPIC instrument; bottom: results for synthetic spectra taking into account
the temperature dependence of the O$_4$ cross section).

Table A15 Average ratios of O$_4$ (d)AMFs derived for different polynomials versus those for
the standard analysis (polynomial degree 5) for the two middle periods on both selected days
(top: results for spectra measured by the MPIC instrument; bottom: results for synthetic
spectra taking into account the temperature dependence of the O$_4$ cross section).

| Degree of polynomial | AMF ratios | | | dAMF ratios | |
|---|---|---|---|---|---|
| | 18 June 2013, 11:00 – 14:00 | 8 July 2013, 7:00 – 11:00 | | 18 June 2013, 11:00 – 14:00 | 8 July 2013, 7:00 – 11:00 |
| **Measured Spectra** | | | | | |
| 4 | 1.04 | 1.02 | | 1.06 | 1.03 |
| 3 | 1.03 | 1.03 | | 1.06 | 1.06 |
| **Synthetic Spectra** | | | | | |
| 4 | 1.00 | 1.00 | | 1.00 | 1.00 |
| 3 | 1.02 | 1.01 | | 1.03 | 1.01 |


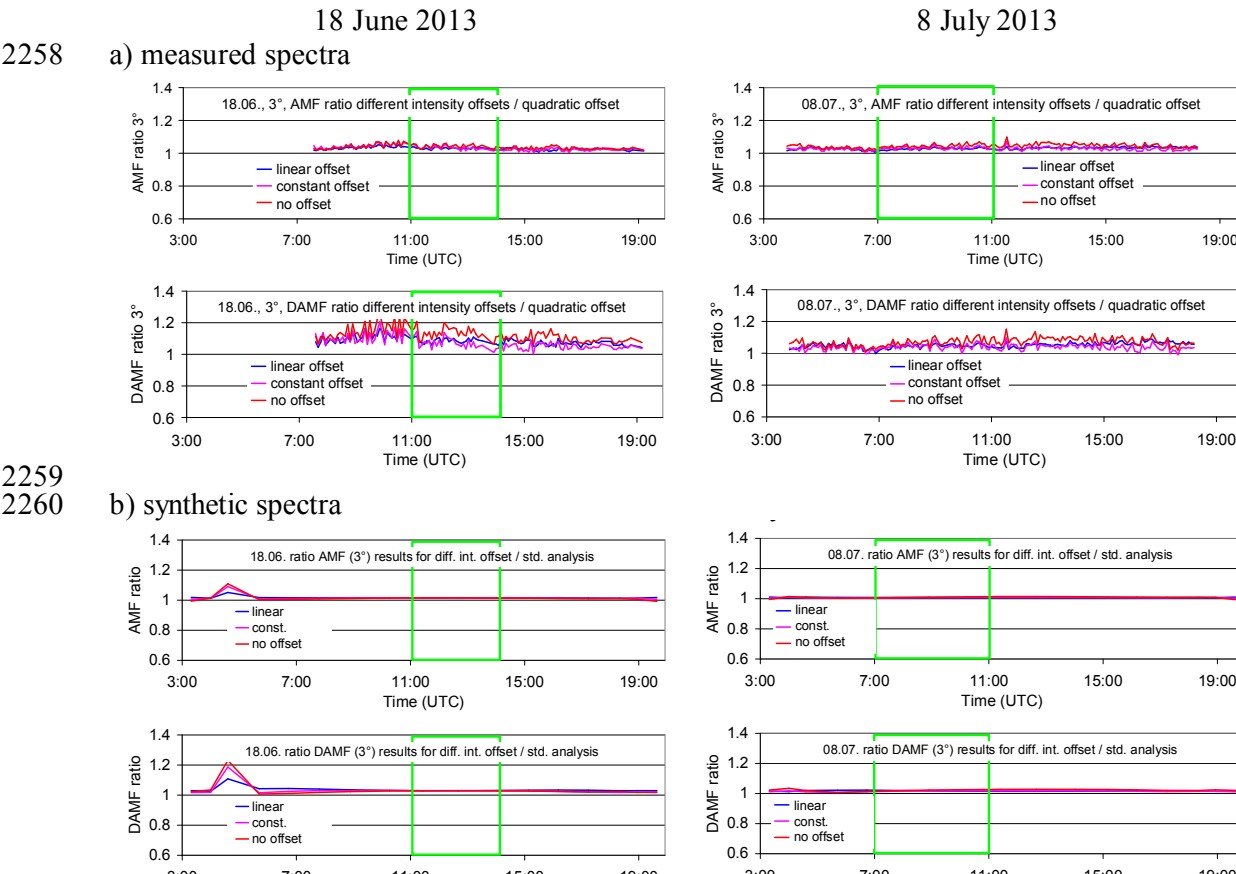

a) measured spectra
b) synthetic spectra
Fig. A17 Ratio of the $O_4$ (d)AMFs derived for different intensity offsets versus those for the
standard analysis (intensity offset of degree 2) for both selected days (top: results for spectra
measured by the MPIC instrument; bottom: results for synthetic spectra taking into account
the temperature dependence of the $O_4$ cross section).

Table A16 Average ratios of $O_4$ (d)AMFs derived for different intensity offsets versus those
for the standard analysis (intensity offset of degree 2) for the two middle periods on both
selected days (top: results for spectra measured by the MPIC instrument; bottom: results for
synthetic spectra taking into account the temperature dependence of the $O_4$ cross section).

| | AMF ratios | | | dAMF ratios | |
|---|---|---|---|---|---|
| Intensity offset | 18 June 2013, 11:00 – 14:00 | 8 July 2013, 7:00 – 11:00 | | 18 June 2013, 11:00 – 14:00 | 8 July 2013, 7:00 – 11:00 |
| **Measured Spectra** | | | | | |
| Linear | 1.04 | 1.03 | | 1.11 | 1.05 |
| Constant | 1.05 | 1.03 | | 1.11 | 1.04 |
| No offset | 1.05 | 1.05 | | 1.16 | 1.07 |
| **Synthetic Spectra** | | | | | |
| Linear | 1.01 | 1.01 | | 1.03 | 1.02 |
| Constant | 1.02 | 1.01 | | 1.03 | 1.02 |
| No offset | 1.02 | 1.01 | | 1.03 | 1.02 |


18 June 2013              8 July 2013

a) measured spectra

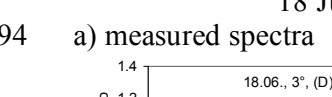
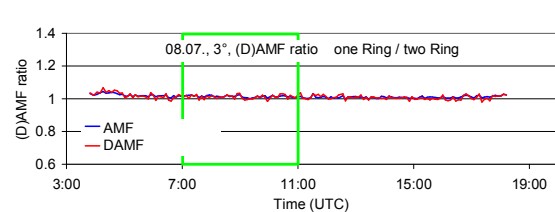

b) synthetic spectra

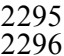
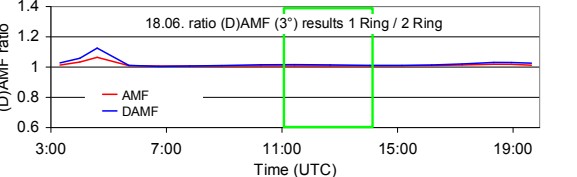
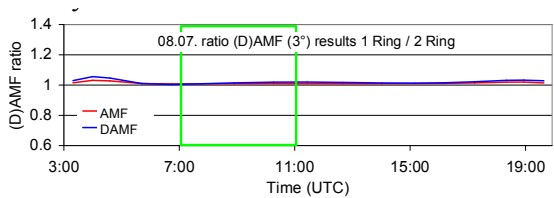

Fig. A18 Ratio of the $O_4$ (d)AMFs derived for the analysis with only one Ring spectrum
versus those for the standard analysis (using two Ring spectra) for both selected days (top:
results for spectra measured by the MPIC instrument; bottom: results for synthetic spectra
taking into account the temperature dependence of the $O_4$ cross section).

Table A17 Average ratios of $O_4$ (d)AMFs derived for the analysis with only one Ring
spectrum versus those for the standard analysis (using two Ring spectra) for the two middle
periods on both selected days (top: results for spectra measured by the MPIC instrument;
bottom: results for synthetic spectra taking into account the temperature dependence of the $O_4$
cross section).

| | AMF ratios | | | dAMF ratios | |
|---|---|---|---|---|---|
| Ring correction | 18 June 2013, 11:00 – 14:00 | 8 July 2013, 7:00 – 11:00 | | 18 June 2013, 11:00 – 14:00 | 8 July 2013, 7:00 – 11:00 |
| **Measured Spectra** | | | | | |
| Only one Ring spectrum | 1.02 | 0.99 | | 1.01 | 0.99 |
| **Synthetic Spectra** | | | | | |
| Only one Ring spectrum | 1.01 | 1.01 | | 1.01 | 1.01 |


18 June 2013                      8 July 2013

a) measured spectra

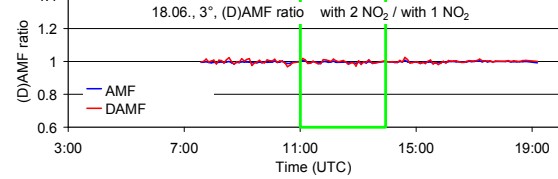 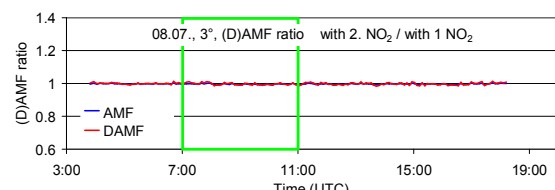

b) synthetic spectra

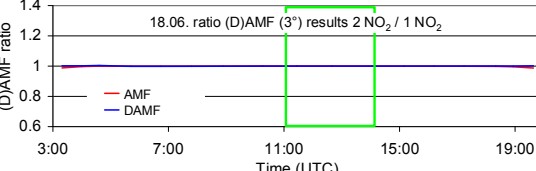 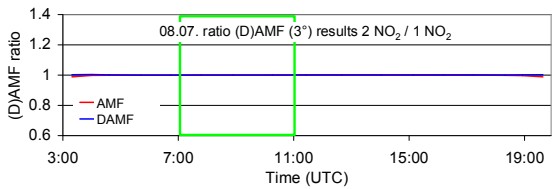

Fig. A19 Ratio of the $O_4$ (d)AMFs derived for the analysis with a second $NO_2$ cross section
(for 220 K) versus those for the standard analysis (only $NO_2$ cross section for 294 K) for both
selected days (top: results for spectra measured by the MPIC instrument; bottom: results for
synthetic spectra taking into account the temperature dependence of the $O_4$ cross section).



Table A18 Average ratios of O$_4$ (d)AMFs derived for the analysis with a second NO$_2$ cross
section (for 220 K) versus those for the standard analysis (only NO$_2$ cross section for 294 K)
for the two middle periods on both selected days (top: results for spectra measured by the
MPIC instrument; bottom: results for synthetic spectra taking into account the temperature
dependence of the O$_4$ cross section).

| NO$_2$ cross sections | AMF ratios | | | dAMF ratios | |
|---|---|---|---|---|---|
| | 18 June 2013, 11:00 – 14:00 | 8 July 2013, 7:00 – 11:00 | | 18 June 2013, 11:00 – 14:00 | 8 July 2013, 7:00 – 11:00 |
| **Measured Spectra** | | | | | |
| 294 & 220 K | 1.00 | 1.00 | | 1.00 | 1.00 |
| **Synthetic Spectra** | | | | | |
| 294 & 220 K | 1.00 | 1.00 | | 1.00 | 1.00 |





18 June 2013          8 July 2013

a) measured spectra

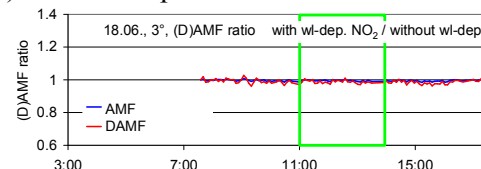 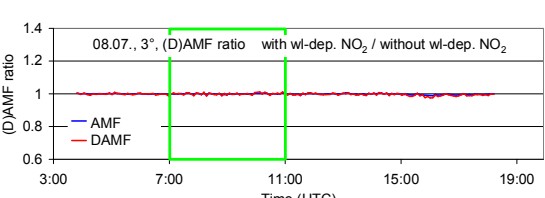


b) synthetic spectra

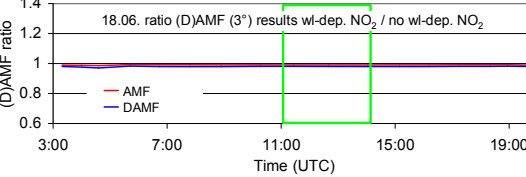 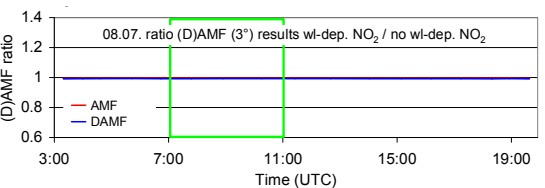


Fig. A20 Ratio of the O$_4$ (d)AMFs derived for the analysis with a second NO$_2$ cross section
(cross section times wavelength) versus those for the standard analysis (only one NO$_2$ cross
section) for both selected days (top: results for spectra measured by the MPIC instrument;
bottom: results for synthetic spectra taking into account the temperature dependence of the O$_4$
cross section).











Table A19 Average ratios of $O_4$ (d)AMFs derived for the analysis with a second $NO_2$ cross
section (cross section times wavelength) versus those for the standard analysis (only one $NO_2$
cross section) for the two middle periods on both selected days (top: results for spectra
measured by the MPIC instrument; bottom: results for synthetic spectra taking into account
the temperature dependence of the $O_4$ cross section).

| | **AMF ratios** | | | **dAMF ratios** | |
|---|---|---|---|---|---|
| $NO_2$ wavelength dependence | 18 June 2013, 11:00 – 14:00 | 8 July 2013, 7:00 – 11:00 | | 18 June 2013, 11:00 – 14:00 | 8 July 2013, 7:00 – 11:00 |
| **Measured Spectra** | | | | | |
| additional cross for wavelength dependence | 1.00 | 1.00 | | 0.99 | 1.00 |
| **Synthetic Spectra** | | | | | |
| additional cross for wavelength dependence | 0.99 | 1.00 | | 0.98 | 0.99 |


18 June 2013                                        8 July 2013

a) measured spectra

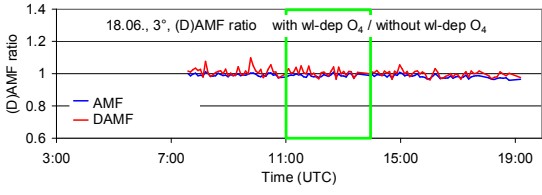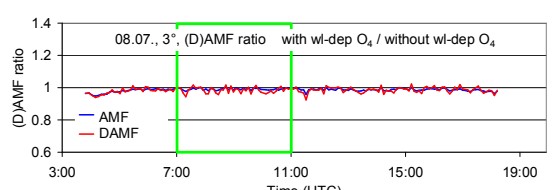

b) synthetic spectra

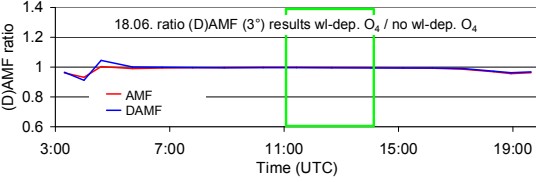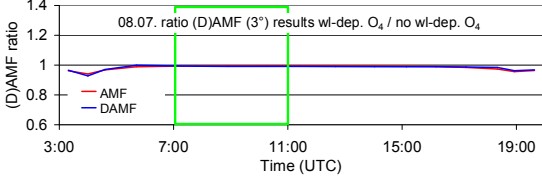

Fig. A21 Ratio of the $O_4$ (d)AMFs derived for the analysis with a second $O_4$ cross section
(accounting for the wavelength dependence) versus those for the standard analysis (only one
$O_4$ cross section) for both selected days (top: results for spectra measured by the MPIC
instrument; bottom: results for synthetic spectra taking into account the temperature
dependence of the $O_4$ cross section).

Table A20 Average ratios of $O_4$ (d)AMFs derived for the analysis with a second $O_4$ cross
section (accounting for the wavelength dependence) versus those for the standard analysis
(only one $O_4$ cross section) for the two middle periods on both selected days (top: results for
spectra measured by the MPIC instrument; bottom: results for synthetic spectra taking into
account the temperature dependence of the $O_4$ cross section).

| $O_4$ wavelength dependence | **AMF ratios** | | | **dAMF ratios** | |
|---|---|---|---|---|---|
| | 18 June 2013, 11:00 – 14:00 | 8 July 2013, 7:00 – 11:00 | | 18 June 2013, 11:00 – 14:00 | 8 July 2013, 7:00 – 11:00 |
| **Measured Spectra** | | | | | |
| additional cross for wavelength dependence | 0.99 | 0.99 | | 1.01 | 0.99 |
| **Synthetic Spectra** | | | | | |
| additional cross for wavelength dependence | 1.00 | 0.99 | | 1.00 | 0.99 |


18 June 2013                                          8 July 2013
a) measured spectra

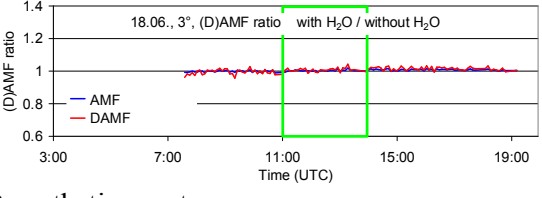 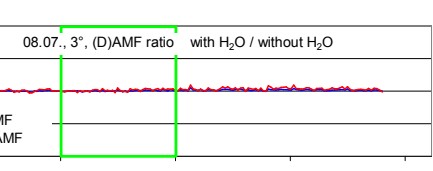

b) synthetic spectra

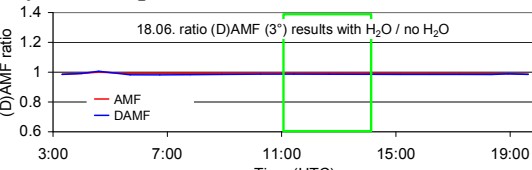 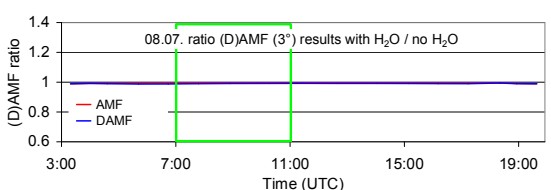

Fig. A22 Ratio of the $O_4$ (d)AMFs derived for the analysis including a $H_2O$ cross section
versus those for the standard analysis (no $H_2O$ cross section) for both selected days (top:
results for spectra measured by the MPIC instrument; bottom: results for synthetic spectra
taking into account the temperature dependence of the $O_4$ cross section).

Table A21 Average ratios of $O_4$ (d)AMFs derived for the analysis including a $H_2O$ cross
section versus those for the standard analysis (no $H_2O$ cross section) for the standard analysis
(only one $O_4$ cross section) for the two middle periods on both selected days (top: results for
spectra measured by the MPIC instrument; bottom: results for synthetic spectra taking into
account the temperature dependence of the $O_4$ cross section).

| | AMF ratios | | | dAMF ratios | |
|---|---|---|---|---|---|
| $H_2O$ cross section | 18 June 2013, 11:00 – 14:00 | 8 July 2013, 7:00 – 11:00 | | 18 June 2013, 11:00 – 14:00 | 8 July 2013, 7:00 – 11:00 |
| **Measured spectra** | | | | | |
| $H_2O$ cross section included | 1.00 | 1.00 | | 1.01 | 1.01 |
| **Synthetic Spectra** | | | | | |
| $H_2O$ cross section included | 0.99 | 1.00 | | 0.99 | 0.99 |


18 June 2013       8 July 2013

a) measured spectra

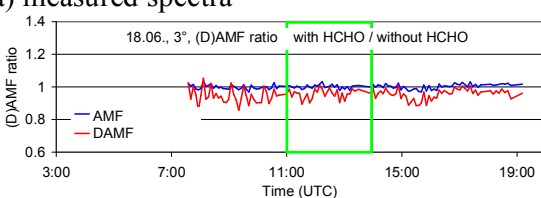 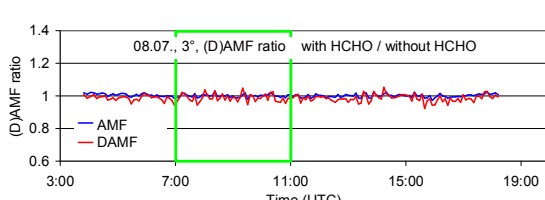

b) synthetic spectra

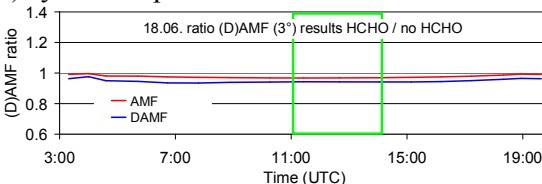 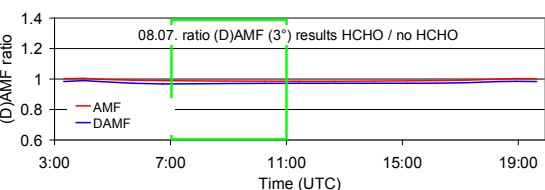

Fig. A23 Ratio of the $O_4$ (d)AMFs derived for the analysis including a HCHO cross section
versus those for the standard analysis (no HCHO cross section) for both selected days (top:
results for spectra measured by the MPIC instrument; bottom: results for synthetic spectra
taking into account the temperature dependence of the $O_4$ cross section).

Table A22 Average ratios of $O_4$ (d)AMFs derived for the analysis including a HCHO cross
section versus those for the standard analysis (no HCHO cross section) for the standard
analysis (only one $O_4$ cross section) for the two middle periods on both selected days (top:
results for spectra measured by the MPIC instrument; bottom: results for synthetic spectra
taking into account the temperature dependence of the $O_4$ cross section).

| | AMF ratios | | | dAMF ratios | |
|---|---|---|---|---|---|
| HCHO cross section | 18 June 2013, 11:00 – 14:00 | 8 July 2013, 7:00 – 11:00 | | 18 June 2013, 11:00 – 14:00 | 8 July 2013, 7:00 – 11:00 |
| **Measured Spectra** | | | | | |
| HCHO cross section included | 1.00 | 1.00 | | 0.96 | 0.98 |
| **Synthetic Spectra** | | | | | |
| HCHO cross section included | 0.97 | 0.99 | | 0.94 | 0.97 |



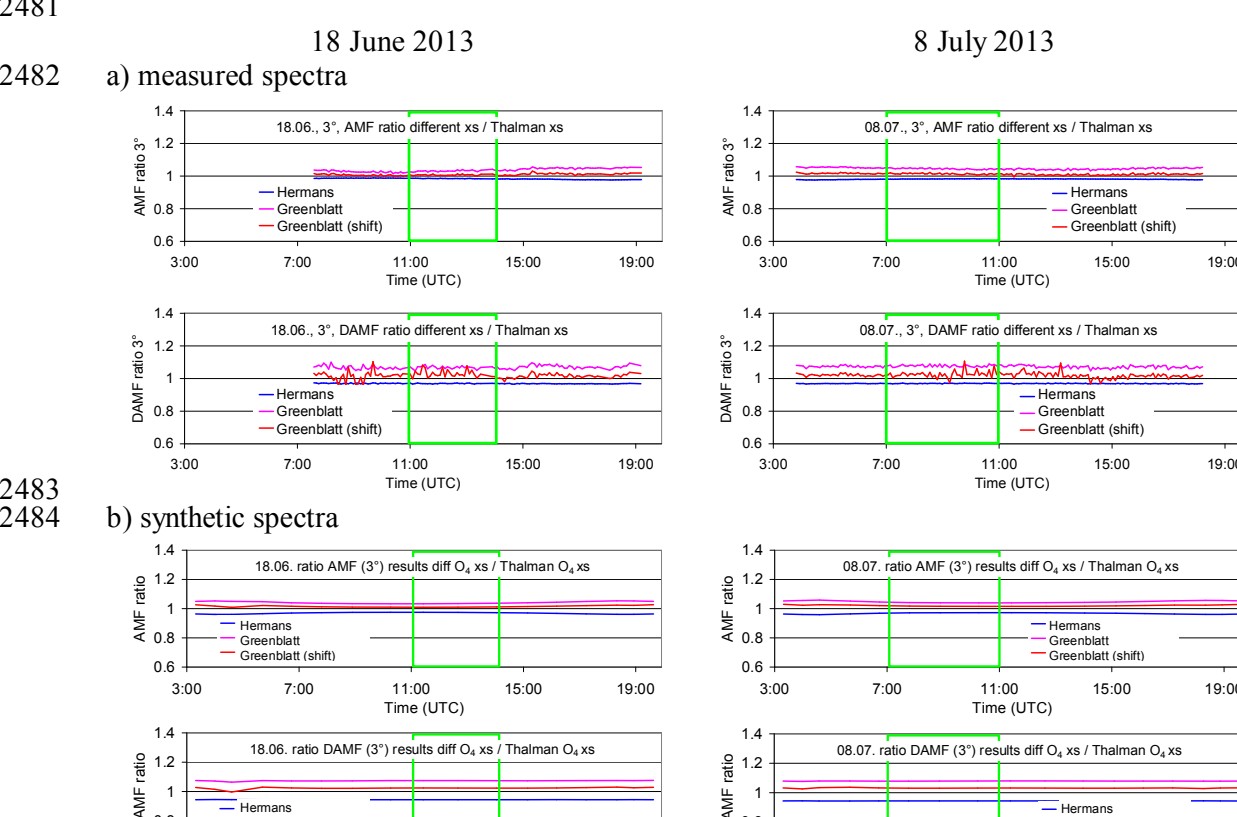

a) measured spectra
b) synthetic spectra
Fig. A24 Ratio of the $O_4$ (d)AMFs derived for the analyses using different $O_4$ cross sections
versus those for the standard analysis (using the Thalman and Volkamer (2013) cross section)
for both selected days (top: results for spectra measured by the MPIC instrument; bottom:
results for synthetic spectra taking into account the temperature dependence of the $O_4$ cross
section).

Table A23 Average ratios of $O_4$ (d)AMFs derived for the analyses using different $O_4$ cross
section versus those for the standard analysis (using the Thalman and Volkamer cross section)
for the standard analysis (only one $O_4$ cross section) for the two middle periods on both
selected days (top: results for spectra measured by the MPIC instrument; bottom: results for
synthetic spectra taking into account the temperature dependence of the $O_4$ cross section).

| | AMF ratios | | | dAMF ratios | |
|---|---|---|---|---|---|
| $O_4$ cross section | 18 June 2013, 11:00 – 14:00 | 8 July 2013, 7:00 – 11:00 | | 18 June 2013, 11:00 – 14:00 | 8 July 2013, 7:00 – 11:00 |
| **Measured spectra** | | | | | |
| Hermans | 0.98 | 0.98 | | 0.97 | 0.97 |
| Greenblatt | 1.03 | 1.04 | | 1.07 | 1.08 |
| Greenblatt shifted | 1.01 | 1.01 | | 1.03 | 1.03 |
| **Synthetic Spectra** | | | | | |
| Hermans | 0.97 | 0.97 | | 0.94 | 0.94 |
| Greenblatt | 1.03 | 1.04 | | 1.07 | 1.08 |
| Greenblatt shifted | 1.01 | 1.02 | | 1.02 | 1.03 |


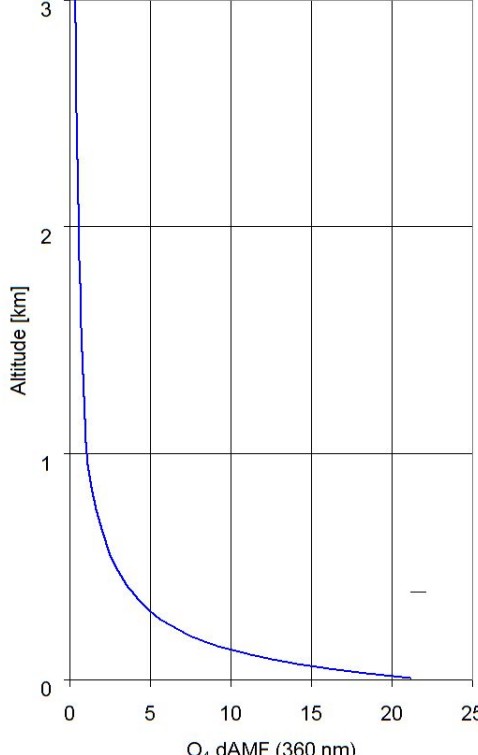

Fig. 25 $O_4$ differential box-AMFs (with 20m vertical resolution) used for the simulation of the temperature-dependent $O_4$ absorption spectra. They are averages of radiative transfer simulations for several scenarios. Simulations are performed for a surface albedo of 6 %, aerosol profiles with constant extinction between 0 and 1000m and different AOD (0.1, 0.3, 0.7) and for all combinations of SZA (40, 60°), relative azimuth angles (0, 90, 180°) and elevation angles (2° and 3°).


18 June 2013                8 July 2013

a) measured spectra

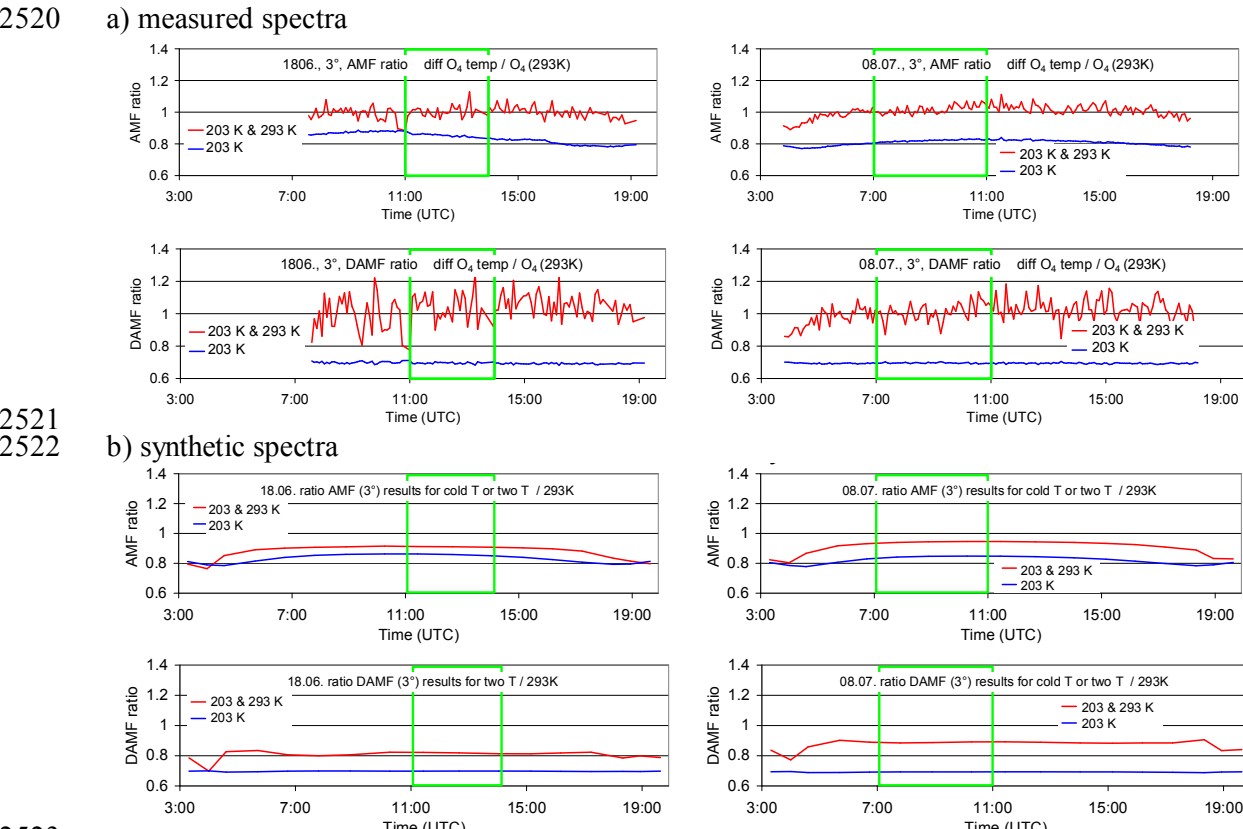

b) synthetic spectra
Fig. A26 Ratio of the $O_4$ (d)AMFs derived for $O_4$ cross sections at different temperatures
(either 203 K or both 203 and 293 K) versus those for the standard analysis (using the $O_4$
cross section for 293 K) for both selected days (top: results for spectra measured by the MPIC
instrument; bottom: results for synthetic spectra taking into account the temperature
dependence of the $O_4$ cross section).

Table A24 Average ratios of $O_4$ (d)AMFs derived $O_4$ cross sections at different temperatures (either 203 K or both 203 and 293 K) versus those for the standard analysis (using the $O_4$ cross section for 293 K) for the two middle periods on both selected days (top: results for spectra measured by the MPIC instrument; bottom: results for synthetic spectra taking into account the temperature dependence of the $O_4$ cross section). For the simultaneous fit of both temperatures also the results for the spectral range 345 – 374 nm (one $O_4$ absorption band) are included.

| $O_4$ cross sections | AMF ratios | | | dAMF ratios | |
|---|---|---|---|---|---|
| | 18 June 2013, 11:00 – 14:00 | 8 July 2013, 7:00 – 11:00 | | 18 June 2013, 11:00 – 14:00 | 8 July 2013, 7:00 – 11:00 |
| **Measured Spectra** | | | | | |
| 203 K | 0.85 | 0.82 | | 0.70 | 0.70 |
| 203 & 293 K | 1.00 | 1.02 | | 1.04 | 1.01 |
| 203 & 293 K (345 – 374 nm) | 0.91 | 1.04 | | 0.95 | 1.02 |
| **Synthetic Spectra** | | | | | |
| 203 K | 0.86 | 0.84 | | 0.70 | 0.69 |
| 203 & 293 K | 0.91 | 0.94 | | 0.82 | 0.89 |
| 203 & 293 K (345 – 374 nm) | 0.99 | 1.00 | | 0.99 | 1.00 |



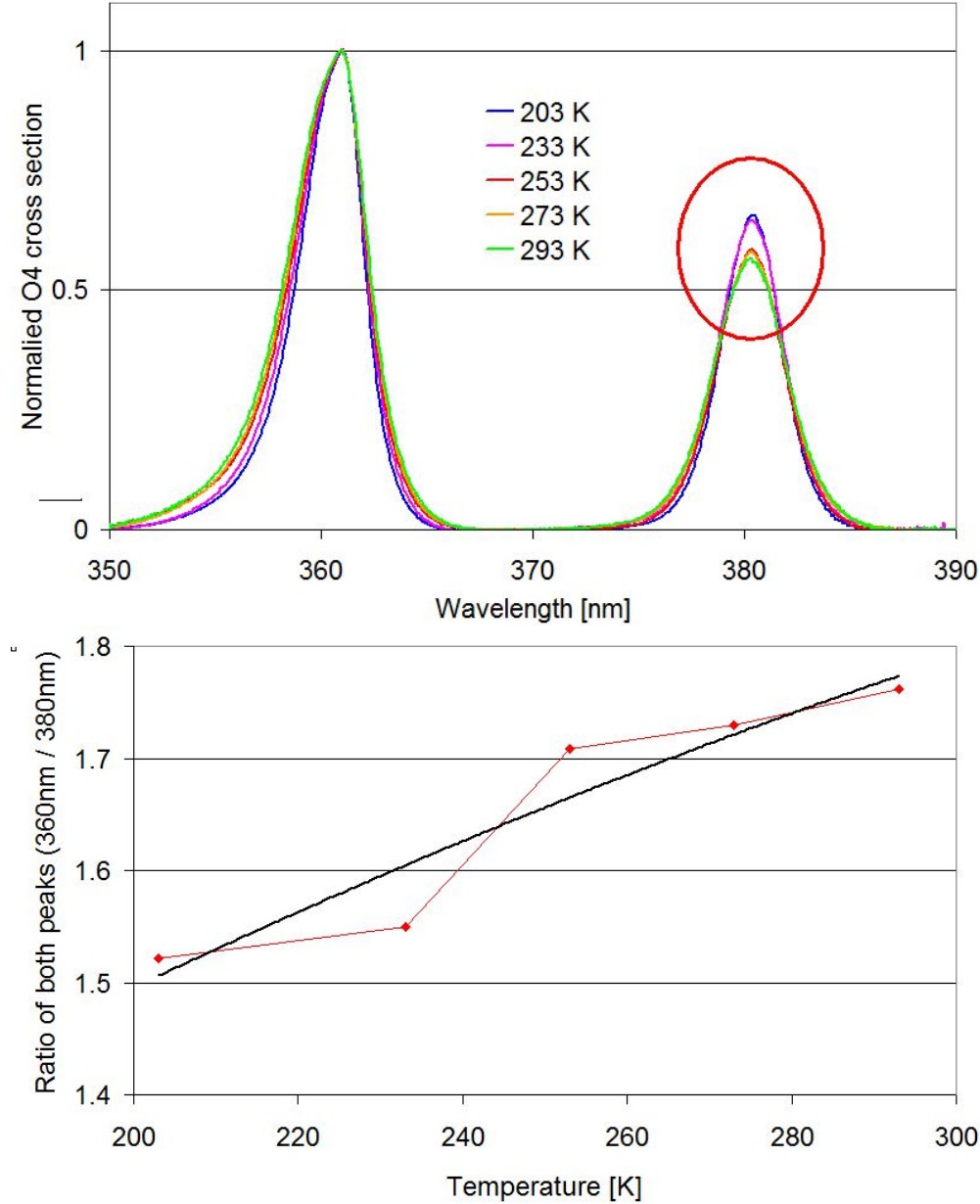

Fig. A27 Top: Comparison of the $O_4$ cross sections from Thalman and Volkamer (2013) for different temperatures. The cross sections are divided by the maximum values at 360 nm. After this normalisation, the resulting values at 380 nm fall into two groups (high values for 203 & 233K, low values for 253, 273, 293K). Bottom: Ratio of the peaks of the $O_4$ cross section at 360 nm and 380 nm as function of temperature (red points). The black curve is a fitted low order polynomial.



a) measured spectra

18 June 2013                                          8 July 2013

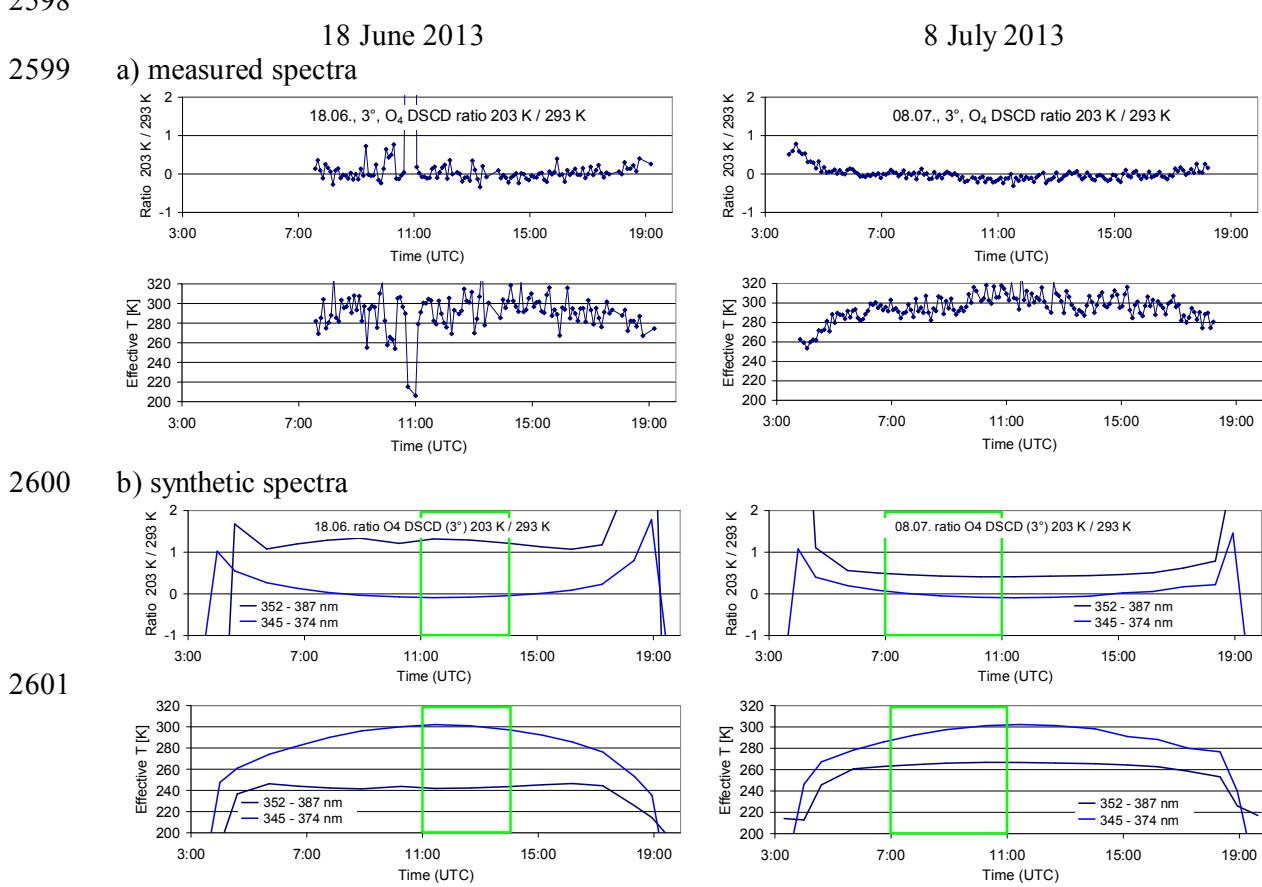

b) synthetic spectra


Fig. A28 Ratio of the derived O$_4$ dSCDs for 203 K and 293 K as well as the derived effective
temperatures for the analyses with both cross sections included.


Table A25 a) Average ratios of $O_4$ (d)AMFs derived from the analysis of MPIC spectra by
different groups versus the analysis of MPIC spectra by MPIC (standard analysis). b) Average
ratios of $O_4$ (d)AMFs derived from spectra of other groups analysed by MPIC versus the
analysis of MPIC spectra by MPIC (using the same analysis settings and spectral range: 335 –
374 nm). c) Average ratios of $O_4$ (d)AMFs derived from spectra of other groups analysed by
the same groups using individual analysis settings versus the analysis of MPIC spectra by
MPIC (standard analysis).

| | **AMF ratios** | | | **dAMF ratios** | |
|---|---|---|---|---|---|
| Measurements / Analysis | 18 June 2013, 11:00 – 14:00 | 8 July 2013, 7:00 – 11:00 | | 18 June 2013, 11:00 – 14:00 | 8 July 2013, 7:00 – 11:00 |
| **a) MPIC spectra analysed by other groups** | | | | | |
| BIRA | 0.96 | 0.98 | | 0.95 | 0.95 |
| IUP-B | 1.03 | 0.98 | | 1.05 | 0.99 |
| INTA | 1.02 | 0.97 | | 1.05 | 0.94 |
| CMA | 0.97 | 0.98 | | 0.98 | 0.95 |
| CSIC | 0.94 | 0.94 | | 0.95 | 0.94 |
| **b) Other spectra analysed by MPIC (335 – 374 nm)** | | | | | |
| BIRA | 0.98 | 0.99 | | 0.89 | 0.95 |
| IUP-B | 1.05 | | | 1.07 | |
| IUP-HD | 0.97 | | | 1.00 | |
| **c) Other spectra analysed by the same groups** | | | | | |
| BIRA | 0.94 | 0.94 | | 0.91 | 0.92 |
| IUP-B | 0.95 | | | 0.88 | |
| IUP-HD | 1.01 | | | 1.04 | |


**Appendix A5 Extraction of aerosol extinction profiles**

In this section, the procedure for the extraction of aerosol extinction profiles is described. The aerosol profiles are derived from the ceilometer measurements (yielding the profile information) in combination with the sun photometer measurements (yielding the vertically integrated aerosol extinction, the aerosol optical depth AOD).

The ceilometer raw data consist of range-corrected backscatter profiles averaged over 15 minutes. The profiles range from the surface to an altitude of 15360m with a height resolution of 15m. Here it is important to note that due to limited overlap of the outgoing Laser beam and the field of view of the telescope, no profile data is available below 180 m. The ceilometer profiles (hourly averages) are shown in Fig. A29 for both selected days.

The AERONET sun photometer data provide the AOD at different wavelengths (340, 360, 440, 500, 675, 870, and 1020 nm) in time intervals of 2 – 25 min if the direct sun is visible.

To determine profiles of aerosol extinction from the ceilometer backscatter data, several processing steps have to be performed. They are described in the sub-sections below. Note that in this section the individual steps are described according to the MPIC procedure. The extracted profiles from other groups differ slightly compared to the results of the MPIC procedure, especially with respect to the altitude above which the extinction was set to zero (see Fig. 9).

**A) Smoothing and extrapolating of the ceilometer backscatter profiles**

First, the ceilometer data are averaged over several hours to reduce the scatter. For that purpose on both days three time periods are identified, for which the backscatter profile show relatively small variations. The profiles for these periods are shown in Fig. A29. In addition to the temporal averaging, the profiles are also vertically smoothed above 2 km. Above altitudes between 5 to 6 km (depending on the period) the (smoothed) ceilometer backscatter profiles become zero. Thus the aerosol extinction profiles above these altitudes are set to zero. Below 180 m above the surface the ceilometer becomes 'blind' for the aerosol extinction because of the insufficient overlap between the outgoing laser beam and the field of view of the telescope. Thus the profiles have to be extrapolated down to the surface. This extrapolation constitutes an important source of uncertainty. To estimate the associated uncertainties, the extrapolation is performed in three different ways:

1) The value below 180 m are set to the value measured at 180m.

2) The values below 180m are linearly extrapolated assuming the same slope below 180 m as between 180m and 240m.

3) The values below 180m are linearly extrapolated by twice the slope between 180m and 240m.


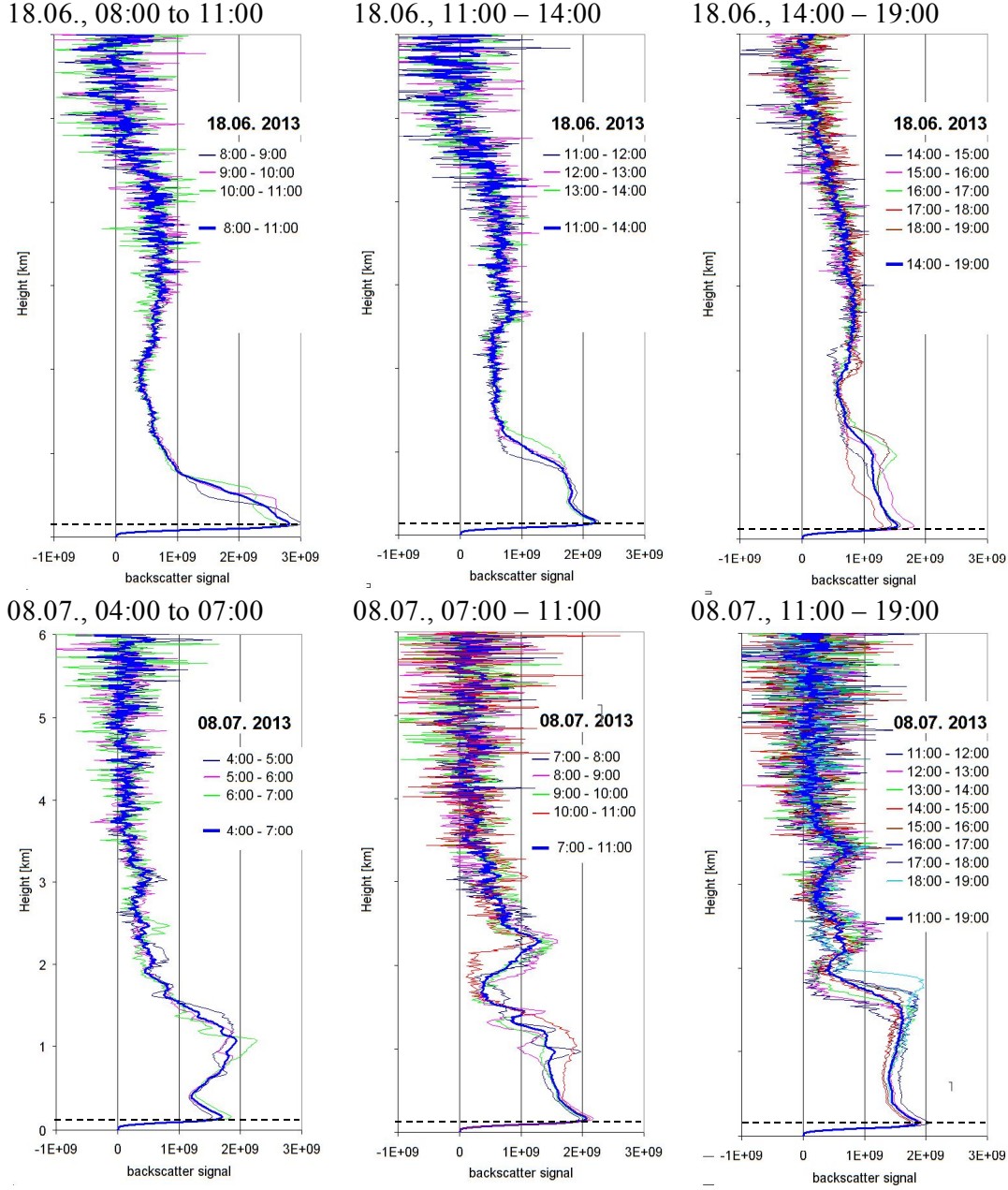

Fig. A29 Range-corrected backscatter profiles (hourly averages) for the three selected periods
on both days. Also the averages over the whole periods are shown (thick lines). Note that the
backscatter signal below 180 m (below the dashed horizontal line) is invalid due to the limited
overlap of the ceilometer instrument.

       **B) Scaling of the Ceilometer profiles by sun photometer AOD at 1020 nm**


The scaling of the ceilometer backscatter profiles by the AOD at 1020 nm is an intermediate
step, which is necessary for the correction of the aerosol self-extinction. The average AOD at
1020 nm for the different selected time periods on both days is shown in Table A26. In that
table also the average values at 380 nm are shown, which are used for a second scaling (see
below).
The backscatter profiles are vertically integrated and then the whole profiles are scaled by the
ratio:

2731       $AOD_{1020nm} / B_{int}$                                                                  (A1)


Here $B_{int}$ indicates the integrated backscatter profile.

Note that the wavelength of the ceilometer measurements (1064 nm) is slightly different from
the sun photometer measurements (1020 nm), but the difference of the AOD is negligible
(typically < 4%).

Table A26 Average AOD at 1020 and 360 nm derived from the sun photometer.

| Time | AOD 1020 nm | AOD 360 nm* |
|---|---|---|
| 18.06.2013, 08:00 - 11:00 | 0.124 | 0.379 |
| 18.06.2013, 11:00 - 14:00 | 0.122 | 0.367 |
| 18.06.2013, 14:00 - 19:00 | 0.118 | 0.296 |
|  |  |  |
| 08.07.2013, 04:00 - 07:00 | 0.045 | 0.295 |
| 08.07.2013, 07:00 - 14:00 | 0.053 | 0.333 |
| 08.07.2013, 11:00 - 19:00 | 0.055 | 0.348 |

*Average of AOD at 340 nm and 380 nm.

2743       **C) Correction of the aerosol extinction**


The photons received by the ceilometer have undergone atmospheric extinction. Here,
Rayleigh scattering can be ignored because of the long wavelength of the ceilometer (optical
depth below 2 km is < 0.001). However, while the extinction due to aerosol scattering is also
small at these long wavelengths it systematically affects the ceilometer signal and has to be
corrected. The extinction correction is performed according to the following formula:

$$\alpha_{i,corr} = \frac{\alpha_i}{\exp\left(-2 \cdot \sum_{j=0}^{i-1} \alpha_{j,corr} \cdot \left(z_j - z_{j-1}\right)\right)}$$     (A2)

Here $\alpha_i$ represent the uncorrected extinction and $\alpha_{i,corr}$ represents the corrected extinction at
height layer i (with $z_i$ is the lower boundary of that height layer). Equation C1 has to be
subsequently applied to all height layers starting from the surface ($z_0$). Note that the factor of
two accounts for the extinction along both paths between the instrument and the scattering
altitude (upward and downward). The extinction correction is performed at a vertical
resolution of 15m.
After the extinction correction, the profiles are scaled by the corresponding AOD at 360 nm
(see table A26). In Fig. A30 the profiles with and without extinction correction are shown.
The extinction correction slightly increases the values at higher altitudes and decreases the
values close to the surface. The effect of the extinction correction is larger on 18 June 2013
(up to 12 %).

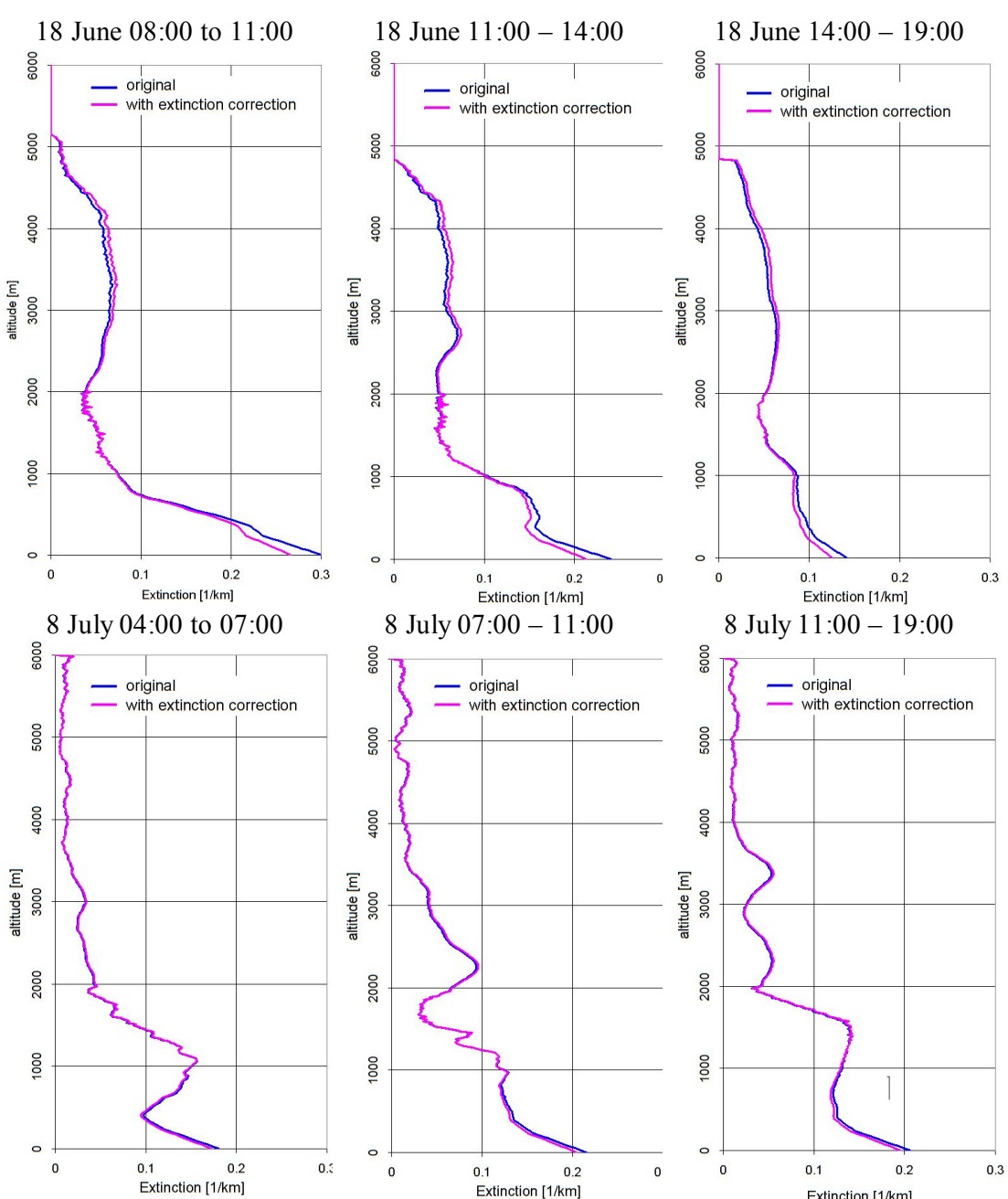

Fig. A30 Comparison of profiles (linear extrapolation below 180 m) without (blue) and with
(magenta) extinction correction. Both profiles are scaled to the same total AOD (at 360 nm)
determined from the sun photometer.


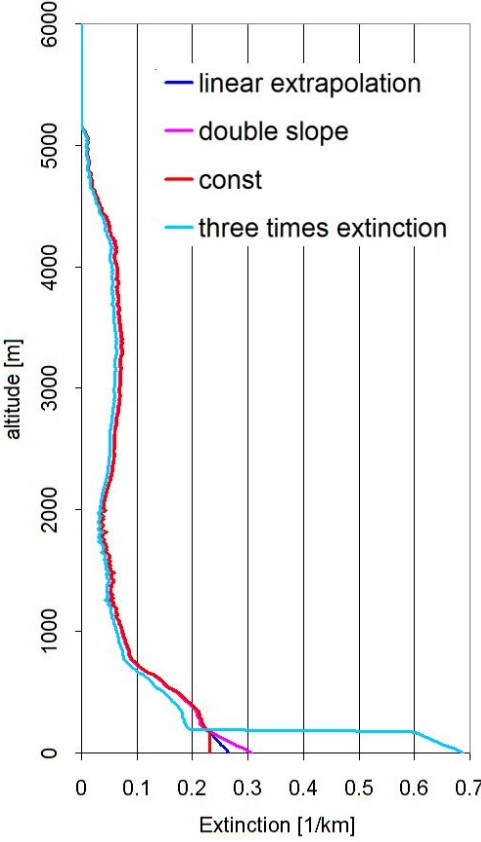

Fig. A31 Aerosol profile (light blue) with extreme extinction close to the surface (below 180
m, the altitude for which the ceilometer is sensitive) extracted for the first period (8:00 –
11:00) on 18 June 2013. Also shown are the profiles extrapolated below 180 as described
above.
**D) Influence of a changing LIDAR ratio with altitude**
For the extraction of the aerosol profiles described above, a fixed LIDAR ratio was assumed,
which implies that the aerosol properties are independent from altitude. However, this is a
rather strong assumption, because it can be expected that the aerosol properties (e.g. the size)
change with altitude. With the available limited information, it is impossible to derive detailed
information about the altitude dependence of the aerosol properties, but it can be quantified
how representative the ceilometer measurements at 1064 nm are for the aerosol extinction
profiles at 360 nm. For these investigations we again focus on the middle periods of both
selected days. From the AERONET Almucantar observations information on the size
distribution for these periods is available (see Fig. A32). On both days two pronounced modes
(fine and coarse mode) are found with a much larger coarse mode fraction on 18 June
compared to 8 July (on 18 June also the coarse mode is broader and shows two distinct
maxima). From the AERONET observations, also separate phase functions for the fine and
coarse mode as well as the relative contributions of both modes to the total aerosol optical
depth at 500 nm are available. On 18 June and 8 July the relative contributions of the coarse
mode fraction to the total AOD at 500 nm are about 39 % and 5 %, respectively (see table
A27). Assuming that the AOD of the coarse mode fraction is independent of wavelength, the
relative contributions of the coarse mode at 360 nm and 1064 nm can be derived (see Table
A27).

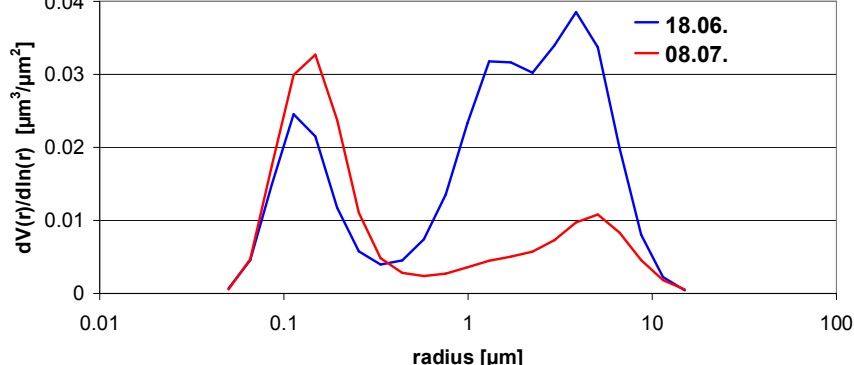

Fig. A32 Size distributions derived from AERONET Almucantar observations on 18 June
(07:24 & 15:34) and 08 July (07:32 & 15:38).
Table A27 Contributions of the coarse mode to the total AOD at different wavelengths
derived from AERONET observations. The relative contributions are calculated assuming
that the AOD of the coarse mode at 500 nm (0.093 and 0.010 on 18 June and 8 July,
respectively) does not depend on wavelength.

| Date | Total AOD 360 nm | Total AOD 500 nm | Total AOD 1064 nm* | Relative contribution of coarse mode 360 nm | Relative contribution of coarse mode 500 nm | Relative contribution of coarse mode 1064 nm |
|---|---|---|---|---|---|---|
| 18 June, 11:00 – 14:00 | 0.37 | 0.242 | 0.119 | 24.9% | 38.7% | 77.7% |
| 08 July, 07:00 – 11:00 | 0.33 | 0.207 | 0.0535 | 3.0% | 4.8% | 18.7% |

*extrapolated from the measurements at 675 nm and 1020 nm)
It is found that on 18 June the coarse mode clearly dominates the AOD at 1064 nm, whereas
on 8 July it only contributes about 20 % to the total AOD. As expected the relative
contributions of the coarse mode to the AOD at 360 nm are much smaller (25 % and 3%).
In the last step the probability of aerosol scattering in backward direction is considered,
because the ceilometer receives scattered light from that direction. For that purpose the ratios
of the optical depths are multiplied by the corresponding values of the normalised phase
functions at 180° and in this way the relative contributions to the backscattered signals from
the coarse mode for both wavelengths and both days are calculated (Table A28). Interestingly,
on 8 July the contributions of the coarse mode to the backscattered signal at both wavelengths
differs by only about 10%. In contrast, on 18 June the difference is much larger.

Table A28 Ratio of phase functions (coarse / fine) in backward direction and relative
contribution of coarse mode to the backscattered signal at both wavelengths

| Date | Ratio phase function at 360 nm | Ratio phase function at 1064 nm | Relative contribution of coarse mode at 360 nm | Relative contribution of coarse mode at 1064 nm |
|---|---|---|---|---|
| 18 June, 11:00 – 14:00 | 1.13 | 0.61 | 27.3% | 68.0% |
| 08 July, 07:00 – 11:00 | 2.7 | 0.99 | 7.8% | 18.3% |

For 8 July, the results can be interpreted in the following way: at 360 nm the aerosol profiles
extracted as described above overestimate the contribution from the coarse mode by about
10%. To estimate the effect of this overestimation we construct modified aerosol extinction
profiles, in which 10% of the total AOD is relocated. Since we expect that the coarse mode
aerosols are usually located at low altitude, we construct 4 different modified profiles (see
Fig. A33) with different altitudes (1.5 km, 1 km, 0.75 km, or 0.5 km), below which 10% of
the aerosol extinction is relocated to altitudes above (assuming that the coarse mode aerosol is
only located below these altitudes). Of course, such a sharp boundary is not very realistic, but
it allows to quantify the overall effect of the relocation. We selected the aerosol profile for 8
July extracted by INTA, which reached up to 7 km (see Fig. 9).  It should be noted that if 10
% of the total AOD is relocated from the lowest layer to only the upper most layer no further
enhancement of the $O_4$ dAMF is found (see appendix A6).

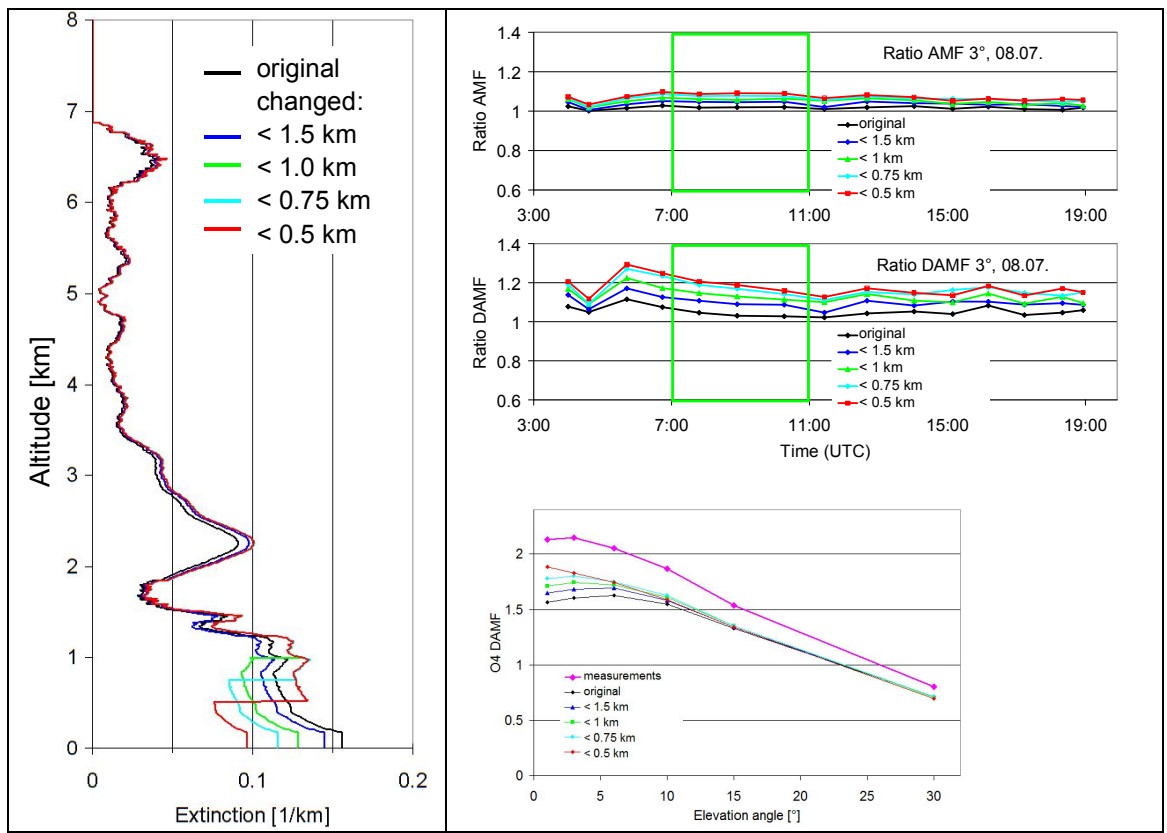

Fig. A33 Left: Modified aerosol profiles for 08 July assuming that the coarse mode aerosol is only located in the lowest part of the atmosphere. Top right: ratios of the (d)AMFs calculated for the modified profiles compared to the dAMFs for the standard settings. With decreasing layer height the (d)AMFs increase systematically, because the aerosol extinction close to the surface decreases. Right bottom: comparison of the measured elevation dependence of the $O_4$ dAMFs for the period 7:00 – 11:00 on 8 July and simulation results for the different profiles.

Table A29 Ratio of the (d)AMFs for the modified profiles versus those of the standard settings

|  | original INTA | coarse mode below 1.5 km | coarse mode below 1 km | coarse mode below 0.75 km | coarse mode below 0.5 km |
|---|---|---|---|---|---|
| AMF | 1.02 | 1.04 | 1.05 | 1.06 | 1.08 |
| dAMF | 1.04 | 1.09 | 1.13 | 1.17 | 1.18 |

For all modified profiles, a systematic increase of the $O_4$ (d)AMFs compared to those for the standard settings is found. For the $O_4$ dAMFs this increase can be up to 18 % (see Table A29. From the comparison of the elevation dependence of the measured and simulated $O_4$ dAMFs (see Fig. A33), we conclude that the aerosol profile with the coarse mode aerosol below 0.75 km is probably the most realistic one. The main conclusion from this section is that the dAMF for 8 July derived from the standard settings probably underestimates the true dAMF by about 17±5 %.

For 18 June we did not perform similarly detailed calculations, because on that day the uncertainties of the aerosol extinction profile caused by the missing sensitivity of the ceilometer below 180 m are much larger than on 8 July. On 18 June also the magnitude of the relocation of the aerosol extinction between different altitudes would be much larger than on 8 July.

**Appendix A6 Influence of elevated aerosol layers on the $O_4$ (d)AMF**

Ortega et al. (2016) showed that for their measurements the consideration of elevated aerosol layers (between about 3 and 5 km) is essential to bring measured and simulated $O_4$ (d)AMFs into agreement. They also used LIDAR measurements at similar wavelengths as the MAX-DOAS observations. In our study, we consider aerosol layers over an even larger altitude range (up to 7 km). Nevertheless, it is interesting to see how the simulated $O_4$ (d)AMFs change if the extinctions at various altitude ranges are changed systematically. Here we chose the aerosol extinction profile extracted by INTA for the period 7:00 to 11:00 on 8 July, because it contains substantial amounts of aerosols in elevated layers (see Fig. 9). During that period three distinct aerosol layers can be identified (see Table A30).

Table A30 Selection of different aerosol layers on 08 July (07:00 – 11:00)

| layer | AOD | Relative contribution to total AOD |
|---|---|---|
| 0 – 1.68 km | 0.186 | 55.4 % |
| 1.68 – 4.9 km | 0.116 | 34.5 % |
| 4.9 – 7 km | 0.035 | 10.4 % |

Then, the extinction of the individual aerosol layers were increased by 40 % compared to the original profile. After that modification the whole profiles are scaled with a constant factor to match the AOD of the sun photometer observations. The modified profiles are then used for the simulation of $O_4$ (d)AMFs. A second set of profiles was created to investigate the effect of extreme relocations: here certain fractions (10%, 25% or 30%) of the total AOD were relocated from the bottom layer to the top layer.

The modified profiles and the ratios of the corresponding $O_4$ (d)AMFs versus the $O_4$ dAMFs of the original profile are shown in Fig. A34. For the $O_4$ AMFs the relocations of the extinction profiles lead to a general increase of the $O_4$ AMFs of up to 20%. For the $O_4$ dAMFs for most modified profiles a strong increase compared to the original profile is found. Only for the profile with an increase of the extinction in the lowest layer a slight decrease is observed. For the profiles with the extreme relocations the increase of the $O_4$ dAMFs reaches almost 50%.

From these results it can be concluded that for a relocation of about 27% almost perfect agreement with the measurements is found (see Fig. A34). For such an aerosol profile simulations and measurements could be brought into agreement without a scaling factor. However, such a large redistribution is not supported by the AERONET inversion products (see appendix A5). It should also be noted that for such a profile, about 73% of the total AOD would be located above about 1.7km. Moreover, for such an aerosol profile it is found that the simulated $O_4$ AMFs for 90° elevation systematically underestimate the measured $O_4$ AMFs at high SZA by about 15% (see Fig. A34), whereas much better agreement is found for the standard settings. The underestimation of the $O_4$ AMFs for 90° elevation is caused by the high aerosol amount at high altitudes, which increases the scattering altitude of the solar photons observed at 90° elevation. A similar effect could be caused by cirrus clouds, but on the selected days there are no indications for such clouds in the ceilometer data.

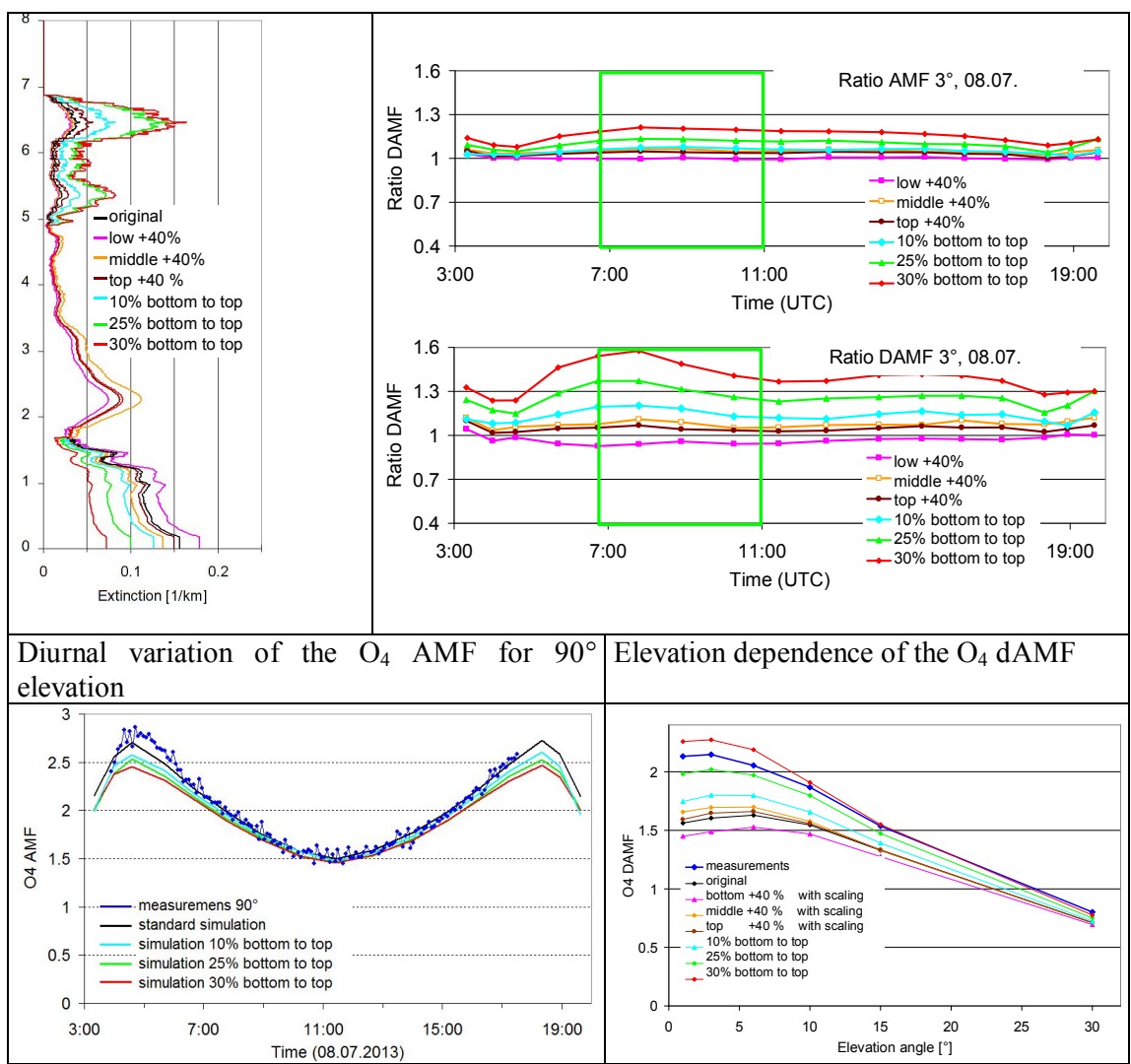

| Diurnal variation of the O$_4$ AMF for 90° elevation | Elevation dependence of the O$_4$ dAMF |
|---|---|

Fig. A34 Top left: Aerosol profiles used for the simulations (see text). Top right: Ratios of the O$_4$ (d)AMFs simulated for the modified profiles versus those of the original profile. Bottom: comparison of the measured diurnal variation (SZA dependence) for 90° elevation (left), and the elevation dependence of the O$_4$ dAMFs for the period 7:00 – 11:00 on 8 July (right).

Table A31 Ratios of (d)AMFs for 8 July 2013 for the modified profiles with respect to the original profile

|  | low +40 % | middle +40 % | top +40 % | 10% bottom to top | 25% bottom to top | 30% bottom to top |
|---|---|---|---|---|---|---|
| AMF | 1.00 | 1.06 | 1.04 | 1.07 | 1.12 | 1.20 |
| dAMF | 0.94 | 1.08 | 1.04 | 1.17 | 1.31 | 1.48 |

2934
2935
2936
2937