# Peer review of "Is a scaling factor required to obtain closure between measured and modelled atmospheric O4 absorptions? – An assessment of uncertainties of measurements and radiative transfer simulations for two days during the MAD-CAT campaign"

_Atmospheric Measurement Techniques, 2018_

## Referee Comment (RC1) · Anonymous Referee #1 · 6 Nov 2018

General comments

This manuscript discusses the statistical significance of the gap between observed and simulated AMFs of O4 on selected two clear-sky days during MADCAT campaign. Thorough and detailed analysis of various factors producing uncertainties in the observed and simulated AMFs was made. The authors pointed out the importance of proper usage of temperature and pressure for the condition, proper account of aerosol optical parameters (phase function, aerosol profile extraction) in the simulation, and standardization of DOAS settings (spectral range, degree of polynomial etc) for obser-

vations. Considering these factors altogether, the authors conclude that the gap was insignificant on one day (June 18) but was significant on other day (July 8), supporting conclusion from some previous works. Recognizing that there is a hot debate in the community if the scaling factor is necessary, the manuscript is valuable since it provides as thorough analyses as ever provided.

Nonetheless, I would like to request revision on the following points. First, I find the studied uncertainties could be classified into two types: those from apparently ill treatment (i.e., 203K O4 cross section, US standard atmosphere without temperature correction, no offset in the DOAS analysis etc) and those unavoidable even with the state-of-the-art analysis. For the purpose of evaluating spread of results from multiple groups and of determining best practice to avoid potential hazard during the analysis, determination of the former type uncertainty helps. But when discussing the significance of the gap between observed and simulated AMFs of O4 critically, only latter type uncertainties should be used. In such a way better control of the determined uncertainties is recommended. Secondly, it should be more clarified in Abstract that the precise determination of the uncertainties (+/- 0.16 and +/-0.12 here) is the main point. Careless readers may not realize the importance. Thirdly, possible influence of horizontal heterogeneity of aerosol optical parameters should be mentioned. When the aerosol abundance over the line of sight is becoming less with distance (which may be likely when instrument is located in a city looking out of it), the observed higher O4 dAMFs might be better explained by considering such inhomogeneity even on July 8. I understand that with 1-D radiative transfer models homogeneity needs to be assumed and detailed discussion would be beyond the scope. However, some simple analysis such as that on spatial distribution of AOD from satellite with a fine resolution maybe possible. Lastly, conciseness should be attained during revision. I would suggest shortening section 4.1 and section 5 (paragraphs before section 5.1).

Overall, I would suggest minor revisions on the general comments above and some specific comments listed below.

[Figure]

Specific comments

1. Line 359. Probably appendix A2?

2. Line 526. US standard atmosphere

3. Figure 10. What are the differences of the first three series, with same legend "HG AP 0.6?"

4. Figure 11. Although the panel is for showing noise influence, the gap related to the main conclusion of this study is well represented as the difference in the O4 optical depths in the first two panels. Such discussion should be added in section 4.3.1.

5. Table A12 in line 1922 is mislabeled. (Table A10)

6. Table A11. MCARTIM

7. Lines 846-848. Second and third points should be exchanged, considering the order of Fig. 14b and c and the following discussion.

8. Line 906. Overall uncertainty calculation deriving 0.12 is not clear. When considering 3% uncertainties for VCD, 6.1% from radiative transfer simulation, and 10.8% from spectral analysis, the overall uncertainty may be 13%. When it is around 0.71, it can be 0.09?

9. Line 944. 8 July

---

## Referee Comment (RC2) · Anonymous Referee #2 · 5 Dec 2018

Wagner et al 2018 address a very important topic of the need of scaling factor to bring MAX-DOAS measured differential slant column densities (dSCD) of oxygen collision complex (O4) retrieved from 352 – 387 nm in agreement with the radiative transfer modeled dSCD at 360 nm. An extensive and very thorough evaluation of the error sources in the DOAS analysis and RT modeling is presented. The authors analyzed data from two time periods (18 June and 8 July 2013) during MADCAT campaign in Mainz, Germany, when time and location coincident MAX-DOAS, aerosol (AERONET, Ceilometer) profile measurements were conducted with a support of additional surface

observations (PM2.5, PM10, temperature, pressure and relative humidity). They iden-
tified "standard" cases for DOAS fitting and for RT model simulations, and a number
of potential scenarios deviating from the standard cases. The authors concluded that
the agreement between the measured and modeled O4 absorption is almost perfect
(within a large error of 16%) on 18 June 2018. On the other hand the measured O4
absorption had to be scaled by 0.71 ($\pm$0.12) to bring in agreement with the modeled
absorption. The cause of the discrepancy was not identified.

This work is very important and is well suited for AMT publication. However, I think the
article will benefit from some reorganization.

Please see attached file for details

Please also note the supplement to this comment:
https://www.atmos-meas-tech-discuss.net/amt-2018-238/amt-2018-238-RC2-
supplement.pdf

———————————————————

[Figure]

**Supplement:**

**Review of: "Is a scaling factor required to obtain closure between measured and modelled atmospheric $O_4$ absorptions? – A case study for two days during the MADCAT campaign (amt-2018-238)" Wagner et al., 2018**

Wagner et al., 2018 address a very important topic of the need of scaling factor to bring MAX-DOAS measured differential slant column densities (dSCD) of oxygen collision complex ($O_4$) retrieved from 352 – 387 nm in agreement with the radiative transfer modeled dSCD at 360 nm. An extensive and very thorough evaluation of the error sources in the DOAS analysis and RT modeling is presented. The authors analyzed data from two time periods (18 June and 8 July 2013) during MADCAT campaign in Mainz, Germany, when time and location coincident MAX-DOAS, aerosol (AERONET, Ceilometer) profile measurements were conducted with a support of additional surface observations ($PM_{2.5}$, $PM_{10}$, temperature, pressure and relative humidity). They identified "standard" cases for DOAS fitting and for RT model simulations, and a number of potential scenarios deviating from the standard cases. The authors concluded that the agreement between the measured and modeled $O_4$ dAMF is almost perfect 1.01 ($\pm$0.16) on 18 June 2018. On the other hand the "measured" $O_4$ dAMF had to be scaled by 0.71 ($\pm$0.12) to bring in agreement with the modeled absorption for standard case DOAS fitting and RT modeling scenarios. The cause of the discrepancy was not identified.

This work is very important and is well suited for AMT publication. However, I think the article will benefit from some reorganization.

**Major comments:**

I think that there are two main topics that the authors are trying to address (I would say each of them is worth a separate publication):

*(1) Is a scaling factor required to obtain closure between measured and modeled atmospheric $O_4$ absorptions – Part A: identifying best-case scenarios based on auxiliary measurements and best practices.*

In this part the best case DOAS fitting scenario and best case RT modeling scenario should be identified based on the best available data to describe atmospheric conditions during the selected periods. Potential sources of errors for ***these particular*** scenarios should be evaluated. For example, for RT modeling:

- Mie scattering phase functions using AERONET inversion data results for size distribution and refractive index real and imaginary parts extrapolated to 360 nm from longer wavelengths (440, 675 nm). Evaluating errors associated with these particular inputs to the RT (e.g using 440 nm inversion results directly?). Please also note that the AERONET level 2.0 inversions are not available during some of the selected periods, potentially due to presence of clouds. Available dates/time are listed below:
    6/18/13 07:24:51
    6/18/13 15:34:32
    6/18/13 16:12:07
    7/8/13   05:16:20
    7/8/13   05:48:33

- Ceilometer backscatter profiles corrected by AERONET CIMEL AOD, and their errors (backscatter to aerosol extinction coefficient profiles conversion, wavelength differences, extrapolation to the surface)
- Radiosonde temperature, pressure and relative humidity measured profiles at fine grid with ECMWF ERA-Interim reanalysis above and their errors (e.g. different groups extraction of the data, usage of MERRA-2 profiles available at better than 1 km resolution near ground and every 3 hours)
- Accounting for polarization and RRS in the RT calculations and their errors (e.g. different models)
- If we consider $O_4$ cross section by Thalman and Volkamer (2013) accurate at all temperatures use T-dependent $O_4$ cross sections for RT calculations.
- Surface albedo from satellite measurement or AERONET inversion at 440 nm (which varies from 2.7 to 4% during the selected times).
- Effect of instrument FOV and pointing error, especially under shallow aerosol layer presence (the fact that measured dSCD at several low VEA are close to each other does not exclude potential error in pointing that has to be accounted for in modeling).

DOAS fitting scenario selected for the standard case can be considered best practice. The only things I would probably recommend changing is the offset from polynomial order 2 to 1 and not applying polynomial at all to the $O_4$ cross section due to its broad band wavelength dependency. In calculating the errors due to the fitting, I would not go to the extreme case of no offset. At low elevation angles the effective $O_4$ temperature is around 270K, I would suggest using $O_4$ cross section at 273K as one of the sensitivity cases.

There is another change I would recommend here – what quantity is actually compared.

Since the actual measurements are ground-based hyperspectral sky radiances the derived variable directly from the measurements without any assumptions about the atmosphere (accept for species effective temperatures) is the differential slant column density (dSCD). There are no passive measurements at the bottom of atmosphere that do not contain $O_4$ absorption, including the reference used in this study (zenith direction). From Beer's law, ignoring wavelength shift, offset and other corrections:

$$\left( \frac{\ln\left( I_{90}^{measured} - I_{VEA}^{measured} \right)}{\sigma_{O4}(T)} \right)_{\lambda \, window} = dSCD_{VEA}^{measured} =$$

$$= \underbrace{SCD_{VEA}^{total} - SCD_{90^o}}_{individual \; components \; are \; not \; measured \; directly}$$

$$dAMF_{VEA} = \frac{dSCD_{VEA}^{measured}}{VCD} = \frac{SCD_{VEA}^{total} - SCD_{90^o}}{VCD} = AMF_{VEA} - AMF_{90^o}$$

From the above discussion AMF and dAMF are quantities derived based on the assumptions made about $AMF_{90}$ and VCD:

$$AMF_{VEA} = dAMF_{VEA} + AMF_{90^o} = \frac{dSCD_{VEA}^{measured}}{VCD} + AMF_{90^o}$$

I believe the paper will benefit if dSCD are compared directly with the RT modeled dSCD in the first section of the paper.

At the end of this section the reader should clearly see based on the best DOAS fitting and relevant to it errors and best atmosphere modeling (with its relevant errors) whether the measured and modeled dSCDs agree and to what extent.

*(2) Is a scaling factor required to obtain closure between measured and modeled atmospheric $O_4$ absorptions – Part B: error analysis to explain potential causes of SF (varying the parameters outside of (1).*

This section can include all the other cases for (d)AMF comparisons. Its main purpose could be to make recommendations and identifying problems with using less realistic atmospheric scenarios in the MAX-DOAS data inversions and DOAS fitting limitations.

**Minor comments:**
1. The paper is very long and difficult to read due to constant references to the appendices and main body figures and tables. Some of the figures and tables can be consolidated or eliminated.
2. Clear days are probably more appropriate to call cloud-free?
3. L 49 … agree within 1% with the corresponding radiative transfer simulations at 360 nm
4. L246: which version of LIDORT is used in this study?
5. L277: rephrase to make clear that the comparison is done between hyperspectral fitting DOAS analysis vs. singe wavelength
6. What is the source of extraterrestrial irradiance used for synthetic spectra simulation?
7. L293: Level 2 data are available now.  It will be good to comment how it compares to level 1.5.
8. L306: Link from the pdf does not work, URL is valid.
9. Abstract refers to the campaign MAD-CAT, other places MADCAT
10. L348: Intensity Offset polynomial of order 2 is quite large. Can you please explain why it was chosen?
11. L903: Can you please explain how 1.01±0.16 and 0.71±0.12 are calculated? Is this for the entire two days and all observation geometries?

Time scale on Fig. 1 for the top panel (A) is unclear.

---

## Author Comment (AC1) · 10 Mar 2019

The replies to the reviewer comments are marked in blue

General comments

This manuscript discusses the statistical significance of the gap between observed and simulated AMFs of O4 on selected two clear-sky days during MADCAT campaign. Thorough and detailed analysis of various factors producing uncertainties in the observed and simulated AMFs was made. The authors pointed out the importance of proper usage of temperature and pressure for the condition, proper account of aerosol optical parameters (phase function, aerosol profile extraction) in the simulation, and standardization of DOAS settings (spectral range, degree of polynomial etc) for observations. Considering these factors altogether, the authors conclude that the gap was insignificant on one day (June 18) but was significant on other day (July 8), supporting conclusion from some previous works. Recognizing that there is a hot debate in the community if the scaling factor is necessary, the manuscript is valuable since it provides as thorough analyses as ever provided.

We thank the reviewer for the positive assessment of our paper and for the good suggestions. We addressed them as described in detail below.

Nonetheless, I would like to request revision on the following points. First, I find the studied uncertainties could be classified into two types: those from apparently ill treatment (i.e., 203K O4 cross section, US standard atmosphere without temperature correction, no offset in the DOAS analysis etc) and those unavoidable even with the stateofthe-art analysis. For the purpose of evaluating spread of results from multiple groups and of determining best practice to avoid potential hazard during the analysis, determination of the former type uncertainty helps. But when discussing the significance of the gap between observed and simulated AMFs of O4 critically, only latter type uncertainties should be used. In such a way better control of the determined uncertainties is recommended.

We agree that such a separation of different types of uncertainties would be helpful. Therefore we added two columns to tables 9 and 10 in which we quantify the uncertainties if optimum settings were used and sufficient independent information was available. For the radiative transfer simulations of the O4 dAMFs the uncertaintes for these optimum settings are about ±4% compared to ±(6 – 9)% for two days of the MAD-CAT campaign. For the spectral analysis the uncertaintes for the optimom settings are about ±6% compared to ±(11-13)% for the two selected days of the MAD-CAT campaign.
These findings indicate that for future campaigns the comparison of measured and simulated $O_4$ absorptions can ptobably be carried out with much better accuracy (if these optimum settings were used). Here it should, however, be noted that the optimum settings for the radiative ransfer simulations will require LIDAR measurements at the same wavelengths as the MAX-DOAS measurements and without a sensitivity gap close to the surface. Such measurements are currently hardly available. This information was added to the new section 4.4.

Secondly, it should be more clarified in Abstract that the precise determination of the uncertainties (+/- 0.16 and +/-0.12 here) is the main point. Careless readers may not realize the importance.

We agree and modified the abstract to make this point more clear. We also changed the title to: 'Is a scaling factor required to obtain closure between measured and modelled atmospheric O4 absorptions? An assessment of uncertainties of measurements and radiative transfer simulations for two days during the MAD-CAT campaign'.

Thirdly, possible influence of horizontal heterogeneity of aerosol optical parameters should be mentioned. When the aerosol abundance over the line of sight is becoming less with distance (which may be likely when instrument is located in a city looking out of it), the observed higher O4 dAMFs might be better explained by considering such inhomogeneity even on July 8. I understand that with 1-D radiative transfer models homogeneity needs to be assumed and detailed discussion would be beyond the scope. However, some simple analysis such as that on spatial distribution of AOD from satellite with a fine resolution maybe possible.

We agree that this is a potentially important aspect. However, for the two selected periods the wind direction and wind speed were rather constant. On 18 June the wind direction was between 80° and 150° wrt North, and the wind speed was about 2 m/s. On 8 July the wind direction was between 70° and 90° wrt North, and the wind speed was about 3 m/s. Thus on 8 July the wind came from almost the same direction at which the instruments were looking. Taking the wind data into account, during the 4 hours of the selected period on 8 July, the air masses moved along a distance of about 40 km. During the 3 hours of the selected period on 18 June, the air masses moved along a distance of about 20 km. These distances are larger than the distances for which the MAX-DOAS observations are sensitive. Since also the AOD and the aerosol extinction profiles were rather constant during both selected periods, we conclude that for the measurements considered here horizontal gradients can not explain the discrepancies between measurements and observations. It should also be noted that the discrepancies were simultaneously observed at all 4 azimuth directions. We added this information to section 4.2.1.

Lastly, conciseness should be attained during revision. I would suggest shortening section 4.1 and section 5 (paragraphs before section 5.1).

We moved several parts of section 4.1 to the appendix. We also shortened the paragraphs before section 5.1.

Overall, I would suggest minor revisions on the general comments above and some specific comments listed below.

Specific comments
1. Line 359. Probably appendix A2?

Corrected

2. Line 526. US standard atmosphere

Corrected

3. Figure 10. What are the differences of the first three series, with same legend "HG AP 0.6?"

The correct labels (0.60, 0.68, and 0.75) were added.

4. Figure 11. Although the panel is for showing noise influence, the gap related to the main conclusion of this study is well represented as the difference in the O4 optical depths in the first two panels. Such discussion should be added in section 4.3.1.

We added the following sentence to section 4.3.1:
'Here it is interesting to note that the ratios of the results for the measured spectrum and the simulated spectra are between 0.68 and 0.74, similar to ratio for the dAMFs on 8 July shown in Table 8.'

5. Table A12 in line 1922 is mislabeled. (Table A10)

Corrected

6. Table A11. MCARTIM

Corrected

7. Lines 846-848. Second and third points should be exchanged, considering the order of Fig. 14b and c and the following discussion.

The order was changed

8. Line 906. Overall uncertainty calculation deriving 0.12 is not clear. When considering 3% uncertainties for VCD, 6.1% from radiative transfer simulation, and 10.8% from spectral analysis, the overall uncertainty may be 13%. When it is around 0.71, it can be 0.09?

Many thanks for this hint! We agree and updated the calculations accordingly (with slightly modified uncertainties, see tables 9 and 10.

9. Line 944. 8 July

Corrected

---

## Author Comment (AC2) · 10 Mar 2019

The replies to the reviewer comments are marked in blue

Wagner et al., 2018 address a very important topic of the need of scaling factor to bring MAX-DOAS measured differential slant column densities (dSCD) of oxygen collision complex (O4) retrieved from 352 – 387 nm in agreement with the radiative transfer modeled dSCD at 360 nm. An extensive and very thorough evaluation of the error sources in the DOAS analysis and RT modeling is presented. The authors analyzed data from two time periods (18 June and 8 July 2013) during MADCAT campaign in Mainz, Germany, when time and location coincident MAX-DOAS, aerosol (AERONET, Ceilometer) profile measurements were conducted with a support of additional surface observations (PM2.5, PM10, temperature, pressure and relative humidity). They identified "standard" cases for DOAS fitting and for RT model simulations, and a number of potential scenarios deviating from the standard cases. The authors concluded that the agreement between the measured and modeled O4 dAMF is almost perfect 1.01 (±0.16) on 18 June 2018. On the other hand the "measured" O4 dAMF had to be scaled by 0.71 (±0.12) to bring in agreement with the modeled absorption for standard case DOAS fitting and RT modeling scenarios. The cause of the discrepancy was not identified.

This work is very important and is well suited for AMT publication. However, I think the article will benefit from some reorganization.

We thank the reviewer for the positive assessment of our paper and for the good suggestions. We addressed most of them as described in detail below.

**Major comments:**

I think that there are two main topics that the authors are trying to address (I would say each of them is worth a separate publication):

*(1) Is a scaling factor required to obtain closure between measured and modeled atmospheric O4 absorptions – Part A: identifying best-case scenarios based on auxiliary measurements and best practices.*

In this part the best case DOAS fitting scenario and best case RT modeling scenario should be identified based on the best available data to describe atmospheric conditions during the selected periods. Potential sources of errors for ***these particular*** scenarios should be evaluated. For example, for RT modeling:

- Mie scattering phase functions using AERONET inversion data results for size distribution and refractive index real and imaginary parts extrapolated to 360 nm from longer wavelengths (440, 675 nm). Evaluating errors associated with these particular inputs to the RT (e.g using 440 nm inversion results directly?). Please also note that the AERONET level 2.0 inversions are not available during some of the selected periods, potentially due to presence of clouds. Available dates/time are listed below:

6/18/13 07:24:51
6/18/13 15:34:32
6/18/13 16:12:07
7/8/13 05:16:20
7/8/13 05:48:33
7/8/13 06:54:34
7/8/13 07:32:12
7/8/13 15:38:04
7/8/13 16:12:13

- Ceilometer backscatter profiles corrected by AERONET CIMEL AOD, and their errors (backscatter to aerosol extinction coefficient profiles conversion, wavelength differences, extrapolation to the surface)
- Radiosonde temperature, pressure and relative humidity measured profiles at fine grid with ECMWF ERA-Interim reanalysis above and their errors (e.g. different groups extraction of the data, usage of MERRA-2 profiles available at better than 1 km resolution near ground and every 3 hours)
- Accounting for polarization and RRS in the RT calculations and their errors (e.g. different models)
- If we consider O4 cross section by Thalman and Volkamer (2013) accurate at all temperatures use T-dependent O4 cross sections for RT calculations.
- Surface albedo from satellite measurement or AERONET inversion at 440 nm (which varies from 2.7 to 4% during the selected times).
- Effect of instrument FOV and pointing error, especially under shallow aerosol layer presence (the fact that measured dSCD at several low VEA are close to each other does not exclude potential error in pointing that has to be accounted for in modeling). DOAS fitting scenario selected for the standard case can be considered best practice. The only things I would probably recommend changing is the offset from polynomial order 2 to 1 and not applying polynomial at all to the O4 cross section due to its broad band wavelength dependency. In calculating the errors due to the fitting, I would not go to the extreme case of no offset. At low elevation angles the effective O4 temperature is around 270K, I would suggest using O4 cross section at 273K as one of the sensitivity cases.

There is another change I would recommend here – what quantity is actually compared. Since the actual measurements are ground-based hyperspectral sky radiances the derived variable directly from the measurements without any assumptions about the atmosphere (accept for species effective temperatures) is the differential slant column density (dSCD).

There are no passive measurements at the bottom of atmosphere that do not contain O4 absorption, including the reference used in this study (zenith direction). From Beer's law, ignoring wavelength shift, offset and other corrections:

$$\left( \frac{\ln\left(I_{90}^{measured} - I_{VEA}^{measured}\right)}{\sigma_{O4}(T)} \right)_{\lambda \, window} = dSCD_{VEA}^{measured} =$$

$$= \underbrace{SCD_{VEA}^{total} - SCD_{90^o}}_{individual \; components \; are \; not \; measured \; directly}$$

$$dAMF_{VEA} = \frac{dSCD_{VEA}^{measured}}{VCD} = \frac{SCD_{VEA}^{total} - SCD_{90^o}}{VCD} = AMF_{VEA} - AMF_{90^o}$$

From the above discussion AMF and dAMF are quantities derived based on the assumptions made about AMF90 and VCD:

$$AMF_{VEA} = dAMF_{VEA} + AMF_{90^o} = \frac{dSCD_{VEA}^{measured}}{VCD} + AMF_{90^o}$$

I believe the paper will benefit if dSCD are compared directly with the RT modeled dSCD in the first section of the paper.
At the end of this section the reader should clearly see based on the best DOAS fitting and relevant to it errors and best atmosphere modeling (with its relevant errors) whether the measured and modeled dSCDs agree and to what extent.
*(2) Is a scaling factor required to obtain closure between measured and modeled atmospheric O4 absorptions – Part B: error analysis to explain potential causes of SF (varying the parameters outside of (1).*
This section can include all the other cases for (d)AMF comparisons. Its main purpose could be to make recommendations and identifying problems with using less realistic atmospheric scenarios in the MAX-DOAS data inversions and DOAS fitting limitations.

We thank the reviewer for this suggestion. We understand the intention, but we decided not to split the paper into two parts. The main reason is that both suggested parts are closely linked and it would thus be difficult for the readers to follow them when split into separate papers. In addition, the suggested part 2 would be rather short and mostly speculative, because the reason for a scaling factor is still not known.
Thus we addressed the suggestion of the reviewer by including a new section (section 5.2 'Which conditions would be needed to bring measurements and simulations on 8 July into agreement?'). In that section changes of the measurement conditions are discussed which could bring measurements and simulations into agreement.

The detailed suggestions of the reviewer given above (for part 1) are addressed below:

In this part the best case DOAS fitting scenario and best case RT modeling scenario should be identified based on the best available data to describe atmospheric conditions during the selected periods. Potential sources of errors for ***these particular*** scenarios should be evaluated. For example, for RT modeling:
- Mie scattering phase functions using AERONET inversion data results for size distribution and refractive index real and imaginary parts extrapolated to 360 nm from longer wavelengths (440, 675 nm). Evaluating errors associated with these particular inputs to the RT (e.g using 440 nm inversion results directly?).

In our opinion, we already selected scenarios for the quantitative comparison which are (at least close to) the optimum choice. On both days we selected periods around noon, for which the measured intensities are high and the variation of the SZA is small. Moreover, during the selected periods, the variation of the ceilometer profiles is relatively small compared to before and after. We added this information to section 3.2.

Many thanks for the information about the available phase functions! We performed sensitivity studies to quantify the effect of the extrapolation of the phase functions. We found that the O4 (d)AMFs hardly change (<1%) if either the phase functions at 440 nm or

extrapolated to 360 nm are used. Similar small changes are found if the phase functions before or after the selected periods are used.

- Ceilometer backscatter profiles corrected by AERONET CIMEL AOD, and their errors (backscatter to aerosol extinction coefficient profiles conversion, wavelength differences, extrapolation to the surface)

As stated above, the variation of the ceilometer backscatter profiles was relatively small during the selected periods.

- Radiosonde temperature, pressure and relative humidity measured profiles at fine grid with ECMWF ERA-Interim reanalysis above and their errors (e.g. different groups extraction of the data, usage of MERRA-2 profiles available at better than 1 km resolution near ground and every 3 hours)

In principle one could use fine grid ECMWF ERA-Interim reanalysis data, but since the uncertainties related to the temperature and pressure profiles are rather small compared to other uncertainties, we did not use additional meteorological data.

- Accounting for polarization and RRS in the RT calculations and their errors (e.g. different models)

As shown in our study, the effects of polarization in the RT are negligible. RRS was taken into account for the synthetic spectra and almost perfect agreement with the simulated O4 (d)AMFs was found. Thus we conclude that the effects of polarization and RRS can be neglected.

- If we consider O4 cross section by Thalman and Volkamer (2013) accurate at all temperatures use T-dependent O4 cross sections for RT calculations.

This is in principle a good idea. However, the effect is probably very small as indicated by the very good agreement of the results from the synthetic spectra and the simulated O4 (d)AMFs.

- Surface albedo from satellite measurement or AERONET inversion at 440 nm (which varies from 2.7 to 4% during the selected times).

The variation of the surface albedo could also be taken into account, especially if it deviates strongly from the 'standard settings'. However, as shown in our study, the influence of small changes (e.g. from 5% to 3%) on the O4 (d)AMFs is rather small (below 1%).

- Effect of instrument FOV and pointing error, especially under shallow aerosol layer presence (the fact that measured dSCD at several low VEA are close to each other does not exclude potential error in pointing that has to be accounted for in modeling).

We agree with the reviewer and performed additional sensitivity studies varying the FOV and also systematically distorting the elevation calibration by ±0.5°. The changes of the O4 (d)AMFs were below 1%. We added this information to the text (section 3.2).

DOAS fitting scenario selected for the standard case can be considered best practice. The only things I would probably recommend changing is the offset from polynomial order 2 to 1 and

not applying polynomial at all to the O4 cross section due to its broad band wavelength dependency.

In our opinion there might be good reasons for increasing the degree of the fitted intensity offset. For example, the relative contribution of spectral stray light could cause an intensity offset in the measured spectra, which changes non-linearly with wavelength. Thus we think it is difficult to give a clear recommendation on the degree of the intensity offset.
We added the following text in section 4.3.2: 'Higher order intensity offsets might compensate for wavelength dependent offsets (e.g. spectral straylight), which can be important for real measurements, while the synthetic spectra do not contain such contributions.'

Concerning the application of the polynomial, there might be a misunderstanding. We included the O4 cross section without any previous high or low pass filtering. Concerning the degree of the DOAS polynomial we see good reaons to use such a polynomial, e.g. that the broad band wavelength dependence of the measured spectra are different for the different elevation angles, and also change with time. The very good agreement between the results of the synthetic spectra and the simulated O4 (d)AMFs indicates that the chosen polynomial degree is not problematic.

In calculating the errors due to the fitting, I would not go to the extreme case of no offset.

We agree. Note that the case without intensity offset was already ignored for calculating the errors in the discussion version of our paper.

At low elevation angles the effective O4 temperature is around 270K, I would suggest using O4 cross section at 273K as one of the sensitivity cases.

In principle we agree with the reviewer here. However, we did not change the O4 cross section because of two reasons:
a) the effect of such small temperature changes is rather small.
b) in most existing studies, O4 cross sections for room temperature were used. Thus we prefer to stay consistent with those studies.

There is another change I would recommend here – what quantity is actually compared. Since the actual measurements are ground-based hyperspectral sky radiances the derived variable directly from the measurements without any assumptions about the atmosphere (accept for species effective temperatures) is the differential slant column density (dSCD).
There are no passive measurements at the bottom of atmosphere that do not contain O4 absorption, including the reference used in this study (zenith direction). From Beer's law, ignoring wavelength shift, offset and other corrections:

$$\left(\frac{\ln\left(I_{90}^{measured} - I_{VEA}^{measured}\right)}{\sigma_{O4}(T)}\right)_{\lambda\ window} = dSCD_{VEA}^{measured} =$$

$$= \underbrace{SCD_{VEA}^{total} - SCD_{90^o}}_{individual\ components\ are\ not\ measured\ directly}$$

$$dAMF_{VEA} = \frac{dSCD_{VEA}^{measured}}{VCD} = \frac{SCD_{VEA}^{total} - SCD_{90^o}}{VCD} = AMF_{VEA} - AMF_{90^o}$$

From the above discussion AMF and dAMF are quantities derived based on the assumptions made about AMF90 and VCD:

$$AMF_{VEA} = dAMF_{VEA} + AMF_{90^o} = \frac{dSCD_{VEA}^{measured}}{VCD} + AMF_{90^o}$$

I believe the paper will benefit if dSCD are compared directly with the RT modeled dSCD in the first section of the paper.

In our opinion, the only difference to your suggestion is that we divide the O4 (d)SCDs by the O4 VCD. Both choices are equivalent. To make the interpretation of the results in units of (d)SCDs easier, we added second y-axes in Figures 2 and 3 in (d)SCD units.

**Minor comments:**

1. The paper is very long and difficult to read due to constant references to the appendices and main body figures and tables. Some of the figures and tables can be consolidated or eliminated.

We understand this concern. However, one important part of the study deals with the quantification of the uncertainties of the spectral analysis and radiative tarnsfer simulations. For readers with interest in the details of the sensitivity studies the figures and tables in the appendix will be important. In contrast, for readers who are mostly interested in the general findings the figures and tables in the main part should be sufficient. We therefore decided not to remove any figures or tables.

2. Clear days are probably more appropriate to call cloud-free?

Changed

3. L 49 … agree within 1% with the corresponding radiative transfer simulations at 360 nm

'at 360 nm' was added

4. L246: which version of LIDORT is used in this study?

Version 3.3. The version is 3.3. This information was added to the text.

5. L277: rephrase to make clear that the comparison is done between hyperspectral fitting DOAS analysis vs. singe wavelength

We added the following text: 'at one wavelength (here: 360 nm)'

6. What is the source of extraterrestrial irradiance used for synthetic spectra simulation?

We used the high resolution solar spectrum from Chance and Kurucz (2010). We added this information and the corresponding reference in section 2.4.

7. L293: Level 2 data are available now. It will be good to comment how it compares to level 1.5.

Many thanks for this hint! The Level-2 data are exactly the same as the Level-1.5 data. We removed the corresponding sentence about the Level 1.5 data from the text.

8. L306: Link from the pdf does not work, URL is valid.

Many thanks for his hint! This link should work after the final copy-editing.

9. Abstract refers to the campaign MAD-CAT, other places MADCAT

Now consistently 'MAD-CAT' is used.

10. L348: Intensity Offset polynomial of order 2 is quite large. Can you please explain why it was chosen?

The following text was added in section 4.3.2:

'Higher order intensity offsets might compensate for wavelength dependent offsets (e.g. spectral straylight), which can be important for real measurements, while the synthetic spectra do not contain such contributions.'

11. L903: Can you please explain how $1.01\pm0.16$ and $0.71\pm0.12$ are calculated? Is this for the entire two days and all observation geometries?

The information was added that the ratio was calculated for the middle period of that day.

Time scale on Fig. 1 for the top panel (A) is unclear.

The corresponding labels were added.

---

## Author Comment (AC3) · 11 Mar 2019

This author comment does not refer to the comments of the 'official' reviewers. It refers to an offline comment from Rainer Volkamer, Ted Koenig and Ivan Ortega on 02.01.2019, after the official discussion phase was closed. The email was directly sent to Thomas Wagner and is thus not stored in the discussion forum. Below this email together with the subsequent email discussion is listed.

The email contained a number of suggestions, from which especially one turned out to be very important for the interpretation of the results in the paper. Rainer Volkamer, Ted Koegig and Ivan Ortega argued that the relative backscatter profile derived from the ceilometer measurements at 1064 nm are probably not representative for the aerosol extinction at 360 nm, because the sensitivity to coarse and fine aerosols at both wavelengths is in general different. This important argument led to extensive additional calculations mainly based on the information available in the AERONET inversion products (mainly the phase functions and optical depths of the coarse and fine mode aerosols at different wavelengths). The new calculations showed that the extracted aerosol extinction profile at 360 nm had to be modified compared to the profile described in the original version of the manuscript. This modification decreased the difference between the simulated and measured $O_4$ (d)AMFs. However, still no agreement between measurements and simulations was found for one of both days (08 July 2013). After the manuscript was updated with the new results, it was sent to Rainer Volkamer, Ted Koegig and Ivan Ortega. They were invited to become co-authors. In the following weeks, a long sequence of email exchange started. Unfortunately, this email discussion eventually turned into a self-repeating, complicated and controversial one. Finally Thomas Wagner came to the conclusion that no agreement could be reached, and the revised version was sent to the other co-authors. After their feedback was received, the updated paper was again sent to Rainer Volkamer, Ted Koegig and Ivan Ortega and co-authorship was again offered (but no response was received).Below, the whole email discussion is chronologically listed. Part of the discussion refers to intermediate versions of the paper, which are thus also made available.
* * *
Comments from Rainer, Ted, Ivan in black
Comments from Thomas in blue
Red: names of pdf-files

Email from Rainer Volkamer, 02.01.2019:

Dear Thomas,
Sorry for the slow response, which are due to a hectic and eventful summer.
The attached our view on the CFO4 paper. I hope you will find these comments useful.
Happy New Year!
-Rainer

Reply to Rainer, Ted, Ivan, email 27.01.2019

Dear Rainer, Ivan, Ted,

many thanks for your valuable comments!
Attached I send you the current version of the revised manuscript together with the replies to both reviewers and to your comments.

Your comments were very helpful, especially with respect to the question of the representativity of the ceilometer measurements at 1020 nm for the MAX-DOAS measurements at 360 nm. The revised version addresses this aspect, and also the effect of elevated aerosol layers.
I want to invite you to become co-author of the paper. If you agree, please send me the correct details of your affiliation(s). In any case, your feedback would be welcome.
I already had asked for an extension of the deadline for the submission of the revised manuscript until 4 February, and I will ask the editorial office again for a further extension. Nevertheless, I would appreciate receiving your feedback within one week from now, because I also have to iterate the manuscript with the other co-authors before re-submission.

Many thanks and best regards,

Thomas

The attached pdf file is O4_scaling_factor_27012019.pdf

The detailed comments were provided in an attachment (Comments on Wagner etal CU-Boulder.docx). The content and the replies are given below:

Comments on Wagner et al. 2018 "Is a scaling factor required to obtain closure between measured and modelled atmospheric O4 absorptions? - A case study for two days during the MADCAT campaign"
By Rainer Volkamer, Ivan Ortega, and Ted Koenig

Dear Rainer, Ivan, and Ted,
Many thanks for your valuable comments. Please find our detailed answers below.

Dear Thomas,
This is a significant body of work, and valuable albeit somewhat inconclusive. We agree the present solution can be viewed as consistent with the measurements. But we also think that there is significant evidence that supports an elevated aerosol layer as a plausible explanation for CFO4 on 8 July.
You mention that Ortega et al. 2016 used a similar approach, but we were missing a connection of the present work with the findings in that study (and other our related papers). We have tried to make it easy to establish this connection by suggesting specific text that could easily be added in the introduction and discussion sections here. Feel free to modify it as you see fit, or let us know if we are missing something in suggesting these changes.

We added part of the suggested text and also sensitivity studies of the effect of elevated layers on the O4 (d)SCDs in the revised version of the manuscript (new appendix A6)

Since we did not really contribute to your work until this late point, we do not feel that we need to be added as co-authors. We would like to see our O4 data that we had submitted compared here, if we were to be added as co-authors. Alternatively, you could just add our names in the acknowledgements.

Your comments on the different sensitivities of ceilometer measurements at 1020 nm to fine and coarse mode aerosols led to important additions to and changes of the manuscript (for details see below). Thus we would like to invite you to become co-authors of the paper.

Sorry for the slow response, which are due to a hectic and eventful summer.
Best wishes,
-Rainer

Specific comments:
1) Abstract, line 35: "many studies, in particular based on direct sun light measurements…". Most studies that concluded on the lack of a need for a correction factor in Table 1 are actually scattered sun light measurements, so this is misleading. Revise language to reflect scattered and direct sun light.

We changed 'in particular' to 'including such'

2) Please add the following studies in Table 1, among "studies that did not apply a scaling factor":
- Volkamer et al., 2015, AMAX-DOAS, 360, 477nm – see Figs. 3 + 4. doi:10.5194/amt-8-2121-2015
- Thalman and Volkamer 2010, CE-DOAS, 477nm – see Figs. 8 + 9. doi: 10.5194/amt-3-1797-2010
Both references were added

3) Abstract, line 37: "Up to now, there is no explanation for the observed discrepancies between measurements and simulations." change to "no broad consensus for an explanation", or eliminate entirely. Note that Ortega et al. 2016 provides the following explanation (quote from the abstract): "However, if in the calculations the aerosol is confined to the surface layer (while keeping AOD constant) we find 0.53<CFO4<0.75, similar to previously reported CFO4. **Our results suggest that elevated aerosol layers, unless accounted for, can cause negative bias in the simulated O4 dSCDs that can explain CFO4.**"

The text was changed as suggested.

Fig. 6 and Table 3 of that paper demonstrate, that - surprisingly - elevated aerosol layers mostly modify the O4 SCD in the lower elevation angles. This is somewhat counterintuitive, but warrants a sensitivity study in your paper in our opinion.

Elevated layers (even at higher altitudes than in Ortega et al. (2016)) were already considered in our paper and could not explain the observed discrepancies on 8 July (see also below). Nevertheless, we added sensitivity studies about the effect of increased aerosol extinction at elevated layers to the paper (new appendix A6). We found that the effect of elevated layers on the $O_4$ (d)AMF is rather small.

4) We agree the present solution is consistent with the measurements. But we also note that the information available (see below) is indicative of an aerosol layer aloft being missed by the ceilometer on 8 July. Could an aerosol layer aloft explain the need for CFO4 on 8 July?
Summary of results in Ortega et al 2016:
- there is no issue with the O4 measurements.
- neglecting layers aloft biases the RTM low for low elevation angles.
- overestimating layers aloft biases the RTM high for low elevation angles.
- there is no bias if the correct profile is represented as input to the RTM.
By extension, we expect there to be a layer aloft on 8 July, but not on 18 June. The higher Angstroem exponent on 8 July is consistent with this expectation, and warrants further discussion in the paper. The below points 7-9 further elaborate on this point.

We now discussed the effect of elevated layers in detail in several parts of the paper, see also the detailed response below.

5) The abstract highlights the "need for more detailed independent aerosol measurements" (line 62). We agree. Consider adding the following text to summarize existing literature on this point: Introduction: "Optical closure studies based on measured O4 SCDs show excellent agreement also in presence of aerosols, when detailed information from independent aerosol measurements is available. Specifically, the aerosol extinction profile inferred from altitude resolved AMAX-DOAS O4 observations agrees very well, and quantitatively, with the aerosol extinction profile measured by collocated airborne High Spectral Resolution Lidar (HSRL) (Volkamer et al. 2015). The HSRL wavelength (532nm) in this study closely resembled that of the O4 wavelength (477nm). HSRL is highly sensitive to quantify aerosols even at the low aerosol extinction in the free troposphere. The existence of elevated aerosol layers has been suggested as a possible explanation for CFO4 (Ortega et al. 2016). Furthermore, when the size distribution and refractive index of aerosols are actively controlled, CE-DOAS O4 observations infer aerosol extinction values that agree very well with Mie calculations of the extinction constrained by these known aerosol properties (Thalman and Volkamer, 2010). Any scaling of the measured O4 SCD by CFO4 smaller unity would lead to systematic high bias in the inferred extinction in either of these studies, which is not observed. These studies thus provide strong evidence from field studies and laboratory experiments, that there is no fundamental limitation to use O4 SCDs to infer aerosol extinction."
Some language on the relation between microphysical properties and macroscopic scattering and extinction is needed to make this point more clearly. Of particular relevance is the aerosol size distribution. At the moment, the only mentioning of the aerosol size distribution is on line 964 in relation to g on 18 June. However, the size distribution strongly impacts g and the Angstroem exponent. In fact, the aerosol size distribution as the underlying property that controls and probably explains differences in the Angstroem exponent on 8 July seems very relevant. This is currently missing, and worth mentioning.

See detailed reply below.

6) line 116: "… similar to Ortega et al. 2016… "– given a similar approach is used, how do the results compare? Some discussion seems appropriate, and is currently missing in Sections 4.1 and 4.2. For example, Ortega et al was the first study to my knowledge that systematically looked at the issue of using realistic density profiles vs a US Standard atmosphere.
Consider adding in Section 4.1: "Ultimately, the accuracy with which O4 concentrations can be calculated is limited by the assumption that O2-O2 is pure collision induced absorption (CIA). If the oxygen concentration profile is well known, the uncertainty due to bound O4 is smaller 0.14% in Earth's atmosphere (Thalman and Volkamer, 2013). By comparison, deviations of air density from the US standard atmosphere (pressure, temperature and humidity) can lead to errors of 15-18% in estimated O4 concentrations (Ortega et al. 2016). Here we investigate different extraction methods… "

At the end of section 4.1 we added the reference to Ortega et al., 2016.
We also added the following text: Ultimately, the accuracy with which $O_4$ concentrations can be calculated is limited by the assumption that $O_4$ ($O_2$-$O_2$) is pure collision induced absorption. If the oxygen concentration profile is well known, the uncertainty due to bound $O_4$ is smaller 0.14% in Earth's atmosphere (Thalman and Volkamer, 2013).

7) line 527 ff: This is a key paragraph in my opinion. We agree with the listed assumptions, but am missing discussion of the uncertain wavelength scaling. In other words, how certain can you be that the profile shape measured at 1064nm actually resembles the profile shape that drives AOD at 360nm? The ceilometer wavelength is sensitive primarily to larger particles, has limited sensitivity aloft, and could miss smaller particles that are expected more abundant aloft. At the same time, smaller particles contribute more effectively to AOD at 360nm (Q factor due to Mie resonances). In this context, it is interesting to note that the Angstroem exponent on 8 July is rather large (Fig. 1C). We suspect that the ceilometer data is less representative of the actual aerosol profile shape at 360nm on 8 July. And that the profile shape from the ceilometer is actually a better proxy at 360nm on 18 June due to the lower Angstroem exponent on that day.  Uncertain wavelength scaling being more important on 8 July seems relevant to the discussion of Section 4.2, and also in Section 5.1, b) aerosol properties.

This is a very good and important remark. We investigated this effect and found that – surprisingly - the ceilometer measurements are actually a better proxy for the aerosol extinction profile on 8 July than on 18 June. The corresponding calculations are added at the end of appendix A5.

8) Section 5.1, and Fig. 1: The aerosol backscatter from the ceilometer indicates lower boundary layer height on July 8 (the day with low PM load). In our opinion, the discrepancy in in-situ PM between both days is indicative that small particles need to be above the boundary layer in order to explain similar AOD [If the boundary layer is shallower and the PM load measured at the surface is lower, than the total aerosol load in the boundary layer is lower by extension. Despite this, July 8 has a similar total AOD. Without aerosol aloft, this would require a greater extinction per unit mass for the boundary layer.] We suspect the non-zero aerosol extinction retrieved consistently at high altitudes on July 8 (in contrast to June 18), coupled with the higher Angstroem exponent that day, provide two important clues to resolve the apparently inconsistent conclusions regarding CF04 on both days. In our opinion the origin for the significant CFO4 on July 8 lies in the shape of the aerosol profile from the ceilometer being propagated as seen in Fig. 8. Note that the ceilometer is not sensitive to smaller particles.

Again we refer to the detailed calculations which were added to appendix A5 of the revised version of the manuscript, which is also copied at the end of this file.
We also added the following text at the end of section 4.2.1:
Finally, we investigated the effect of changing aerosol optical properties with altitude (changing LIDAR ratio). Such effects are in partivular important if the wavelength of the ceilomter measurements (1020 nm) differs largely from that of the MAX-DOAS observations (360 nm). Based on the partitioning in fine and coarse mode aerosols derived from the sun photometer observations, as well as the corresponding phase functions and optical depths, the sensitivity of the ceilometer to fine mode aerosols can be estimated (for details see appendix A5). While for 18 June the contribution of the fine mode to the ceilometer signal is about 32% on 8 July it is much larger (about 82 %). Thus it can be concluded that the aerosol extinction profile derived rrom the ceilometer is largely representative for the fine mode aerosols on that day. Nevertheless, the remaining uncertainties of the aerosol extinction profile at 360 nm together with the assumption that the coarse aerosols are probably located close to the surface led to a repartitioning of parts of the aerosol extinction profile (extracted assuming a constant LIDAR ratio). This repartitioning led to a decrease of the aerosol extinction close to the surface which is balanced by an increase at higher altitudes (see Fig. A34). The O4 dAMFs calculated for the modified profile are by about 15 % larger than those for the standard settings (for details see appendix A5).

Ortega et al emphasizes the importance of the aerosol profile shape for a given AOD. **We concluded in that paper that profile shape uncertainty is actually more important than air density uncertainty as drivers for CFO4 (Section 3.4 in Ortega, and their Fig 5 and Fig 7).** In our opinion, a similar sensitivity study that varies the profile shape at constant AOD is needed here, essentially redistributing a fraction of the AOD to the layer that is visible in the ceilometer data near 7km on 8 July.

We added such sensitivity studies to the revised version of the manuscript (new appendix A6). The results indicate that the effect of redistributing large parts of the aerosol extinction profile leads to rather small changes of the $O_4$ dAMFs (between –7 and +7 %) for rather large changes of the aerosol extinction (+40%) at different layers.

9) In Section 5.1, b) aerosol properties:
- Please add the fraction (%) of the AOD that resides below 1km, between 1-2km, and above 2km for both days.
- It would be interesting to add a few sentences how the aerosol profile shapes compare with the climatology for elevated aerosol layers - see section 3.3 in Ortega for context. How do the days during MADCAT compare with the days studied during TCAP?

Since for the measurements selected in this study elevated layers can not explain the discrepancy between measurements and simulations, we think it is not important to add this information. Nevertheless, we compared the aerorosl profiles for the selected days with those in Ortega et al. (2016). At the end of section 4.2.1 the following text was added:
The effect of elevated aerosol layers (see Ortega et al., 2016) was further investigated by systematic sensitivity studies (appendix A6). On both selected days enhanced aerosol extinction was found at elevated layers (Fig. 9). Compared to those reported by Ortega et al. (2016) the profiles extracted in this study reach even up to higher altitudes. For the investigation of the effect of changes of the aerosol extinction at different altitudes, the aerosol extinction profile on 8 July was subdivided into 3 layers (0-1.7 km; 1.7 – 4.9 km; 4.9 – 7 km), and the extinction in the individual layers was increased by +20% or + 40 %. It was found that even a strong increase of the aerosol extinction at high altitudes by 40% leads only to an increase of the O4 dAMFs by 7 %. Here it should be noted that on 8 July no indications for such a strong underestimation of the aerosol extinction at high altitudes are found.

**D) Influence of a changing LIDAR ratio with altitude**

For the extraction of the aerosol profiles described above, a fixed LIDAR ratio was assumed, which implies that the aerosol properties are independent from altitude. However, this is a rather strong assumption, because it can be expected that the aerosol properties (e.g. the size) change with altitude. With the available limited information, it is impossible to derive detailed information about the altitude dependence of the aerosol properties, but it can be how representative the ceilometer measurements at 1020 nm are for the aerosol extinction profiles at 360 nm. For these investigations we again focus on the middle periods of both selected days. From the AERONET Almucantar observations information on the size distribution for these periods is available (see Fig. A32). On both days two pronounced modes (fine and coarse mode) are found with a much larger coarse mode fraction on 18 June compared to 8 July. From the AERONET observations, also separate phase functions for the fine and coarse mode as well as the relative contribution of both modes to the total aerosol optical depth at 500 nm are available. On 18 June and 8 July the relative contributions to the total AOD at 500 nm are 40 % and 5 %,

respectively. Assuming that the AOD of the coarse mode fraction is independent on wavelength, the relative contributions of the coarse mode at 360 nm and 1040 nm can be derived (see Table A27).

[Figure]

Fg. A32 Size distributions derived from AERONET Almucatar observations on 18 June (07:24 & 15:34) and 08 July (07:32 & 15:38).

Table A27 Contribution of the coarse mode to the total AOD at different wavelengths

| Date | Total AOD 360 nm | Total AOD 1020 nm | Relative contribution of coarse mode 360 nm | Relative contribution of coarse mode 1020 nm |
|---|---|---|---|---|
| 18 June, 11:00 – 14:00 | 0.37 | 0.12 | 24.9% | 77.7% |
| 08 July, 07:00 – 11:00 | 0.33 | 0.055 | 3.0% | 18.1% |

It is found that on 18 June the coarse mode clearly dominates the AOD at 1020 nm, whereas on 8 July it only contributes about 20 % to the total AOD. As expected the relative contributions of the coarse mode to the AOD at 360 nm are much smaller (25 % and 3%).

In the last step the probability of aerosol scattering in backward direction is considered, because the ceilometer receives scattered light from that direction. For that purpose the ratios of the optical depths are multiplied by the corresponding values of the normalised phase functions at 180° and in this way the relative contributions to the backscattered signals from the coarse mode for both wavelenghs and both days are calculated (Table A28). Interestingly, on 8 July the contributions of the coarse mode to the backscattered signal at both wavelengths differs only by about 10%. In contrast, on 18 June the difference is much larger.

Table A28 Ratio of phase functions (coarse / fine) in backward direction and relative contribution of coarse mode to the backscattered signal at both wavelengths

| Date | Ratio phase function at 360 nm | Ratio phase function at 1020 nm | Relative contribution of coarse mode at 360 nm | Relative contribution of coarse mode at 1020 nm |
|---|---|---|---|---|
| 18 June, 11:00 – 14:00 | 1.13 | 0.61 | 27.3% | 68.0% |
| 08 July, 07:00 | 2.7 | 0.99 | 7.8% | 18.0% |

| – 11:00 | | | | |
|---------|--|--|--|--|

For 8 July, the results can be interpreted in the following way: at 360 nm the aerosol profiles extracted as described above overestimate the contribution from the coarse mode by about 10%. To estimate the effect of this overestimation we construct modified aerosol extinction profiles, in which 10% of the total AOD is relocated. Since we expect that the coarse mode aerosols are usually located at low altitude, we construct 4 different modified profiles (see Fig. A33) with different altitudes (1.5 km, 1 km, 0.75 km, or 0.5 km), below which 10% of the aerosol extinction is relocated to altitudes above (assuming that the coarse mode aerosol is only located below these altitudes). Of course, such a sharp boundary is not very realistic, but it allows to quantify the overall effect of the relocation. Here it should be noted that we selected the aerosol profile for 8 July extracted by INTA which reached up to 7 km (see Fig. 9).

[Figure]

Fig. A33 Left: Modified aerosol profiles for 08 July assuming that the coarse mode aerosol is only located in the lowest part of the atmosphere. Top right: ratios of the (d)AMFs calculated for the modified profiles compared to the dAMFs for the standard settings. With decreasing layer height the (d)AMFs increase systematically, because the aerosol extinction close to the surface decreases. Righ bottom: comparison of the measured elevation dependence of the $O_4$ dAMFs for the period 7:00 – 11:00 on 8 July and simulation results for the different profiles.

Table A29 Ratio of the (d)AMFs for the modified profiles versus those of the standard settings

| | original | coarse   mode | coarse   mode | coarse   mode | coarse   mode |
|--|----------|-------------|-------------|-------------|-------------|

|  | INTA | below 1.5 km | below 1 km | below 0.75 km | below 0.5 km |
|---|---|---|---|---|---|
| AMF | 1.02 | 1.04 | 1.05 | 1.06 | 1.08 |
| dAMF | 1.04 | 1.09 | 1.13 | 1.17 | 1.18 |

For all modified profiles, a systematic increase of the $O_4$ (d)AMFs compared to those for the standard settings is found. For the $O_4$ dAMFs this increase can be up to 18 % (see Table A29. From the comparison of the elevation dependence of the measured and simulated $O_4$ dAMFs (see Fig. A33), we conclude that the aerosol profile with the coarse mode aerosol below 0.75 km is probably the most realistic one. The main conclusion from this section ist that the dAMFs for 8 July derived from the standard settings probably underestimates the true dAMF by about 15 $\pm$5 %.

For 18 June we did not perform similarly detailed calculations, because on that day the uncertainties of the aerosol extinction profile caused by the missing sensitivity of the ceilometer below 180 m are much larger than on 8 July. On 18 June also the magnitude of the relocation of the aerosol extinction between different altitudes would be much larger than on 8 July.
* * *
Email from Rainer and Ted, 02.02.2019
* * *
Lieber Thomas,

thank you very much for the revised manuscript. The additions seem promising, but in our opinion there is one additional sensitivity study that warrants to be added in Appendix 6. Can you please add a case, where you redistribute 10% of the AOD on 8 July from the lowest layer to the upper layer. Specifically, the partial column AOD would look as follows:

| Alt_low | Alt_up | Ext_INTA | Ext_redist |
|---|---|---|---|
| 0 | 1.68 | 0.186 | 0.1523 |
| 1.68 | 4.9 | 0.116 | 0.116 |
| 4.9 | 7 | 0.035 | 0.0687 |

Please also include the EA dependence of the O4 dAMF in Figure A34.

And since the Figure is already quite busy, it would be helpful to digest the results for the O4 AMFs, and O4 dAMFs in form of a new Table A31.

We believe that the addition of the size distribution from both days is a key improvement to the paper. I have discussed with Ted, and we are not sure that we agree with everything that is said in Appendix 5. But in essence we agree with the main conclusion, that the profile shape at 1020nm is a poor indicator for that at 360nm. Following your argument in Table A28, up to 40% of the AOD should be able to redistribute (18 June). This does not seem to be sensible in light of the size distributions in Fig. A32, which support the mismatch in ceilometer wavelengths is much less of an issue on 18 June than on 8 July. This does not yet appear to transpire from the discussion in Appendix 5. But I hope the above sensitivity study will help move that discussion along.

We will send more detailed comments in due time, but wanted to get back to you in a timely manner.

Thanks for all your efforts, and I hope this email finds you well.

Best wishes,
-Rainer & Ted

Email from Thomas, 03.02.2019

Dear Rainer and Ted,

many thanks for your feedback and further suggestions. Please find my detailed replies below.

> thank you very much for the revised manuscript. The additions seem promising, but in our opinion there is one additional sensitivity study that warrants to be added in Appendix 6. Can you please add a case, where you redistribute 10% of the AOD on 8 July from the lowest layer to the upper layer. Specifically, the partial column AOD would look as follows:

| Alt_low | Alt_up | Ext_INTA | Ext_redist |
|---------|--------|----------|------------|
| 0 | 1.68 | 0.186 | 0.1523 |
| 1.68 | 4.9 | 0.116 | 0.116 |
| 4.9 | 7 | 0.035 | 0.0687 |

Good point! The simulations were done, please see the results below. The ratios of dAMFs for the new profile ('from bottom to top') are similar to those of the profile '< 0.75 km'. This information is now mentioned in the manuscript in appendix A5

[Figure]

[Figure]

Please also include the EA dependence of the O4 dAMF in Figure A34.

The elevation dependence was added, see figures below:

[Figure]

[Figure]

And since the Figure is already quite busy, it would be helpful to digest the results for the O4 AMFs, and O4 dAMFs in form of a new Table A31.

The Table was added:

Table A31 Ratios of (d)AMFs for 8 July 2013 for the modified profiles with respect to the original profile

|  | low 140 % | middle 140 % | top 140 % |
|---|---|---|---|
| ratio AMF without scaling | 0.95 | 1.03 | 1.03 |
| ratio dAMF without scaling | 0.85 | 1.02 | 1.02 |
| ratio AMF with scaling | 1.00 | 1.06 | 1.04 |
| ratio dAMF with scaling | 0.94 | 1.08 | 1.04 |

We believe that the addition of the size distribution from both days is a key improvement to the paper. I have discussed with Ted, and we are not sure that we agree with everything that is said in Appendix 5. But in essence we agree with the main conclusion, that the profile shape at 1020nm is a poor indicator for that at 360nm. Following your argument in Table A28, up to 40% of the AOD should be able to redistribute (18 June). This does not seem to be sensible in light of the size distributions in Fig. A32, which support the mismatch in ceilometer wavelengths is much less of an issue on 18 June than on 8 July. This does not yet appear to transpire from the discussion in Appendix 5. But I hope the above sensitivity study will help move that discussion along.

We will send more detailed comments in due time, but wanted to get back to you in a timely manner.

Could you already estimate when you will send me the more detailed comments? The current deadline is on 4 February. I will ask for a further extension today.

Many thanks!

Thomas
* * *
Email from Rainer, 03.02.2019

Thanks Thomas,

A quick clarification & request:
Please add the results from the new profile shape also on Table A31 and Fig A34 for direct comparison with those data. That would be most helpful.
Please also comment on our point regarding your argument supporting why redistribution of extinction should be limited to 10% (and not 40%? or 80%?)...
It seems that by redistributing only 10% of the partial AOD to higher altitudes, about half (or slightly more) of the correction factor is explained on 8 July. The effect of adding extinction aloft (while keeping AOD constant) increases the dAMF in the higher EAs,  while the opposite is observed in the lower EAs if extinction is added near the surface. After scaling to normalize AOD, adding extinction aloft is relatively more efficient at closing the gap. The latest profile shape is in fact the closest of all dAMFs, better even than the 0.75km case for EA ~4.5 and larger.
This bears the question then: How much AOD would need to be redistributed on 8 July in order to obtain closure?
We will make it a priority to get back to you, but asking for a further extension seems a good idea.
Greetings on a sunny Sunday morning - it's Superbowl in the NFL today - Patriots vs LA RAMs. Meaning the city, trails and slopes should be empty this afternoon... ;)

-Rainer
* * *
Emai from Thomas, 04.02.2019

Dear Rainer,

from table A28 it is found that for 8 July the coarse mode fraction contributes 18 % to the ceilometer signal at 1020 nm, while it would contribute 8 % to a ceilometer signal at 360nm.
That means there is a difference of 10% of the total AOD measured at 1020 nm, which would not be seen at 360 nm. These 10% of the coarse mode contribution could be anywhere in the atmospheric column, but most probable close to the surface. This is also supported by the elevation dependence. The elevation dependence for the profile where 10% from below 750 m is relocated to above 750 m fits best to the measurements. Results for relocations from below 500m to above 500m also those for relocation from below 1.68 km to above 4.9 km (your suggestions) don't fit. I hope this makes the argument more clear.
You are right that the results for the latest profile shape is in fact the closest of all dAMFs, better even than the 0.75km case for EA ~4.5 and larger. However, the complete elevation dependence does not fit to the measurements.
There is another point: you wrote: It seems that by redistributing only 10% of the partial AOD to higher altitudes, about half (or slightly more) of the correction factor is explained on 8 July. This is not really

true. The ratio between measurements and simulations (without the relocation) is 0.71. If it is multiplied by 1.15 one gets: 0.82.
I will add the results for the new profile shape to Table A31 and Fig A34

Best regards,

Thomas

Email from Rainer, 05.02.2019

Hi Thomas,

can you send us the McArtim files and O4 measurements for both days?

We probably want to run some simulations here ourselves to have an effective conversation. McArtim3 is also what we usually use. Its a good exercise for Chris (cc here) to setup calculations for the 18.6. and 8.7 case study days and inform further discussions.

Please also send the size distribution files from Aeronet, and any info on the complex refractive index (and its variation with size if available).

Maybe I am missing something, but Table A28 discusses % units of coarse mode contributions to backscatter signal, which is not the same as % units AOD. I appreciate what you are trying to do here. But if your argument is applied to 18 June data, 40% of AOD can be redistributed at 360nm on 18 June (change from 27.3 to 68%). This is four times larger flexibility to redistribute extinction, and would be at odds with the primary message that I take away from Tables A27 & A28, which is that 1) there is less of a need to extrapolate wavelength on 18 June than on 8 July, 2) aerosol profiles at 1020nm make a relatively larger contribution to control extinction also at 360nm on 18 June than 8 July, and 3) the fact that no correction factor is needed on 18 June.

It will help to have the data to play with it... thanks for you soon response.

Thanks,
-Rainer

Email from Thomas, 05.02.2019

Dear Rainer,

please find my response to your points below:

On 05.02.2019 01:30, Rainer Volkamer wrote:
        Hi Thomas,

can you send us the McArtim files and O4 measurements for both days?
We probably want to run some simulations here ourselves to have an effective conversation. McArtim3 is also what we usually use. Its a good exercise for Chris (cc here) to setup calculations for the 18.6. and 8.7 case study days and inform further discussions.
Please also send the size distribution files from Aeronet, and any info on the complex refractive index (and its variation with size if available).

All input data of the first comparison round are available at the MADCAT web page, see http://joseba.mpch-mainz.mpg.de/Comparison.htm
Additional AERONET inversion data were provided in my email from 10 May 2017 (including you as an addressee). I will re-send this email again in the next minute. Please let me know if you need something else.
However, I don't want to wait for the results of these additional simulations for the paper to be submitted. Detailed comparison studies between different RTM were already performed and are an important part of the paper. Also many sensitivity studies covering a large variety of settings (including your recent suggestions) were performed and are an important part of the paper. You were always included in the respective emails, but I never got feedback from your group during the last two years. Additional RTM exercises will lead to further delays of the paper, but one can not expect significantly new findings.
The present study is not very conclusive, and the question about a scaling factor can not be answered. More future comparison studies will be needed to address this issue (as stated in the paper).
Thus I want to ask you if you (and Ivan and Ted) can agree to become co-authors of the paper in its current form. If not, I will mention your contributions to the paper in the acknowledgments. Please send me your feedback within the next days.

Maybe I am missing something, but Table A28 discusses % units of coarse mode contributions to backscatter signal, which is not the same as % units AOD.

Yes, this is true. And this is the reason why the respective fractions are compared for both wavelengths. If the fractions were exactly the same, the ceilometer measurements at 1020 nm would be perfectly representative for the aerosol profile at 360 nm. If they were different by 100%, then from the ceilometer measurements at 1020 nm no information about that at 360 nm could be retrieved.

I appreciate what you are trying to do here. But if your argument is applied to 18 June data, 40% of AOD can be redistributed at 360nm on 18 June (change from 27.3 to 68%). This is four times larger flexibility to redistribute extinction, and would be at odds with the primary message that I take away from Tables A27 & A28, which is that 1) there is less of a need to extrapolate wavelength on 18 June than on 8 July, 2) aerosol profiles at 1020nm make a relatively larger contribution to control extinction also at 360nm on 18 June than 8 July, and 3) the fact that no correction factor is needed on 18 June.

On 18 June, indeed the ceilometer measurement is much less representative for the aerosol extinction at 360 nm. This is a surprising finding, but by using the detailed AERONET inversion products, this it what
* * *
Email from Ivan, 05.02.2019

Dear Thomas,
I should check this email more often, I am sorry I have missed discussions about your analysis. I am still trying to catch up with everything. My very initial input was about what Rainer just suggested. From the very beginning I noticed that you use similar extinction values in the boundary layer for both days, even larger extinction on July 8, although it is clear that the surface mass loading is significantly lower on July 8. I also have notice that you use zero extinction above 6-7 km or so, but in reality there might be some extinction, maybe assuming 1x10-3 or so might be more realistic, maybe is not important?. I am still trying to understand why the correction factor is not needed in one day and needed on another day. I think either the state of the atmosphere is not well characterized yet when correction factor is needed or purely luck when the correction is not needed.

Thanks for all the hard work getting this manuscript out.

Best,

Ivan
* * *
Email from Rainer, 05.02.2019

Dear Thomas,

thanks for re-sending the link to the files. I understand your desire to wrap this up.

However, I still see a disconnect between the in-situ PM mass loadings and near surface extinction values. I see a strong motivation to digest this relevant information in form of a further sensitivity study. The settings are in the last column of the below table:

| Alt_low | Alt_up | Ext_INTA | Ext_redist_10% | Ext_redist_30% |
|---------|--------|----------|----------------|----------------|
| 0 | 1.68 | 0.186 | 0.1523 | 0.085 |
| 1.68 | 4.9 | 0.116 | 0.116 | 0.116 |
| 4.9 | 7 | 0.035 | 0.0687 | 0.136 |

The issue is that PM10 and PM2.5 mass loadings near the surface in Fig. 1D are significantly (factor 2) lower on 8 July compared to 18 June, while the extinction profiles in Fig. 9 have very

similar extinction near the surface. The additional sensitivity study is needed to make the interpretation of the overall dataset more coherent. At the same time, I expect an improvement in the dAMFs for all EAs, based on the results of the first new profile shape. This would be significant!

I was trying to help save you time by offering to involve Chris. But after giving it some thought, the above case is probably sufficient to finalize our thinking about this paper.

If you agree, please add the results into Table A31 and Fig A34, and also archive the AMFs and dAMFs for all data in Fig. A34 (all EAs) in form of a new Table (similar to Table A29). I liked to see the results before sending detailed comments, as I certainly see potential to "expect significantly new findings" from these RTM calculations.

I have a dental procedure tomorrow, but should be back online on Friday. If we see the revised manuscript with the above changes by then we should be able to send our detailed comments by early next week. Sound good?

Regards,
-Rainer

Email from Thomas, 06.02.2019

Dear Rainer,

the differences in the in situ pm measurements on both days are in fact an important point. However, since the ceilometer is blind below 180m it is difficult to make a direct connection between the ceilometer measurements and the in situ data. Here the AERONET inversion products become important. From the AERONET inversion products for 8 July it is found that 10% of the total integrated extinction (but not 30%) 'could' be redistributed.

Your suggested case of a redistribution from the lowest layer to the upper layer of 30% can indeed bring measurements and simulations into closer agreement, see figure below (in fact a redistribution of about 28 % (not shown) would lead to an even better agreement. I think it makes sense to add this information to the paper.

Nevertheless, the AERONET inversion products don't support the assumption of a re-distribution of 30% of the total AOD from the lowest to the highest layer. Remember that only 18% of the total ceilometer signal is caused by the coarse mode aerosols.
Concerning the differences between both days, I think there is a simple explanation: since the ceilometer is blind below 180 m, it is very probable that on 18 June a much higher concentration of coarse aerosols exists below 180m. Note especially the large amount of pm10 on that day, which is unlikely to be lifted up to high altitudes. The assumption of large aerosol extinction below 180m on 18 June would also lead to an underestimation of the measured $O_4$ (d)AMFs by the simulations (see Fig. A4 and Fig. A34). But this is of course a speculation and can not be further quantified based on the existing measurement data.
As stated above, for me it makes sense to include a) the results for the modified profile (relocation of

28%) to the paper, and b) mention the fact that such an assumption is not supported by the AERONET inversion results.
I hope you can agree to this procedure.
And don't forget, this paper will not explain the world. One important aim of the study is to provide guidelines to improve further comparison studies.

Best regards,

Thomas

[Figure]

Email from Rainer, 06.02.2019

Dear Thomas,

a potential temperature profile should be able to tell the height of the first inversion. A significant gradient below 200m would lead to a significant gradient in the measured O4 dAMFs between the 1 and 3EA, while in fact both angles show near identical dAMFs. I do not see any

evidence to support a strong gradient below 180m, certainly not on 8 July (Fig 2), but also not really on 18 June. It is reasonably easy, and worth corroborating this point by calculating potential temperature profiles for both days.

I do not have an issue with the calculations shown in Appendix 5D. But I do not follow relevancy for redistribution of AOD at 360nm. Your argument equates a %signal contribution of the coarse mode with a %AOD redistribution. Why the focus on the coarse mode in the first place? Large particles are more likely near the surface (some may be aloft). Aloft its likely fine particles. If I was to take a guess on what size particles is responsible for "redistribution of AOD" its the fine particles. Quantifying detector signal from the coarse mode carries no information about how aerosols are distributed, or can be redistributed. In my opinion, Table A28 should be constructed from a perspective of fine particles, as they dominate optical properties at 360nm and AOD.

All that Appendix 5D is saying in my opinion, is that on either day small particles contribute to the ceilometer signal at 1064nm. And that the contribution from small particles to the AOD at 360nm is sufficient to justify redistribution of 30% AOD. In particular, the contribution of fine particles to the AOD at 360nm is 73% (18 June) and 92% (8 July) at 360nm (ignoring caveats from the lack of knowledge about wavelength dependent refractive index, questions about whether Mie theory applies, etc).

The wavelength extrapolation adds uncertainty to the profile shape at 360nm. This is significant, since the Mie resonances at 360nm are likely very different than at 1064nm. Meaning that a ceilometer measurement at 360nm would look very different. One would need to know the wavelength-, size- and altitude- resolved refractive index to relate an extinction profile from 1064nm to 360nm. In lack of that information, the profile shape measured at 1064nm is a crude guess on that at 360nm. I have to respectfully disagree that "From the AERONET inversion products for 8 July it is found that 10% of the total integrated extinction (but not 30%) 'could' be redistributed."

A minor point: All calculations in Appendix 5 should be done at 1064nm, with the AOD from 1020nm extrapolated to 1064nm. There is confusion also in other parts of the paper about the 1020/1064nm wavelengths pair (it certainly confused me at first). But profile information is extrapolated from 1064 to 360nm, not 1020 to 360nm. Not a biggie in the big picture, but there is some confusion here.

I wanted to get back to you before my dentist knocks me out for the rest of the day. And I look forward to seeing the potential temperature profiles, if available, and the revised manuscript. Ivan, Ted, Chris - feel free to add to this.

Cheers,
-Rainer

Email from Thomas, 06.02.2019

On 06.02.2019 18:52, Rainer Volkamer wrote:

> Dear Thomas,
>
>
> a potential temperature profile should be able to tell the height of the first
> inversion. A significant gradient below 200m would lead to a significant gradient
> in the measured O4 dAMFs between the 1 and 3EA, while in fact both angles
> show near identical dAMFs. I do not see any evidence to support a strong gradient
> below 180m, certainly not on 8 July (Fig 2), but also not really on 18 June. It is
> reasonably easy, and worth corroborating this point by calculating potential
> temperature profiles for both days.

Unfortunately, there is no potential temperature profile data available. ECMWF data are at a rather coarse grid and are thus not representative for the local conditions.
On 18 June the dAMFs for 1° are smaller than for 3° and 6°. This indicates that high aerosol extinction is located close to the surface. But as mentioned yesterday, this is a speculation and can not further be quantified based on the available data.

> I do not have an issue with the calculations shown in Appendix 5D. But I do not
> follow relevancy for redistribution of AOD at 360nm. Your argument equates a
> %signal contribution of the coarse mode with a %AOD redistribution. Why the
> focus on the coarse mode in the first place? Large particles are more likely near
> the surface (some may be aloft). Aloft its likely fine particles. If I was to take a
> guess on what size particles is responsible for "redistribution of AOD" its the fine
> particles. Quantifying detector signal from the coarse mode carries no information
> about how aerosols are distributed, or can be redistributed. In my opinion, Table
> A28 should be constructed from a perspective of fine particles, as they dominate
> optical properties at 360nm and AOD.

Both perspectives (from coarse or fine mode aerosols) are equivalent. Taking the 'fine mode perspective', on 8 July the fine mode contributes 82% to 1064 nm and 92% to 360 nm.
The difference of 10% (either 92% - 82% or 18 % - 8%) is what matters.

> All that Appendix 5D is saying in my opinion, is that on either day small particles
> contribute to the ceilometer signal at 1064nm. And that the contribution from
> small particles to the AOD at 360nm is sufficient to justify redistribution of 30%
> AOD. In particular, the contribution of fine particles to the AOD at 360nm is 73%
> (18 June) and 92% (8 July) at 360nm (ignoring caveats from the lack of
> knowledge about wavelength dependent refractive index, questions about whether
> Mie theory applies, etc).

Still, on 8 July the difference is 10%.

The wavelength extrapolation adds uncertainty to the profile shape at 360nm. This is significant, since the Mie resonances at 360nm are likely very different than at 1064nm. Meaning that a ceilometer measurement at 360nm would look very different. One would need to know the wavelength-, size- and altitude- resolved refractive index to relate an extinction profile from 1064nm to 360nm. In lack of that information, the profile shape measured at 1064nm is a crude guess on that at 360nm. I have to respectfully disagree that "From the AERONET inversion products for 8 July it is found that 10% of the total integrated extinction (but not 30%) 'could' be redistributed."

I still think my arguments are correct.
There is another indication that the '30% redistribution profile' does not fit to the measurements. I compared the measured diurnal variation of the O4 SCDs at 90° elevation to the simulations (see below). For the '30% redistribution profile' too much light is scattered from high altitudes leading to smaller O4 AMFs for high SZA.

A minor point: All calculations in Appendix 5 should be done at 1064nm, with the AOD from 1020nm extrapolated to 1064nm. There is confusion also in other parts of the paper about the 1020/1064nm wavelengths pair (it certainly confused me at first). But profile information is extrapolated from 1064 to 360nm, not 1020 to 360nm. Not a biggie in the big picture, but there is some confusion here.

This is correct, and I also discovered this slight inconsistency today. It will of course be corrected in the final version. However, it is a very small effect. The coarse mode contribution on 8 July changes from 18.1 to 18.3 %. On 18 June, it is completely negligible.

I wanted to get back to you before my dentist knocks me out for the rest of the day. And I look forward to seeing the potential temperature profiles, if available, and the revised manuscript. Ivan, Ted, Chris - feel free to add to this.

It seems to me that we will not come to an agreement about the aerosol profiles. Therefore I suggest that the editor and reviewers should decide. Of course your comments and my replies will be made available to them and also be made available for the public at the discussion site. If you still want to send me your detailed comments, they are still welcome.

Many thanks,

I hope the surgery at the dentist will not be too painful and you will recover soon.

Thomas

[Figure]
* * *
email from Ted, 07.02.2019

Hello Thomas,

I present my thinking at more length below, but skipping to the conclusions: the SZA dependence highlights the 30% redistribution is not ultimately fully consistent, and I further suspect that a fully consistent solution is not easily found. Rather, these sensitivity studies can be presented and framed to highlight that discrepancies between modeled and measured O4 can be explained by such changes which cannot be ruled out by available data. This points to the potential to leverage the angle specific O4 dAMFs and SZA dependent dAMFs in conjunction with certain assumptions to make adjustments and perhaps reach a fully consistent solution. That is beyond the scope of the paper. Still, I think the sensitivity studies can highlight that while such a exercise is challenging, at present poorly constrained, and perhaps impractical, it is not impossible.

I outline my thinking below, my apologies for the length:

As I understand the analysis in the manuscript, an angstom exponent was derived for a given point in time, and then applied to the entire ceiliometer profile. The extinction for monodisperse large particles is relatively flat with wavelength, whereas for smaller particles the extinction changes more rapidly with wavelength. However, the coarse fine dichotomy is not the only concern, see for instance Schuster et al., 2006. While the precise size distribution of coarse mode aerosol does not change the angstrom exponent, the specifics of particle distributions in the fine mode can act as a strong lever on the angstrom exponent.

The absolute contribution of fine mode aerosol provides some measure of the expected inaccuracy of adapting an extinction profile from a different wavelength. I don't think that the difference in the relative contribution of the aerosol modes to the different measurements is a relevant metric for this effect. Unfortunately I don't have a firm constraint to offer beyond the fact that it should be more important when more aerosol volume is in the fine mode.

In this context it is not surprising that June 18 has a correction factor closer to 1 than July 8, because overall the aerosol are larger and therefore a constant angstrom exponent with altitude is more likely to be closer to the truth. For July 8, while aerosol size distribution profiles in the atmosphere are complex they generally tend to get smaller and narrower rising through the troposphere. Both these effects increase the angstrom exponent and as such there is expected to be a general tendency that when transferring an extinction profile from longer to shorter wavelengths that it will be relatively enhanced at higher altitudes. Atmospheric layering of course also plays a role.

The sensitivity studies in Appendix 6 are therefore consistent with expected effects in the absence of better constraints highlighting layers especially. I cannot offer a corollary to your ~10% bound, but I don't believe the ~30% effect can be completely rejected either. If another sensitivity study can further illustrate the principle while ignoring any layers, a naive smooth altitude dependence of scaling might serve i.e multiply the extinction profile by [(1-x) + (2x/7 km-1) * altitude], where x between between 0.1 and 0.3 should serve to illustrate.

Please let me know if there is anything compelling which I am overlooking. Perhaps we will not reach agreement. In any case, I would appreciate your thoughts on this perspective and framing, it would be useful in determining some of the specific comments. Thank you for bringing this extensive exercise together and for your responsiveness these last days.

Best Wishes,

Ted
* * *
Email from Thomas, 07.02.2019

Dear Ted,
many thanks for your feedback!
I want to clarify my general view: I think we all agree that there are uncertainties about the aerosol extinction profiles.
To decide which extinction profile might be the most probable, we can use the following information:
a) the ceilometer data and the AERONET inversion products
b) the elevation dependence
c) the SZA dependence
In addition to these observations, we can assume that coarse mode aerosol is probably located at lower altitudes than the fine mode aerosols.
All of these observations and assumptions have their uncertainties. Nevertheless, taking all information into account, I conclude that the scenario of a 10% redistribution is the most probable.
The results of the sensitivity studies for the different profiles and their compatibility with the above stated observations and assumptions will of course be provided in the paper.
Then not only the editor and the reviewers, but also the readers can reach their own conclusion on what they think is most probable.

I hope you can agree to that procedure.

Please find my response to the individual points below.

Best regards,

Thomas

On 07.02.2019 03:39, Theodore Konstantinos Koenig wrote:

Hello Thomas,

I present my thinking at more length below, but skipping to the conclusions: the SZA dependence highlights the 30% redistribution is not ultimately fully consistent, and I further suspect that a fully consistent solution is not easily found. Rather, these sensitivity studies can be presented and framed to highlight that discrepancies between modeled and measured O4 can be explained by such changes which cannot be ruled out by available data. This points to the potential to leverage the angle specific O4 dAMFs and SZA dependent dAMFs in conjunction with certain assumptions to make adjustments and perhaps reach a fully consistent solution. That is beyond the scope of the paper. Still, I think the sensitivity studies can highlight that while such a exercise is challenging, at present poorly constrained, and perhaps impractical, it is not impossible.

see my general comments above

I outline my thinking below, my apologies for the length:

As I understand the analysis in the manuscript, an angstom exponent was derived for a given point in time, and then applied to the entire ceiliometer profile.

This is not exactly true. The angstrom exponent was determined for the selected period. Also it is not applied to the entire ceilometer profile. The altitude dependence of the size distribution (and thus the angstrom exponent) is implicitly accounted for by the re-distribution of 10% of the total extinction.

The extinction for monodisperse large particles is relatively flat with wavelength, whereas for smaller particles the extinction changes more rapidly with wavelength. However, the coarse fine dichotomy is not the only concern, see for instance Schuster et al., 2006. While the precise size distribution of coarse mode aerosol does not change the angstrom exponent, the specifics of particle distributions in the fine mode can act as a strong lever on the angstrom exponent.

Of course this is true. However, the wavelength dependencies are intrinsically taken into account by the use of the phase functions for fine and coarse mode aerosols derived from the AERONET inversion. This information is not perfect, but describes best the aerosol properties during that day.

The absolute contribution of fine mode aerosol provides some measure of the expected inaccuracy of adapting an extinction profile from a different wavelength. I don't think that the difference in the relative contribution of the aerosol modes to the different measurements is a relevant metric for this effect. Unfortunately I don't have a firm constraint to offer beyond the fact that it should be more important when more aerosol volume is in the fine mode.

Intuitively, I had the same expectations at the beginning. Nevertheless, by taking the optical depths and the phase functions of fine and coarse mode aerosols into account, it turns out that on 8 July even at the rather large wavelength of the ceilometer measurements the fine mode dominates the ceilometer signal (82%). I think this is the key point and tells us that on 8 July the ceilometer measurements at 1064 nm are a very good proxy for the aerosol extinction profile shape at 360 nm.

In this context it is not surprising that June 18 has a correction factor closer to 1 than July 8, because overall the aerosol are larger and therefore a constant angstrom exponent with altitude is more likely to be closer to the truth. For July 8, while aerosol size distribution profiles in the atmosphere are complex they generally tend to get smaller and narrower rising through the troposphere. Both these effects increase the angstrom exponent and as such there is expected to be a general tendency that when transferring an extinction profile from longer to shorter wavelengths that it will be relatively enhanced at higher altitudes. Atmospheric layering of course also plays a role.

Of course, I agree that in general the size distribution varies with altitude. This is what our whole discussion is about. But I think this is the case for both days: We should expect that the size of the aerosols in general decreases with altitude. The important difference is that on 8 July the relative contribution from the coarse mode to the ceilometer signal is much larger than on 18 June which complicates the quantitative interpretation.

The sensitivity studies in Appendix 6 are therefore consistent with expected effects in the absence of better constraints highlighting layers especially. I cannot offer a corollary to your ~10% bound, but I don't believe the ~30% effect can be completely rejected either. If another sensitivity study can further illustrate the principle while ignoring any layers, a naive smooth altitude dependence of scaling might serve i.e multiply the extinction profile by [(1-x) + (2x/7 km-1) * altitude], where x between between 0.1 and 0.3 should serve to illustrate.

Initially, I also had this thought. Such a smooth altitude dependence is surely more realistic than a re-distribution between layers. However, I decided to use the more extreme re-distributions between layers for the sensitivity studies because of two reasons:
a) we have no information on the altitude dependence of the fine and coarse mode fractions. All assumed re-distributions are simply assumptions (of course with some plausibility)

b) from the extreme scenarios the overall magnitude of the effect can be estimated, and that is what matters.

Please let me know if there is anything compelling which I am overlooking. Perhaps we will not reach agreement. In any case, I would appreciate your thoughts on this perspective and framing, it would be useful in determining some of the specific comments. Thank you for bringing this extensive exercise together and for your responsiveness these last days.

My current plan is to prepare an updated version of the manuscript in the next two days and send it to you. If your detailed feedback contains further fundamental points, it would be good to know these points before I prepare the updated version. I want to avoid too many iterations.

Many thanks,

Thomas
* * *
Email from Rainer, 08.02.2019

On 2/7/2019 5:52 PM, Rainer Volkamer wrote:

Dear Thomas,

sorry for another lengthy email. I have added my comments below your initial text to Ted, as well as below your responses to Teds comments:

On 2/7/2019 5:33 AM, Thomas Wagner wrote:

Dear Ted,
many thanks for your feedback!
I want to clarify my general view: I think we all agree that there are uncertainties about the aerosol extinction profiles.
To decide which extinction profile might be the most probable, we can use the following information:
a) the ceilometer data and the AERONET inversion products
b) the elevation dependence
c) the SZA dependence

Missing here is d) near surface PM levels on 18 June are significantly (factor ~2 times?) higher than on 8 July. To reproduce this gradient in surface PM between both days it is necessary to redistribute 30% of the AOD from lower to higher altitudes. I further elaborate on synergies between b and d to inform this below.

In absence of potential temperature profiles, information from b) is helpful to assess whether the near surface PM is expected to be highly localized near the surface, or is indeed representative also at altitudes above ~200m. I elaborate below.

During MADCAT, the effective pathlength of photons at 350nm has been quantified as 7km in the lower angles (Ortega et al. 2015; doi:10.5194/amt-8-2371-2015). This distance corresponds to an altitude of 120m for EA1, 367m for EA3, and 735m for EA6 for the effective last scattering event. The sensitivity studies shown in Fig. A33 of Wagner et al. reveal that O4 dAMFs should be sensitive to assess aerosol gradients over these altitudes. In particular, if there is a sharp gradient at 500m, the O4 dAMFs for EA1 are systematically larger than at EA3 (consistent with the tail-shape of the box-AMFs expected for these EAs, and the above altitudes for the last scattering event). However, no such behavior is observed in the measurements. In fact, the measured O4 dAMFs slightly decrease from EA3 to EA1. And this shape in EA splits of the O4 dAMFs is very well reproduced based on the ceilometer shape information (consistent with 82% of the ceilometer signal actually originating from the fine mode also at 1064nm). The EA split among measured O4 dAMFs is well reproduced at all EAs by RTM.

This is only consistent with 1) the absence of sharp gradients below 500m, and suggests the PM gradients measured near the surface are in fact a good proxy. The gradient in surface PM strongly suggests a gradient in the surface extinction of a factor of 2 is expected between both days (compare Fig.1 and Fig. 9). This is only achieved if 2) 28% of the lower AOD are redistributed to higher altitudes. Any lower number would overestimate surface extinction. Note that surface extinction is probably the only place where MAX-DOAS can constrain altitude resolved extinction well. Finally, 3) if 28% of AOD are redistributed the measured and predicted O4 dAMFs values agree quantitatively at all EAs.

The information from b and d combined thus provide strong experimental evidence in support of the hypothesis that uncertain aerosol vertical profiles are the primary cause for the correction factor on 8 July.

I consider this evidence as fully consistent also with the information provided in Appendix 5D, which shows that the fine mode aerosol is responsible for the major share of the signal detected by the ceilometer on 8 July, but not on 18 June.

Interestingly, the inferred profiles vary much more strongly on 8 July, and do not vary much on 18 June, possibly providing an important clue on what is driving the different behavior between both case study days (by affecting the initialization of RTM). In this context, I liked to point out that in Ortega et al. 2016 uncertainty due to wavelength scaling of the aerosol extinction profile is minimized (airborne HSRL_532nm was compared with O4 at 477 and 360nm in both Ortega et al. 2016 and Volkamer et al. 2015). Generally speaking, no correction factor was needed if information about aerosols aloft was well characterized in our previous

work where HSRL was available. As you know, HSRL overcomes the fundamental limitation of characterizing sub-Rayleigh aerosol by measuring Rayleigh back-scatter directly, which greatly enhances the aerosol contrast in air where aerosol extinction becomes sub-Rayleigh. Sub-Rayleigh aerosol extinction becomes an issue with interpreting ceilometer data, which require to define a "zero" aerosol aloft to decouple aerosols. There is a fundamental limitation in that the ceilometer cannot measure sub-Rayleigh aerosol. In our aircraft campaigns comparing HSRL and AMAX-O4 inferred aerosol extinction, both sensors find aerosols typically become sub-Rayleigh at altitudes above 4-6km (compare e.g., Fig. 3 in Volkamer et al. 2015). This behavior we have observed over continents, and over oceans, and it is further generally also the altitude range where the ceilometer profiles during MADCAT are close or below the Rayleigh extinction (you could calculate the Rayleigh extinction line, and add it into Fig. 9). This is probably the reason why the extinction profile shape extracted from identical ceilometer data by different groups varies so much at altitude (Fig. 9).

In summary, I see no information in this paper that would not be compatible with the explanation presented in Ortega et al. 2016. And in fact, your paper makes an important contribution in that it helps establish that uncertain aerosol profiles aloft have probably a larger uncertainty than has previously been recognized.

> In addition to these observations, we can assume that coarse mode aerosol is probably located at lower altitudes than the fine mode aerosols.
> All of these observations and assumptions have their uncertainties. Nevertheless, taking all information into account, I conclude that the scenario of a 10% redistribution is the most probable.

Taking also the information from d) into account, a larger redistribution is justified. See above.

> The results of the sensitivity studies for the different profiles and their compatibility with the above stated observations and assumptions will of course be provided in the paper.

I agree.

> Then not only the editor and the reviewers, but also the readers can reach their own conclusion on what they think is most probable.

> I hope you can agree to that procedure.

I would strongly advise against an approach that involves the Editor. Let alone you are the Editor in Chief of the Journal, and this could open all kinds of worms... I see no reason not to resolve this before submission. Its mostly language really, as I see it. And input from the co-authors could also be helpful.

Please find my response to the individual points below.

I am adding some short responses below as well.

Best regards,

Thomas

On 07.02.2019 03:39, Theodore Konstantinos Koenig wrote:

Hello Thomas,

I present my thinking at more length below, but skipping to the conclusions: the SZA dependence highlights the 30% redistribution is not ultimately fully consistent, and I further suspect that a fully consistent solution is not easily found. Rather, these sensitivity studies can be presented and framed to highlight that discrepancies between modeled and measured O4 can be explained by such changes which cannot be ruled out by available data. This points to the potential to leverage the angle specific O4 dAMFs and SZA dependent dAMFs in conjunction with certain assumptions to make adjustments and perhaps reach a fully consistent solution. That is beyond the scope of the paper. Still, I think the sensitivity studies can highlight that while such a exercise is challenging, at present poorly constrained, and perhaps impractical, it is not impossible.

see my general comments above

I outline my thinking below, my apologies for the length:

As I understand the analysis in the manuscript, an angstom exponent was derived for a given point in time, and then applied to the entire ceiliometer profile.

This is not exactly true. The angstrom exponent was determined for the selected period. Also it is not applied to the entire ceilometer profile. The altitude dependence of the size distribution (and thus the angstrom exponent) is implicitly accounted for by the re-distribution of 10% of the total extinction.

It is not possible in my opinion to recover information at 360nm accurately from measurements at 1064nm. We also have more direct evidence that supports a larger re-distribution of AOD.

The extinction for monodisperse large particles is relatively flat with wavelength, whereas for smaller particles the extinction changes more rapidly with wavelength. However, the coarse fine dichotomy is not the only concern, see for instance [Schuster et al., 2006](). While the precise size distribution of coarse mode aerosol does not change the angstrom exponent, the specifics of particle distributions in the fine mode can act as a strong lever on the angstrom exponent.

Of course this is true. However, the wavelength dependencies are intrinsically taken into account by the use of the phase functions for fine and coarse mode aerosols derived from the AERONET inversion. This information is not perfect, but describes best the aerosol properties during that day.

I agree that you have done what can be done, Thomas. But a case with high wavelength dependence (8 July) should result in a larger uncertainty due to wavelength scaling than a case with a lower wavelength dependence (18 June). I think nobody would argue that

a measurement at 360nm would be more valuable to inform 360nm than a measurement at 1064nm -- but it seems to me that your argument in Appendix 5D can be misunderstood that way. We can agree to disagree here.

> The absolute contribution of fine mode aerosol provides some measure of the expected inaccuracy of adapting an extinction profile from a different wavelength. I don't think that the difference in the relative contribution of the aerosol modes to the different measurements is a relevant metric for this effect. Unfortunately I don't have a firm constraint to offer beyond the fact that it should be more important when more aerosol volume is in the fine mode.

> Intuitively, I had the same expectations at the beginning. Nevertheless, by taking the optical depths and the phase functions of fine and coarse mode aerosols into account, it turns out that on 8 July even at the rather large wavelength of the ceilometer measurements the fine mode dominates the ceilometer signal (82%). I think this is the key point and tells us that on 8 July the ceilometer measurements at 1064 nm are a very good proxy for the aerosol extinction profile shape at 360 nm.

I agree - and had made a similar point in my email in suggesting to construct Table A28 from a perspective of the fine mode. Note that Mie resonances of fine mode particles happen at the wavelengths around the O4 observations. They do not happen at the wavelengths where the ceilometer strongly interacts. The ceilometer thus does not constrain the Q (extinction enhancement) of fine mode particles well, even though it is sensitive to fine aerosols. The exact wavelength and magnitude of Q depends on the refractive index and many other parameters (see earlier email), which strongly vary with wavelength. And all of this introduces uncertainty that goes well beyond the scope of this paper.

Its your paper, Thomas, but I strongly advise against putting too much faith into the calculations in Appendix 5D.

> In this context it is not surprising that June 18 has a correction factor closer to 1 than July 8, because overall the aerosol are larger and therefore a constant angstrom exponent with altitude is more likely to be closer to the truth. For July 8, while aerosol size distribution profiles in the atmosphere are complex they generally tend to get smaller and narrower rising through the troposphere. Both these effects increase the

angstrom exponent and as such there is expected to be a general tendency that when transferring an extinction profile from longer to shorter wavelengths that it will be relatively enhanced at higher altitudes. Atmospheric layering of course also plays a role.

Of course, I agree that in general the size distribution varies with altitude. This is what our whole discussion is about. But I think this is the case for both days: We should expect that the size of the aerosols in general decreases with altitude. The important difference is that on 8 July the relative contribution from the coarse mode to the ceilometer signal is much larger than on 18 June which complicates the quantitative interpretation.

See above. I agree its complicated.

The sensitivity studies in Appendix 6 are therefore consistent with expected effects in the absence of better constraints highlighting layers especially. I cannot offer a corollary to your ~10% bound, but I don't believe the ~30% effect can be completely rejected either. If another sensitivity study can further illustrate the principle while ignoring any layers, a naive smooth altitude dependence of scaling might serve i.e multiply the extinction profile by [(1-x) + (2x/7 km-1) * altitude], where x between between 0.1 and 0.3 should serve to illustrate.

Initially, I also had this thought. Such a smooth altitude dependence is surely more realistic than a re-distribution between layers. However, I decided to use the more extreme re-distributions between layers for the sensitivity studies because of two reasons:
a) we have no information on the altitude dependence of the fine and coarse mode fractions. All assumed re-distributions are simply assumptions (of course with some plausibility)
b) from the extreme scenarios the overall magnitude of the effect can be estimated, and that is what matters.

I agree with all that is said here. But I do think the combination of b) and d) above holds new merit that should be considered. It supports redistribution out of the surface layer. Since the ceilometer is sensitive mostly to fine particles, and faces the fundamental limitation of loosing sensitivity for sub-Rayleigh aerosols, a redistribution into the higher aerosol layer is plausible.

Note that we did not optimize elevated layers using information from a) yet. There would be lots of room to optimize this distribution, and i.e. elevated layers, based on the SZA dependence in future work. I think this is worth pointing out in the section on recommendations in the revised manuscript.

> Please let me know if there is anything compelling
> which I am overlooking. Perhaps we will not reach
> agreement. In any case, I would appreciate your
> thoughts on this perspective and framing, it would be
> useful in determining some of the specific comments.
> Thank you for bringing this extensive exercise together
> and for your responsiveness these last days.

> My current plan is to prepare an updated version of the manuscript in
> the next two days and send it to you. If your detailed feedback contains
> further fundamental points, it would be good to know these points
> before I prepare the updated version. I want to avoid too many
> iterations.

I liked to resonate Ted comments, and thank you for your responsiveness, and your patience.

Its a massive piece of work, with many interlocking pieces. Its at present also a very complicated paper to read. I am hoping that our discussions, albeit lengthy at times, are helpful, and can be used to simplify the paper. I look forward to seeing the revised version.

-Rainer
* * *
email from Thomas, 11.02.2019

Dear Rainer, dear all,

please find attached the updated version of the paper. The changes compared to the previous version are in sections 4.2.1, 5.2, and appendices A5 and A6.
Please let me know if you can agree to this version. Then I will send it around to the other co authors.

Concerning the last email from Rainer, I don't want to respond to each individual point, because the communication is already quite complicated. Below I give my feedback to the points which - in my view - are the most important ones:

a) what can we learn from the in situ measurements? In my opinion we can use them only in a qualitative way for the comparison between both days (as already discussed in the paper). But it

is not possible to make a direct quantitative link between the in situ measurements and the ceilometer profiles, because different quantities are measured (backscatter signal versus aerosol mass concentration).

b) Rainer states that he agrees that we disagree. I also agree to that. Overall, the input from your group has led to large improvements of the paper, especially with respect to the shape of the aerosol extinction profile and its uncertainties, and to the recommendations for future comparison exercises. I hope that you find the discussion of the aerosol profiles and their uncertainties in the revised version acceptable for you.

c) Rainer suggests that the discussion between him/his group and me should not be shared with the editor. I must say that I strongly disagree. It is one important feature of AMT that important discussions have to be made available to the editor, to the reviewers, and also on the discussion web page.

Best regards,

Thomas

The attached pdf file is: O4_scaling_factor_10022019.pdf
* * *
email from Ivan, 11.0.2019

Dear Thomas,

Thanks for sending a revised version. It was hard to track down everything based on emails. I included a few comments using the annotation tools in Adobe Reader (see attachment). Below are some general comments:

- The appendix is quite long. Sometimes I had to look for key information in the appendix when, in my opinion, it should be included in the main text. Especially, regarding how the aerosol extinction profile at 360nm was derived. I think all assumptions should be included clearly in the main text instead of directing the reader to the appendix several times in a single paragraph.

- In the abstract you mention:

"*One important recommendation for future studies is that aerosol profile data should be measured at the same wavelengths as the MAX-DOAS measurements.*"

Similarly in the conclusion:

"*one important  quality of the aerosol data sets is crucial to constrain the radiative transfer simulations. For example, it is*

*recommended that LIDAR instruments are operated at wavelengths close to those of the MAX-DOAS".*

I fully agree. Note that in Ortega et al. (2016) this approach was already used. In that study, we used highly resolved independent extinction profiles measured at 355, 532, and 1064 nm from HSRL, i.e., no assumptions about construction of extinction profiles. However, detail information in that regard is missing in the manuscript. Ortega et al. (2016) and Volkamer et al are mentioned in the manuscript but it is not recognized the approach used and the key HSRL products used.

- As far as I can tell from the manuscript the only parameter that brings SF to unity is if aerosol extinction aloft is included, is that correct?. Maybe I am missing another factor that brings the SF to unity?. Leaving behind assumptions and whether this is true or not I would mention parameters that bring SF to unity in the abstract/conclusions and of course that more measurements are needed, as you already mentioned. I am mentioning this because still you mention in the abstract that *"Besides the inconsistent comparison results for both days, also no explanation for a O4 scaling factor could be derived in this study"*. It is hard to reconcile this though. It is clear that the SF is not explained by O4 MAX-DOAS measurements, but it has to be something in the state of the atmosphere causing the need of SF.  In Ortega et al (2016) we concluded that independent highly-resolved profiles were needed and elevated aerosol layers were identified and if not accounted for the SF < 1 is needed. I am not saying this is the case always, but it has to be something in the state of the atmosphere causing this the forward model.

- Following up above, elevated aerosol layers are really more frequent that we thought, see Berg et al. (2015) and references therein. https://agupubs.onlinelibrary.wiley.com/doi/full/10.1002/2015JD023848

- I am not really sure but have you seen if using CALIPSO extinction profiles might help?. Again, I am not sure if there is an overpass or if they measure in boundary layer and even if they can be used we might have the same issues.

Thanks for all this important work and I apologize for not sending comments before.

Greeting to you & your group,

Ivan

The attached pdf file is O4_scaling_factor_10022019_io.pdf

Email from Thomas, 12.02.2019

Dear Ivan,

many thanks for your feedback! Please find my replies below.

Please let me know until 13 February if you agree to be co-author of the paper in the current form (including the changes described below). I have to send the updated version to all other co-authors to receive their feedback before I submit the revised version of the paper.

Rainer, Ted, please also let me know until 13 February if you agree to be co-author of the paper.

Many thanks,

Thomas

On 11.02.2019 20:51, Ivan Ortega wrote:

Dear Thomas,

Thanks for sending a revised version. It was hard to track down everything based on emails. I included a few comments using the annotation tools in Adobe Reader (see attachment). Below are some general comments:

- The appendix is quite long. Sometimes I had to look for key information in the appendix when, in my opinion, it should be included in the main text. Especially, regarding how the aerosol extinction profile at 360nm was derived. I think all assumptions should be included clearly in the main text instead of directing the reader to the appendix several times in a single paragraph.

I would prefer to leave the structure as it is. I understand your concern, but there is so much information in the paper that a lot of details have to be put to the appendix. Nevertheless, in the main text it is clearly stated how the details can be found.

- In the abstract you mention:

"*One important recommendation for future studies is that aerosol profile data should be measured at the same wavelengths as the MAX-DOAS measurements.*"

Similarly in the conclusion:

"*one important quality of the aerosol data sets is crucial to constrain the radiative transfer simulations. For example, it is recommended that LIDAR instruments are operated at wavelengths close to those of the MAX-DOAS*".

I fully agree. Note that in Ortega et al. (2016) this approach was already used. In that study, we used highly resolved independent extinction profiles measured at 355, 532, and 1064 nm from HSRL, i.e., no assumptions about construction of extinction profiles. However, detail information in that regard is missing in the manuscript. Ortega et al. (2016) and Volkamer et al are mentioned in the manuscript but it is not recognized the approach used and the key HSRL products used.

In the parts of the text, where it is stated that it is important to use LIDAR measurements at the same wavelength, the reference to Ortega et al., 2016 was added.

- As far as I can tell from the manuscript the only parameter that brings SF to unity is if aerosol extinction aloft is included, is that correct?.

No, that is not correct, see section 5.2. There several potential reasons for the discrepancies are listed.

Maybe I am missing another factor that brings the SF to unity?. Leaving behind assumptions and whether this is true or not I would mention parameters that bring SF to unity in the abstract/conclusions and of course that more measurements are needed, as you already mentioned. I am mentioning this because still you mention in the abstract that *"Besides the inconsistent comparison results for both days, also no explanation for a O4 scaling factor could be derived in this study"*. It is hard to reconcile this though. It is clear that the SF is not explained by O4 MAX-DOAS measurements, but it has to be something in the state of the atmosphere causing the need of SF.

I disagree here. It is not clear that the reason has to be something in the atmosphere. Also high levels of instrument straylight or wrong O4 cross sections could explain the differences, see section 5.2.

In Ortega et al (2016) we concluded that independent highly-resolved profiles were needed and elevated aerosol layers were identified and if not accounted for the SF < 1 is needed. I am not saying this is the case always, but it has to be something in the state of the atmosphere causing this the forward model.

I think it is not clear that it has to be something in the atmosphere, see comment above.

- Following up above, elevated aerosol layers are really more frequent that we thought, see Berg et al. (2015) and references therein.
https://agupubs.onlinelibrary.wiley.com/doi/full/10.1002/2015JD023848

This might be the case, and it is indeed an interesting finding. But I don't see the relevance for this study. Here two days were selected, and all relevant available information is considered.

- I am not really sure but have you seen if using CALIPSO extinction profiles might help?. Again, I am not sure if there is an overpass or if they measure in boundary layer and even if they can be used we might have the same issues.

Unfortunately, Mainz is not seen by CALIOP

Thanks for all this important work and I apologize for not sending comments before.

Greeting to you & your group,

Ivan

Please find below my replies to the individual comments in the pdf. There have been a few comments without text. Maybe my pdf reader has problems here. Please let me know if I missed something important.

Comment 1: This still reads as if only direct sun observations found no need of correction factor. I would change by: "However, many studies came to opposite conclusion, that there is no need for a scaling factor"

My intention was to mention that even direct sun light measurements came to that conclusions, because such measurements are not affected by AMF uncertainties. I thus still think it would make sense to keep the formulation as it is.

Comment 2: Ortega et al. (2016) already presented a study where MAX-DOAS O4 and aerosol extinction were measured at the same wavelength. This important description is missing in the current manuscript.

The reference to Ortega et al. 2016 is given later at several parts in the text. References should be avoided in the abstract. So I prefer not add a reference to Ortega et al., 2016 there.

Comment 3: After this paragraph. I also suggest to include a short description of the methodology followed by studies where SF is unity, i.e., Volkamer et al (20) and Ortega et al. (2016) used independent highly resolved extinction profiles. in Ortega et al. (2016) aerosol extinction was measured at 355, 532, and 1064 nm.

I don't see the need to add such descriptions here.

Comment 4: it is strange to see ? in this equation

The question mark indicates the question whether the expected equality is true.

Comment 5: A description of how this was concluded is missing. Is it based on the 1064 nm ceilometer signal?

The description is given in appendix A5. I see no need to add more information here.

Comment 6: It may be too late to re-arrange, but it would have been nice to read first how the atmospheric conditions were derived before reading about the need of SF.

I see no need to add this information here. What would be gained from it? All relevant detailed information is given later.

Comment 7: Including wavelength of the ceilometer and AERONET is important but missing here. Also, assumptions about extrapolating ceilometer to extinction profiles are 360nm is missing

Please note that the paragraph starts with 'In short, the ceilometer measurements....' This indicates that only the basic principle is described here. In the next sentences the link to appendix A5 is given, where all further details are provided.

Comment 8: Not sure why is set to zero?. Realistically, the aerosol extinction would not be zero.

The reason is stated in the remainder of the sentence: '...because of the further increasing scatter and the usually small extinctions.'

Comment 9: Maybe I am missing an explanation but I don't see the value of comparing extracted exticntion profiles from different groups if all of them are constructed the same way, i.e., scaled by AOD and shape of the ceilometer at 1064nm. I would think extracted exticntion profiles using different methods, based on current independent measurements would be better.

The value of this comparison is to investigate the effects of different procedures. This is important information because not only the fundamental assumptions matter, but also the details of the extraction.

Comment 10: How was this derived?. Instead of showing key information in the appendix, I suggest to include it here.

I prefer to leave the structure as it is, because the derived results matter in the main text. The details for the interested readers are given in the appendix.

Comment 11: Again, I think mentioning that highly resolved and independent extinction profiles were measured in Ortega et al (2016), without assumptions about wavelength dependency is missing

Here the point is the altitude range. I don't see why information on the wavelength is this important here.

Comment 12: Again, Ortega et al. (2016) already use this approach. HSRL measured extinction profiles at 355, 532, and 1064 nm.

The reference to Ortega et al., 2016 was added

Comment 12: Ortega et al. (2016) already use this approach.

The reference to Ortega et al., 2016 was added

Comment 13: Again, one key aspect of Ortega et al. (2016) is that they use HSRL extinction profiles at 355, 532, 1064 nm products.

This information is added.
* * *
email from Rainer, 14.02.2019

Dear Thomas,

find attached the comments from Ted and me combined into a single file. Some short replies to your latest summary is below.

On 2/10/2019 4:00 PM, Thomas Wagner wrote:

> Dear Rainer, dear all,
>
> please find attached the updated version of the paper. The changes compared to the previous version are in sections 4.2.1, 5.2, and appendices A5 and A6.
> Please let me know if you can agree to this version. Then I will send it around to the other co authors.

I am fine with your changes in Sections 4.2.1, 5.2, and made some additions to reflect what I found was missing. I also added comments in section 4.3.5 "Effect of the temperature dependence of the O4 cross section". Please take a look, and let us know you are on board with the suggested changes.

> Concerning the last email from Rainer, I don't want to respond to each individual point, because the communication is already quite complicated. Below I give my feedback to the points which - in my view - are the most important ones:
>
> a) what can we learn from the in situ measurements? In my opinion we can use them only in a qualitative way for the comparison between both days (as already discussed in

the paper). But it is not possible to make a direct quantitative link between the in situ measurements and the ceilometer profiles, because different quantities are measured (backscatter signal versus aerosol mass concentration).

Surface PM scales as volume, and so does surface extinction. So I think my argument carries merit. It is true that in-situ / column comparisons are always complicated, but this is probably partially mitigated if temporal averages are compared in a relative sense (as I did). But ok to frame this as a qualitative argument (as done in Section 5.2)

> b) Rainer states that he agrees that we disagree. I also agree to that. Overall, the input from your group has led to large improvements of the paper, especially with respect to the shape of the aerosol extinction profile and its uncertainties, and to the recommendations for future comparison exercises. I hope that you find the discussion of the aerosol profiles and their uncertainties in the revised version acceptable for you.

I am glad you feel that way. It was an interesting and somewhat open ended discussion. I see nothing in this paper that contradicts our own earlier work on the topic (incl. Thalman and Volkamer, 2010; Thalman and Volkamer, 2013; Spinei et al., 2015; Volkamer et al., 2015; Ortega et al. 2016). Several of these papers include data from collocated airborne multi-wavelength HSRL, near surface extinction all the way to the surface, and comparisons at multiple O4 wavelengths. That is not a trivial statement.
I am fine to be a co-author (with the attached changes).

> c) Rainer suggests that the discussion between him/his group and me should not be shared with the editor. I must say that I strongly disagree. It is one important feature of AMT that important discussions have to be made available to the editor, to the reviewers, and also on the discussion web page.

This is not what I said. You are misrepresenting my email.
My point is that differences are best sorted among all co-authors first, and ideally are reflected in the paper.
Feel free to use for following text for the purposes of circulating to co-authors (I'd need to think a bit more if I was to write for a permanent archive such as AMTD):
The remaining disagreement ranks around the importance of lacking vertically resolved aerosol properties, and uncertainties specific to MADCAT. Boulder and MPI agree that Appendix A5 provides an interesting and useful semi-quantitative argument about the origin of ceilometer signal at 1064nm. MPI claims a quantitative and low uncertainty for inferred aerosol vertically resolved information at 360nm from AERONET measured column properties. Boulder notes that the uncertainty due to wavelength scaling needs to hold for both days, and the argument in Appendix A5 implies a four times smaller uncertainty on a day when AOD varies much more strongly with wavelength (8 July), than on a day when AOD varies weakly with wavelength (18 June). Boulder further notes that less than 3 times higher uncertainty holds potential to explain the correction factor quantitatively on 8 July for all EAs in form of elevated aerosol layers. The resulting aerosol distribution has not been further optimized for SZA effects. Boulder notes that 60-70% AOD to reside above 1km is fully consistent with previous observations during TCAP, where multi wavelength airborne HSRL measurements constrain aerosol extinction all the way to

the surface, and elevated aerosol layers are key to providing closure on O4 (Ortega et al., 2016). Boulder further points out that the low uncertainty estimate provided in Appendix A5 does not resolve an inconsistency that still exists between the relative abundance of near surface PM (lower on 8 July), and the near surface aerosol extinction (roughly constant) between both days, and that the factor of 3 higher uncertainty holds potential to resolve this inconsistency. An attempt has been made to reflect this discussion in the revised Section 5.2.

Of course I would also be happy to post a public comment to this effect in the AMT discussion forum, if needed.

I have made an attempt to add suggestions in the file also on other points that I did not find in your above summary.

Best regards,

-Rainer

The attached pdf file is O4_scaling_factor_10022019_TKK_RMV-1.pdf

Individual replies to the comments from Rainer and Ted in the pdf (O4_scaling_factor_10022019_TKK_RMV-1.pdf).

Comment from Ted, page 1:

Name removed, because it is not expected that Ted and the others agree to be co-author of the current version. They will be asked again if they want to be co-author after feedback from all other co-authors has become available.

Comment from Rainer, page 1: see above

Comment from Ted, page 1: see above

Comment from Ted, page 2: This is 0.81 below.

Corrected (further corrected tp 0.82, because slightly wrong factor for the profile '0.75km' was used.

Comment from Rainer, page 2: do you mean "should be collected"?

The text was changed to 'should be collected and used'

Comment from Ted, page 3: For our internal discussions we have found the form of dSCD/VCD for dAMF most simple to consider. Could it perhaps be added as further equality?

In principle, this could be added. But I think, the equation in its current form is more consistent.

Comment from Ted, page 7: 'n' was added

Comment from Ted, page 8: 'to' deleted

Comment from Rainer, page 12: Add: These deviations are lower than during the case study days in Ortega et al. (2016), where deviations between observed and calculated O4 profiles in the U.S standard atmosphere were found to be 13-18%.

Why should this be added here? It is later discussed that of course the deviations can be stronger for different locations and seasons. There the reference to Ivan's paper was added.

Comment from Rainer, page 12: the suggested sentence was added: 'This assumption reflects a practical limitation of the ceilometer likely responsible for the larger variability in the profile shape aloft by different groups.'

Comment from Ted, page 12: The sentence was changed to 'This assumption reflects a practical limitation of the ceilometer likely responsible for the larger variability in the profile shape aloft by different groups.'

Comment from Ted, page 12: The suggested sentence was added: 'This effect is further examined in Appendix A6'

Comment from Ted, page 13: I do not agree with this conclusion. Because Appendix A5 leverages the AERONET size distributions and AOD which are column properties I do agree that the column properties of the fine mode aerosol are well represented. I do not think a statement can be made regarding the profile. Can we present the information without this specific statement?

I still think the statement is correct.

Comment from Rainer, page 13: I disagree with this statement. While backscatter signal at 1064nm originates largely from fine mode aerosoll, there is no profile shape information at 360nm that can be recovered from column properties.

I think this statement is correct. This is one important point where we disagree.

Comment from Ted, page 13: I would rather consider the 10% and 30% redistribution from lowest layer to highest layer cases quoted in the main text. Understanding that we disagree as to whether the latter is reasonable, would the following language be acceptable?

"... subdivided into 3 layers (0-1.7 km; 1.7-4.9 km; 4.9 - 7 km), and extinction was redistributed from the lowest layer to the highest layer. It was found that redistributing 30% of total AOD this way increased the O4 dAMFs by 25%. However, such redistribution cannot be specifically justified."

I use 25% here as I don't have a more accurate number.

I think the case with the 25% or 30% redistribution fits best to section 5.2

Comment from Rainer, page 13: , where extinction is less well constrained (i.e., assumptions of zero aerosol extinction give rise to significant variability above 5km on 8 July, compare Fig. 9).

I think it makes no sense to add this statement here. The uncertainty in the study of Ortega et al., 2016 is probably even larger, because the maximum altitude of the profiles was even lower.

Comment from Ted, page 14: appendix is now consistently written in lower case.

Comment from Ted, page 14: Might be worth referencing the primary viewing direction briefly, since the sensitivity to the phase function is in part a function of the prevailing solar relative azimuth angle.

I think it is not necessary to add this information here. The azimuth angle was provided in the general description of the measurements. Also, the sacttering angle also depends on the solar zenith angle.

Comment from Rainer page 17: corrected

Comment from Rainer, page 18: since simulated spectra have access to complete information, the larger difference in synthetic data cannot be a problem with the cross-sections.

The synthetic spectra used the cross sections for all temperatures. However, they show a slight inconsistency as function of temperature. Thus in the syntehtic spectra the temperature dependence is not as smooth as it (very probably) should be. The temperature range of this inconsistency (~210 – 265 K) corresponds to a large atmospheric altitude range (~5 – 10 km).

The non smooth temperature dependence of the $O_4$ cross sections are likely to explain the observed inconsistent fit results, because the fit results largely improve if only one $O_4$ absorption band is used in the fit.

It must be a problem introduced by the noise, which I read somewhere is larger than noise in the measurements.

Actually, here results for spectra without noise were shown. This information was added to the text (sections 4.3.1 and 4.3.2).

Comment from Rainer, page 18: This is not an "inconsistency" in our data, but rather the wavelength dependent dsigma/dT is a "feature" of the spectra.

The word inconsistency suggests something is wrong with the spectra. While indeed the xs is a physical property. The inconsistency here is introduced by the DOAS fit, and driven by differences in dsigma/dT between different O4 bands that leads to bias if two bands with different dsigma/dT are fitted as part of a single fit window.

I think that's what you are trying to say here. But it is not what the text currently says.

I think the non smooth temperature dependene shown in Fig. A27 indicates a (slight) inconsistency. Of course the Thalman and Volkamer O4 cross sections are a very useful data sets, which helped a lot in improving the O4 spectral analyses compared to earlier O4 cross section measurements.

But this non-smooth temperature dependence is not what should be expected.

Comment from Rainer, page 18: 'Thalman and Volkamer' corrected

Comment from Rainer, page 18: The temperature dependence of the peak xs at 380nm is shown in Fig. S2 of T&V2013. It looks continuous there.

This is not true. Have a closer look at the values for 380 nm. There is exactly the same slight inconsistency found between 233K and 253K. The reference to Fig. S2 is added to the text.

dsigma/dT for each band behaves a little differently, i.e., larger at 380nm than at 360nm. For some bands, the peak xs is flat down to 253K, and a change in the band shape kicks in only at lower temperatures. When you renormalize at 360nm, you are transferring some of the 360nm behavior to 380nm, which distorts the picture at 380nm.

The normalisation was applied to make the ratio between both peeks more clearly visible.

The caption to Fig. A27 should clearly state that the Figure is constructed from a perspective of a least-squares DOAS fit which weights the peak_sigma more strongly than the band integral absorption.

Probably there is a misunderstanding here. Fig. A27 is simply showing values directly calculated from the original cross sections. No DOAS is applied.

Note that Fig. A27 was updated. In the original version the wrong temperature (223K instaed of 233K) was shown. This mistake is now corrected.

Comment from Rainer, page 18: I disagree with this statement. The change in the peak sigma is compensated by narrowing of the line, leaving the band integral independent of temperature. Compare Fig. 4 in T&V2013.

This does not change the fact that the ratio of the peaks shows a slight inconsistency.

Comment from Rainer, page 18: I am not sure I can agree with this statement. Also for a single band, the least-squares nature of a DOAS fit will weigh peak_xs more strongly than band integral. The situation has thus not changed fundamentally.

If a cross section with two separate peaks is fitted to a spectrum, in which the ratio of both peaks is different, the fit tries to find a compromise for both peaks (one will be over, the other underestimated). The resulting residual is large. If only one peak is fitted, this problem does not occur.

An improvement arises from the fact that dsigma/dT is lower at 360nm than 380nm, and better defined for a single band; furthermore, the temperature dependence in peak-sigma is partially compensated by the lack of a temperature dependence in the band integral (exactly speaking, dsigma/dT is not constant, but itself a function of temperature... ).

I still think that the above eplanation is correct.

Comment from Rainer, page 18: , and wavelength dependent

I still belive that the statement is correct as it is.

Comment from Rainer, page 19: After reading it several times, I think the last four lines here really belong into the next paragraph.

I still think the text is correct here as it is.

The larger difference for synthetic data surprises me. One important difference being noise, which may be shielding band-shape differences, and mislead the DOAS fit into wavelength dependent differences in dsigma/dT as described in the previous paragraph.

As stated above, the synthetic spectra without noise were used here.

Comment from Rainer, page 19: This is not obvious to me.

It seems to me then that there is no difference expected between synthetic and measured data; in particular, the synthetic spectra are based on complete information. The differences are thus surprising. The differences could be due to either the other cross-sections, or noise, or a combination of the two.

Indeed, the differences are surprising, but can be understood as explained in the text..

Comment from Rainer, page 19: An alternative explanation for the different behavior of synthetic and measured spectra is noise.

The band shape effect is tough to unravel experimentally from noise at the UV wavelengths. We tried in the cold uFT... see Fig. 5 in Spinei et al. 2015.

As stated above, the synthetic spectra without noise were used here.

An important evidence is also Fig. S2 in the Supplement of T&V2013, where the peak-sigma is compared with the balloon profiles from Klaus Pfeilsticker. The agreement is remarkeable at all temperatures. However, noise in the ballon spectra did not reveal a band-shape change, which lead Klaus to attribute the change in peak sigma (T) to a change in the equilibrium constant of O4 (we now know its a band-shape change, and the integral absorption is independent of temperature).

As stated above, the slight inconsistency is also seen in Fig. S2 in the Supplement of T&V2013

(the Fig. S2 of the Supplement is copied below; the magenta ellipse was added to mark the (slight) inconsistency)

[Figure]

Figure S2: Comparison of Peak $\sigma_{O4\text{-}CIA}$ of this work to Figure 3 of Pfeilsticker et al. 2001. The red triangles represent our data, while the black circles are measurements from a balloon-borne DOAS instrument, combined with the room temperature cross-section of Greenblatt et al. (black diamonds).

Can noise be ruled out to explain the different behavior in the synthetic data? I think its worth mentioning here.

See above

Comment from Rainer, page 19: you mean to say "unity"?

No, 'zero' is correct

Comment from Rainer, page 19: wavelength dependent differences in the temperature dependence

I still think 'inconsistency' is correct.

Comment from Ted, page 20: I understand this to the random uncertainty only, but that may be worth saying more explicitly.

OK, this information is added.

Comment from Ted, page 21: corrected

Comment from Ted, page 22: There is not much discussion of July 8. I would add a sentence addressing Appendix 6 here. Perhaps:

"Significant redistribution of aerosol extinction to high altitudes on July 8 results in ratios of simulated and measured dAMFs not significantly different from unity."

I think this still fits under the subsection heading even if it is more explicitly dealt with in the next subsection.

I think this statement does not fit well here. This section is on the differences between both days. The uncertainty about the profile shape of the extinction at 360 nm is much larger on 18 June than on 8 July.

Comment from Rainer, page 22: possibly unrealistic

The text was changed to: 'This section describes possible (but probably unrealistic) changes…'

Comment from Ted, page 22: I am not aware of a specific lack of agreement of contradiction.

From the AERONET inversion products a 10% redistribution was obtained, but not 27%.

Comment from Rainer, page 23: No attempts were made to optimize layers aloft in this respect. At the same time, the lower near surface extinction in this scenario is qualitatively consistent with the lower PM mass loadings measured near the surface on 8 July, which appear to be at odds with the rather constant near surface extinction between both days.

Only with the very large 27% redistribution the measured O4 dAMFs could be matched. But this scenario systematically understimates the zenith observations at large SZA.

It is only possible to 'optimise' the one aspect or the other.

Comment from Ted, page 23: I would cite here also that in situ aerosol measurements at the distributed site are consistent.

Good point! This information was added.

Comment from Ted, page 23: This would need to manifest differently between the two days and consistently so across the different instruments, correct? That would require some common environmental cause for increased straylight or else be vanishingly improbable.

Comment from Rainer, page 23: I agree. Consider removing these sentences.

In principle I agree to this argument. However, on 18 June the uncertainties of the aerosol extinction profile are much larger than on 8 July. So we don't have to expect similar results on both days.

Comment from Ted, page 24: Several nested subclauses here, and I think an unnecessary comma. Perhaps rephrase to: "However, as long as the reason for this deviation is not understood, it is unclear how ..."

OK, corrected.

Comment from Ted, page 26: Replaced by more specific table footnotes below?

Not clear what is suggested. It seems to me that everything is OK here.

Comment from Rainer, page 27: V15 should be listed as separate from S15. Note that they deal with different case studies. V15 is focused on an evaluation of O4-inferred aerosol (case "with aerosol"), while S15 is focused on an evaluation of O4 in a Rayleigh atmosphere. Maybe add a footnote to this effect here.

Both references are now separately liested.

Comment from Ted, Page 90: corrected

Comment from Ted, page 92: 'along' inserted

Comment from Ted, page 92: changed as suggested

Comment from Rainer, page 93: Please show both case study days up to 8km. The scale seems to be cut off on 8 July.

These are examples of the MPIC extraction which set the extinction to zero above 6 km.

At the beginning of appendix A5 the following information was added: 'Note that in this section the individual steps are described according to the MPIC procedure. The extracted profiles from other groups differ slightly compared to the results of the MPIC procedure, especially with respect to the altitude above which the extinction was set to zero (see Fig. 9).'

Comment from Rainer, page 96: a modest enhancement of the O4 dAMF is found in the elevated EAs (see appendix A6).

In both cases the enhancement is the same (+17%). The text (wrongly stating +15%) was corrected accordingly.

Comment from Ted, page 97: corrected

Comment from Ted, page 97: corrected: 15% => 17%

Comment from Ted, page 98: One or more numbers to compare with the 3% and 7% effects quoted above would be helpful. Similarly, regarding my suggestion to quote one such number in the main text.

Not clear what is suggested here.

Comment from Ted, page 98: corrected

Comment from Ted, page 98: As an aside, I think the remaining shortcomings in fact indicate that there is unleveraged information from the MAX-DOAS measurements, but that is beyond the scope of this work.

Not clear to me, what exactly is meant here.

Email from Thomas, 17.02.2019

Dear Rainer,

many thanks for your feedback!

It seems that we can not come to an agreement about the extraction of the profile shape. I understand your email that you will not agree to be co-author if your suggested changes are not implemented.

I have now sent the revised manuscript as well as the responses to the reviewer comments to the co-authors. I will also send them the protocol of our discussions. After I have received their feedback, I will send the manuscript again to you. Maybe you can then still agree to become co-author(s).

Best regards,

Thomas

Email from Thomas, 09.03.2019

Dear Rainer, Ivan, Ted,

attached I send you the updated version of the paper based on the feedback of the other co-authors. Please have a look at it and let me know until Sunday if you agree to be co-author.

Also attached is the protocol of our email exchanges, which will be uploaded to the discussion page.

Many thanks for your feedback!

Thomas

---

## Author Comment (AC4) · 11 Mar 2019

[revised manuscript text omitted]
together with the assumption that the coarse aerosols are probably located close to the surface
led to a repartitioning of parts of the aerosol extinction profile (extracted assuming a constant
LIDAR ratio). This repartitioning led to a decrease of the aerosol extinction close to the
surface which is balanced by an increase at higher altitudes (see Fig. A34). The $O_4$ dAMFs
calculated for the modified profile are by about 15 % larger than those for the standard
settings (for details see appendix A5).
The effect of elevated aerosol layers (see Ortega et al., 2016) was further investigated by
systematic sensitivity studies (appendix A6). On both selected days enhanced aerosol
extinction was found at elevated layers (Fig. 9). Compared to those reported by Ortega et al.
(2016) the profiles extracted in this study reach even up to higher altitudes. For the
investigation of the effect of changes of the aerosol extinction at different altitudes, the
aerosol extinction profile on 8 July was subdivided into 3 layers (0-1.7 km; 1.7 – 4.9 km; 4.9
– 7 km), and the extinction in the individual layers was increased by +20% or + 40 %. It was
found that even a strong increase of the aerosol extinction at high altitudes by 40% leads only
to an increase of the $O_4$ dAMFs by 7 %. Here it should be noted that on 8 July no indications
for such a strong underestimation of the aerosol extinction at high altitudes are found.

[revised manuscript text omitted]

**5.2 Recommendations**
Based on the findings of this comparison study, recommendations for similar future studies
are derived. Part of them are also of interest for the interpretation of $O_4$ measurements in
general.
a) VCD calculation
Temperature and pressure profiles representative for individual days should be used. If such
profiles are not available, also profiles extrapolated from surface measurements can be used.
They are not 'perfect' but usually the associated errors are at the percent level. The vertical
grid for the integration of the $O_4$ profile should not be coarser than 100m. The integration
should be carried out up to an altitude of at least 30 km. The exact height of the instrument
position needs to be taken into account.
b) Radiative transfer simulations
If available appropriate phase functions (e.g. from Mie calculations) should be used. Here it is
important to note that even if appropriate asymmetry parameters are available, the often used
HG parameterisation becomes very imprecise for forward scattering geometries.
c) Spectral analysis
The spectral range should cover the two $O_4$ bands at 360 and 380 nm. An intensity offset
should be included in the analysis. If the surface temperature differs strongly (more than 25K)
from 300K the effect of the temperature dependence of the $O_4$ absorption should be
considered.
d) Preferred scenarios for future studies
In particular the uncertainties related to aerosols should be minimised. For example,
measurements at rather low AOD (≤0.1) and with low temporal variability should be selected.
Aerosol profiles should be derived from LIDARs/ceilomters which are sensitive down to very
shallow altitudes (low overlap ranges). If possible, Raman LIDARs or high spectral
resolution LIDARs (HSRL) should be used, because from such observations the aerosol
extinction profile can be derived without the assumption of a LIDAR ratio. Also sun
photometer measurements should be available. Besides AOD and the Ångström parameter
also information on the phase function and single scattering albedo from these measurements
should be used.
It would be interesting to cover other meteorological conditions (e.g. low temperatures),
viewing geometries (e.g. low SZA), surface albedos (e.g. snow and ice) and wavelengths (e.g.
477, 577, and 630 nm).
In order to minimise the effects of instrumental properties, the instruments should be well
calibrated and should have low straylight levels. At least two instruments should be operated

[revised manuscript text omitted]

Then, the extinction of the individual aerosol layers were increased by either 20 % or 40 % compared to the original profile. These profiles (referred to as 'without scaling') were used for the simulation of $O_4$ (d)AMFs). A further set of $O_4$ (d)AMFs was simulated for the same profiles, after they were scaled by a constant factor to match the AOD of the original extinction profile (referred to as 'with scaling'). The modified profiles and the ratios of the corresponding $O_4$ DAMFs versus the $O_4$ dAMFs of the original profile are shown in Fig. A34. For the unscaled profiles the $O_4$ dAMFs strongly decrease (by about 30%) if the extinction in the lowest layer is increased. If the extinction in the middle or upper layer is increased a slight increase (about 3 %) of the $O_4$ dAMFs is found. For the scaled profiles different results are found, because the increase of the extinction in one layer is now balanced by a decrease of the aerosol extinction in the other layers.  If the extinction in the lowest layer is increased by 40%, the $O_4$ dAMFs still decrease, but only by about 7%. If the extinction in the middle or upper layer is increased the $O_4$ dAMFs increase by about 3 % and 7 %, respectively.

[Figure]

Fig. A34 Left: Aerosol profiles used for the simulations (see text). Right: Ratios of the $O_4$
(d)AMFs simulated for the modified profiles versus those of the original profile.

---

## Author Comment (AC5) · 11 Mar 2019

[revised manuscript text omitted]

As long as the reason for this deviation is
not understood, it is, however, unclear, how representative these findings are for other
measurements (e.g. from other platforms, at other locations/seasons, for other aerosol loads,
and other wavelengths). Thus further studies spanning a large variety of measurement
conditions and also including other wavelengths are recommended.
****

aerosol concentrations of large particles on 18 June. Also a larger forward peak of the derived
aerosol phase function is found for 18 June. Both effects probably cause larger uncertainties
on 18 June.
c) spectral analysis
Larger uncertainties of the spectral analysis are found for 18 June compared to 8 July. This
finding was surprising, but was also partly reproduced by the analysis of the synthetic spectra.
One possible explanation is the smaller wavelength dependence of aerosol scattering at low
altitudes on 18 June, which mainly affects measurements at low elevation angles. When
analysed versus a zenith reference, for which the broad band wavelength dependency is much
stronger (because of the larger contribution from Rayleigh scattering), larger deviations can
be expected (e.g. because of differences of instrumental straylight, or the different detector
saturation levels). On 18 June also higher (about doubled) $NO_2$ and HCHO concentrations are
present compared to 8 July possibly leading to increased spectral interferences with the $O_4$
absorption, but this effect is expected to be small.
**5.2 Recommendations**
Based on the findings of this comparison study, recommendations for similar future studies
are derived. Part of them are also of interest for the interpretation of $O_4$ measurements in
general.
a) VCD calculation
Temperature and pressure profiles representative for individual days should be used. If such
profiles are not available, also profiles extrapolated from surface measurements can be used.
They are not 'perfect' but usually the associated errors are at the percent level. The vertical
grid for the integration of the $O_4$ profile should not be coarser than 100m. The integration
should be carried out up to an altitude of at least 30 km. The exact height of the instrument
position needs to be taken into account.
b) Radiative transfer simulations
If available appropriate phase functions (e.g. from Mie calculations) should be used. Here it is
important to note that even if appropriate asymmetry parameters are available, the often used
HG parameterisation becomes very imprecise for forward scattering geometries.
c) Spectral analysis
The spectral range should cover the two $O_4$ bands at 360 and 380 nm. An intensity offset
should be included in the analysis. If the surface temperature differs strongly (more than 25K)
from 300K the effect of the temperature dependence of the $O_4$ absorption should be
considered.
d) Preferred scenarios for future studies
In particular the uncertainties related to aerosols should be minimised. For example,
measurements at rather low AOD ($\leq$0.1) and with low temporal variability should be selected.
Aerosol profiles should be derived from LIDARs/ceilometers which are sensitive down to very
shallow altitudes (low overlap ranges). If possible, Raman LIDARs or high spectral-
resolution LIDARs (HSRL) should be used, because from such observations the aerosol
extinction profile can be derived without the assumption of a LIDAR ratio. Also sun
photometer measurements should be available. Besides AOD and the Ångström parameter

**Acknowledgments**

We are thankful for several external data sets which were used in this study: Temperature and pressure profiles from the ERAInterim reanalysis data set were provided by the European Centre for Medium-Range Weather Forecasts. In situ measurements of trace gas and aerosol concentrations as well as meteorological data were performed by the environmental monitoring services of the States of Rhineland-Palatinate and Hesse (http://www.luft-rlp.de and https://www.hlnug.de/themen/luft/luftmessnetz.html). We thank M. O. Andreae and Günther Schebeske for operating the Ceilometer and the AERONET instrument at the Max Planck Institute for Chemistry.

**Tables**

[revised manuscript text omitted]
 original extinction profile (referred to as 'with scaling'). A third set of profiles was created assuming that a certain fraction of the total AOD was relocated from the bottom layer to the top layer. Here fractions of 10%, 25% and 30% were assumed.

The modified profiles and the ratios of the corresponding $O_4$ DAMFs versus the $O_4$ dAMFs of the original profile are shown in Fig. A34. For the unscaled profiles the $O_4$ dAMFs strongly decrease (by about 30%) if the extinction in the lowest layer is increased. If the extinction in the middle or upper layer is increased a slight increase (about 3 %) of the $O_4$ dAMFs is found. For the scaled profiles different results are found, because the increase of the extinction in one layer is now balanced by a decrease of the aerosol extinction in the other layers.  If the extinction in the lowest layer is increased by 40%, the $O_4$ dAMFs still decrease, but only by about 7%. If the extinction in the middle or upper layer is increased the $O_4$ dAMFs increase by about 3 % and 7 %, respectively (see Table A31). For the profiles in which a certain fraction of the total AOD was relocated from the bottom to the top layer, the $O_4$ dAMFs increase strongly compared to those of the standard profiles. If 10% of the total AOD were relocated the increase is similar to that for the modified profile 'below 0.75km' in appendix A5. However, if 25% or 30% of the total AOD were relocated, the $O_4$ dAMFs increase much stronger. For a relocation of about 27% almost perfect agreement with the measurements is found (see Fig. A34). That means for such an aerosol profile simulations and measurements are in agreement wthout the need for a scaling factor. However, it should be noted that such a large redistribution is not supported by the AERONET inersion products (see appendix A5). Here it should be noted that for such a profile, about 73% of the total AOD would be located above about 1.7km. Also, for such aerosol profiles the simulated $O_4$ AMFs for 90° elevation systematically underestimate the measured $O_4$ AMFs at high SZA by about 15% (see Fig. A34), whereas much better agreement is found for the standard settings. The understimation is caused by the high aerosol extinction at high altitudes, which increase the scattering altitude of the solar photons observed at 90° elevation.

[Figure]

Fig. A34 Top left: Aerosol profiles used for the simulations (see text). Top right: Ratios of the $O_4$ (d)AMFs simulated for the modified profiles versus those of the original profile. Bottom: comparison of the measured diurnal variation (SZA dependence) for 90° elevation, and the elevation dependence of the $O_4$ dAMFs for the period 7:00 – 11:00 on 8 July.

Table A31 Ratios of (d)AMFs for 8 July 2013 for the modified profiles with respect to the
original profile

| | low +40 % | middle +40 % | top +40 % | 10% bottom to top | 25% bottom to top | 30% bottom to top |
|---|---|---|---|---|---|---|
| ratio AMF without scaling | 0.95 | 1.03 | 1.03 | | | |
| ratio dAMF without scaling | 0.85 | 1.02 | 1.02 | | | |
| ratio AMF with scaling | 1.00 | 1.06 | 1.04 | 1.07 | 1.12 | 1.20 |
| ratio dAMF with scaling | 0.94 | 1.08 | 1.04 | 1.17 | 1.31 | 1.48 |

---

## Author Comment (AC6) · 11 Mar 2019

The comment was uploaded in the form of a supplement:
https://www.atmos-meas-tech-discuss.net/amt-2018-238/amt-2018-238-AC6-
supplement.pdf

---

## Author Comment (AC7) · 11 Mar 2019

The comment was uploaded in the form of a supplement:
https://www.atmos-meas-tech-discuss.net/amt-2018-238/amt-2018-238-AC7-supplement.pdf

---

## Author Response (AR2)

Dear Jochen,

many thanks for your assessment and suggestions! We adressed all points as detailed below (our answers in blue).

Best regards,

Thomas
* * *
Overall, the manuscript is long but relatively easy to read. I only have a number of technical issues I think you should address before it is published:

In Table 9 you use "?" in two cells in the LIDAR ratio row. I don't think it will be clear to a reader what this means. If the calculations were not performed, then it is more appropriate to use "n/a" to express that the results are not available. There may be other alternatives, but whatever you use has to be clear to the reader.

We replaced the ,?' by , not quantified**' and added the following informatiom below the table:
**uncertainty was not assessed for 18 June 2013, because thecontributions from the coarse and fine mode at both wavelenghs arevery different (see Tab. A28). The uncertainty is thus much larger than on 08 July 2013.

Figure 8, x-axis label in left plot: Change ".. signa" to "…signal".

Corrected

Fig 13: Can you please check the y-axis label vs caption. The label is "SCD(fit)/SCD(RTM)" while the cation refers to a (d)AMF ratio. Those are not necessarily the same quantity. So which one are you plotting?

We corrected the y-axis to dAMF(fit) / dAMF (RTM)

In the figure labels you use (d)AMF, (D)AMF, and DAMF, in the text its (d)AMF and dAMF, while equation 3) defines it as dAMF. You may want to go over the text and figures and make sure you use the same name throughout the masnucript.

We changed the labels consistently to dAMF or (d)AMF We also added the following clarification after equation 3:

Note that in this paper the following notations are used:
AMF:          air mass factor
dAMF:         differential air mass factor
(d)AMF:       air mass factor and/or differential air mass factor
(similar notations are used for the (d)SCDs)

Figure A1: Units are missing

The units are added.

Line 1987 in revised manuscript: change "surface-near" to "near-surface"

Corrected

Figures A9, A12, A15a, lower right plots: Should be dAMF as y-axis label

Corrected